# Conditional Diffusion Models are Minimax-Optimal and Manifold-Adaptive for Conditional Distribution Estimation

**Rong Tang**
The Hong Kong University of Science and Technology
martang@ust.hk

**Lizhen Lin& Yun Yang**
University of Maryland, College Park
{lizhen01,yy84}@umd.edu

## Abstract

We consider a class of conditional forward-backward diffusion models for conditional generative modeling, that is, generating new data given a covariate (or control variable). To formally study the theoretical properties of these conditional generative models, we adopt a statistical framework of *distribution regression* to characterize the large sample properties of the conditional distribution estimators induced by these conditional forward-backward diffusion models. Here, the conditional distribution of data is assumed to smoothly change over the covariate. In particular, our derived convergence rate is minimax-optimal under the total variation metric within the regimes covered by the existing literature. Additionally, we extend our theory by allowing both the data and the covariate variable to potentially admit a low-dimensional manifold structure. In this scenario, we demonstrate that the conditional forward-backward diffusion model can adapt to both manifold structures, meaning that the derived estimation error bound (under the Wasserstein metric) depends only on the intrinsic dimensionalities of the data and the covariate.

## 1 Introduction

Conditional distribution estimation aims to estimate the distribution (or its density if exists) of a response variable $Y$ given some covariate or predictor variable $X$, which is a fundamental problem in statistics with wide applicability in finance, economics (Li & Racine, 2007), biology Krishnaswamy et al. (2014) and social science, to name just a few. The conditional distribution provides a full characterization of the dependence structure of the response variable on the predictors, which allows one to gain deeper insights about the data characteristics beyond those from a simple mean regression model, such as capturing uncertainty and addressing multiple-modality. Conditional density estimation has received significant attention from both statistics and machine learning community with proposed estimators ranging from classical nonparametric estimates such as those based on smoothing techniques (Rosenblatt, 1969; Fan & Yim, 2004; Holmes et al., 2007; Bashtannyk & Hyndman, 2001), Bayesian nonparametric estimates (Norets & Pati, 2017), to some recent methods that utilize deep neural networks (Rothfuss et al., 2019).

Although there is a rich literature on conditional distribution estimation, many existing methods, such as the classical nonparametric estimators based on kernel smoothing (Bashtannyk & Hyndman, 2001; Izbicki & Lee, 2016; Li et al., 2022), suffer from some limitations. One notable drawback of these classical methods is the requirement for the existence of a conditional density function, which is often violated when the response variable $Y$ contains discrete components or itself is a high-dimensional object with low-dimensional structures. As a result, most classical methods can only deal with data with small dimensions, and their performance deteriorates quickly when the dimension increases, thus they suffer from the curse of dimensionality. In addition, these classical estimators generally do not have the ability to adapt to any potential intrinsic structure, such as the manifold structure of the data, a characteristic of many modern high-dimensional datasets. For distribution or density estimation in the unconditional setting, estimators based on deep generative models appear to overcome the aforementioned challenges. Constructing a distribution estimator implicitly by specifying its data-generating process naturally allows singular structures in the data. Additionally, an emerging body of literature on the theoretical understanding of deep generative models, including diffusion-based

models (Chae et al., 2023; Dahal et al., 2022; Chen et al., 2023a; Tang & Yang, 2024), demonstrates that such models provide an estimator of the underlying distribution with convergence rates dependent only on the intrinsic dimension of the data.

Motivated by advancements in deep generative modeling, in this work we explore *conditional diffusion models* based on deep neural networks for conditional distribution estimation, accounting for possible low-dimensional manifold structures on either (or both) the covariate $X$ and response $Y$. Unlike other generative model estimation procedures, such as GANs (Goodfellow et al., 2014) and variational auto-encoders (Kingma & Welling, 2013), which explicitly incorporate low-dimensional structures by operating in or maintaining a low-dimensional latent space, diffusion models operate directly in the original ambient data space. Therefore, it is both interesting and important to formally study whether they can still adapt to any low-dimensional structure, if present. Towards these goals, we consider conditional distribution estimators implicitly defined through a class of conditional forward-backward diffusion models with conditional score matching. We investigate theoretical properties of such estimators through a finite-sample analysis of their statistical error bounds with respect to various metrics and examine their dependence on the intrinsic dimension and certain smoothness characteristics of the data. The key findings and contributions of our work can be summarized in the following:

- Our rates are minimax-optimal under the total variation metric in the classical setting when the conditional distribution admits a smooth density function that also varies smoothly across different covariate values.
- Our models encompass unconditional distribution estimation and nonparametric mean regression as special cases. When restricted to the former, our derived estimation error bounds achieve the minimax rate under both the total variation and the Wasserstein metrics. For the latter, our rates recover the classical minimax rate of nonparametric regression under the $L_2$ risk.
- Our results show that conditional diffusion estimators are adaptive to intrinsic manifold structures when either (or both) the covariate $X$ and response $Y$ are concentrated around some lower-dimensional manifold; thus, our model can handle high-dimensional *distribution regression* with covariates exhibiting low-dimensional structures.

**Other related works.** There is vast literature on nonparametric conditional distribution estimation. In addition to smoothing-based methods such as the ones employ kernel smoothing or local polynomial regression (Fan & Yim, 2004), there are other approached based on mixture model (Bishop, 2006) , Gaussian processes (Payne et al., 2019; Dutordoir et al., 2018), and nonparametric Bayes (Chung & Dunson, 2009; Dunson et al., 2007), among others. There has been a recent line of work that utilizes deep generative approach for conditional sampling such as Zhou et al. (2022) and Liu et al. (2021). Zhou et al. (2022) utilizes conditional GAN based approach and derived a consistent conditional density estimator but no convergence rates or error bounds are provided. There is also a growing body of theoretical research on diffusion generative models, though most have not considered the conditional setting as we do, such as Oko et al. (2023); Chen et al. (2023a); Wang et al. (2024); Li & Yan (2024); De Bortoli et al. (2021); Lee et al. (2022); Chen et al. (2022); Lee et al. (2023); Chen et al. (2023b); Tang & Yang (2024); Li et al. (2024b) and Li et al. (2024a). Among these, Oko et al. (2023); Chen et al. (2023a) and Tang & Yang (2024) study the approximation error and generalization ability of diffusion models for estimating (unconditional) distributions when data exhibits low-dimensional structures, which are shown to attain the minimax optimality in the 1-Wasserstein metric for distributions supported on low-dimensional hyperplanes (Oko et al., 2023) and general submanifolds (Tang & Yang, 2024). Some works, such as Li & Yan (2024) and De Bortoli (2022), explicitly leverage low-dimensional data structures during the generative sampling phase of implementing diffusion models. In scenarios where a covariate is available, some recent works (Chen et al., 2024; Fu et al., 2024) explore the theoretical properties of conditional diffusion models; however, they do not account for manifold structures when deriving their convergence rates.

## 2 Forward-Backward Diffusion Model and its Conditional Variant

In this section, we will begin by reviewing the forward-backward diffusion model for (unconditional) distribution estimation. After that, we will introduce its adaptations for estimating conditional distributions under the statistical framework of distribution regression.

## 2.1 FORWARD-BACKWARD DIFFUSION MODEL WITH SCORE MATCHING

Forward-backward diffusion models (see, e.g., Ho et al. (2020); Song et al. (2020); Nichol & Dhariwal (2021); Song & Ermon (2019)) have emerged as a new state-of-the-art class of generative models for estimating and generating samples from an underlying data distribution $\mu^*$ on a data space $\mathcal{M}_Y \subset \mathbb{R}^{D_Y}$. In a typical forward-backward diffusion model, two diffusion processes are utilized collaboratively: one process is designed for the estimation of a time-dependent score function describing the direction towards a high data probability region, while the other is for generating samples through a time-inhomogeneous process, based on the estimated score functions. Consequently, this model overcomes the slow convergence issues (e.g., due to the multimodality of $\mu^*$) commonly observed in models that rely solely on a single diffusion process, such as Langevin diffusion. Throughout the remainder of this paper, the term "diffusion model" specifically refers to the forward-backward diffusion model.

More concretely, the first diffusion process in the diffusion model, often referred to as the forward diffusion, employs a simple diffusion starting from $\mu^*$ that admits a closed-form solution and converges exponentially quickly to its limiting distribution. In this paper, we focus on the commonly used Ornstein–Uhlenbeck (OU) process as the forward process, which gradually injects Gaussian noise into the data and is described as a stochastic differntial equation (SDE)

$$\mathrm{d}\overrightarrow{Y}_t = -\delta_t \overrightarrow{Y}_t \, \mathrm{d}t + \sqrt{2\delta_t} \, \mathrm{d}B_t, \ \overrightarrow{Y}_0 \sim \mu^*, \tag{1}$$

where $\{B_t \, : \, t > 0\}$ denotes the standard Brownian motion in $\mathbb{R}^{D_Y}$ and $\{\delta_t \, : \, t \geq 0\}$ is some (possibly time-dependent) drift coefficient. Note that the OU process admits the closed form solution $\overrightarrow{Y}_t = m_t \overrightarrow{Y}_0 + \int_0^t \frac{m_t}{m_s} \sqrt{2\delta_s} \, \mathrm{d}B_s$; thus, the conditional distribution of $\overrightarrow{Y}_t$ given $\overrightarrow{Y}_0 = y$ is $\mathcal{N}(m_t y, \sigma_t^2 I_{D_Y})$, where $m_t = \exp\left(-\int_0^t \delta_s \, \mathrm{d}s\right)$ and $\sigma_t^2 = 1 - m_t^2$. Therefore, the marginal distribution of $\overrightarrow{Y}_t$, denoted as $p_t$, converges exponentially quickly to its limiting distribution $p_\infty = \mathcal{N}(0, I_{D_Y})$ under the Kullback–Leibler divergence.

The second diffusion process in the diffusion model, usually called the backward diffusion, reverses the forward diffusion and can be written as the following SDE,

$$\mathrm{d}\overleftarrow{Y}_t = \left[\delta_{T-t} \overleftarrow{Y}_t + 2\delta_{T-t} \nabla \log p_{T-t}(\overleftarrow{Y}_t)\right] \mathrm{d}t + \sqrt{2\delta_{T-t}} \, \mathrm{d}B_t, \ \overleftarrow{Y}_0 \sim p_T. \tag{2}$$

Under mild conditions on $\mu^*$ (Song et al., 2020; Haussmann & Pardoux, 1986) (valid for our setting), the distribution of $\overleftarrow{Y}_t$ is $p_{T-t}$, so that $\overleftarrow{Y}_T \sim p_0 = \mu^*$. Since $p_T$ is close to $p_\infty = \mathcal{N}(0, I_{D_Y})$, one can instead initialize the backward diffusion using the easy-to-sample distribution $p_\infty$, i.e. set $\overleftarrow{Y}_0 \sim \mathcal{N}(0, I_{D_Y})$. The drift term of the backward diffusion depends on the time dependent score function $\nabla \log p_t$ defined through the forward diffusion; therefore, the forward and the backward diffusions together constitutes a generative model for sampling from $\mu^*$.

Equations (1) and (2) define the forward-backward diffusion model at the population-level. In a standard statistical setting, we utilize independent and identically distributed (i.i.d.) samples $\{Y_i\}_{i=1}^n$ from $\mu^*$ to estimate the time-dependent score function $\nabla \log p_t$ in the backward diffusion. The estimation is achieved by the so-called score matching (Song & Ermon, 2019; Vincent, 2011). Specifically, one first numerically simulates for some sufficiently large time horizon $T$ from a sample-level forward process $\{y_t \, : \, t \in [0, T]\}$, which is SDE (1) initialized at the empirical distribution of the data, that is, $y_0 \sim \widehat{\mu}_n = n^{-1} \sum_{i=1}^n \delta_{Y_i}$, with $\delta_y$ denoting the point mass (Dirac) measure at a point $y$. One then uses a score approximating map $S_\theta(y, t)$ over space and time, indexed by a parameter $\theta$, e.g., (deep) neural networks with controlled depth and number of non-zero parameters, to estimate the true underlying score function $\nabla \log p_t(y)$, by minimizing the following ($L_2$-)score matching risk (over $\theta$):

$$\int_\tau^T \mathbb{E}_{y_t \sim p_t(\cdot \,|\, y_0), \, y_0 \sim \widehat{\mu}_n} \left[\|S_\theta(y_t, t) - \nabla \log p_t(y_t \,|\, y_0)\|^2\right] \lambda(t) \, \mathrm{d}t,$$

where $p_t(\cdot \,|\, y)$ denotes the distribution of $\overrightarrow{Y}_t$ in forward diffusion (1) initialized at $\overrightarrow{Y}_0 = y$ for any $y \in \mathcal{M}_Y$. Here, $\lambda(t)$ is a weighting function (over time), and $\tau$ is an early-stopping threshold for preventing the explosion (singularity) of the score function as $t \to 0$ commonly employed in practice (Song & Ermon, 2020; Oko et al., 2023). Equivalently, this score estimation step can be efficiently carried out in practice by simulating a trajectory $\{y_t \, : \, t \geq 0\}$ from SDE (1) starting from each data

point $Y_i$, that is, $y_t \sim p_t(\cdot \,|\, Y_i)$. One then uses $S_\theta(y, t)$ to match the ensemble of all sample score functions $\nabla \log p_t(y_t \,|\, Y_i)$ over all $n$ simulated trajectories, by minimizing the following empirical risk function (over $\theta$):

$$\frac{1}{n} \sum_{i=1}^{n} \int_{\tau}^{T} \mathbb{E}_{y_t \sim p_t(\cdot \,|\, Y_i)} \big[ \|S_\theta(y_t, t) - \nabla \log p_t(y_t \,|\, Y_i)\|^2 \big] \lambda(t) \, \mathrm{d}t. \tag{3}$$

We will adopt this statistical formulation of score matching to facilitate our theoretical analysis, leveraging tools from statistical learning theory.

Finally, let $\widehat{S}(x, t) = S_{\widehat{\theta}}(x, t)$ denote the resulting score estimator. The distribution estimator of $\mu^*$ based on the forward-backward diffusion model is then $\widehat{p}_{T-\tau}$, where $\widehat{p}_t$ represents the distribution of $\overleftarrow{Y}_t^\dagger$ for $t \in [0, T-\tau]$, and $\overleftarrow{Y}_t^\dagger$ follows the SDE below with a plugged-in score,

$$\mathrm{d}\overleftarrow{Y}_t^\dagger = \big[\delta_{T-t} \overleftarrow{Y}_t^\dagger + 2\delta_{T-t}\widehat{S}(\overleftarrow{Y}_t^\dagger, T-t)\big] \, \mathrm{d}t + \sqrt{2\delta_{T-t}} \, \mathrm{d}B_t, \quad \overleftarrow{Y}_0^\dagger \sim \mathcal{N}(0, I_{D_Y}). \tag{4}$$

## 2.2 Conditional forward-backward diffusion model with conditional score matching

A notable characteristic of diffusion models is their flexibility in incorporating a covariate or control variable, denoted as $X \in \mathcal{M}_X \subset \mathbb{R}^{D_X}$, to guide the generation of new data $Y \in \mathcal{M}_Y \subset \mathbb{R}^{D_Y}$. This can be equivalently formulated as the statistical problem of generating samples from the conditional distribution $\mu^*_{Y|x}$ of $Y$ given $X = x$ for any covariate value $x \in \mathcal{M}_X$. To facilitate the borrowing of information across different covariate values, it is commonly assumed that the conditional distribution $\mu^*_{Y|x}$ varies smoothly with $x \in \mathcal{M}_X$. This assumption underlies a statistical framework often referred to as *distribution (density) regression* in the literature (Bashtannyk & Hyndman, 2001; Izbicki & Lee, 2016; Li et al., 2022). Distribution regression expands the classical (nonparametric) mean regression by estimating not only the conditional expectation $\mathbb{E}[Y|X = x]$ as a smooth function of $x$, but also the entire conditional distribution $\mu^*_{Y|x}$ that varies smoothly with $x$. Compared to classical distribution regression methods based on kernel smoothing, which require $\mu^*_{Y|x}$ to admit a density function (thus termed density regression in the early literature, see Bashtannyk & Hyndman (2001); Izbicki & Lee (2016)), conditional diffusion model-based methods are more flexible. They can be more generally applicable to cases where $\mu^*_{Y|x}$ is supported on a low-dimensional manifold and is therefore singular; see Section 3.2 for details.

A natural way to convert a diffusion model into a conditional diffusion model for sampling from $\mu^*_{Y|x}$ is to replace the (unconditional) score function $\nabla \log p_t$ in the backward diffusion with some conditional score function $\nabla \log p_t(\cdot \,|\, x)$ satisfying $p_0(\cdot \,|\, x) = \mu^*_{Y|x}(\cdot)$ and $p_T(\cdot \,|\, x) = p_\infty(\cdot)$. Earlier literature considers the so-called classifier guidance method for estimating the conditional score using Bayes' rule (especially when covariate $x$ is discrete or categorical; see, e.g., Dhariwal & Nichol (2021); Song et al. (2020)):

$$\nabla \log p_t(y_t \,|\, x) = \nabla \log p_t(y_t) + \nabla \log c_t(x \,|\, y_t),$$

Here, the first term, $\nabla \log p_t(y_t)$, is the unconditional score function defined in the forward diffusion (1) in the unconditional diffusion model, which can be estimated via score matching. The second term, $\nabla \log c_t(x \,|\, y_t)$, is the likelihood function of an external "classifier" trained to predict $x$ from $y_t$. In other words, the classifier guidance method incorporates information from the covariate value $x$ into the unconditional score function through the gradient of an external classifier to guide the backward diffusion for generating samples from $\mu^*_{Y|x}$.

In this work, for the sake of theoretical simplicity and to avoid the need to analyze and quantify the statistical accuracy of an external classifier, we consider a classifier-free method, which at the population level directly applying a *conditional score matching* method (Hyvärinen & Dayan, 2005; Vincent, 2011; Batzolis et al., 2021; Tashiro et al., 2021) based on simulating $\overrightarrow{Y}_t$ from the same (marginal) forward diffusion

$$\mathrm{d}\overrightarrow{Y}_t = -\delta_t \overrightarrow{Y}_t \, \mathrm{d}t + \sqrt{2\delta_t} \, \mathrm{d}B_t, \quad \overrightarrow{Y}_0 \sim \mu^*_Y. \tag{5}$$

Here, $\mu^*_Y$ denotes the marginal distribution of $Y$. Consider a generic conditional score approximating map $S_\theta(y, x, t)$ over space, covariate value and time, indexed by a parameter $\theta$. In the conditional

score matching step, we minimize the following ($L_2$-)conditional score matching risk over $\theta$:

$$\int_{\tau}^{T} \mathbb{E}_{y_t \sim p_t(\cdot \,|\, y_0),\, (x,y_0) \sim \mu_{X,Y}^*} \left[ \| S_\theta(y_t, x, t) - \nabla \log p_t(y_t \,|\, y_0) \|^2 \right] \lambda(t)\, dt, \tag{6}$$

where $\mu_{X,Y}^*$ denotes the joint distribution of $(X, Y)$, and $p_t(\cdot \,|\, y)$ is the distribution of $\overrightarrow{Y}_t$ in forward diffusion (5) initialized at $\overrightarrow{Y}_0 = y$ for any $y \in \mathcal{M}_Y$. Here, the early stopping threshold $\tau$ and the weighting function $\lambda(t)$ are defined as before. It is straightforward to show (see Lemma C.12 in Appendix C.2) that if $S_\theta(y, x, t)$ can range over all possible conditional score functions, then the global minimizer of the preceding risk function is precisely the true underlying conditional score function $\nabla \log p_t(y_t \,|\, x)$. Here, $p_t(y_t \,|\, x) := \mathbb{E}_{y_0 \sim \mu_{Y|x}^*}[p_t(y_t \,|\, y_0)]$ denotes the conditional distribution of $\overrightarrow{Y}_t$ given $X = x$ after marginalizing out $\overrightarrow{Y}_0$, where $(X, \overrightarrow{Y}_0) \sim \mu_{X,Y}^*$ and $\overrightarrow{Y}_t$ follows forward diffusion (5) starting from $\overrightarrow{Y}_0$.

The corresponding conditional backward diffusion for sampling from $\mu_{Y|x}^*$ is then given by

$$d\overleftarrow{Y}_{t|x} = \left[ \delta_{T-t} \overleftarrow{Y}_{t|x} + 2\delta_{T-t} \nabla \log p_{T-t}(\overleftarrow{Y}_{t|x} \,|\, x) \right] dt + \sqrt{2\delta_{T-t}}\, dB_t, \quad \overleftarrow{Y}_{0|x} \sim p_T(\cdot \,|\, x). \tag{7}$$

Here, we added a subscript $x$ in the notation $\overleftarrow{Y}_{t|x}$ to indicate that, unlike the forward process (5), the backward diffusion process is $x$-dependent. Similar to the (unconditional) diffusion model, choosing a sufficiently large $T$ can guarantee $p_T(\cdot \,|\, x) \approx p_\infty(\cdot) = N(0, I_{D_Y})$, which is independent of $x$. We will refer to equations (5) and (7) as the *conditional (forward-backward) diffusion model* for the remainder of the paper.

To estimate the conditional score $\nabla \log p_t(\cdot \,|\, x)$ using i.i.d. observations $\{(X_i, Y_i)\}_{i=1}^n$ sampled from the joint distribution $\mu_{X,Y}^*$ of $(X, Y)$ under the statistical framework of distribution regression, one can again minimize the following empirical version of conditional score matching risk (6),

$$\frac{1}{n} \sum_{i=1}^{n} \int_{\tau}^{T} \mathbb{E}_{y_t \sim p_t(\cdot | Y_i)} \left[ \| S_\theta(y_t, X_i, t) - \nabla \log p_t(y_t \,|\, Y_i) \|^2 \right] \lambda(t)\, dt. \tag{8}$$

Finally, let $\widehat{S}(y, x, t) = S_{\widehat{\theta}}(y, x, t)$ denote the corresponding conditional score estimator. For each $x \in \mathcal{M}_X$, the conditional distribution estimator of $\mu_{Y|x}^*$ based the conditional forward-backward diffusion model is then $\widehat{p}_{T-\tau}(\cdot \,|\, x)$, where $\widehat{p}_t(\cdot \,|\, x)$ is the distribution of $\overleftarrow{Y}_{t|x}^{\dagger}$ for $t \in [0, T - \tau]$, and $\overleftarrow{Y}_{t|x}^{\dagger}$ follows the SDE below with a plugged-in conditional score,

$$d\overleftarrow{Y}_{t|x}^{\dagger} = \left[ \delta_{T-t} \overleftarrow{Y}_{t|x}^{\dagger} + 2\delta_{T-t} \widehat{S}(\overleftarrow{Y}_{t|x}^{\dagger}, x, T - t) \right] dt + \sqrt{2\delta_{T-t}}\, dB_t, \quad \overleftarrow{Y}_{0|x}^{\dagger} \sim \mathcal{N}(0, I_{D_Y}). \tag{9}$$

## 2.3 NEURAL NETWORK CLASS FOR CONDITIONAL SCORE FUNCTION APPROXIMATION

**Definition** (neural network class): A class of neural networks $\Phi(H, W, R, B, V)$ with height $H$, width vector $W = (W_1, W_2, \ldots, W_{H+1})$, sparsity $R$, norm constraint $B$, and function norm constraint $V$ is defined as $\Phi(H, W, R, B, V) = \big\{ f(\cdot) = (A^{(H)} \mathrm{ReLU}(\cdot) + b^{(H)}) \circ \cdots \circ (A^{(2)} \mathrm{ReLU}(\cdot) + b^{(2)}) \circ (A^{(1)} x + b^{(1)})$, so that $A^{(i)} \in \mathbb{R}^{W_i \times W_{i+1}}; b^{(i)} \in \mathbb{R}^{W_{i+1}}; \sum_{i=1}^{H} (\|A^{(i)}\|_0 + \|b^{(i)}\|_0) \leq R; \max_i \|A^{(i)}\|_\infty \vee \|b^{(i)}\|_\infty \leq B; \|f\|_\infty \leq V \big\}$, where $\mathrm{ReLU}(x) = \max\{0, x\}$ is the rectified linear unit activation function and the $\max$ function is applied elementwise to a vector.

According to Oko et al. (2023) and Tang & Yang (2024), the smoothing effect of gradually injecting Gaussian noise into the data distribution during the forward diffusion (5) suggests that the optimal size of the neural network for effectively approximating $\nabla \log p_t(\cdot, |, x)$ should decrease as $t$ increases. This observation motivates us to consider a neural network class whose size diminishes over time. For technical convenience, we discretize the time and adopt the following piece-wise constant complexity neural network class, as utilized in Tang & Yang (2024):

$$\mathcal{S}_{NN} = \left\{ S(y, x, t) = \sum_{i=1}^{\mathcal{I}} S_i(y, x, t) \cdot \mathbf{1}\,(t_{i-1} \leq t < t_i) \,\Big|\, S_i \in \Phi(H_i, W_i, R_i, B_i, V_i),\, i \in [\mathcal{I}] \right\},$$

where $\tau = t_0 < t_1 < \cdots < t_{\mathcal{I}} = T$, $\frac{t_{i+1}}{t_i} = 2$ for any $i \in [\mathcal{I}]$, and $\tau = 2^{-\mathcal{I}} T$ for some $\mathcal{I}$ to be determined later. We have also conducted a simulation study (see Appendix A) to demonstrate

the effectiveness of this theoretically guided neural network architecture compared to a standard single ReLU neural network (across both space and time). In this experiments, we consider cases where, given the covariate $X$, the response $Y$ is supported on different (tilted) ellipses depending on the values of the covariate. Consistent with our theoretical findings, the simulations show that incorporating the piecewise structure into the neural network results in a more accurate estimation of the conditional distribution. Recall that $\widehat{S}(y, x, t)$ denotes the conditional score estimator, defined as the minimizer of the conditional score matching risk (8) over the class $\mathcal{S}_{NN}$ with the weight function $\lambda(t) = t$ (although any other weights such as $\lambda(t) \equiv 1$ would also suffice). We define a (truncated) estimator $\widehat{\mu}_{Y|x}$ for $\mu^*_{Y|x}$ as the distribution of $\overleftarrow{Y}^\dagger_{T-\tau|x} \cdot \mathbf{1}(\|\overleftarrow{Y}^\dagger_{T-\tau|x}\|_\infty \leq L)$, where $(L, T)$ are large enough constants so that $\mathcal{M}_Y \subset \mathbb{B}_{\mathbb{R}^{D_Y}}(0, L/2)$ and $p_T(\cdot, |, x) \approx \mathcal{N}(0, I_{D_Y})$. Here, we truncate the random variable $\overleftarrow{Y}^\dagger_{T-\tau|x}$ to guarantee a bounded support for the induced distribution estimator $\widehat{\mu}_{Y|x}$, which is solely for technical reasons.

## 3 MAIN THEORETICAL RESULTS

In this section, we present our main theoretical results characterizing the statistical accuracy of the conditional diffusion model for conditional distribution estimation (or distribution regression) under two scenarios. In the first scenario, we consider the classical density regression setting where the conditional distribution $\mu^*_{Y|x}$ admits a density function relative to the Lebesgue measure of the data space $\mathcal{M}_Y \subset \mathbb{R}^{D_Y}$. We derive the convergence rate of the estimator under both the total variation and the Wasserstein metrics. In particular, our derived convergence rate is minimax-optimal under the total variation metric within the regime covered by the existing literature Li et al. (2022) (see Remark 1 for further details), and extends to a broader regime. In the second scenario, we consider a high-dimensional distribution regression setting where both the response variable $Y \in \mathbb{R}^{D_Y}$ and the covariate variable $X \in \mathbb{R}^{D_X}$ reside in high-dimensional ambient spaces characterized by large $D_Y$ and $D_X$. However, the covariate space $\mathcal{M}_X$ of $X$ has an intrinsic (or effective) dimension $d_X$ that is significantly smaller than $D_X$. Furthermore, given any $x \in \mathcal{M}_X$, the corresponding data space of $Y$, denoted as $\mathcal{M}_{Y|x}$, can be $x$-dependent and also has a small intrinsic dimension $d_Y < D_Y$. We demonstrate that the conditional diffusion model effectively adapts to the underlying manifold structures of both the data and the covariate variable. Specifically, we show that the convergence rate of the estimator depends solely on the intrinsic dimensions $(d_Y, d_X)$, rather than the ambient dimensions $(D_Y, D_X)$.

In the following, we denote $d_{\mathrm{TV}}(\mu, \nu)$ and $W_1(\mu, \nu)$ as the respective total variation distance and the 1-Wasserstein distance between two distributions $\mu$ and $\nu$. We denote $\mathcal{M} = \{(x, y) : x \in \mathcal{M}_X, y \in \mathcal{M}_{Y|x}\}$ as the joint space of $(X, Y)$ and $\mathcal{M}_Y = \bigcup_{x \in \mathcal{M}_X} \mathcal{M}_{Y|x}$ as the (marginal) data space. We use the notation $a \vee b$ and $a \wedge b$ to denote the respectively shorthand of $\max\{a, b\}$ and $\min\{a, b\}$. For a sequence $\{a_n : n \geq 1\}$, we use $\Theta(a_n)$ to indicate the order of $a_n$ up to a multiplicative constant as $n \to \infty$, and $\widetilde{\Theta}(a_n)$ to indicate the order of $a_n$ up to a multiplicative constant and logarithmic terms of $n$. Similarly, we use $\mathcal{O}(a_n)$ and $\widetilde{\mathcal{O}}(a_n)$ to indicate at most of order $a_n$.

### 3.1 CLASSICAL DENSITY REGRESSION IN EUCLIDEAN SPACE

In this subsection, we consider the classical density regression setting where both the covariate space $\mathcal{M}_X \subset \mathbb{R}^{D_X}$ and the data space $\mathcal{M}_Y \subset \mathbb{R}^{D_Y}$ are compact subsets (with open interiors) of the Euclidean spaces, and the conditional distribution $\mu^*_{Y|x}$ admits a density function, denoted as $\mu^*(y \,|\, x)$, relative to the Lebesgue measure of $\mathbb{R}^{D_Y}$. For simplicity, we assume $\mathcal{M}_Y = [-1, 1]^{D_Y}$ and $\mathcal{M}_X = [-1, 1]^{D_X}$. In order to derive a non-asymptotic bound to the expected total variation distance and Wasserstein distance between the conditional distribution estimator $\widehat{\mu}_{Y|X}$ and the target $\mu^*_{Y|X}$, with the expectation taken over $X \sim \mu^*_X$, we impose certain smoothness condition to the condition density function $\mu^*(y \,|\, x)$ relative to $(y, x)$ as in the classical density regression literature (Li et al., 2022; Bilodeau et al., 2023). Specifically, we assume that $\mu^*(y, |, x)$, as a function of $(y, x)$, is $C^{\alpha_Y, \alpha_X}$-smooth, where $\alpha_Y$ and $\alpha_X$ quantify the respective smoothness in the response variable $y$ and the covariate $x$. Note that a function $f(y, x)$ being $C^{\alpha_Y, \alpha_X}$-smooth implies that, around any point $(y_0, x_0)$, there exists a local polynomial approximation of $f$, with an approximation error of order $\mathcal{O}(\|y - y_0\|^{\alpha_Y} + \|x - x_0\|^{\alpha_X})$; a rigorous definition can be found in Appendix B. Formally, we make the following assumptions.

**Assumption A** (smoothness and lower boundness of $\mu^*_{Y|x}$): For each $x \in \mathcal{M}_X$, the conditional distribution $\mu^*_{Y|x}$ admits a density function $\mu^*(y \,|\, x)$ that is $C^{\alpha_Y, \alpha_X}$-smooth in $(y, x)$. Moreover, there exists a positive constant $c$ so that $\mu^*(y \,|\, x) \geq c$ holds for any $x \in \mathcal{M}_X, y \in \mathcal{M}_Y$.

**Assumption B** (regularity of the drift coefficient): The drift coefficient $\delta_t$ is infinitely differentiable and there exist positive constants $c_1, c_2$ so that $c_1 \leq \delta_t \leq c_2$ for any $t \geq 0$.

Here the lower bound requirement of $\mu^*(y \,|\, x)$ is a commonly made assumption for distribution estimation in the classical density regression literature, and is also imposed in Oko et al. (2023); Tang & Yang (2024) for analyzing (unconditional) diffusion models. In practical applications, the drift coefficient $\delta_t$ is typically chosen as a positive constant independent of $t$, thus naturally satisfying Assumption B.

**Theorem 1** (Density regression in Euclidean space). *Suppose Assumptions A and B are satisfied. Let $\varepsilon_1 = n^{-1 / \left( 2\alpha_Y + D_Y + \frac{\alpha_Y}{\alpha_X} D_X \right)}$. If we take $\tau = \widetilde{\Theta}(\varepsilon_1^{2(\alpha_Y+1)})$, $T = \Theta(\log n)$, and neural network sizes satisfying $H_i = \Theta(\log^4 n)$, $\|W_i\|_\infty = \widetilde{\Theta}\big(n^{\frac{D_X}{2\alpha_X + D_X}} t_i^{\frac{-\alpha_X D_Y}{2\alpha_X + D_X}} \wedge \varepsilon_1^{-D_Y - \frac{\alpha_Y D_X}{\alpha_X}}\big)$, $R_i = \widetilde{\Theta}\big(n^{\frac{D_X}{2\alpha_X + D_X}} t_i^{\frac{-\alpha_X D_Y}{2\alpha_X + D_X}} \wedge \varepsilon_1^{-D_Y - \frac{\alpha_Y D_X}{\alpha_X}}\big)$, $B_i = \exp(\Theta(\log n^4))$ and $V_i = \Theta\big(\sqrt{\frac{\log n}{t_i \wedge 1}}\big)$ for $i \in [\mathcal{I}]$[1] with $\mathcal{I} = \log_2(\frac{T}{\tau})$, then it holds with probability at least $1 - n^{-1}$ that*

$$\mathbb{E}_{x \sim \mu^*_X}\big[ d_{\mathrm{TV}}(\widehat{\mu}_{Y|x}, \mu^*_{Y|x}) \big] = \widetilde{\mathcal{O}}\bigg( n^{-\frac{1}{2 + \frac{D_X}{\alpha_X} + \frac{D_Y}{\alpha_Y}}} \bigg), \tag{10}$$

*and* $$\mathbb{E}_{x \sim \mu^*_X}\big[ W_1(\widehat{\mu}_{Y|x}, \mu^*_{Y|x}) \big] = \widetilde{\mathcal{O}}\bigg( n^{-\frac{1}{2 + \frac{D_X}{\alpha_X}}} \vee n^{-\frac{1 + \frac{1}{\alpha_Y}}{2 + \frac{D_X}{\alpha_X} + \frac{D_Y}{\alpha_Y}}} \bigg). \tag{11}$$

**Remark 1.** *In the special case when $\alpha_X \in [0, 1]$, Li et al. (2022) shows that a well-designed kernel-based estimator can achieve the same convergence rate (10) in the total variation metric as our conditional diffusion model-based estimator; furthermore, this rate is shown to be minimax-optimal. Therefore, our result implies the minimax-optimality of the conditional diffusion model for density regression in the regime where $\alpha_X \in [0, 1]$, although our upper bound is also applicable to $\alpha_X > 1$.*

**Remark 2.** *When specializing to the unconditional case with no covariate (that is, taking $D_X = 0$ in Theorem 1), our derived estimation error bounds (10) and (11) reduce respectively to the minimax rate of (unconditional) distribution estimation under the total variation metric and the Wasserstein metric (Liang, 2021; Tang & Yang, 2023). However, unlike the upper bound proofs in Liang (2021); Tang & Yang (2023), which rely on generative adversarial network (GAN) type estimators, our proof demonstrates that the diffusion model is also minimax-optimal for distribution estimation. In particular, our results recover those from Oko et al. (2023) as a special case (by taking $D_X = 0$).*

**Remark 3.** *The derived $W_1$ error bound (11) comprises two terms $n^{-\alpha_X / (2\alpha_X + D_X)}$ and $n^{-(\alpha_Y + 1)/(2\alpha_Y + \frac{D_X \alpha_Y}{\alpha_X} + D_Y)}$. The first term resembles the classical minimax rate of nonparametric regression under the $L_2$ risk and can be interpreted as mainly capturing the estimation error related to learning the dependence of the response variable $Y$ on the covariate $X$, so that it only depends on the smoothness and intrinsic dimension of $X$. Technically, this term arises from the approximation of the conditional score function for large time $t$, where finer details of the conditional distribution in $Y$ have been smoothed out and only the global dependence on $X$ matters. The second term reflects the estimation error of recovering the entire conditional distribution of $Y$ given $X$, and depends on characteristics related to the response variable $Y$, such as the smoothness $\alpha_Y$ of the conditional density function and the dimension of $Y$. Interestingly, the derived rate suggests a phase transition phenomenon: if the dimension of the response variable $D_Y$ satisfies $D_Y \leq 2 + \frac{D_X}{\alpha_X}$, then the estimation error under the $W_1$ metric remains of order $n^{-\alpha_X / (2\alpha_X + D_X)}$ regardless of the smoothness level $\alpha_Y$, and the $W_1$ estimation error is dominated by the error of capturing the global dependence of the response variable $Y$ on the covariate variable $X$; otherwise, the $W_1$ estimation error is influenced by both the smoothness $\alpha_Y$ of conditional density on $Y$ and the smoothness $\alpha_X$, which captures the finer details of the conditional distribution of $Y$ given $X$.*

**Remark 4.** *A recent related work Fu et al. (2024) also explores theoretical properties of conditional diffusion model, and show the minimax optimality of diffusion model under the total variation distance.*

---

[1] Here we use the notation $[\mathcal{I}] = \{1, 2, \cdots, \mathcal{I}\}$.

*In our work, we allow the conditional distribution of $Y$ given $X$ to have difference smoothness levels $\alpha_Y$ and $\alpha_X$ on the response $Y$ and covariate $X$; in comparison, Fu et al. (2024) assumes the two smoothness levels are the same. The varying smoothness levels can allow for the applicability of the results in more general settings. For instance, when specializing to the mean regression case, where the conditional distribution is a Gaussian distribution centered at the evaluation of an $\alpha_X$ smooth regression function over $\mathbb{R}^{D_X}$, our derived estimation error bound (10) under $d_{\mathrm{TV}}$ (that is, taking $\alpha_Y \to \infty$ in Theorem 1) can recover the classical minimax rate $n^{-\frac{\alpha_X}{2\alpha_X + D_X}}$ of nonparametric regression under the $L_2$ risk.*

## 3.2 High-dimensional distribution regression with low-dimensional manifold structures

In this subsection, we consider the case where both the covariate space $\mathcal{M}_X$ and the response space $\mathcal{M}_Y$ may have low-dimensional structures in their respective ambient spaces $\mathbb{R}^{D_X}$ and $\mathbb{R}^{D_Y}$. For the covariate space $\mathcal{M}_X$, the low-dimensional structure is imposed in terms of its upper Minkowski dimension, which is related to the growth of its packing number (see Assumption C). This low-dimensional structure is notably less stringent than a typical manifold assumption, as it does not require any smoothness properties of $\mathcal{M}_X$. For the response space $\mathcal{M}_Y$, since we allow the conditional distribution $\mu^*_{Y|x}$ to have different supports, denoted as $\mathcal{M}_{Y|x}$, for each $x \in \mathcal{M}_X$, we can decompose $\mathcal{M}_Y$ as $\bigcup_{x \in \mathcal{M}_X} \mathcal{M}_{Y|x}$. In our theory, we require each "section" $\mathcal{M}_{Y|x}$ to be a smooth submanifold in $\mathbb{R}^{D_Y}$; additionally, we require $\mathcal{M}_{Y|x}$ to vary smoothly with $x$ (see Assumption D). We list the concrete assumptions as follows.

**Assumption C** (intrinsic dimension of $\mathcal{M}_X$): $\mathcal{M}_X$ is compact set in $\mathbb{R}^{D_X}$ and there exist constants $(C_1, C_2)$ so that for any $\varepsilon > 0$, any $\varepsilon_1 \in (0, \varepsilon)$, and any $x \in \mathcal{M}$, we have[2] $\mathbf{M}(\mathbb{B}_{\mathcal{M}_X}(x, \varepsilon), \|\cdot\|, \varepsilon_1) := \max\{m : \exists \varepsilon_1\text{-packing of } B_{\mathcal{M}_X}(x, \varepsilon) \text{ of size } m\} \le C_2(\frac{\varepsilon_1}{\varepsilon})^{-d_X}$.

This assumption naturally holds if $\mathcal{M}_X$ is a compact subset of a $d_X$-dimensional hyperplane, or more generally, a compact $d_X$-dimensional submanifold embedded in $\mathbb{R}^{D_X}$ with its reach[3] bounded away from zero. Therefore, the constant $d_X$ in the assumption can be interpreted as the intrinsic dimension of $\mathcal{M}_X$. Next, we introduce our assumption on the conditional distribution $\mu^*_{Y|X}$. For easy understanding, we present an informal assumption here and postpone the more rigorous and detailed version to Appendix B in the supplement. Recall that $\mathcal{M}$ denotes the joint space of $(X, Y)$.

**Assumption D** (smoothness of $\mu^*_{Y|X}$, informal version): For any $x \in \mathcal{M}_X$, $\mathcal{M}_{Y|x}$ is a $d_Y$-dimensional submanifold in $\mathbb{R}^{D_Y}$, and $\mu^*_{Y|x}$ admits a density with respect to the volume measure of $\mathcal{M}_{Y|x}$, which is uniformly lower bounded away from zero. Moreover, for any $\omega = (x_0, y_0) \in \mathcal{M}$ and $x \in \mathbb{B}_{\mathcal{M}_X}(x_0, r_0)$, there exists an encoder-decoder pair $(Q^\omega_x(y), G^\omega_x(z))$, such that $Q^\omega_x(\cdot)$ maps $y \in \mathbb{B}_{\mathcal{M}_{Y|x}}(y_0, r_0)$ to a low-dimensional latent variable $z \in \mathbb{R}^{d_Y}$, and $G^\omega_x(\cdot)$ reconstructs the data $y$ through the latent variable $z$. Here, the decoder $G^\omega_x(z)$ is $C^{\beta_Y, \beta_X}$-smooth in $(z, x)$; and the induced (local) conditional density function $v^\omega(z|x)$ of the latent variable $z$, as the pushforward measure through the encoder $Q^\omega_x(y)$ of the restriction of the measure $\mu^*_{Y|x}$ onto $\mathbb{B}_{\mathcal{M}_{Y|x}}(y_0, r_0)$, is $C^{\alpha_Y, \alpha_X}$-smooth in $(z, x)$.

Constants $(\beta_Y, \beta_X)$ in Assumption D quantify the smoothness of the submanifold $\mathcal{M}_{Y|x}$, which is the image (range) of the decoder $G^\omega_x(z)$. Specifically, index $\beta_Y$ characterizes the smoothness level of the manifold $M_{Y|x}$ supporting the response variable for any fixed $x \in \mathcal{M}_X$; and index $\beta_X$ characterizes the smoothness level of the section manifold $M_{Y|x}$ in $x$, that is, how similar $\mathcal{M}_{Y|x}$ and $\mathcal{M}_{Y|x'}$ are when $x$ is close to $x'$ in $\mathcal{M}_X$. In contrast, constants $(\alpha_Y, \alpha_X)$ in Assumption D quantify the smoothness of the conditional distribution $\mu^*_{Y|x}$, or more precisely, the corresponding conditional density function on its supporting manifold $\mathcal{M}_{Y|x}$. Specifically, the index $\alpha_Y$ characterizes the smoothness level of the conditional density function in the response variable $Y$ for any fixed $x \in \mathcal{M}_X$, while the index $\alpha_X$ captures how smoothly the conditional density function changes with $x$. The

---

[2] A set $P \subseteq S$ is a $\varepsilon$-packing of $S$ if for every $x, x' \in P$ we have $\|x - x'\| > \varepsilon$.

[3] The reach of a closed subset $A \subset \mathbb{R}^D$ is defined as $\tau_A = \inf_{p \in A} \mathrm{dist}(p, \mathrm{Med}(A)) = \inf_{z \in \mathrm{Med}(A)} \mathrm{dist}(z, A)$, where $\mathrm{dist}(z, A) = \inf_{p \in A} \|p - z\|$ denotes the distance function to $A$, and $\mathrm{Med}(A)$ is the medial axis of $A$ consisting of the points that have at least two nearest neighbors.

constant $d_Y$ in the assumption can be viewed as the intrinsic dimension of the response variable $Y$. Similar to Tang & Yang (2024), the requirements on the conditional distribution $\mu_{Y|x}^*$ in Assumption D are stated in a local manner since a manifold, as a topological space, is only locally defined. In fact, many common manifolds, such as spheres, do not admit a global parameterization (or encoder-decoder representation).

**Theorem 2** (Distribution regression on manifolds). *Suppose Assumptions B, C and D are satisfied with $\beta_Y \geq \alpha_Y \vee 1 + 1$ and $\beta_X \geq \alpha_X + \frac{\alpha_X}{\alpha_Y}$. Let $\varepsilon_1 = n^{-1/\left(2\alpha_Y + d_Y + \frac{\alpha_Y}{\alpha_X}d_X\right)}$. If we take $\tau = \widetilde{\Theta}(\varepsilon_1^{2(\alpha_Y + 1)})$, $T = \Theta(\log n)$, and neural network sizes satisfying $H_i = \Theta(\log^4 n)$, $\|W_i\|_\infty = \widetilde{\Theta}\big(n^{\frac{d_X}{2\alpha_X + d_X}} t_i^{\frac{-\alpha_X d_Y}{2\alpha_X + d_X}} \wedge \varepsilon_1^{-d_Y - \frac{\alpha_Y d_X}{\alpha_X}}\big)$, $R_i = \widetilde{\Theta}\big(n^{\frac{d_X}{2\alpha_X + d_X}} t_i^{\frac{-\alpha_X d_Y}{2\alpha_X + d_X}} \wedge \varepsilon_1^{-d_Y - \frac{\alpha_Y d_X}{\alpha_X}}\big)$, $B_i = \exp(\Theta(\log n^4))$ and $V_i = \Theta\big(\sqrt{\frac{\log n}{t_i \wedge 1}}\big)$ for $i \in [\mathcal{I}]$ with $\mathcal{I} = \log_2(\frac{T}{\tau})$, then it holds with probability at least $1 - n^{-1}$ that*

$$\mathbb{E}_{x \sim \mu_X^*}\left[W_1\big(\widehat{\mu}_{Y|x}, \mu_{Y|x}^*\big)\right] = \widetilde{\mathcal{O}}\left(n^{-\frac{1}{2 + \frac{d_X}{\alpha_X}}} \vee n^{-\frac{1 + \frac{1}{\alpha_Y}}{2 + \frac{d_X}{\alpha_X} + \frac{d_Y}{\alpha_Y}}}\right).$$

**Remark 5.** *Since $\widehat{\mu}_{Y|x}$ and $\mu_{Y|x}^*$ are almost surely mutually singular measures (supporting on different submanifolds), the total variation metric is always 1 and, therefore, not suitable for quantifying their closeness. Consequently, our error bound is stated only in terms of the Wasserstein metric. In fact, even in the (unconditional) distribution estimation case, Tang & Yang (2023) shows that no estimator can achieve estimation consistency under the total variation metric.*

**Remark 6.** *Theorem 2 demonstrates that the statistical accuracy of the conditional diffusion model depends solely on the intrinsic dimensions $(d_X, d_Y)$ rather than the ambient dimensions $(D_X, D_Y)$, modulo multiplicative constants and logarithmic terms. This indicates that the conditional diffusion model can adapt to the low-dimensional manifold structures in both the response and the covariate variables. In particular, when there is no low-dimensional manifold structures (i.e., $D_X = d_X$ and $D_Y = d_Y$), the $W_1$ error bound in Theorem 2 recovers the $W_1$ error bound in Theorem 1 in the classical density regression.*

**Remark 7.** *The same remarks after Theorem 1 in the previous subsection also apply: when specializing to the unconditional case with no covariate (that is, taking $D_X = 0$ in Theorem 1), our error bound reduces to the minimax rate of (unconditional) distribution estimation under the Wasserstein metric (Tang & Yang, 2023) with a (sufficiently smooth) manifold structure; when specializing to the mean regression case, our error bound can recover the classical convergence rate $n^{-\frac{\alpha_X}{2\alpha_X + d_X}}$ of nonparametric regression when the covariate $X$ is supported on a $d_X$-dimensional submanifold (Yang & Dunson, 2016; Jiao et al., 2023).*

**Remark 8.** *Several works (Chen et al., 2023a; Oko et al., 2023) have also studied the unconditional diffusion model with a low-dimensional structure, where the data lies in a subspace. However, our work addresses a more general setting in which the manifold is unknown and can be highly nonlinear. In particular, for a linear subspace, the (unconditional) diffusion process can be decomposed into the tangent part and orthogonal part, so that the subspace estimation error and the estimation error of the distribution on the subspace can be decoupled and analyzed separately. In comparison, for a nonlinear manifold, such a decomposition does not exist, and the manifold estimation error and distribution estimation error are coupled in a complicated manner. Furthermore, due to the nonlinearity, in our approximation error analysis using neural networks, we have to locally approximate a class of projection operators of the nonlinear manifold that changes cross the manifold, rather than approximating a single global projection operator onto a linear subspace.*

## 3.3 PROOF HIGHLIGHTS

Since Theorem 2 extends Theorem 1 by incorporating manifold structures, we will only outline the proof for the former in this subsection. All missing definitions, formal assumptions, and detailed proofs are provided in the appendices of the supplementary material for this paper.

Our strategy for bounding the distribution estimation error mainly follows the pipeline of Oko et al. (2023); Tang & Yang (2024). First, we construct a specific neural network within the class $\mathcal{S}_{NN}$ to approximate the true conditional score function $\nabla \log p_t(\cdot, |, x)$ with controlled error, which is summarized in the following lemma.

**Lemma 1** (Score approximation error by neural network class). *Under the same neural network sizes* $\{(H_i, W_i, R_i, B_i, V_i)\}_{i=1}^{\mathcal{I}}$ *and choices of* $\tau$, $T$ *as in Theorem 2, there exists neural network* $\phi_i(w, x, t) \in \Phi(H_i, W_i, R_i, B_i, V_i)$ *for any* $i \in [\mathcal{I}]$ *so that*

$$
\mathbb{E}_{\mu_X^*} \left[ \int_{t_{i-1}}^{t_i} \int_{\mathbb{R}^D} \left\| \phi_i(w, x, t) - \nabla \log p_t(\cdot \,|\, x)(w) \right\|^2 p_t(\cdot \,|\, x)(w) \, \mathrm{d}w \, \mathrm{d}t \right]
$$

$$
= \begin{cases}
\widetilde{\mathcal{O}}\left( n^{-\frac{2\alpha_Y}{2\alpha_Y + d_Y + \frac{\alpha_Y}{\alpha_X} d_X}} + t_i^{-1} n^{-\frac{2 \cdot (\beta_Y \wedge \frac{\beta_X \alpha_Y}{\alpha_X})}{2\alpha_Y + d_Y + \frac{\alpha_Y}{\alpha_X} d_X}} \right), & if \, \tau \le t_i < n^{-\frac{2}{2\alpha_Y + d_Y + d_X \frac{\alpha_Y}{\alpha_X}}}; \\
\widetilde{\mathcal{O}}\left( n^{-\frac{2\alpha_X}{2\alpha_X + d_X}} t_i^{-\frac{\alpha_X d_Y}{2\alpha_X + d_X}} \right), & if \, n^{-\frac{2}{2\alpha_Y + d_Y + d_X \frac{\alpha_Y}{\alpha_X}}} \le t_i \le T.
\end{cases}
$$

The primary technical challenge in proving Lemma 1 arises from the fact that the space $\mathcal{M}_{Y|x}$ is a general (possibly nonlinear) manifold that depends on $x$. To address this, we partition the joint space $\mathcal{M}$ of $(X, Y)$ into small pieces with varying resolution levels in $X$ and $Y$, tailored to the smoothness levels $(\alpha_Y, \alpha_X)$, dimensions $(d_Y, d_X)$, and times $t_i$. Within each pieces, we carefully construct local polynomials to approximate the local charts (i.e., the decoders $G_x^\omega(z)$ defined in Assumption D) of $\mathcal{M}_{Y|x}$. The actual proof is much more involved and delicate in order to optimally balance between the approximating neural network size and the approximate error; see Appendix C for details. Based on Lemma 1, we can now utilize the complexity of $\mathcal{S}_{NN}$ to control the generalization error for our conditional score estimator $\widehat{S}$, which minimizes the empirical score matching risk (8). The result is summarized as follows.

**Lemma 2** (Score matching generalization error). *It holds with probability at least* $1 - n^{-1}$ *that,*

$$
\mathbb{E}_{\mu_X^*} \left[ \int_{t_{i-1}}^{t_i} \int_{\mathbb{R}^{D_Y}} \left\| \widehat{S}(w, x, t) - \nabla \log p_t(\cdot \,|\, x)(w) \right\|^2 p_t(\cdot \,|\, x)(w) \, \mathrm{d}t \, \mathrm{d}w \right]
$$

$$
\lesssim \min_{S \in \mathcal{S}_i} \mathbb{E}_{\mu_X^*} \left[ \int_{t_{i-1}}^{t_i} \int_{\mathbb{R}^{D_Y}} \left\| S(w, x, t) - \nabla \log p_t(\cdot \,|\, x)(w) \right\|^2 p_t(\cdot \,|\, x)(w) \, \mathrm{d}t \, \mathrm{d}w \right]
$$

$$
+ \frac{R_i H_i \log \left\{ R_i H_i \|W_i\|_\infty (B_i \vee 1) \, n \right\} (\log n)^2}{n}, \quad \textit{for each } i \in [\mathcal{I}].
$$

Oko et al. (2023) derived a similar result in the context of (unconditioned) distribution estimation without $x$; in addition, their error bound is not a high probability bound but instead takes another expectation with respect to the randomness of $\widehat{S}$. In contrast, our proof for the conditional distribution estimation requires a *high probability bound*. We utilize more technical tools in empirical process theory, such as the localization and peeling techniques Wainwright (2019), to derive such a high probability bound as in Lemma 2. The rest of the analysis is similar to a standard analysis for score-based diffusion models Song & Ermon (2019); Chen et al. (2022); Oko et al. (2023), where we apply Girsanov's theorem to relate the distribution estimation error with the obtained $L_2$ score estimation error.

## 4 CONCLUSION

In this study, we investigate the theoretical properties of conditional forward-backward diffusion estimators within the statistical framework of distribution regression. Our results identify the primary sources of error in conditional distribution estimation using conditional diffusion models and include earlier results on unconditional distribution estimation and nonparametric mean regression as special cases. Notably, our findings demonstrate that although (conditional) diffusion models operate directly in the original ambient data space and do not explicitly incorporate low-dimensional structures, the resulting conditional distribution estimators can still adapt to intrinsic manifold structures when either (or both) the covariate $X$ and response $Y$ are concentrated around a lower-dimensional manifold. Our analysis also offers practical guidance for designing the neural network approximation family to optimally control different types of errors. This includes recommendations for the architecture of the neural network, as well as how the network's size (depth, width, sparsity, etc.) should depend on various problem characteristics, such as sample size, smoothness levels, and intrinsic dimensions.

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
