# Supplementary Materials for "Conditional Diffusion Models are Minimax-Optimal and Manifold-Adaptive for Conditional Distribution Estimation"

**Notation**: We adopt the notations in the manuscript, and further introduce the following additional notations for the technical proofs. For neural network class $\Phi(H, W, R, B, V)$, we write $\Phi(H, W, R, B) = \Phi(H, W, R, B, \infty)$ if there is no constraint on the function norm. For any $U \subset \mathcal{M}$, we denote $U_X = \{x \in \mathcal{M}_X : \text{there exists } y \in \mathcal{M}_y \text{ so that } (x, y) \in U\}$, $U_{Y|x} = \{y \in \mathcal{M}_{Y|x} : (x, y) \subset U\}$ and $U_Y = \bigcup_{x \in U_X} U_{Y|x}$. For any measure $\nu$ on $\mathcal{Z}$ and map $G : \mathcal{Z} \to \mathcal{X}$, the pushforward measure $\mu = G_{\#}\nu$ is defined as the unique measure on $\mathcal{X}$ such that $\mu(A) = \nu(G^{-1}(A))$ holds for any measurable set $A$ on $\mathcal{X}$. We use $N(\mathcal{F}, \tilde{d}, \epsilon)$ to denote the $\epsilon$-covering number of function space $\mathcal{F}$ with respect to pseudo-metric $\tilde{d}$. For any positive integer $m$, we use the shorthand $[m] := \{1, \cdots, m\}$. For $\alpha \in \mathbb{R}$, the floor and ceiling functions are denoted by $\lfloor \alpha \rfloor$ and $\lceil \alpha \rceil$, indicating rounding $\alpha$ to the next smaller and larger integer. For two sequences $\{a_n\}$ and $\{b_n\}$, we use the notation $a_n \lesssim b_n$ and $a_n \gtrsim b_n$ to mean $a_n \leq Cb_n$ and $a_n \geq Cb_n$, respectively, for some constant $C > 0$ independent of $n$. In addition, $a_n \asymp b_n$ means that both $a_n \lesssim b_n$ and $a_n \gtrsim b_n$ hold. We use $\|\cdot\|_\infty$ to denote the usual vector $\ell_\infty$ norm (i,e., for $x = (x_1, x_2, \cdots, x_d)$, $\|x\|_\infty = \max_i |x_i|$) and reserve $\|\cdot\|$ for the vector $\ell_2$ norm. For a vector $x \in \mathbb{R}^d$, we use $x_i$ to denote its $i$th element. For a multi-index $j = (j_1, \cdots, j_d) \in \mathbb{N}_0^d = \{(j_1, \cdots, j_d) \mid \forall i \in [d], j_i \in \mathbb{N}_0\}$, we define $|j| = \sum_{i=1}^d j_i$ and $j! = \prod_{i=1}^d j_i!$. For two vectors $x, y \in \mathbb{R}^d$, we use $(x - y)^j$ to denote $\prod_{i=1}^d (x_i - y_i)^{j_i}$. For a multivariate function $f : \mathbb{R}^{d_1} \times \mathbb{R}^{d_2} \to \mathbb{R}$ and multi-indexes $j_1 \in \mathbb{N}_0^{d_1}$, $j_2 \in \mathbb{N}_0^{d_2}$ we use $f^{(j_1, j_2)}$ to denote its mixed partial derivative $\frac{\partial^{|j_1| + |j_2|} f(x,y)}{\partial x^{j_{11}} \cdots \partial x^{j_{1d_1}} \partial y^{j_{21}} \cdots \partial y^{j_{2d_2}}}$. Throughout, $C, c, C_0, c_0, C_1, c_1, L, L_0, L_1, \ldots$ are generically used to denote positive constants whose values might change from one line to another, but are independent from everything else.

## A  Simulation

To empirically demonstrate the adaptiveness of the conditional diffusion model to manifold structures, we conduct a simulation study. We first generate a dataset of $(x, \phi)$, where $x$ is generated from a uniform distribution over $[0, 1]$ and $\phi$ is generated from a uniform distribution over $[0, 2\pi]$. Given $x$ and $\phi$, the response $Y$ is generated by the process

$$Y = ((R + r\cos(x))\cos(\phi), (R + r\sin(x))\sin(\phi), r\sin(x)),$$

where $R = 1$, $r = 1.5$. Then we aim at estimating $Y|x$ and $Y|\phi$. In both examples, we have $D_X = 1$, $D_Y = 3$, and $d_Y = 1$. In particular, given $x$, the response $Y$ is supported on an ellipse whose radius depends on $x$. When given $\phi$, the response $Y$ is supported on a section of a "tilted" ellipse depending on $\phi$.

We then generate $n = 30000$ i.i.d. data points and fit the conditional diffusion model to both examples. We consider two types of conditional score families $\mathcal{S}_{NN}$. The first approach directly models the conditional score function using a ReLU neural network with two hidden layers, where the hidden layer widths are $(128, 64)$. We refer to this approach as NN. The second approach models the conditional

score family using a piecewise ReLU neural network, based on our theoretical results. Specifically, we consider

$$\mathcal{S}_{NN} = \left\{ S(y,x,t) = \sum_{i=1}^{\mathcal{I}} S_i(y,x,t) \cdot \mathbf{1}\,(t_{i-1} \leq t < t_i) \right\}, \tag{1}$$

where $\mathcal{I} = 5$, $t_0 = 0.001$, $t_1 = 0.1$, $t_2 = 0.2$, $t_3 = 0.4$, $t_4 = 1$, $t_5 = 2$. For each $i \in [5]$, $S_i$ is a ReLU neural network with two hidden layers, and the size of the hidden layers decreases with $i$. Specifically, for $i$ range from 1 to 5, the hidden layer widths are given by $(64, 64)$, $(64, 32)$, $(32, 32)$, $(32, 16)$ and $(16, 16)$, respectively. We refer to this approach as Piecewise NN. The two conditional score families considered above have a comparable number of training parameters. The plots of the generated data from the conditional diffusion model given $x = 0$, $x = 0.5$, and $x = 1$ are shown in Figure 1, and the plots of conditional diffusion model given $\phi = 0$, $\phi = 0.5$, $\phi = 1$ are shown in Figure 2. It is evident that for different values of $x$ (and $\phi$), the generated response $Y$ concentrates around distinct ellipse (and tilted ellipse). This indicates that the conditional diffusion model effectively captures both the covariate information and the underlying manifold structure. Table 1 presents the MMD (Maximum Mean Discrepancy) distance[1] between the generated data and the true conditional distribution for different values of $x$ (and $\phi$), as well as the average across all values of $x$ (and $\phi$). Consistent with our theoretical results, introducing the piecewise structure to the neural network results in a smaller MMD distance.

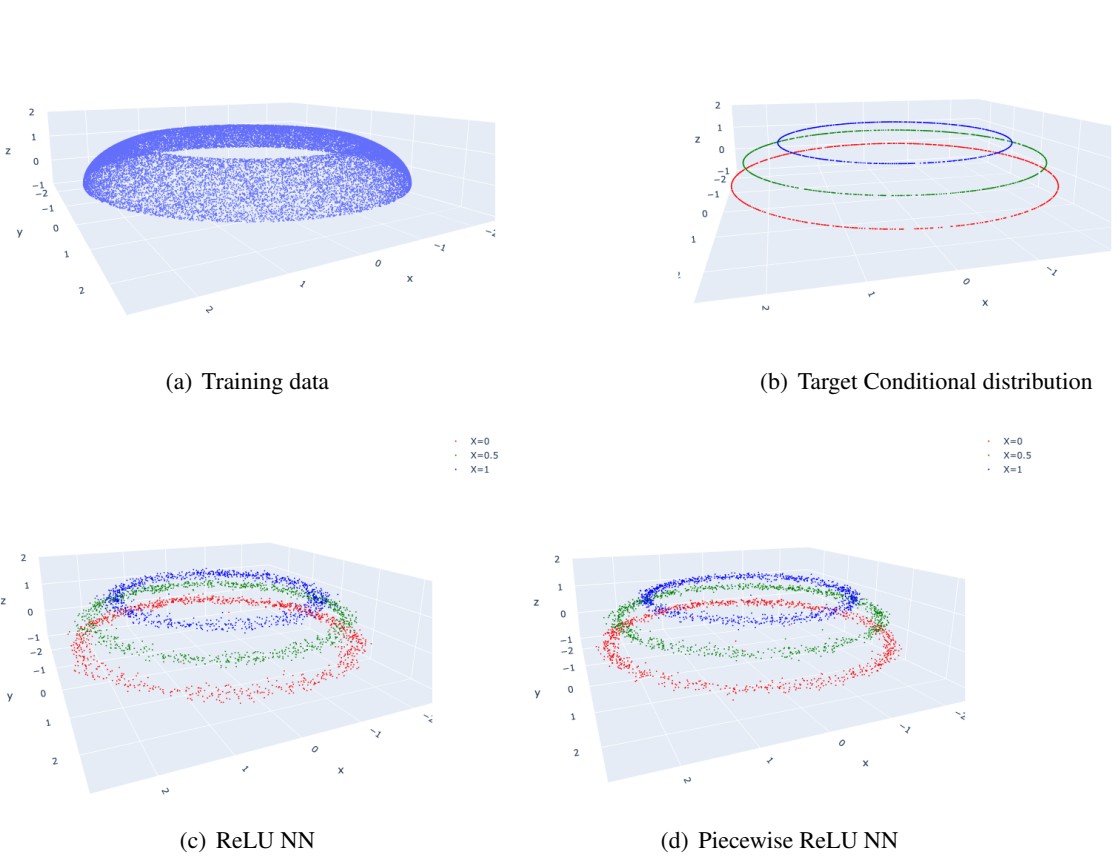

(a) Training data

(b) Target Conditional distribution

(c) ReLU NN

(d) Piecewise ReLU NN

Figure 1: Figure (a) shows the training dataset. Figure (b) displays samples from the (true) conditional distribution $Y \mid x = 0$ (red dots), $Y \mid x = 0.5$ (green dots), and $Y \mid x = 1$ (blue dots). Figure (c) presents samples generated from the conditional diffusion model with the conditional score modeled by a ReLU neural network. Figure (d) illustrates samples generated from the conditional diffusion model with the conditional score modeled by the piecewise ReLU neural network defined in equation (1).

---

[1]We consider the MMD distance associated with the RBF kernel $\exp(-\|x - y\|^2/6)$.

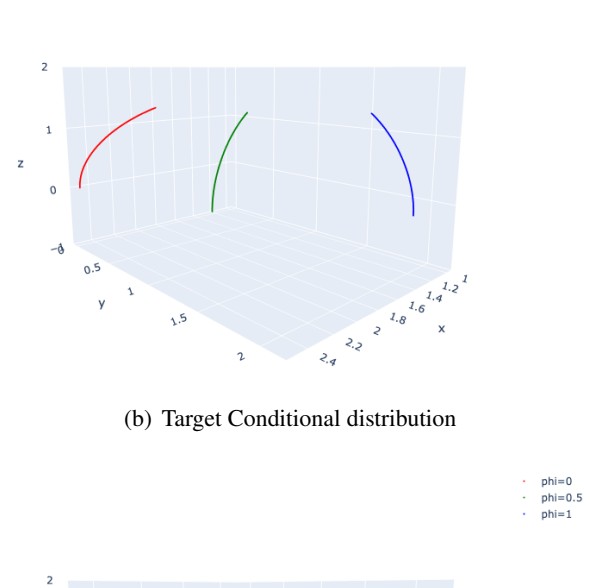

|  |  |
|---|---|
| (a) Training data | (b) Target Conditional distribution |

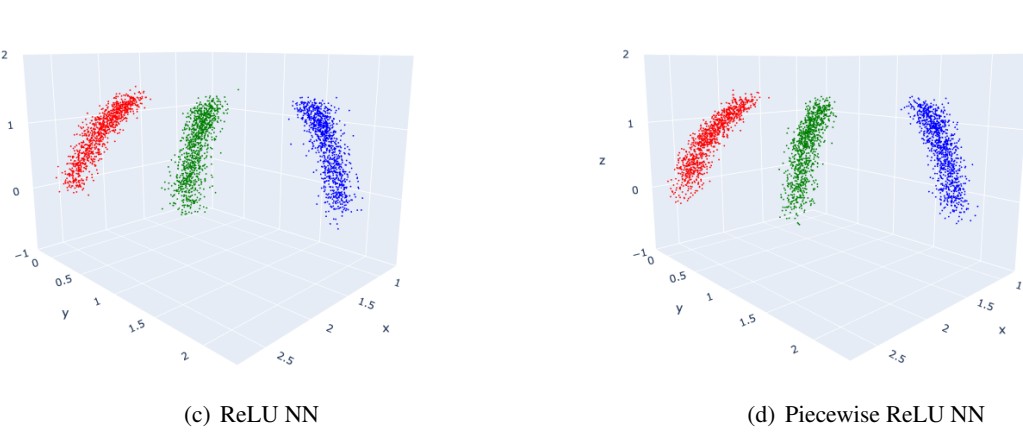

|  |  |
|---|---|
| (c) ReLU NN | (d) Piecewise ReLU NN |

Figure 2: Figure (a) shows the training dataset. Figure (b) displays samples from the (true) conditional distribution $Y \mid \phi = 0$ (red dots), $Y \mid \phi = 0.5$ (green dots), and $Y \mid \phi = 1$ (blue dots). Figure (c) presents samples generated from the conditional diffusion model with the conditional score modeled by a ReLU neural network. Figure (d) illustrates samples generated from the conditional diffusion model with the conditional score modeled by the piecewise ReLU neural network defined in equation (1).

Table 1: The table presents the MMD distances between the conditional diffusion model estimator and the target conditional distribution. The first three rows show the MMD distances for the conditional distributions $Y \mid x = 0$, $Y \mid x = 0.5$, $Y \mid x = 1$ and $Y \mid \phi = 0$, $Y \mid \phi = 0.5$, $Y \mid \phi = 1$. The last row displays the expected conditional MMD distance, where the expectation is taken with respect to $x$ sampled from a uniform distribution on $[0, 1]$ and $\phi$ sampled from uniform over $[0, 2\pi]$.

|  | Piecewise NN | NN |  | Piecewise NN | NN |
|---|---|---|---|---|---|
| $x = 0$ | 0.0023 | 0.0032 | $\phi = 0$ | 0.0002 | 0.0034 |
| $x = 0.5$ | 0.0013 | 0.0014 | $\phi = 0.5$ | 0.0009 | 0.0013 |
| $x = 1$ | 0.0007 | 0.0017 | $\phi = 1$ | 0.0014 | 0.0028 |
| $x \sim Unif[0, 1]$ | 0.0021 | 0.0031 | $\phi \sim Unif[0, 2\pi]$ | 0.0008 | 0.0015 |

# B  Missing Definition and Assumption

We begin by defining a class of smooth multivariate functions.

**Definition 1.** *The class $C_L^{\alpha_1,\alpha_2}(U_1, U_2)$ consists of functions $f : U_1 \times U_2 \to \mathbb{R}$ with $U_1 \subset \mathbb{R}^{d_1}$, $U_2 \subset \mathbb{R}^{d_2}$ so that for any $(x_0, y_0) \in U_1 \times U_2$, and each index $(j_1, j_2) \in \mathcal{J}_{\alpha_1,\alpha_2}^{d_1,d_2} = \{j_1 \in \mathbb{N}_0^{d_1}, j_2 \in \mathbb{N}_0^{d_2} : |j_1| + \frac{\alpha_1}{\alpha_2}|j_2| < \alpha_1\}$, there is a number $f_{(j_1,j_2)}(x_0, y_0) \in [-L, L]$ so that for any $(x, y) \in U_1 \times U_2$,*

$$\left| f(x, y) - \sum_{(j_1,j_2) \in \mathcal{J}_{\alpha_1,\alpha_2}^{d_1,d_2}} \frac{f_{(j_1,j_2)}(x_0, y_0)}{j_1! j_2!}(x - x_0)^{j_1}(y - y_0)^{j_1} \right| \le L(\|x - x_0\|^{\alpha_1} + \|y - y_0\|^{\alpha_2}).$$

*For $d > 1$, we use $C_{L,d}^{\alpha_1,\alpha_2}(U_1, U_2) = \{f = (f_1, f_2, \cdots, f_d) : U_1 \times U_2 \to \mathbb{R}^d : \forall i \in [d], f_i \in C_L^{\alpha_1,\alpha_2}(U_1, U_2)\}$ to denote the vector-valued function space counterpart. We call a (vector-valued) function $f = (f_1, f_2, \cdots, f_d) : U_1 \times U_2 \to \mathbb{R}^d$ being $C^{\alpha_1,\alpha_2}$-smooth if there exists a constant $L$ so that $f \in C_{L,d}^{\alpha_1,\alpha_2}(U_1, U_2)$.*

If $\alpha_X \le 1$, then a conditional density function $p(y|x)$ is $C^{\alpha_Y,\alpha_X}$-smooth if, for any given $x$, $p(y|x)$ is $\alpha_Y$-Hölder smooth with respect to the variable $y$, and for any $y$ and pair $x, x'$, $|p(y|x) - p(y|x')| \le L\|x - x'\|^{\alpha_X}$. This corresponds to the conditional density class considered in [1]. When $\alpha_X > 1$, the $C^{\alpha_Y,\alpha_X}$ smoothness of a function can be verified by checking the existence and smoothness of its partial derivatives, as summarized in the following lemma.

**Lemma B.1.** *For a function $f : U_1 \times U_2 \to \mathbb{R}$, if there exists a function $\overline{f} : \mathbb{R}^{d_1} \times \mathbb{R}^{d_2} \to \mathbb{R}$ so that $\overline{f}|_{U_1 \times U_2} = f$ and*

$$\sum_{(j_1,j_2) \in \mathcal{J}_{\alpha_1,\alpha_2}^{d_1,d_2}} \sup_{(x,y) \in \mathbb{R}^{d_1} \times \mathbb{R}^{d_2}} |\overline{f}^{(j_1,j_2)}(x, y)| + \sum_{\substack{(j_1,j_2) \in \mathcal{J}_{\alpha_1,\alpha_2}^{d_1,d_2} \\ \frac{|j_1|+1}{\alpha_1} + \frac{|j_2|}{\alpha_2} \ge 1}} \sup_{\substack{x,x_0 \in \mathbb{R}^{d_1}, y \in \mathbb{R}^{d_2} \\ x \ne x_0}} \frac{|\overline{f}^{(j_1,j_2)}(x, y) - \overline{f}^{(j_1,j_2)}(x_0, y)|}{\|x - x_0\|^{\alpha_1 - |j_1| - \frac{\alpha_1}{\alpha_2}|j_2|}}$$

$$+ \sum_{\substack{(j_1,j_2) \in \mathcal{J}_{\alpha_1,\alpha_2}^{d_1,d_2} \\ \frac{|j_1|}{\alpha_1} + \frac{|j_2|+1}{\alpha_2} \ge 1}} \sup_{\substack{x \in \mathbb{R}^{d_1}, y, y_0 \in \mathbb{R}^{d_2} \\ y \ne y_0}} \frac{|\overline{f}^{(j_1,j_2)}(x, y) - \overline{f}^{(j_1,j_2)}(x, y_0)|}{\|y - y_0\|^{\alpha_2 - |j_2| - \frac{\alpha_2}{\alpha_1}|j_1|}} \le L,$$

*then there exists a constant $L_1$ so that $f \in C_{L_1}^{\alpha_1,\alpha_2}(U_1, U_2)$.*

*Proof.* For any $(x, y), (x_0, y_0) \in U_1 \times U_2$, we have

$$\left| \overline{f}(x, y) - \sum_{\substack{j_2 \in \mathbb{N}_0^{d_2} \\ |j_2| < \alpha_2}} \frac{\overline{f}^{(0,j_2)}(x, y_0)}{j_2!}(y - y_0)^{j_2} \right|$$

$$= \left| \sum_{\substack{j_2 \in \mathbb{N}_0^{d_2} \\ |j_2| = \lfloor \alpha_2 \rfloor}} \frac{\lfloor \alpha_2 \rfloor}{j_2!} \int_0^1 (1 - t)^{\lfloor \alpha_2 \rfloor - 1} \left( \overline{f}^{(0,j_2)}(x, y_0 + t(y - y_0)) - \overline{f}^{(0,j_2)}(x, y_0) \right) \mathrm{d}t \cdot (y - y_0)^{j_2} \right|$$

$$= \mathcal{O}(\|y - y_0\|^{\alpha_2}).$$

Moreover, we have

$$\left| \sum_{\substack{j_2 \in \mathbb{N}_0^{d_2} \\ |j_2| < \alpha_2}} \frac{\overline{f}^{(0,j_2)}(x, y_0)}{j_2!} (y - y_0)^{j_2} - \sum_{\substack{j_2 \in \mathbb{N}_0^{d_2} \\ |j_2| < \alpha_2}} \sum_{\substack{j_1 \in \mathbb{N}_0^{d_1} \\ |j_1| + \frac{\alpha_1}{\alpha_2}|j_2| < \alpha_1}} \frac{\overline{f}^{(j_1,j_2)}(x_0, y_0)}{j_2!} (x - x_0)^{j_1} (y - y_0)^{j_2} \right|$$

$$= \left| \sum_{\substack{j_2 \in \mathbb{N}_0^{d_2} \\ |j_2| < \alpha_2}} \sum_{\substack{j_1 \in \mathbb{N}_0^{d_1} \\ |j_1| = \lfloor \alpha_1 - \frac{\alpha_1}{\alpha_2}|j_2| \rfloor}} \frac{\lfloor \alpha_1 - \frac{\alpha_1}{\alpha_2}|j_2| \rfloor}{j_1! j_2!} \int_0^1 (1-t)^{\lfloor \alpha_1 - \frac{\alpha_1}{\alpha_2}|j_2| \rfloor - 1} \left( \overline{f}^{(j_1,j_2)}(x_0 + t(x - x_0), y_0) - \overline{f}^{(j_1,j_2)}(x_0, y_0) \right) \mathrm{d}t \right.$$

$$\left. \cdot (x - x_0)^{j_1} (y - y_0)^{j_2} \right|$$

$$= \mathcal{O}\left( \sum_{\substack{j_2 \in \mathbb{N}_0^{d_2} \\ |j_2| < \alpha_2}} \sum_{\substack{j_1 \in \mathbb{N}_0^{d_1} \\ |j_1| = \lfloor \alpha_1 - \frac{\alpha_1}{\alpha_2}|j_2| \rfloor}} \|x - x_0\|^{\alpha_1 - \frac{\alpha_1}{\alpha_2}|j_2|} \|y - y_0\|^{|j_2|} \right) = \mathcal{O}(\|x - x_0\|^{\alpha_1} + \|y - y_0\|^{\alpha_2}),$$

where the last inequality uses the Young's inequality for products. Therefore, by choosing $f_{(j_1, j_2)}(x_0, y_0) = \overline{f}^{(j_1,j_2)}(x_0, y_0)$, we can get

$$\left| f(x, y) - \sum_{(j_1, j_2) \in \mathcal{J}_{\alpha_1, \alpha_2}^{d_1, d_2}} \frac{f_{(j_1,j_2)}(x_0, y_0)}{j_1! j_2!} (x - x_0)^{j_1} (y - y_0)^{j_1} \right|$$

$$= \left\| \overline{f}(x, y) - \sum_{\substack{j_2 \in \mathbb{N}_0^{d_2} \\ |j_2| < \alpha_2}} \sum_{\substack{j_1 \in \mathbb{N}_0^{d_1} \\ |j_1| + \frac{\alpha_1}{\alpha_2}|j_2| < \alpha_1}} \frac{\overline{f}^{(j_1,j_2)}(x_0, y_0)}{j_2!} (x - x_0)^{j_1} (y - y_0)^{j_2} \right\| = \mathcal{O}(\|x - x_0\|^{\alpha_1} + \|y - y_0\|^{\alpha_2}).$$

$\square$

Note that when $\alpha_1 = \alpha_2 = \alpha$, the function class considered in Lemma B.1 is the conventional Hölder class with Hölder exponent being $\alpha$. Then we present the formal version of Assumption D for the analysis of the conditional diffusion model under the manifold assumption.

**Assumption D** (Smoothness of $\mu_{Y|X}^*$): For any $x \in \mathcal{M}_X$, $\mathcal{M}_{Y|x}$ is a compact boundaryless $d_Y$-dimensional submanifold embedded in $\mathbb{R}^{D_Y}$ with reach bounded away from zero. Moreover, there exists positive constants $(c_1, c_2, r_0, r_1, L, L_1, L_2, L_3)$ so that for any $w = (x_0, y_0) \in \mathcal{M}$, there exists a open set $U^\omega$ on $\mathcal{M}$ so that

1. $\mathbb{B}_{\mathcal{M}_X}(x_0, r_0) \subset U_X^\omega$ and for any $x \in U_X^\omega$, $\mathbb{B}_{\mathcal{M}_{Y|x}}(y_0, r_0) \subset U_{Y|x}^\omega$.

2. For any $x \in U_X^\omega$, there exists a uniformly $L$-Lipschitz diffeomorphism $Q_x^\omega$ that maps $U_{Y|x}^\omega$ to $\mathbb{B}_{\mathbb{R}^{d_Y}}(0, r_1)$ with inverse $[Q_x^\omega]^{-1}$ so that $Q_{x_0}^\omega(y_0) = 0$ and

   (a) The function $G^\omega : \mathbb{B}_{\mathbb{R}^{d_Y}}(0, r_1) \times U_X^\omega \to U_Y^\omega$ defined as $G^\omega(z, x) = [Q_x^\omega]^{-1}(z)$ satisfies that $G^\omega \in C_{L_1, D_Y}^{\beta_Y, \beta_X}(\mathbb{B}_{\mathbb{R}^{d_Y}}(0, r_1), U_X^\omega)$.

   (b) For any $x \in \mathcal{M}_X$, $\mu_{Y|x}^*$ have a density $f(y|x)$ with respect to the volume measure of $\mathcal{M}_{Y|x}$ so that $f(y|x)$ is uniformly lower bounded by $c_1$. Moreover, the function $v^\omega : \mathbb{B}_{\mathbb{R}^{d_Y}}(0, r_1) \times U_X^\omega \to \mathbb{R}$ defined by $v^\omega(z, x) = f([Q_x^\omega]^{-1}(z)|x) \cdot \sqrt{\det(J_x(z)^T J_x(z))}$ with $J_x(z) = J_{[Q_x^\omega]^{-1}}(z)$ being the Jacobian matrix of $[Q_x^\omega]^{-1}(\cdot)$ evaluated at $z$, satisfies that $v^\omega \in C_{L_2}^{\alpha_Y, \alpha_X}(\mathbb{B}_{\mathbb{R}^{d_Y}}(0, r_1), U_X^\omega)$.

**Remark B.1.** *Intuitively, this assumption implies that, locally around any point $\omega = (x_0, y_0) \in \mathcal{M}$, there exists an encoder-dencoder pair $(G^\omega(z, x), Q_x^\omega(y))$. Here, $Q_x^\omega(\cdot)$ maps $y \in \mathbb{B}_{\mathcal{M}_{Y|x}}(y_0, r_0)$ to a low dimensional code $z \in \mathbb{R}^{d_Y}$, and $G^\omega(\cdot, x)$ reconstruct the data $y$ from the code $z$. A large $\beta_Y$ implies*

that, for any given $x \in \mathcal{M}_X$, the manifold $M_{Y|x}$ is smooth. On the other hand, a large $\beta_X$ indicates that the space $M_{Y|x}$ changes smoothly with $x$, meaning $\mathcal{M}_{Y|x}$ and $\mathcal{M}_{Y|x'}$ will be similar if $x$ is close to $x'$ in Euclidean distance. Moreover, $v^\omega(z|x) = v^\omega(z, x)$ is the conditional density function of the (local) latent vector, that is, the density of the push forward measure $[Q_x^\omega]_\# (\mu_{Y|x}^*|_{U_{Y|x}^\omega})$ with respect to the Lebesgue measure on $\mathbb{R}^{d_Y}$, where $\mu_{Y|x}^*|_{U_{Y|x}^\omega}$ is the measure of $\mu_{Y|x}^*$ constrained on $U_{Y|x}^\omega$, defined by $\frac{\mathrm{d}\mu_{Y|x}^*|_{U_{Y|x}^\omega}}{\mathrm{d}\mu_{Y|x}^*} = \mathbf{1}(y \in U_{Y|x}^\omega)$. Therefore, a large $\alpha_Y$ indicates that the conditional density function will be smooth for any fixed $x$, while a large $\alpha_X$ means that the density function changes smoothly in $x$. Under Assumption D, we can express $\mu_{Y|X}^*$ as a mixture of (deterministic) conditional generative model, as shown in the following lemma.

**Lemma B.2.** *(Expressing $\mu_{Y|X}^*$ as conditional generative model) For any $r$ with $0 < r \leq r_0$, it holds that*

1. *For any $(x^*, y^*) \in \mathcal{M}$ and $x \in \mathbb{B}_{\mathcal{M}_X}(x^*, r_0)$, it holds for any measurable function $g : \mathcal{M}_{Y|x} \to \mathbb{R}$ that*

$$\mathbb{E}_{y \sim \mu_{Y|x}^*}[g(y) \cdot \mathbf{1}(y \in \mathbb{B}_{\mathcal{M}_{Y|x}}(y^*, r))] = \int_{\mathbb{B}_{\mathbb{R}^{d_Y}}(0, r_1)} g(G^*(z, x)) \mathbf{1}(G^*(z, x) \in \mathbb{B}_{\mathcal{M}_{Y|x}}(y^*, r)) v^*(z|x) \, \mathrm{d}z,$$

*where $G^* = G^\omega$ and $v^* = v^\omega$ with $\omega = (x^*, y^*)$.*

2. *For any function $\rho : \mathbb{R} \to [0, 1]$ satisfies when $t \in [0, 1]$, $\rho(t) = 1$ and when $t \in [2, \infty)$, $\rho(t) = 0$. Let $\{(x_k^*, y_k^*)\}_{k=1}^K \subset \mathcal{M}$ be a $\frac{r}{2}$-cover of $\mathcal{M}$. Then for any $\widetilde{r}$ satisfies $r \leq \widetilde{r} \leq r_0$, denote*

$$\rho_k(x, y) = \rho\left(\frac{4\|x - x_k^*\|^2}{\widetilde{r}^2}\right) \rho\left(\frac{4\|y - y_k^*\|^2}{\widetilde{r}^2}\right)$$

*and $\widetilde{\rho}_k = \rho_k / \sum_{k=1}^K \rho_k$, it holds for any measurable function $g : \mathcal{M}_{Y|x} \to \mathbb{R}$ that*

$$\mathbb{E}_{y \sim \mu_{Y|x}^*}[g(y)] = \sum_{k=1}^K \int_{\mathbb{B}_{\mathbb{R}^{d_Y}}(0, r_1)} g(G_{[k]}^*(z, x)) \widetilde{\rho}_k(x, G_{[k]}^*(z, x)) v_{[k]}^*(z|x) \, \mathrm{d}z,$$

*where $G_{[k]}^* = G^\omega$ and $v_{[k]}^* = v^\omega$ with $\omega = (x_k^*, y_k^*)$.*

# C Proof of Theorem 2

Denote $\mu_X^*$ as the distribution of $X$, and $\mu^* = \mu_X^* \mu_{Y|X}^*$ as the joint distribution of $(X, Y)$. We only need to consider the case where $\beta_Y = \alpha_Y \vee 1 + 1$ and $\beta_X = \alpha_X + \frac{\alpha_X}{\alpha_Y}$. Set

$$\tau = (n^{-\frac{1}{2\alpha_Y + d_Y + \frac{\alpha_Y}{\alpha_X} d_X}} \sqrt{\log n})^{2(\alpha_Y + 1)}.$$

We have the following lemma that relate the generalization error of the conditional score function $\nabla \log p_{t|x}(w)$ to the generalization error of $\mu_{Y|x}^*$.

**Lemma C.1.** *Suppose for any $t \in [\tau, T]$, $\sup_{w \in \mathbb{R}^{D_Y}} \sup_{x \in \mathcal{M}_X} \|\widehat{S}(w, x, t)\| \lesssim \sqrt{\frac{\log n}{t \wedge 1}}$, then let $\mu_{Y|x}^\tau$ be the distribution of $\overleftarrow{Y}_{\tau|x}$, we have*

$$\mathbb{E}_{\mu_X^*}[\mathrm{TV}\left(\widehat{\mu}_{Y|x}^\tau, \mu_{Y|x}^*\right)] \lesssim \frac{1}{n} + \mathbb{E}_{\mu_X^*}[\mathrm{TV}(\mu_{Y|x}^*, \mu_{Y|x}^\tau)]$$

$$+ \mathbb{E}_{\mu_X^*}\left[\sum_{i=0}^{K-1} \sqrt{\int_{t_i}^{t_{i+1}} \int_{\mathbb{R}^{D_Y}} \left\|\widehat{S}(w, x, t) - \nabla \log p_{t|x}(w)\right\|^2 p_{t|x}(w) \, \mathrm{d}w \, \mathrm{d}t}\right].$$

*and*

$$\mathbb{E}_{\mu_X^*}[W_1(\widehat{\mu}_{Y|x}, \mu_{Y|x}^*)] \lesssim \frac{1}{n} + \tau^{\frac{1}{2}}$$

$$+ \mathbb{E}_{\mu_X^*}\left[\sum_{i=0}^{K-1}\sqrt{((t_i \log n) \wedge 1) \int_{t_i}^{t_{i+1}} \int_{\mathbb{R}^{D_Y}} \left\|\widehat{S}(w, x, t) - \nabla \log p_{t|x}(w)\right\|^2 p_{t|x}(w)\, \mathrm{d}w\, \mathrm{d}t}\right].$$

The proof of Lemma C.1 directly follows from Lemma D.7 of [2] and Lemma B.2 of [4]. Then the following lemma provides upper bounds to the score approximation error.

**Lemma C.2.** *For any $t \in [\underline{t}, \overline{t}]$ with $1 < \frac{\overline{t}}{\underline{t}} \le 2$:*

1. *If $\tau \le \underline{t} < n^{-\frac{2}{2\alpha_Y + d_Y + d_X \frac{\alpha_Y}{\alpha_X}}}$, there exists a neural network $\phi_{score}(w, x, t) \in \Phi(H, W, R, B, V)$ satisfying*

$$\mathbb{E}_{\mu_X^*}\left[\int_{\underline{t}}^{\overline{t}} \int_{\mathbb{R}^{D_Y}} \left\|\phi_{score}(w, x, t) - \nabla \log p_{t|x}(w)\right\|^2 p_{t|x}(w)\, \mathrm{d}w \mathrm{d}t\right]$$

$$= \widetilde{\mathcal{O}}\left(\frac{\varepsilon_2^{2\beta_X} + \varepsilon_1^{2\beta_Y}}{\underline{t}} + \varepsilon_1^{2\alpha_Y}\right),$$

   *where $\varepsilon_1 = n^{-\frac{1}{2\alpha_Y + d_Y + \frac{\alpha_Y}{\alpha_X} d_X}}$ and $\varepsilon_2 = \varepsilon_1^{\frac{\alpha_Y}{\alpha_X}}$. Here $H$, $W$, $R$, $B$ and $V$ are evaluated as $H = \Theta(\log^4 n)$, $\|W\|_\infty = \widetilde{\Theta}(\varepsilon_1^{-d_Y} \varepsilon_2^{-d_X})$, $R = \widetilde{\Theta}(\varepsilon_1^{-d_Y} \varepsilon_2^{-d_X})$, $B = \exp(\Theta(\log^4 n))$ and $V = \Theta(\sqrt{\frac{\log n}{\underline{t}}})$.*

2. *If $n^{-\frac{2}{2\alpha_Y + d_Y + d_X \frac{\alpha_Y}{\alpha_X}}} \le \underline{t} \le (\log n)^{-3}$, there exists a neural network $\phi_{score}(w, x, t) \in \Phi(H, W, R, B, V)$ satisfying*

$$\mathbb{E}_{\mu_X^*}\left[\int_{\underline{t}}^{\overline{t}} \int_{\mathbb{R}^{D_Y}} \left\|\phi_{score}(w, x, t) - \nabla \log p_{t|x}(w)\right\|^2 p_{t|x}(w)\, \mathrm{d}w \mathrm{d}t\right] = \widetilde{\mathcal{O}}(\varepsilon_2^{2\alpha_X}),$$

   *with $\varepsilon_2 = \widetilde{\Theta}(n^{-\frac{1}{2\alpha_X + d_X}} \underline{t}^{-\frac{d_Y}{4\alpha_X + 2d_X}})$. Here $H$, $W$, $R$, $B$ and $V$ are evaluated as $H = \Theta(\log^4 n)$, $\|W\|_\infty = \widetilde{\Theta}(\underline{t}^{-\frac{d_Y}{2}} \varepsilon_2^{-d_X})$, $R = \widetilde{\Theta}(\underline{t}^{-\frac{d_Y}{2}} \varepsilon_2^{-d_X})$, $B = \exp(\Theta(\log^4 n))$ and $V = \Theta(\sqrt{\frac{\log n}{\underline{t}}})$.*

3. *If $(\log n)^{-3} \le \underline{t} \le T = \Theta(\log n)$, there exists a neural network $\phi_{score}(w, x, t) \in \Phi(H, W, R, B, V)$ satisfying*

$$\mathbb{E}_{\mu_X^*}\left[\int_{\underline{t}}^{\overline{t}} \int_{\mathbb{R}^{D_Y}} \left\|\phi_{score}(w, x, t) - \nabla \log p_{t|x}(w)\right\|^2 p_{t|x}(w)\, \mathrm{d}w \mathrm{d}t\right] = \widetilde{\mathcal{O}}(n^{-\frac{2\alpha_X}{2\alpha_X + d_X}}).$$

   *Here $H$, $W$, $R$, $B$ and $V$ are evaluated as $H = \Theta(\log^4 n)$, $\|W\|_\infty = \widetilde{\Theta}(n^{\frac{d_X}{2\alpha_X + d_X}})$, $R = \widetilde{\Theta}(n^{\frac{d_X}{2\alpha_X + d_X}})$, $B = \exp(\Theta(\log^4 n))$ and $V = \Theta(\sqrt{\frac{\log n}{\underline{t} \wedge 1}})$.*

Then denote $\mathcal{S}_i = \Phi(H_i, W_i, R_i, B_i, V_i)$, by Lemma C.2, we have for any $i \in [\mathcal{I}]$,

$$((t_i \log n) \wedge 1) \cdot \min_{S \in \mathcal{S}_i} \mathbb{E}_{\mu_X^*}\left[\int_{t_{i-1}}^{t_i} \int_{\mathbb{R}^{D_Y}} \left\|S(w, x, t) - \nabla \log p_{t|x}(w)\right\|^2 p_{t|x}(w)\, \mathrm{d}t \mathrm{d}w\right]$$

$$= \begin{cases} \widetilde{\mathcal{O}}\left(n^{-\frac{2 + \frac{2}{\alpha_Y}}{2 + \frac{d_X}{\alpha_X} + \frac{d_Y}{\alpha_Y}}}\right), & \tau \le t_{i-1} < n^{-\frac{2}{2\alpha_Y + d_Y + d_X \frac{\alpha_Y}{\alpha_X}}}; \\ \widetilde{\mathcal{O}}\left(n^{-\frac{2\alpha_X}{2\alpha_X + d_X}} + n^{-\frac{2 + \frac{2}{\alpha_Y}}{2 + \frac{d_X}{\alpha_X} + \frac{d_Y}{\alpha_Y}}}\right) & n^{-\frac{2}{2\alpha_Y + d_Y + d_X \frac{\alpha_Y}{\alpha_X}}} \le t_{i-1} < \frac{c}{\log n} \\ \widetilde{\mathcal{O}}\left(n^{-\frac{2\alpha_X}{2\alpha_X + d_X}}\right) & \frac{c}{\log n} \le t_{i-1} \le T \end{cases}$$

Then we derive the following oracle inequality for bounding the estimation error.

**Lemma C.3.** *It holds with probability larger than $1 - \frac{1}{n}$ that for any $i \in [\mathcal{I}]$,*

$$
\mathbb{E}_{\mu_X^*} \Big[ \int_{t_{i-1}}^{t_i} \int_{\mathbb{R}^{D_Y}} \| \widehat{S}(w, x, t) - \nabla \log p_{t|x}(w) \|^2 \, p_{t|x}(w) \, \mathrm{d}t \mathrm{d}w \Big]
$$
$$
\lesssim (\log n)^2 \frac{R_i H_i \log (R_i H_i \| W_i \|_\infty (B_i \vee 1) n)}{n}
$$
$$
+ \min_{S \in \mathcal{S}_i} \mathbb{E}_{\mu_X^*} \Big[ \int_{t_{i-1}}^{t_i} \int_{\mathbb{R}^{D_Y}} \| S(w, x, t) - \nabla \log p_{t|x}(w) \|^2 \, p_{t|x}(w) \, \mathrm{d}t \mathrm{d}w \Big]
$$

So by combining all pieces, we have

$$
\mathbb{E}_{\mu_X^*} [W_1(\widehat{\mu}_{Y|x}, \mu_{Y|x}^*)] \lesssim \tau^{\frac{1}{2}} + \frac{1}{n}
$$
$$
+ \mathbb{E}_{\mu_X^*} \sum_{i=1}^{\mathcal{I}} \sqrt{((t_i \log n) \wedge 1) \int_{t_{i-1}}^{t_i} \int_{\mathbb{R}^{D_Y}} \left\| \widehat{S}(w, x, t) - \nabla \log p_{t|x}(w) \right\|^2 p_{t|x}(w) \, \mathrm{d}w \, \mathrm{d}t}
$$
$$
\lesssim \tau^{\frac{1}{2}} + \sum_{i=1}^{\mathcal{I}} (\log n) \sqrt{((t_i \log n) \wedge 1) \frac{R_i H_i \log (R_i H_i \| W_i \|_\infty (B_i \vee 1) n)}{n}}
$$
$$
+ \sum_{i=1}^{\mathcal{I}} \sqrt{((t_i \log n) \wedge 1) \min_{S \in \mathcal{S}_i} \mathbb{E}_{\mu_X^*} \Big[ \int_{t_{i-1}}^{t_i} \int_{\mathbb{R}^{D_Y}} \| S(w, x, t) - \nabla \log p_{t|x}(w) \|^2 \, p_{t|x}(w) \, \mathrm{d}t \mathrm{d}w \Big]}
$$
$$
= \widetilde{\mathcal{O}} \Big( n^{-\frac{\alpha_X}{2\alpha_X + d_X}} + n^{-\frac{\alpha_Y + 1}{2\alpha_Y + d_Y + d_X \frac{\alpha_Y}{\alpha_X}}} \Big).
$$

## C.1  Proof of Lemma C.2

To begin with, we introduce the following lemma, which states that it is sufficient to approximate the score function $\nabla \log p_{t|x}(w)$ only for values of $w$ that are in close proximity to the manifold $\mathcal{M}_{Y|x}$.

**Lemma C.4.** *If* $\sup_{w \in \mathbb{R}^{D_Y}, x \in \mathcal{M}_X} \sup_{t \in [\tau, T]} [\| S(w, x, t) \|_\infty \sigma_t] \le c\sqrt{\log n}$. *Then, there exist constants* $(c_0, c_1, c_2, c_3)$ *so that for any $i \in [\mathcal{I}]$ and $t \in [t_{i-1}, t_i]$ with $1 < \frac{t_i}{t_{i-1}} \le 2$,*

1. *Denote* $\mathrm{dist}(w, \mathcal{M}_{Y|x})$ *as the distance of point $w \in \mathbb{R}^{D_Y}$ to manifold $\mathcal{M}_{Y|x}$. Then for any $x \in \mathcal{M}_X$,*

$$
\int_{\mathbb{R}^{D_Y}} \left\| \nabla \log p_{t|x}(w) - S(w, x, t) \right\|^2 p_{t|x}(w) \, \mathrm{d}w
$$
$$
\le \int_{\mathbb{R}^{D_Y}} \left\| \nabla \log p_{t|x}(w) - S(w, x, t) \right\|^2 p_{t|x}(w) \cdot 1 \left( \mathrm{dist}(w, \mathcal{M}_{Y|x}) \le c_0 \sigma_{t_{i-1}} \sqrt{\log n} \right) \mathrm{d}w
$$
$$
+ (1 + c^2) \cdot c_1 \frac{1}{n^2}.
$$

2. *For any $x \in \mathcal{M}_X$ and $w \in \mathbb{R}^{D_Y}$ satisfying $\mathrm{dist}(w, \mathcal{M}_{Y|x}) \le c_0 \sigma_{t_{i-1}} \sqrt{\log n}$, we have*

   (a) $\| \nabla \log p_{t|x}(w) \|_\infty \le c_2 \frac{\sqrt{\log n}}{\sigma_{t_{i-1}}}$.

   (b) $(2\pi \sigma_t^2)^{\frac{D}{2}} p_{t|x}(w) \ge n^{-c_3}$.

Then the following lemma gives a way for constructing covering set to $\mathcal{M}_{Y|X}$.

**Lemma C.5.** *Under Assumptions C and D, there exist positive constants $r, \overline{r}, L_1$ satisfying $r \le \frac{r_0}{4}$ and $\overline{r} \le \frac{r_1}{2}$, so that for any $\varepsilon_1 \in (0, \overline{r}), \varepsilon_2 \in (0, r)$, let $\mathcal{N}_{\varepsilon_1}^Z$ be an $\varepsilon_1$-cover of $\mathbb{B}_{\mathbb{R}^{d_Y}}(0, \overline{r})$, $\mathcal{N}_{\varepsilon_2}^X$ be an $\varepsilon_2$-cover of $\mathcal{M}_X$ and $\{(x_k^*, y_k^*)\}_{k=1}^K \subset \mathcal{M}$ be any $r$-cover of $\mathcal{M}$, we have*

1. for any $k \in [K]$ and $x \in U_X^{(x_k^*, y_k^*)}$, $z \in \mathbb{B}_{\mathbb{R}^{d_Y}}(0, r_1)$,

$$\lambda_{\min}\big(J_{G^{(x_k^*, y_k^*)}(\cdot, x)}(z)^T J_{G^{(x_k^*, y_k^*)}(\cdot, x)}(z)\big) \geq \frac{1}{L^2}.$$

2. for any $k \in [K]$ and $(x, y) \in \mathcal{M}$ so that $\|x - x_k^*\| \leq r$ and $\|y - y_k^*\| \leq r$, it holds that $Q_x^{(x_k^*, y_k^*)}(y) \in \mathbb{B}_{\mathbb{R}^{d_Y}}(0, \bar{r})$.

3. for any $(x, y) \in \mathcal{M}$, there exists $x^* \in \mathcal{N}_{\varepsilon_2}^X$, $k \in \mathcal{K}_{x^*} = \{k \in [K] : \|x^* - x_k^*\| \leq 2r\}$ and $z^* \in \mathcal{N}_{\varepsilon_1}^Z$ so that $\|x - x^*\| \leq \varepsilon_2$ and $\|G^{(x_k^*, y_k^*)}(z^*, x) - y\| \leq L_1 \varepsilon_1$.

4. for any $x^* \in \mathcal{N}_{\varepsilon_2}^X$ and $k \in \mathcal{K}_{x^*}$, if $x \in \mathcal{B}_{\mathcal{M}_X}(x^*, \varepsilon_2)$ and $z \in \mathbb{B}_{\mathbb{R}^{d_Y}}(0, \bar{r})$, then $\|G^{(x_k^*, y_k^*)}(z, x) - y_k^*\| \leq \frac{r_0}{2}$.

In the following analysis, we will fix an $r$-cover $\{(x_k^*, y_k^*)\}_{k=1}^K \subset \mathcal{M}$ of $\mathcal{M}$: and let $\mathcal{K}_{x^*} = \{k \in [K] : \|x^* - x_k^*\| \leq 2r\}$, where $r$ is the positive constant required in Lemma C.5. For each $k \in [K]$, we denote $U_{[k]}^* = U^{(x_k^*, y_k^*)}$,

$$G_{[k]}^*(z, x) = G^{(x_k^*, y_k^*)}(z, x), \qquad Q_{[k]}^*(y, x) = Q_x^{(x_k^*, y_k^*)}(y),$$

and

$$v_{[k]}^*(z|x) = v^{(x_k^*, y_k^*)}(z|x),$$

where $U^{(x_k^*, y_k^*)}$, $G^{(x_k^*, y_k^*)}(z, y)$, $Q_x^{(x_k^*, y_k^*)}(y)$ and $v^{(x_k^*, y_k^*)}(z|x)$ are defined in Assumption D with $\omega = (x_k^*, y_k^*)$. Then let us fix a time interval $t \in [\underline{t}, \overline{t}]$ where $1 < \frac{\overline{t}}{\underline{t}} \leq 2$. According to Lemma C.4, it suffices to focus on approximating the score function for $t \in [\underline{t}, \overline{t}]$, $x \in \mathcal{M}_X$ and $w \in \mathbb{R}^{D_Y}$ with $\mathrm{dist}(w, \mathcal{M}_{Y|x}) \leq c_0 \sigma_t \sqrt{\log n}$. Our first objective is to demonstrate that if there are neural networks capable of accurately approximating $\nabla \log p_{t|x}(w)$ for $(x, w)$ lies within local neighborhoods in $\mathcal{M}$, then there exists a neural network capable of providing a reliable approximation of $\nabla \log p_{t|x}(w)$ for all $(x, w)$ satisfying $x \in \mathcal{M}_X$ and $\mathrm{dist}(w, \mathcal{M}_{Y|x}) \leq c_0 \sigma_t \sqrt{\log n}$, this is summarized in the following lemma.

**Lemma C.6.** *Suppose $\tau \leq \underline{t} \leq T$, $\varepsilon_2 \in (0, r)$, and $\varepsilon_1 \geq \sigma_t \sqrt{\log n} + \varepsilon_2^{\beta_X}$. Let $\mathcal{N}_{\varepsilon_2}^X = (\widetilde{x}_1, \widetilde{x}_2, \cdots, \widetilde{x}_{J_2})$ be one of the largest $\varepsilon_2$-packing of $\mathcal{M}_X$, and let $\mathcal{N}_{\varepsilon_1}^Z = (\widetilde{z}_1, \widetilde{z}_2, \cdots, \widetilde{z}_{J_1})$ be one of the largest $\varepsilon_1$-packing of $\mathbb{B}_{\mathbb{R}^{d_Y}}(0, \bar{r})$. Then if for any $j_1 \in [J_1]$ and $j_2 \in [J_2]$, $k \in \mathcal{K}_{\widetilde{x}_{j_2}}$, there exists a neural network $\phi_{kj_1j_2}^*(w, x, t) \in \Phi\big(H, W, R, B, \Theta(\frac{\sqrt{\log n}}{\sigma_t})\big)$ so that for any $t \in [\underline{t}, \overline{t}]$, $x \in \mathbb{B}_{\mathcal{M}_X}(\widetilde{x}_{j_2}, \sqrt{2}\varepsilon_2)$ and $w \in \mathbb{R}^{D_Y}$ with $\|w - G_{[k]}^*(\widetilde{z}_{j_1}, x)\| \leq \sqrt{2}(2\varepsilon_1 L_1 + c_0 \sigma_t \sqrt{\log n})$ and $\mathrm{dist}(w, \mathcal{M}_{Y|x}) \leq c_0 \sigma_t \sqrt{\log n}$, it holds that*

$$\|\phi_{kj_1j_2}^*(w, x, t) - \nabla \log p_{t|x}(w)\|_\infty \leq \varepsilon.$$

*Then there exists a neural network $\phi_{\mathrm{score}}(w, x, t) \in \Phi\big(H_1, W_1, R_1, B_1, \Theta(\frac{\sqrt{\log n}}{\sigma_t})\big)$ with $H_1 = \Theta(H + \log^2 n)$, $\|W_1\|_\infty = \Theta(J_1 J_2(\|W\|_\infty + \log n) + \log^3 n)$, $R_1 = \Theta(J_1 J_2(R + \log n) + \log^4 n)$ and $B_1 = \exp(\Theta(\log^2 n)) \vee B$, so that for any $t \in [\underline{t}, \overline{t}]$, $x \in \mathcal{M}_X$ and $w \in \mathbb{R}^{D_Y}$ satisfying $\mathrm{dist}(x, \mathcal{M}_{Y|x}) \leq c_0 \sigma_t \sqrt{\log n}$, it holds that*

$$\|\phi_{\mathrm{score}}(w, x, t) - \nabla \log p_{t|x}(w)\|_\infty \lesssim \varepsilon + \frac{1}{n}.$$

Now recall that

$$\nabla \log p_{t|x}(w) = \frac{\nabla p_{t|x}(w)}{p_{t|x}(w)},$$

where

$$\nabla p_{t|x}(w) = (2\pi \sigma_t^2)^{-\frac{D}{2}} \mathbb{E}_{y \sim \mu_{Y|x}^*} \left[ \exp\left( -\frac{\|w - m_t y\|^2}{2\sigma_t^2} \right) \cdot -\frac{(w - m_t y)}{\sigma_t^2} \right],$$

and

$$p_{t|x}(w) = (2\pi\sigma_t^2)^{-\frac{D}{2}} \mathbb{E}_{y\sim\mu_{Y|x}^*} \left[ \exp\left( -\frac{\|w - m_t y\|^2}{2\sigma_t^2} \right) \right],$$

with $m_t = \exp\left(-\int_0^t \beta_s \, ds\right)$ and $\sigma_t^2 = 1 - m_t^2$, which satisfies $1 - m_t \asymp t \wedge 1$ and $\sigma_t \asymp \sqrt{t \wedge 1}$. By statement 2 of Lemma C.4, there exists a large enough constant $c_2$, so that for any $t \in [\underline{t}, \overline{t}]$, $x \in \mathcal{M}_X$, $w \in \mathbb{R}^{D_Y}$ with $\mathrm{dist}(w, \mathcal{M}_{Y|x}) \leq c_0 \sigma_{\underline{t}} \sqrt{\log n}$, and any partition $\{\mathcal{A}_{(x,w)}, \mathcal{M}_{Y|x} \backslash \mathcal{A}_{(x,w)}\}$ of $\mathcal{M}_{Y|x}$ satisfying $\{y \in \mathcal{M}_{Y|x} : \|y - w\| \leq c_2 \sigma_{\underline{t}} \sqrt{\log n}\} \subset \mathcal{A}_{w,x}$, it holds that

$$\left\| \nabla \log p_{t|x}(w) - \frac{1}{\sigma_t} \cdot \frac{\mathbb{E}_{y\sim\mu_{Y|x}^*}\left[\exp\left(-\frac{\|w - m_t y\|^2}{2\sigma_t^2}\right) \cdot -\frac{(w - m_t y)}{\sigma_t} \mathbf{1}(y \in \mathcal{A}_{(x,w)})\right]}{\mathbb{E}_{y\sim\mu_{Y|x}^*}\left[\exp\left(-\frac{\|w - m_t y\|^2}{2\sigma_t^2}\right) \mathbf{1}(y \in \mathcal{A}_{(x,w)})\right]} \right\| \leq \frac{1}{n}. \quad (2)$$

We will approximate $\nabla \log p_{t|x}(w)$ by constructing suitable sets $\mathcal{A}_{(x,w)}$ and considering the approximation of $\mathbb{E}_{y\sim\mu_{Y|x}^*}\left[\exp\left(-\frac{\|w - m_t y\|^2}{2\sigma_t^2}\right) \cdot -\frac{(w - m_t y)}{\sigma_t} \mathbf{1}(y \in \mathcal{A}_{(x,w)})\right]$ and $\mathbb{E}_{y\sim\mu_{Y|x}^*}\left[\exp\left(-\frac{\|w - m_t y\|^2}{2\sigma_t^2}\right) \mathbf{1}(y \in \mathcal{A}_{(x,w)})\right]$ separately. To achieve this, we will utilize the following lemmas concerning the approximation theorem of neural networks for some standard functions.

**Lemma C.7.** *(Lemma 3.3 in [2]) There exist neural networks $\phi_m(t), \phi_\sigma(t) \in \Phi(H, W, B, R)$ that approximates $m_t$ and $\sigma_t$ up to $\varepsilon$ for all $t \geq 0$, where $H = \mathcal{O}\left(\log^2\left(\varepsilon^{-1}\right)\right), \|W\|_\infty = \mathcal{O}\left(\log^3\left(\varepsilon^{-1}\right)\right), R = \mathcal{O}\left(\log^4\left(\varepsilon^{-1}\right)\right)$, and $B = \exp\left(\mathcal{O}\left(\log^2\left(\varepsilon^{-1}\right)\right)\right)$.*

**Lemma C.8.** *(Lemma F.12 in [2]) Take $\varepsilon > 0$ arbitrarily. There exists a neural network $\phi_{\exp} \in \Phi(H, W, R, B)$ such that*

$$\sup_{x,x'\geq 0} \left| e^{-x'} - \phi_{\exp}(x) \right| \leq \varepsilon + |x - x'|$$

*holds, where $H = \mathcal{O}\left(\log^2 \varepsilon^{-1}\right), \|W\|_\infty = \mathcal{O}\left(\log \varepsilon^{-1}\right), R = \mathcal{O}\left(\log^2 \varepsilon^{-1}\right), B = \exp\left(\mathcal{O}\left(\log^2 \varepsilon^{-1}\right)\right)$. Moreover, $|\phi_{\exp}(x)| \leq \varepsilon$ for all $x \geq \log 3\varepsilon^{-1}$.*

**Lemma C.9.** *(Lemma F.6 in [2]) Let $d \geq 2, C \geq 1, 0 < \varepsilon_{error} \leq 1$. For any $\varepsilon > 0$, there exists a neural network $\phi_{mult}(x_1, x_2, \cdots, x_d) \in \Phi(H, W, R, B)$ with $H = \mathcal{O}\left(\log d\left(\log \varepsilon^{-1} + d\log C\right)\right), \|W\|_\infty = 48d, R = \mathcal{O}\left(d\log \varepsilon^{-1} + d\log C\right), B = C^d$ such that*

$$\left| \phi_{mult}\left(x_1', x_2', \cdots, x_d'\right) - \prod_{d'=1}^{d} x_{d'} \right| \leq \varepsilon + dC^{d-1}\varepsilon_{error}, \text{ for all } x \in [-C, C]^d \text{ and } x' \in \mathbb{R} \text{ with } \|x - x'\|_\infty \leq \varepsilon_{error},$$

*and $|\phi_{mult}(x)| \leq C^d$ for all $x \in [-C, C]$. Note that some of $x_i, x_j (i \neq j)$ can be shared. For $\prod_{i=1}^{I} x_i^{\omega_i}$ with $\omega_i \in \mathbb{Z}_+ (i = 1, 2, \cdots, I)$ and $\sum_{i=1}^{I} \omega_i = d$, there exists a neural network satisfying the same bounds as above, and the network is denoted by $\phi_{mult}(x; \omega)$.*

**Lemma C.10.** *(Lemma F.7 in [2]) For any $0 < \varepsilon < 1$, there exists $\phi_{rec} \in \Phi(H, W, R, B)$ with $H \leq \mathcal{O}\left(\log^2 \varepsilon^{-1}\right), \|W\|_\infty = \mathcal{O}\left(\log^3 \varepsilon^{-1}\right), R = \mathcal{O}\left(\log^4 \varepsilon^{-1}\right)$, and $B = \mathcal{O}\left(\varepsilon^{-2}\right)$ such that*

$$\left| \phi_{rec}\left(x'\right) - \frac{1}{x} \right| \leq \varepsilon + \frac{|x' - x|}{\varepsilon^2}, \quad \text{for all } x \in \left[\varepsilon, \varepsilon^{-1}\right] \text{ and } x' \in \mathbb{R}.$$

We are then ready to construct neural networks for approximating $\nabla \log p_{t|x}(w)$ for $t \in [\underline{t}, \overline{t}]$, where $\underline{t}$ takes on different values.

### C.1.1 Case 1: $c(\log n)^{-1} \leq \underline{t} \leq T$ with $c$ being a samll enough positive constant

Set $\varepsilon_2 = n^{-\frac{1}{2\alpha_X + d_X}}$ and let $\mathcal{N}_{\varepsilon_2}^X$ be one of the largest $\varepsilon_2$-packing of $\mathcal{M}_X$, we have $J_2 := |\mathcal{N}_{\varepsilon_2}^X| \lesssim \varepsilon_2^{-d_X}$. Then take an arbitrary $\widetilde{x} \in \mathcal{N}_{\varepsilon_2}^X$, we claim that

**Claim 1.** *For any* $0 \leq \delta < \frac{\frac{\alpha_X}{\alpha_Y} \wedge \frac{1}{2}}{2\alpha_X + d_X}$, *let* $\varepsilon_1 = c_1 \, n^{-\delta}(\log n)^{-1}$, $\mathscr{L}_1 = \frac{\alpha_X \log \varepsilon_2}{\log(\varepsilon_1^2) - \log(\sigma_{\underline{t}}^2)}$, $\mathscr{L}_2 = \frac{\alpha_X \log \varepsilon_2}{\log(\varepsilon_1 \sqrt{\log n}) - \log(\sigma_{\underline{t}})}$ *and* $\mathscr{L} = \mathscr{L}_1 \mathscr{L}_2 \cdot (2\mathscr{L}_1 + 2\mathscr{L}_2 + 1)\binom{\mathscr{L}_2 + D_Y}{D_Y}\binom{(2\mathscr{L}_1 + 2\mathscr{L}_2 + 1)\lfloor \beta_X \rfloor + \lceil 2\beta_X \rceil + \lfloor \alpha_X \rfloor + D_X}{D_X}$, *then when* $c_1$ *is small enough, there exists a network* $\phi^*(w, x, t) \in \Phi(H, W, R, B, V)$ *with* $H = \Theta(\log^4 n)$, $\|W\|_\infty = \Theta(\log^6 n \mathscr{L} \varepsilon_1^{-d_Y})$, $R = \Theta(\log^8 n \mathscr{L} \varepsilon_1^{-d_Y})$, $B = \exp(\Theta(\log^4 n))$ *and* $V = \Theta(\sqrt{\frac{\log n}{\underline{t} \wedge 1}})$, *so that for any* $t \in [\underline{t}, \overline{t}]$ *and* $(x, w) \in \{x \in \mathcal{M}_X, w \in \mathbb{R}^{D_Y} : \|x - \widetilde{x}\| \leq \varepsilon_2, \text{dist}(w, \mathcal{M}_{Y|x}) \leq c_0 \sigma_{\underline{t}} \sqrt{\log n}\}$,

$$\|\phi^*(w, x, t) - \nabla \log p_{t|x}(w)\|_\infty = \mathcal{O}(\varepsilon_2^{\alpha_X} \log n).$$

Then by $J_2 = \mathcal{O}(\varepsilon_2^{-d_X})$ and Lemma C.6, by setting $\delta = 0$ in Claim 1, we can get that there exists a neural network $\phi_{\text{score}}(w, x, t) \in \Phi(H, W, R, B, V)$ with $H = \Theta(\log^4 n)$, $\|W\|_\infty = \widetilde{\Theta}(n^{\frac{d_X}{2\alpha_X + d_X}})$, $R = \widetilde{\Theta}(n^{\frac{d_X}{2\alpha_X + d_X}})$, $B = \exp(\Theta(\log^4 n))$ and $V = \Theta(\sqrt{\frac{\log n}{\underline{t} \wedge 1}})$ so that for any $t \in [\underline{t}, \overline{t}]$ and $(x, w) \in \{x \in \mathcal{M}_X, w \in \mathbb{R}^{D_Y} : \text{dist}(w, \mathcal{M}_{Y|x}) \leq c_0 \sigma_{\underline{t}} \sqrt{\log n}\}$,

$$\|\phi_{\text{score}}(w, x, t) - \nabla \log p_{t|x}(w)\|_\infty = \widetilde{\mathcal{O}}(\varepsilon_2^{\alpha_X}).$$

So by Lemma C.4, we have

$$\mathbb{E}_{\mu_X^*}\left[ \int_{\underline{t}}^{\overline{t}} \int_{\mathbb{R}^{D_Y}} \left\| \phi_{\text{score}}(w, x, t) - \nabla \log p_{t|x}(w) \right\|^2 p_{t|x}(w) \, \mathrm{d}w \mathrm{d}t \right] = \widetilde{\mathcal{O}}(n^{-\frac{2\alpha_X}{2\alpha_X + d_X}}).$$

Now we show Claim 1. Recall $\mathcal{K}_{\widetilde{x}} = \{k \in [K] : \|\widetilde{x} - x_k^*\| \leq 2r\}$. Then consider $\varepsilon_1 = c_1 n^{-\delta}(\log n)^{-1}$ and let $\mathcal{N}_{\varepsilon_1}^Z = \{\widetilde{z}_1, \widetilde{z}_2, \cdots, \widetilde{z}_{J_1}\}$ be one of the largest $\varepsilon_1$-packing of $\mathbb{B}_{\mathbb{R}^{d_Y}}(0, \overline{r})$, where $\overline{r}$ is the positive constant required in Lemma C.5. We have $|J_1| \lesssim \varepsilon_1^{-d_Y}$. Then we define

$$\mathcal{N} = \{(\widetilde{x}, y) : y = G_{[k]}^*(z, \widetilde{x}), k \in \mathcal{K}_{\widetilde{x}}, z \in \mathcal{N}_{\varepsilon_1}^Z\}.$$

For any $(x, y) \in \mathcal{M}$ so that $\|x - \widetilde{x}\| \leq \varepsilon_2 < r$, since $\{(x_k^*, y_k^*)\}_{k=1}^K$ is an $r$-cover of $\mathcal{M}$, there exists $k \in \mathcal{K}_{\widetilde{x}}$ so that $\|x - x_k^*\| \leq r$ and $\|y - y_k^*\| \leq r$. So by lemma C.5, it holds that $Q_{[k]}^*(y, x) \in \mathbb{B}_{\mathbb{R}^{d_Y}}(0, \overline{r})$ and there exists $z \in \mathcal{N}_{\varepsilon_1}^Z$ so that $\|z - Q_{[k]}^*(y, x)\| \leq \varepsilon_1$. Thus,

$$
\begin{aligned}
\|y - G_{[k]}^*(z, \widetilde{x})\| &= \|G_{[k]}^*(Q_{[k]}^*(y, x), x) - G_{[k]}^*(z, \widetilde{x})\| \\
&\leq \|G_{[k]}^*(Q_{[k]}^*(y, x), x) - G_{[k]}^*(z, x)\| + \|G_{[k]}^*(z, x) - G_{[k]}^*(z, \widetilde{x})\| \\
&\leq C\varepsilon_1.
\end{aligned}
\tag{3}
$$

Then let $J := |\mathcal{N}|$ and write $\mathcal{N} = (\omega_1 = (\widetilde{x}, y_1), \omega_2 = (\widetilde{x}, y_2), \cdots, \omega_J = (\widetilde{x}, y_J))$, we have for any $x \in \mathbb{B}_{\mathcal{M}_X}(\widetilde{x}, \varepsilon_2)$ and $y \in \mathcal{M}_{Y|x}$, there exists $\omega_j = (\widetilde{x}, y_j)$ so that $\|y - y_j\| \leq C\varepsilon_1$. So we consider a partition of unity $\widetilde{\rho}_j(\cdot) = \rho_j(\cdot) / \sum_{j=1}^J \rho_j(\cdot)$ with

$$\rho_j(y) = \rho\left(\frac{\|y - y_j\|^2}{C^2 \varepsilon_1^2}\right)$$

and $\rho$ is the following smooth transition function

$$
\rho(t) = \begin{cases}
0 & |t| \geq 2 \\
1 & |t| \leq 1 \\
\frac{1}{1 + \exp(\frac{2x - 3}{x^2 - 3x + 2})} & 1 < t < 2 \\
\frac{1}{1 + \exp(\frac{-2x - 3}{x^2 + 3x + 2})} & -2 < t < -1.
\end{cases}
\tag{4}
$$

Then for any $x \in \mathbb{B}_{\mathcal{M}_X}(\widetilde{x}, \varepsilon_2)$ and $w \in \mathbb{R}^{D_Y}$ so that $\mathrm{dist}(w, \mathcal{M}_{Y|x}) \leq c_0 \sigma_t \sqrt{\log n}$, by Lemma B.2, we have

$$
\mathbb{E}_{y \sim \mu^*_{Y|x}} \left[ \exp \left( -\frac{\|w - m_t y\|^2}{2\sigma_t^2} \right) \cdot -\frac{(w - m_t y)}{\sigma_t} \right]
$$

$$
= \sum_{j=1}^{J} \mathbb{E}_{y \sim \mu^*_{Y|x}} \left[ \widetilde{\rho}_j(y) \exp \left( -\frac{\|w - m_t y\|^2}{2\sigma_t^2} \right) \cdot -\frac{(w - m_t y)}{\sigma_t} \right]
$$

$$
= \sum_{j=1}^{J} \mathbb{E}_{y \sim \mu^*_{Y|x}} \left[ \widetilde{\rho}_j(y) \mathbf{1}(y \in \mathbb{B}_{\mathcal{M}_{Y|x}}(y_j, \sqrt{2} C \varepsilon_1)) \exp \left( -\frac{\|w - m_t y\|^2}{2\sigma_t^2} \right) \cdot -\frac{(w - m_t y)}{\sigma_t} \right]
$$

$$
= \sum_{j \in [J]} \int_{\mathbb{B}_{\mathbb{R}^{d_Y}}(0, r_1)} \exp \left( -\frac{\|w - m_t G^{\omega_j}(z, x)\|^2}{2\sigma_t^2} \right) \cdot -\frac{(w - m_t G^{\omega_j}(z, x))}{\sigma_t} \widetilde{\rho}_j(G^{\omega_j}(z, x)) v^{\omega_j}(z|x) \, \mathrm{d}z.
$$

Then notice that for any $(x, y) \in \mathcal{M}$ so that $\|x - \widetilde{x}\| \leq \varepsilon_2$ and $\widetilde{\rho}_j(y) \neq 0$,

$$
\begin{aligned}
&\|G^{\omega_j}(0, x) - y\| \\
&\leq \|G^{\omega_j}(0, x) - y_j\| + \|y_j - y\| \\
&= \|G^{\omega_j}(0, x) - G^{\omega_j}(0, \widetilde{x})\| + \|y_j - y\| \\
&\lesssim \varepsilon_1 + \varepsilon_2^{\beta_X \wedge 1},
\end{aligned}
$$

and thus

$$
\|Q_x^{\omega_j}(y)\| = \|Q_x^{\omega_j}(y) - Q_x^{\omega_j}(G^{\omega_j}(0, x))\| \lesssim \varepsilon_1 + \varepsilon_2^{\beta_X \wedge 1} \lesssim \varepsilon_1.
$$

So we have

$$
\mathbb{E}_{y \sim \mu^*_{Y|x}} \left[ \exp \left( -\frac{\|w - m_t y\|^2}{2\sigma_t^2} \right) \cdot -\frac{(w - m_t y)}{\sigma_t} \right]
$$

$$
= \sum_{j \in [J]} \int_{\mathbb{B}_{\mathbb{R}^{d_Y}}(0, C_1 \varepsilon_1)} \exp \left( -\frac{\|w - m_t G^{\omega_j}(z, x)\|^2}{2\sigma_t^2} \right) \cdot -\frac{(w - m_t G^{\omega_j}(z, x))}{\sigma_t} \widetilde{\rho}_j(G^{\omega_j}(z, x)) v^{\omega_j}(z|x) \, \mathrm{d}z.
$$

Notice that $G^{\omega_j}(z, x)$ is $C_{D_Y}^{\beta_Y, \beta_X}$-smooth, let

$$
G^{\omega_j}|_{\widetilde{x}}(z, x) = \sum_{\substack{j_1 \in \mathbb{N}_0^{D_X} \\ |j_1| < \beta_X}} \frac{G^{\omega_j}_{(0_{d_Y}, j_1)}(z, \widetilde{x})}{j_1!} (x - \widetilde{x})^{j_1},
$$

we have

$$
\|G^{\omega_j}(z, x) - G^{\omega_j}|_{\widetilde{x}}(z, x)\| \lesssim \|x - \widetilde{x}\|^{\beta_X} \lesssim \varepsilon_2^{\beta_X}.
$$

So based on the decomposition

$$
\|w - m_t G^{\omega_j}(z, x)\|^2 = \|w - m_t G^{\omega_j}|_{\widetilde{x}}(0, x)\|^2 + 2 \langle w - m_t G^{\omega_j}|_{\widetilde{x}}(0, x), m_t G^{\omega_j}|_{\widetilde{x}}(0, x) - m_t G^{\omega_j}(z, x) \rangle
$$
$$
+ \|m_t G^{\omega_j}|_{\widetilde{x}}(0, x) - m_t G^{\omega_j}(z, x)\|^2,
$$

we can obtain

$$\sum_{j\in[J]}\int_{\mathbb{B}_{\mathbb{R}^{d_Y}}(0,C_1\varepsilon_1)}\exp\left(-\frac{\|w-m_tG^{\omega_j}(z,x)\|^2}{2\sigma_t^2}\right)\cdot-\frac{(w-m_tG^{\omega_j}(z,x))}{\sigma_t}\widetilde{\rho}_j(G^{\omega_j}(z,x))v^{\omega_j}(z|x)\,\mathrm{d}z$$

$$=\sum_{j\in[J]}\int_{\mathbb{B}_{\mathbb{R}^{d_Y}}(0,C_1\varepsilon_1)}\exp\left(-\frac{\|m_tG^{\omega_j}|_{\widetilde{x}}(0,x)-m_tG^{\omega_j}(z,x)\|^2}{2\sigma_t^2}\right)$$

$$\cdot\exp\left(-\frac{\langle w-m_tG^{\omega_j}|_{\widetilde{x}}(0,x),m_tG^{\omega_j}|_{\widetilde{x}}(0,x)-m_tG^{\omega_j}(z,x)\rangle}{\sigma_t^2}\right)$$

$$\cdot-\frac{(w-m_tG^{\omega_j}(z,x))}{\sigma_t}\widetilde{\rho}_j(G^{\omega_j}(z,x))v^{\omega_j}(z|x)\,\mathrm{d}z\cdot\exp\left(-\frac{\|w-m_tG^{\omega_j}|_{\widetilde{x}}(0,x)\|^2}{2\sigma_t^2}\right).$$

Similarly, we have

$$\mathbb{E}_{y\sim\mu^*_{Y|x}}\left[\exp\left(-\frac{\|w-m_ty\|^2}{2\sigma_t^2}\right)\right]$$

$$=\sum_{j\in[J]}\int_{\mathbb{B}_{\mathbb{R}^{d_Y}}(0,C_1\varepsilon_1)}\exp\left(-\frac{\|w-m_tG^{\omega_j}(z,x)\|^2}{2\sigma_t^2}\right)\widetilde{\rho}_j(G^{\omega_j}(z,x))v^{\omega_j}(z|x)\,\mathrm{d}z$$

$$=\sum_{j\in[J]}\int_{\mathbb{B}_{\mathbb{R}^{d_Y}}(0,C_1\varepsilon_1)}\exp\left(-\frac{\|m_tG^{\omega_j}|_{\widetilde{x}}(0,x)-m_tG^{\omega_j}(z,x)\|^2}{2\sigma_t^2}\right)$$

$$\cdot\exp\left(-\frac{\langle w-m_tG^{\omega_j}|_{\widetilde{x}}(0,x),m_tG^{\omega_j}|_{\widetilde{x}}(0,x)-m_tG^{\omega_j}(z,x)\rangle}{\sigma_t^2}\right)$$

$$\cdot\widetilde{\rho}_j(G^{\omega_j}(z,x))v^{\omega_j}(z|x)\,\mathrm{d}z\cdot\exp\left(-\frac{\|w-m_tG^{\omega_j}|_{\widetilde{x}}(0,x)\|^2}{2\sigma_t^2}\right).$$

Notice that for any $j\in[J]$, $x\in\mathbb{B}_{\mathcal{M}_X}(\widetilde{x},\varepsilon_2)$, $w\in\mathbb{R}^{D_Y}$ with $\mathrm{dist}(w,\mathcal{M}_{Y|x})\le c_0\sigma_{\underline{t}}\sqrt{\log n}$ and $t\in[\underline{t},\overline{t}]$, we have

$$\frac{\|w-m_tG^{\omega_j}|_{\widetilde{x}}(0,x)\|^2}{2\sigma_t^2}\lesssim\log n,$$

and for any $\|z\|\lesssim\varepsilon_1$,

$$\frac{\|m_tG^{\omega_j}|_{\widetilde{x}}(0,x)-m_tG^{\omega_j}(z,x)\|^2}{2\sigma_t^2}\lesssim\frac{\varepsilon_1^2+\varepsilon_2^{2\beta_X}}{\sigma_t^2}\lesssim\frac{\varepsilon_1^2}{\sigma_t^2}=\mathcal{O}(1),$$

and

$$\left|\frac{\langle w-m_tG^{\omega_j}|_{\widetilde{x}}(0,x),m_tG^{\omega_j}|_{\widetilde{x}}(0,x)-m_tG^{\omega_j}(z,x)\rangle}{\sigma_t^2}\right|=\mathcal{O}(\frac{\varepsilon_1}{\sigma_{\underline{t}}}\sqrt{\log n})=\mathcal{O}(1). \tag{5}$$

Moreover, there exists a small enough constant $c$ so that when $\|z\|\le c\varepsilon_1$,

$$\|G^{\omega_j}(z,x)-y_j\|=\|G^{\omega_j}(z,x)-G^{\omega_j}(0,\widetilde{x})\|\le C\varepsilon_1,$$

which implies that $\widetilde{\rho}_j(G^{\omega_j}(z,x))$ is bounded away from zero. We can then obtain

$$\int_{\mathbb{B}_{\mathbb{R}^{d_Y}}(0,C_1\varepsilon_1)}\exp\left(-\frac{\|m_tG^{\omega_j}|_{\widetilde{x}}(0,x)-m_tG^{\omega_j}(z,x)\|^2}{2\sigma_t^2}\right)$$

$$\cdot\exp\left(-\frac{\langle w-m_tG^{\omega_j}|_{\widetilde{x}}(0,x),m_tG^{\omega_j}|_{\widetilde{x}}(0,x)-m_tG^{\omega_j}(z,x)\rangle}{\sigma_t^2}\right)\widetilde{\rho}_j(G^{\omega_j}(z,x))v^{\omega_j}(z|x)\,\mathrm{d}z\asymp\varepsilon_1^{d_Y},$$

$$\left\|\frac{w-m_tG^{\omega_j}|_{\widetilde{x}}(z,x)}{\sigma_t}\right\|\lesssim\sqrt{\log n},$$

and

$$\exp\left(-\frac{\|w - m_t G^{\omega_j}|_{\widetilde{x}}(0, x)\|^2}{2\sigma_t^2}\right) \gtrsim n^{-C_1},$$

for some positive constant $C_1$. Therefore, if for any $j \in [J]$, there exist neural networks $\phi_j^{[1]}(w, x, t)$, $\phi_j^{[2]}(w, x, t)$ and $\phi_j^{[3]}(w, x, t)$ and number $\epsilon = o(1)$ so that for any $x \in \mathbb{B}_{\mathcal{M}_X}(\widetilde{x}, \varepsilon_2)$, $w \in \mathbb{R}^{D_Y}$ with $\mathrm{dist}(w, \mathcal{M}_{Y|x}) \leq c_0 \sigma_{\underline{t}} \sqrt{\log n}$ and $t \in [\underline{t}, \overline{t}]$,

$$\left\| \int_{\mathbb{B}_{\mathbb{R}^{d_Y}}(0, C_1 \varepsilon_1)} \exp\left(-\frac{\|m_t G^{\omega_j}|_{\widetilde{x}}(0, x) - m_t G^{\omega_j}(z, x)\|^2}{2\sigma_t^2}\right) \right.$$
$$\cdot \exp\left(-\frac{\langle w - m_t G^{\omega_j}|_{\widetilde{x}}(0, x), m_t G^{\omega_j}|_{\widetilde{x}}(0, x) - m_t G^{\omega_j}(z, x)\rangle}{\sigma_t^2}\right) \tag{6}$$
$$\left. \cdot -\frac{(w - m_t G^{\omega_j}(z, x))}{\sigma_t} \widetilde{\rho}_j(G^{\omega_j}(z, x)) v^{\omega_j}(z|x) \, \mathrm{d}z - \phi_j^{[1]}(w, x, t) \right\|_\infty \lesssim \varepsilon_1^{-d_Y} \epsilon \sqrt{\log n},$$

$$\left| \int_{\mathbb{B}_{\mathbb{R}^{d_Y}}(0, C_1 \varepsilon_1)} \exp\left(-\frac{\|m_t G^{\omega_j}|_{\widetilde{x}}(0, x) - m_t G^{\omega_j}(z, x)\|^2}{2\sigma_t^2}\right) \right.$$
$$\cdot \exp\left(-\frac{\langle w - m_t G^{\omega_j}|_{\widetilde{x}}(0, x), m_t G^{\omega_j}|_{\widetilde{x}}(0, x) - m_t G^{\omega_j}(z, x)\rangle}{\sigma_t^2}\right) \widetilde{\rho}_j(G^{\omega_j}(z, x)) v^{\omega_j}(z|x) \, \mathrm{d}z \tag{7}$$
$$\left. - \phi_j^{[2]}(w, x, t) \right| \lesssim \varepsilon_1^{-d_Y} \epsilon,$$

and

$$\left| \exp\left(-\frac{\|w - m_t G^{\omega_j}|_{\widetilde{x}}(0, x)\|^2}{2\sigma_t^2}\right) - \phi_j^{[3]}(w, x, t) \right| \lesssim n^{-C_1} \epsilon. \tag{8}$$

Then it holds that

$$\left\| \nabla \log p_{t|x}(w) - \frac{1}{\sigma_t} \cdot \frac{\sum_{j=1}^J \phi_j^{[1]}(w, x, t) \phi_j^{[3]}(w, x, t)}{\sum_{j=1}^J \phi_j^{[2]}(w, x, t) \phi_j^{[3]}(w, x, t)} \right\|_\infty \lesssim \log n \cdot \epsilon. \tag{9}$$

Now we build such neural networks. Notice that $v^{\omega_j}(z|x)$ is $C^{\alpha_Y, \alpha_X}$-smooth, let

$$v^{\omega_j}|_{\widetilde{x}}(z|x) = \sum_{\substack{j_1 \in \mathbb{N}_0^{D_X} \\ |j_1| < \alpha_X}} \frac{v_{(0, j_1)}^{\omega_j}(z|\widetilde{x})}{j_1!} (x - \widetilde{x})^{j_1},$$

we have

$$\|v^{\omega_j}(z|x) - v^{\omega_j}|_{\widetilde{x}}(z|x)\| \lesssim \|x - \widetilde{x}\|^{\alpha_X} \lesssim \varepsilon_2^{\alpha_X}.$$

Furthermore, for any $x \in \mathbb{B}_{\mathcal{M}_X}(\widetilde{x}, \varepsilon_2)$, $z \in \mathbb{B}_{\mathbb{R}^{d_Y}}(0, C_1 \varepsilon_1)$ and $w \in \mathbb{R}^{D_Y}$ with $\mathrm{dist}(w, \mathcal{M}_{Y|x}) \leq c_0 \sigma_{\underline{t}} \sqrt{\log n}$, it holds that

$$\left| \frac{\|m_t G^{\omega_j}|_{\widetilde{x}}(0, x) - m_t G^{\omega_j}(z, x)\|^2}{2\sigma_t^2} - \frac{\|m_t G^{\omega_j}|_{\widetilde{x}}(0, x) - m_t G^{\omega_j}|_{\widetilde{x}}(z, x)\|^2}{2\sigma_t^2} \right| \lesssim \log n \cdot \varepsilon_1 \cdot \varepsilon_2^{\beta_X};$$

$$\left| \frac{\langle w - m_t G^{\omega_j}|_{\widetilde{x}}(0, x), m_t G^{\omega_j}|_{\widetilde{x}}(0, x) - m_t G^{\omega_j}(z, x)\rangle}{\sigma_t^2} \right.$$
$$\left. - \frac{\langle w - m_t G^{\omega_j}|_{\widetilde{x}}(0, x), m_t G^{\omega_j}|_{\widetilde{x}}(0, x) - m_t G^{\omega_j}|_{\widetilde{x}}(z, x)\rangle}{\sigma_t^2} \right| \lesssim \log n \cdot \varepsilon_2^{\beta_X}$$

$$\left\| \frac{(w - m_t G^{\omega_j}(z, x))}{\sigma_t} - \frac{(w - m_t G^{\omega_j}|_{\widetilde{x}}(z, x))}{\sigma_t} \right\| \lesssim \sqrt{\log n} \cdot \varepsilon_2^{\beta_X}.$$

Moreover, by equation (3), for any $z \in \mathbb{B}_{\mathbb{R}^{d_Y}}(0, C_1\varepsilon_1)$, $x \in \mathbb{B}_{\mathcal{M}_X}(\widetilde{x}, \varepsilon_2)$ and $j \in [J]$,

$$\sum_{k=1}^{J} \rho_k(G^{\omega_j}(z, x)) \geq 1.$$

So we have

$$\left|\widetilde{\rho}_j(G^{\omega_j}(z, x)) - \widetilde{\rho}_j(G^{\omega_j}|_{\widetilde{x}}(z, x))\right|$$
$$\leq \left|\frac{\rho_j(G^{\omega_j}(z, x))}{\sum_{k=1}^{J} \rho_k(G^{\omega_j}(z, x))} - \frac{\rho_j(G^{\omega_j}|_{\widetilde{x}}(z, x))}{\sum_{k=1}^{J} \rho_k(G^{\omega_j}(z, x))}\right| + \left|\frac{\rho_j(G^{\omega_j}|_{\widetilde{x}}(z, x))}{\sum_{k=1}^{J} \rho_k(G^{\omega_j}(z, x))} - \frac{\rho_j(G^{\omega_j}|_{\widetilde{x}}(z, x))}{\sum_{k=1}^{J} \rho_k(G^{\omega_j}|_{\widetilde{x}}(z, x))}\right|$$
$$\lesssim \frac{\varepsilon_2^{\beta_X}}{\varepsilon_1}.$$

Then for any $z \in \mathbb{B}_{\mathbb{R}^{d_Y}}(0, C_1\varepsilon_1)$, denote $\widetilde{\rho}_j^z(x) = \widetilde{\rho}_j(G^{\omega_j}|_{\widetilde{x}}(z, x))$ as a function of $x$, $\widetilde{\rho}_j^z(\cdot)$ is $C^{\infty}$ around $\widetilde{x}$ with $|\widetilde{\rho}_j^{z(j_1)}(\widetilde{x})| \lesssim \varepsilon_1^{-|j_1|}$. So set $\delta_1 = \lceil \frac{\beta_X \log \varepsilon_2}{\log \varepsilon_2 - \log \varepsilon_1} \rceil \leq \lceil 2\beta_X \rceil$, we have

$$\widetilde{\rho}_j(G^{\omega_j}|_{\widetilde{x}}(z, x)) = \widetilde{\rho}_j^z(x) = \sum_{\substack{j_1 \in \mathbb{N}_0^{D_X} \\ |j_1| < \delta_1}} \frac{\widetilde{\rho}_j^{z(j_1)}(\widetilde{x})}{j_1!}(x - \widetilde{x})^{j_1} + O(\varepsilon_2^{\beta_X}).$$

Then denote

$$\widetilde{\rho}_j^z|_{\widetilde{x}}(x) = \sum_{\substack{j_1 \in \mathbb{N}_0^{D_X} \\ |j_1| < \delta_1}} \frac{\widetilde{\rho}_j^{z(j_1)}(\widetilde{x})}{j_1!}(x - \widetilde{x})^{j_1}.$$

By combining all pieces, we can obtain

$$\left\|\int_{\mathbb{B}_{\mathbb{R}^{d_Y}}(0, C_1\varepsilon_1)} \exp\left(-\frac{\|m_t G^{\omega_j}|_{\widetilde{x}}(0, x) - m_t G^{\omega_j}(z, x)\|^2}{2\sigma_t^2}\right)\right.$$
$$\cdot \exp\left(-\frac{\langle w - m_t G^{\omega_j}|_{\widetilde{x}}(0, x), m_t G^{\omega_j}|_{\widetilde{x}}(0, x) - m_t G^{\omega_j}(z, x)\rangle}{\sigma_t^2}\right)$$
$$\cdot -\frac{(w - m_t G^{\omega_j}(z, x))}{\sigma_t}\widetilde{\rho}_j(G^{\omega_j}(z, x))v^{\omega_j}(z|x)\, \mathrm{d}z$$
$$- \int_{\mathbb{B}_{\mathbb{R}^{d_Y}}(0, C_1\varepsilon_1)} \exp\left(-\frac{\|m_t G^{\omega_j}|_{\widetilde{x}}(0, x) - m_t G^{\omega_j}|_{\widetilde{x}}(z, x)\|^2}{2\sigma_t^2}\right)$$
$$\cdot \exp\left(-\frac{\langle w - m_t G^{\omega_j}|_{\widetilde{x}}(0, x), m_t G^{\omega_j}|_{\widetilde{x}}(0, x) - m_t G^{\omega_j}|_{\widetilde{x}}(z, x)\rangle}{\sigma_t^2}\right)$$
$$\left.\cdot -\frac{(w - m_t G^{\omega_j}|_{\widetilde{x}}(z, x))}{\sigma_t}\widetilde{\rho}_j^z|_{\widetilde{x}}(x)v^{\omega_j}|_{\widetilde{x}}(z|x)\, \mathrm{d}z\right\|$$
$$\lesssim \sqrt{\log n} \cdot \varepsilon_1^{-d_Y-1}\varepsilon_2^{\beta_X} + \sqrt{\log n} \cdot \varepsilon_1^{-d_Y}\varepsilon_2^{\alpha_X}$$
$$\lesssim \sqrt{\log n} \cdot \varepsilon_1^{-d_Y}\varepsilon_2^{\alpha_X},$$

where the last inquality uses $\varepsilon_1 \geq \varepsilon_2^{\frac{\alpha_X}{\alpha_Y} \wedge \frac{1}{2}} \geq \varepsilon_2^{\frac{\alpha_X}{\alpha_Y}} = \varepsilon_2^{\beta_X - \alpha_X}$. Since for any $-1 < z < 1$, we have $|\exp(z) - \sum_{l=0}^{\mathscr{L}} \frac{z^l}{l!}| \leq e \frac{|z|^{\mathscr{L}+1}}{(\mathscr{L}+1)!} \leq e\left(\frac{|z|e}{\mathscr{L}+1}\right)^{\mathscr{L}+1}$. Set $\mathscr{L}_1 = \frac{\alpha_X \log \varepsilon_2}{\log(\varepsilon_1^2) - \log(\sigma_t^2)}$ and $\mathscr{L}_2 = \frac{\alpha_X \log \varepsilon_2}{\log(\varepsilon_1\sqrt{\log n}) - \log(\sigma_{\underline{t}})}$, using inequality (5), we have

$$\left|\exp\left(-\frac{\langle w - m_t G^{\omega_j}|_{\widetilde{x}}(0, x), m_t G^{\omega_j}|_{\widetilde{x}}(0, x) - m_t G^{\omega_j}|_{\widetilde{x}}(z, x)\rangle}{\sigma_t^2}\right)\right.$$
$$\left. - \sum_{l=0}^{\mathscr{L}_2}(-1)^l \frac{\langle w - m_t G^{\omega_j}|_{\widetilde{x}}(0, x), m_t G^{\omega_j}|_{\widetilde{x}}(0, x) - m_t G^{\omega_j}|_{\widetilde{x}}(z, x)\rangle^l}{l!(\sigma_t)^{2l}}\right| \lesssim \varepsilon_2^{\alpha_X}.$$

$$\left| \exp\left( -\frac{\|m_t G^{\omega_j}|_{\widetilde{x}}(0,x) - m_t G^{\omega_j}|_{\widetilde{x}}(z,x)\|^2}{2\sigma_t^2} \right) \right.$$

$$\left. - \sum_{l=0}^{\mathscr{L}_1} (-1)^l \frac{\|m_t G^{\omega_j}|_{\widetilde{x}}(0,x) - m_t G^{\omega_j}|_{\widetilde{x}}(z,x)\|^{2l}}{2^l l! (\sigma_t)^{2l}} \right| \lesssim \varepsilon_2^{\alpha_X}.$$

Therefore,

$$\left\| \int_{\mathbb{B}_{\mathbb{R}^{d_Y}}(0,C_1\varepsilon_1)} \exp\left( -\frac{\|m_t G^{\omega_j}|_{\widetilde{x}}(0,x) - m_t G^{\omega_j}|_{\widetilde{x}}(z,x)\|^2}{2\sigma_t^2} \right) \right.$$

$$\cdot \exp\left( -\frac{\langle w - m_t G^{\omega_j}|_{\widetilde{x}}(0,x), m_t G^{\omega_j}|_{\widetilde{x}}(0,x) - m_t G^{\omega_j}|_{\widetilde{x}}(z,x) \rangle}{\sigma_t^2} \right)$$

$$\cdot -\frac{(w - m_t G^{\omega_j}|_{\widetilde{x}}(z,x))}{\sigma_t} \widetilde{\rho}_j^z|_{\widetilde{x}}(x) v^{\omega_j}|_{\widetilde{x}}(z|x) \, \mathrm{d}z$$

$$- \int_{\mathbb{B}_{\mathbb{R}^{d_Y}}(0,C_1\varepsilon_1)} \sum_{l=0}^{\mathscr{L}_1} (-1)^l \frac{\|m_t G^{\omega_j}|_{\widetilde{x}}(0,x) - m_t G^{\omega_j}|_{\widetilde{x}}(z,x)\|^{2l}}{l!(\sigma_t)^{2l}}$$

$$\cdot \sum_{l=0}^{\mathscr{L}_2} (-1)^l \frac{\langle w - m_t G^{\omega_j}|_{\widetilde{x}}(0,x), m_t G^{\omega_j}|_{\widetilde{x}}(0,x) - m_t G^{\omega_j}|_{\widetilde{x}}(z,x) \rangle^l}{l!(\sigma_t)^{2l}}$$

$$\left. \cdot -\frac{(w - m_t G^{\omega_j}|_{\widetilde{x}}(z,x))}{\sigma_t} \widetilde{\rho}_j^z|_{\widetilde{x}}(x) v^{\omega_j}|_{\widetilde{x}}(z|x) \, \mathrm{d}z \right\|_\infty$$

$$\lesssim \varepsilon_2^{\alpha_X} \varepsilon_1^{-d} \sqrt{\log n}.$$

Notice that we can write

$$\int_{\mathbb{B}_{\mathbb{R}^{d_Y}}(0,C_1\varepsilon_1)} \sum_{l=0}^{\mathscr{L}_1} (-1)^l \frac{\|m_t G^{\omega_j}|_{\widetilde{x}}(0,x) - m_t G^{\omega_j}|_{\widetilde{x}}(z,x)\|^{2l}}{2^l l!(\sigma_t)^{2l}}$$

$$\cdot \sum_{l=0}^{\mathscr{L}_2} (-1)^l \frac{\langle w - m_t G^{\omega_j}|_{\widetilde{x}}(0,x), m_t G^{\omega_j}|_{\widetilde{x}}(0,x) - m_t G^{\omega_j}|_{\widetilde{x}}(z,x) \rangle^l}{l!(\sigma_t)^{2l}}$$

$$\cdot -\frac{(w - m_t G^{\omega_j}|_{\widetilde{x}}(z,x))}{\sigma_t} \widetilde{\rho}_j^z|_{\widetilde{x}}(x) v^{\omega_j}|_{\widetilde{x}}(z|x) \, \mathrm{d}z$$

$$= \int_{\mathbb{B}_{\mathbb{R}^{d_Y}}(0,C\varepsilon_1)} \sum_{l=0}^{\mathscr{L}_1} \frac{(-1)^l m_t^{2l}}{2^l l!(\sigma_t)^{2l}} \Big\| \sum_{j_1 \in \mathbb{N}_0^{D_X}, |j_1| < \beta_X} \big( G_{(0,j_1)}^{\omega_j}(0,\widetilde{x}) - G_{(0,j_1)}^{\omega_j}(z,\widetilde{x}) \big) \frac{(x - \widetilde{x})^{j_1}}{j_1!} \Big\|^{2l}$$

$$\cdot \sum_{l=0}^{\mathscr{L}_2} \frac{(-1)^l m_t^l}{l!(\sigma_t)^{2l}} \Big\langle w - m_t \sum_{j_1 \in \mathbb{N}_0^{D_X}, |j_1| < \beta_X} G_{(0,j_1)}^{\omega_j}(0,\widetilde{x}) \frac{(x - \widetilde{x})^{j_1}}{j_1!}, \sum_{j_1 \in \mathbb{N}_0^{D_X}, |j_1| < \beta_X} \big( G_{(0,j_1)}^{\omega_j}(0,\widetilde{x}) - G_{(0,j_1)}^{\omega_j}(z,\widetilde{x}) \big) \frac{(x - \widetilde{x})^{j_1}}{j_1!} \Big\rangle^l$$

$$\cdot -\frac{w - m_t \sum_{j_1 \in \mathbb{N}_0^{D_X}, |j_1| < \beta_X} G_{(0,j_1)}^{\omega_j}(z,\widetilde{x}) \frac{(x-\widetilde{x})^{j_1}}{j_1!}}{\sigma_t} \cdot \sum_{j_1 \in \mathbb{N}_0^{D_X}, |j_1| < \delta_1} \frac{\widetilde{\rho}_j^{(j_1)}(\widetilde{x})}{j_1!}(x - \widetilde{x})^{j_1} \cdot \sum_{j_1 \in \mathbb{N}_0^{D_X}, |j_1| < \alpha_X} \frac{v_{(0,j_1)}^{\omega_j}(z|\widetilde{x})}{j_1!}(x - \widetilde{x})^{j_1} \, \mathrm{d}z$$

$$= \sum_{l_1=0}^{\mathscr{L}_1} \sum_{l_2=0}^{\mathscr{L}_2} \left( \frac{1}{\sigma_t} \right)^{2l_1 + 2l_2 + 1} \sum_{0 \le k \le 2l_1 + 2l_2 + 1} m_t^k \sum_{i \in \mathbb{N}_0^{D_Y}, |i| \le l_2+1} w^{(i)} \sum_{s \in \mathbb{N}_0^{D_X}, |s| \le (2l_1 + 2l_2 + 1)\lfloor \beta_X \rfloor + \lfloor \delta_1 \rfloor + \lfloor \alpha_X \rfloor} a_{l_1 l_2 kil} \cdot x^{(s)},$$

where $a_{l_1 l_2 kil} \in \mathbb{R}^{D_Y}$ and $\left( \frac{1}{\sigma_t} \right)^{2l_1 + 2l_2 + 1} a_{l_1 l_2 kil} \lesssim \exp(\mathcal{O}(\log^2 n))$. Therefore, using Lemmas C.7, C.8, C.9 and C.10, we

1. Approximate $m_t$ by $\phi_m(t) \in \Phi(H, W, R, B)$ with $H = \Theta(\log^4 n)$, $\|W\|_\infty = \Theta(\log^6 n)$, $R = \Theta(\log^8 n)$ and $B = \exp(\Theta(\log^4 n))$.

2. Approximate $\sigma_t$ by $\phi_\sigma(t) \in \Phi(H, W, R, B)$ with $H = \Theta(\log^4 n)$, $\|W\|_\infty = \Theta(\log^6 n)$, $R = \Theta(\log^8 n)$ and $B = \exp(\Theta(\log^4 n))$.

3. Approximate $\frac{1}{x}$ by $\phi_{rec}(x) \in \Phi(H, W, R, B)$ with $H = \Theta(\log^4 n)$, $\|W\|_\infty = \Theta(\log^6 n)$, $R = \Theta(\log^8 n)$ and $B = \exp(\Theta(\log^4 n))$.

4. For vector $x \in \mathbb{R}^{D_X}$, approximate $x^{(i)}$ by $\phi_{vpower}^{[D_X]}(x; i) \in \Phi(H, W, R, B)$ with $H = \Theta(\log^2 n \cdot \log \log n)$, $\|W\|_\infty = \Theta(\log n)$, $R = \Theta(\log^3 n)$ and $B = \exp(\Theta(\log n))$.

5. For vector $w \in \mathbb{R}^{D_Y}$, approximate $w^{(i)}$ by $\phi_{vpower}^{[D_Y]}(w; i) \in \Phi(H, W, R, B)$ with $H = \Theta(\log^2 n \cdot \log \log n)$, $\|W\|_\infty = \Theta(\log n)$, $R = \Theta(\log^3 n)$ and $B = \exp(\Theta(\log n \cdot \log \log n))$.

6. For $x \in \mathbb{R}$, Approximate $x^a$ by $\phi_{power}(x; a) \in \Phi(H, W, R, B)$ with $H = \Theta(\log^2 n \log \log n)$, $\|W\|_\infty = \Theta(\log n)$, $R = \Theta(\log^3 n)$ and $B = \exp(\Theta(\log n \cdot \log \log n))$.

7. For $x, y \in \mathbb{R}$, Approximate $x \cdot y$ by $\phi_{mult}(x, y) \in \Phi(H, W, R, B)$ with $H = \Theta(\log^2 n)$, $\|W\|_\infty = \Theta(1)$, $R = \Theta(\log^2 n)$ and $B = \exp(\Theta(\log^2 n))$.

We have for any $x \in \mathbb{B}_{\mathcal{M}_X}(\widetilde{x}, \varepsilon_2)$, $w \in \mathbb{R}^{D_Y}$ with $\text{dist}(w, \mathcal{M}_{Y|x}) \le c_0 \sigma_{\underline{t}} \sqrt{\log n}$ and $t \in [\underline{t}, \overline{t}]$,

$$
\left\| \sum_{l_1=0}^{\mathcal{L}_1} \sum_{l_2=0}^{\mathcal{L}_2} (\frac{1}{\sigma_t})^{2l_1+2l_2+1} \sum_{0 \le k \le 2l_1+2l_2+1} m_t^k \sum_{i \in \mathbb{N}_0^{D_Y}, |i| \le l_2+1} w^{(i)} \sum_{s \in \mathbb{N}_0^{D_X}, |s| \le (2l_1+2l_2+1)\lfloor \beta_X \rfloor + \lfloor \delta_1 \rfloor + \lfloor \alpha_X \rfloor} a_{l_1 l_2 k i l} \cdot x^{(s)} \right.
$$

$$
- \sum_{l_1=0}^{\mathcal{L}_1} \sum_{l_2=0}^{\mathcal{L}_2} \sum_{0 \le k \le 2l_1+2l_2+1} \sum_{i \in \mathbb{N}_0^{D_Y}, |i| \le l_2+1} \sum_{s \in \mathbb{N}_0^{D_X}, |s| \le (2l_1+2l_2+1)\lfloor \beta_X \rfloor + \lfloor \delta_1 \rfloor + \lfloor \alpha_X \rfloor} a_{l_1 l_2 k i l}
$$

$$
\left. \cdot \phi_{mult}\left( \phi_{mult}\left( \phi_{mult}\left( \phi_{power}\left(\phi_{rec}(\phi_\sigma(t)); 2l_1 + 2l_2 + 1\right), \phi_{power}\left(\phi_m(t); k\right)\right), \phi_{vpower}^{[D_Y]}(\omega; i) \right), \phi_{vpower}^{[D_X]}(x; s) \right) \right\|_\infty
$$

$$
\lesssim \varepsilon_2^{\alpha_X} \varepsilon_1^{-d} \sqrt{\log n}.
$$

Therefore, based on Lemmas F.1-F.3 in [2] for the concatenation and parallelization of neural networks, let $\mathcal{L} = \mathcal{L}_1 \mathcal{L}_2 \cdot (2\mathcal{L}_1 + 2\mathcal{L}_2 + 1)\binom{\mathcal{L}_2+D_Y}{D_Y}\binom{(2\mathcal{L}_1+2\mathcal{L}_2+1)\lfloor\beta_X\rfloor+\delta_1+\lfloor\alpha_X\rfloor+D_X}{D_X}$, there exists networks $\phi_j^{[1]}(w, x, t) \in \Phi(H, W, R, B)$ with $H = \Theta(\log^4 n)$, $\|W\|_\infty = \Theta(\log^6 n \mathcal{L})$, $R = \Theta(\log^8 n \mathcal{L})$, $B = \exp(\Theta(\log^4 n))$ so that (6) holds with $\epsilon = \mathcal{O}(\varepsilon_2^{\alpha_X})$. Similarly, there exists a neural network $\phi_j^{[2]}(w, x, t)$ with the same size as $\phi_j^{[1]}(w, x, t)$ so that (7) holds with $\epsilon = \mathcal{O}(\varepsilon_2^{\alpha_X})$. For the term $\exp\left(-\frac{\|w - m_t G^{\omega_j}|_{\widetilde{x}}(0,x)\|^2}{2\sigma_t^2}\right)$, using Lemmas C.7-C.10, there exists $\phi_j^{[3]}(w, x, t) \in \Phi(H, W, R, B)$ with $H = \Theta(\log^4 n)$, $\|W\|_\infty = \Theta(\log^6 n)$, $R = \Theta(\log^8 n)$, $B = \exp(\Theta(\log^4 n))$ so that (8) holds with $\epsilon = \mathcal{O}(\varepsilon_2^{\alpha_X})$. Then using (9) and Lemmas C.4, C.9, C.10, we can obtain

$$
\left\| \max\left\{ \frac{-c_2\sqrt{\log n}}{\sigma_{\underline{t}}}, \min\left\{ \frac{c_2\sqrt{\log n}}{\sigma_{\underline{t}}}, \phi_{mult}\left( \phi_{rec}\left(\phi_\sigma(t)\right), \phi_{mult}\left(\sum_{j=1}^J \phi_j^{[1]}(w, x, t)\phi_j^{[3]}(w, x, t), \right. \right. \right. \right. \right.
$$

$$
\left. \left. \left. \left. \phi_{rec}\left(\sum_{j=1}^J \phi_j^{[2]}(w, x, t)\phi_j^{[3]}(w, x, t)\right)\right)\right)\right\}\right\} - \nabla \log p_{t|x}(w) \right\|_\infty = \mathcal{O}(\log n \cdot \varepsilon_2^{\alpha_X}).
$$

The desired result then follows from Lemmas F.1-F.3 in [2] for the concatenation and parallelization of neural networks.

**Remark C.1.** *Here, choosing $\varepsilon_1 = (\log n)^{-1}$ will result in a factor of $(\log n)^{D_Y}$ due to the orders of $\mathcal{L}_1$ and $\mathcal{L}_2$ being $\Theta(\log n)$. This can lead to problems when $D_Y$ is significantly large. To address this issue, we can instead choose $\varepsilon_1 = n^{-\delta}(\log n)^{-1}$, where $\delta > 0$ is a small number. With this choice, $\mathcal{L}_1$ and $\mathcal{L}_2$ will be of constant order, and the influence on the neural network size will be $\mathcal{O}(n^{\delta d_Y})$.*

### C.1.2 Case 2: $c' n^{-2\delta}(\log n)^{-3} \le \underline{t} \le c \log^{-1} n$ with $0 \le \delta < \frac{\frac{\alpha_X}{\alpha_Y} \wedge \frac{1}{2}}{2\alpha_X + d_X}$ and $c'$ is a small enough positive constant

Set $\varepsilon_1 = \sigma_{\underline{t}} \sqrt{\log n}$ and $\varepsilon_2 = n^{-\frac{1}{2\alpha_X + d_X}}$. Let $\mathcal{N}_{\varepsilon_2}^X$ be one of the largest $\varepsilon_2$-packing of $\mathcal{M}_X$ and $\mathcal{N}_{\varepsilon_1}^Z$ be one of the largest $\varepsilon_1$-packing of $\mathbb{B}_{\mathbb{R}^{d_Y}}(0, \overline{r})$. Then we have $J_1 := |\mathcal{N}_{\varepsilon_1}^Z| \lesssim \varepsilon_1^{-d_Y}$ and $J_2 := |\mathcal{N}_{\varepsilon_2}^X| \lesssim$

$\varepsilon_2^{-d_X}$. Now we take an arbitrary $\widetilde{x} \in \mathcal{N}_{\varepsilon_2}^X$, $k \in \mathcal{K}_{\widetilde{x}} = \{k \in [K] \ : \ \|\widetilde{x} - x_k^*\| \leq 2r\}$ and $\widetilde{z} \in N_{\varepsilon_1}^Z$, consider set

$$\mathscr{S}_{k\widetilde{x}\widetilde{z}} = \Big\{(x,w) \ : \ x \in \mathbb{B}_{\mathcal{M}_X}(\widetilde{x}, \sqrt{2}\varepsilon_2), w \in \mathbb{R}^{D_Y},$$
$$\|w - G_{[k]}^*(\widetilde{z}, x)\| \leq C\sigma_{\underline{t}}\sqrt{\log n}, \operatorname{dist}(w, \mathcal{M}_{Y|x}) \leq c_0\sigma_{\underline{t}}\sqrt{\log n}\Big\},$$

we claim that

**Claim 2.** *For any* $0 \leq \delta < \frac{\frac{\alpha_X}{\alpha_Y} \wedge \frac{1}{2}}{2\alpha_X + d_X}$, *let* $\widetilde{\varepsilon}_1 = c_1 \sigma_{\underline{t}} \frac{n^{-\delta}}{\sqrt{\log n}}$, $\mathscr{L}_1 = \frac{\alpha_X \log \varepsilon_2}{\log(\varepsilon_1^2) - \log(\sigma_{\underline{t}}^2)}$, $\mathscr{L}_2 = \frac{\alpha_X \log \varepsilon_2}{\log(\varepsilon_1\sqrt{\log n}) - \log(\sigma_{\underline{t}})}$ *and* $\mathscr{L} = \mathscr{L}_1 \mathscr{L}_2 \cdot (2\mathscr{L}_1 + 2\mathscr{L}_2 + 1)\binom{\mathscr{L}_2 + D_Y}{D_Y}\binom{(2\mathscr{L}_1 + 2\mathscr{L}_2 + 1)\lfloor \beta_X \rfloor + \lfloor \alpha_X \rfloor + D_X}{D_X}$, *then when* $c_1$ *is small enough, there exist a network* $\phi^*(w, x, t) \in \Phi\big(H, W, R, B, \Theta(\frac{\sqrt{\log n}}{\sigma_{\underline{t}}})\big)$ *with* $H = \Theta(\log^4 n)$, $\|W\|_\infty = \Theta\big(\log^6 n\mathscr{L}(\frac{\varepsilon_1}{\widetilde{\varepsilon}_1})^{d_Y}\big)$, $R = \Theta\big(\log^8 n\mathscr{L}(\frac{\varepsilon_1}{\widetilde{\varepsilon}_1})^{d_Y}\big)$, $B = \exp(\Theta(\log^4 n))$, *so that for any* $(x, w) \in \mathscr{S}_{k\widetilde{x}\widetilde{z}}$, *and* $t \in [\underline{t}, \overline{t}]$,

$$\|\phi^*(w, x, t) - \nabla \log p_{t|x}(w)\|_\infty = \mathcal{O}(\varepsilon_2^{\alpha_X} \log n).$$

Then similar as Case 1, by setting $\delta = 0$, using Lemmas C.4 and C.6, we can get that there exists a neural network $\phi_{\text{score}}(w, x, t) \in \Phi(H, W, R, B, V)$ with $H = \Theta\big(\log^4 n\big)$, $\|W\|_\infty = \widetilde{\Theta}\big(n^{\frac{d_X}{2\alpha_X + d_X}}\big)$, $R = \widetilde{\Theta}\big(n^{\frac{d_X}{2\alpha_X + d_X}}\big)$, $B = \exp\big(\Theta(\log^4 n)\big)$ and $V = \Theta(\sqrt{\frac{\log n}{\underline{t} \wedge 1}})$ so that for any $t \in [\underline{t}, \overline{t}]$ and $(x, w) \in \{x \in \mathcal{M}_X, w \in \mathbb{R}^{D_Y} \ : \ \operatorname{dist}(w, \mathcal{M}_{Y|x}) \leq c_0\sigma_{\underline{t}}\sqrt{\log n}\}$,

$$\mathbb{E}_{\mu_X^*}\Big[\int_{\underline{t}}^{\overline{t}} \int_{\mathbb{R}^{D_Y}} \big\|\phi_{\text{score}}(w, x, t) - \nabla \log p_{t|x}(w)\big\|^2 p_{t|x}(w) \, \mathrm{d}w\mathrm{d}t\Big] = \widetilde{\mathcal{O}}(n^{-\frac{2\alpha_X}{2\alpha_X + d_X}}).$$

By combining the result from Case 1, we can deduce the third statement in Lemma C.2.

Now we show Claim 2. When $t \leq c \log^{-1} n$ for a small enough $c$, we have for any $(x, w) \in \mathscr{S}_{k\widetilde{x}\widetilde{z}}$,

$$\{y \in \mathcal{M}_{Y|x} \ : \ \|y - w\| \leq c_2\sigma_{\underline{t}}\sqrt{\log n}\}$$
$$\subset \{y \in \mathcal{M}_{Y|x} \ : \ \|y - G_{[k]}^*(\widetilde{z}, x)\| \leq c_3\sigma_{\underline{t}}\sqrt{\log n}\}$$
$$\subset \{y = G_{[k]}^*(z, x) \ : \ \|z - \widetilde{z}\| \leq c_4\varepsilon_1\}.$$

Let $\widetilde{\varepsilon}_1 = c_1 \sigma_{\underline{t}} \frac{n^{-\delta}}{\sqrt{\log n}}$ and let $\overline{\mathcal{N}}_{\widetilde{\varepsilon}_1}^{\widetilde{z}} = \{\overline{z}_1, \overline{z}_2, \cdots, \overline{z}_J\}$ be one of the largest $\widetilde{\varepsilon}_1$-packing of $\mathbb{B}_{\mathbb{R}^{d_Y}}(\widetilde{z}, c_4\varepsilon_1)$, we have $|J| = |\overline{\mathcal{N}}_{\widetilde{\varepsilon}_1}^{\widetilde{z}}| \lesssim (\frac{\widetilde{\varepsilon}_1}{\varepsilon_1})^{-d_Y}$. Then we consider $\widetilde{\rho}_j(\cdot) = \rho_j(\cdot)/\sum_{j=1}^J \rho_j(\cdot)$ with

$$\rho_j(y) = \rho(\frac{\|z - \overline{z}_j\|^2}{\widetilde{\varepsilon}_1^2})$$

and $\rho$ is the transition function defined in (4). Then for any $(x, w) \in \mathscr{S}_{k\widetilde{x}\widetilde{z}}$, by Lemma B.2, we have

$$\mathbb{E}_{y \sim \mu_{Y|x}^*}\left[\exp\left(-\frac{\|w - m_t y\|^2}{2\sigma_t^2}\right) \cdot -\frac{(w - m_t y)}{\sigma_t}\mathbf{1}\left(y \in \{y = G_{[k]}^*(z, x) \ : \ \|z - \widetilde{z}\| \leq c_4\varepsilon_1\}\right)\right]$$

$$= \int_{\mathbb{B}_{\mathbb{R}^{d_Y}}(\widetilde{z}, c_4\varepsilon_1)} \exp\left(-\frac{\left\|w - m_t G_{[k]}^*(z, x)\right\|^2}{2\sigma_t^2}\right) \cdot -\frac{\left(w - m_t G_{[k]}^*(z, x)\right)}{\sigma_t}v_{[k]}^*(z|x) \, \mathrm{d}z$$

$$= \sum_{j=1}^J \int_{\mathbb{B}_{\mathbb{R}^{d_Y}}(\widetilde{z}, c_4\varepsilon_1)} \exp\left(-\frac{\left\|w - m_t G_{[k]}^*(z, x)\right\|^2}{2\sigma_t^2}\right) \cdot -\frac{\left(w - m_t G_{[k]}^*(z, x)\right)}{\sigma_t}\widetilde{\rho}_j(z)v_{[k]}^*(z|x) \, \mathrm{d}z$$

$$= \sum_{j=1}^J \int_{\mathbb{B}_{\mathbb{R}^{d_Y}}(\widetilde{z}, c_4\varepsilon_1) \cap \mathbb{B}_{\mathbb{R}^{d_Y}}(\overline{z}_j, \sqrt{2}\widetilde{\varepsilon}_1)} \exp\left(-\frac{\left\|w - m_t G_{[k]}^*(z, x)\right\|^2}{2\sigma_t^2}\right) \cdot -\frac{\left(w - m_t G_{[k]}^*(z, x)\right)}{\sigma_t}\widetilde{\rho}_j(z)v_{[k]}^*(z|x) \, \mathrm{d}z.$$

Notice that $G^*_{[k]}(z, x)$ is $C^{\beta_Y, \beta_X}_{D_Y}$-smooth, let

$$G^*_{[k]}|_{\widetilde{x}}(z, x) = \sum_{\substack{j_1 \in \mathbb{N}_0^{D_X} \\ |j_1| < \beta_X}} \frac{G^*_{[k](0, j_1)}(z, \widetilde{x})}{j_1!} (x - \widetilde{x})^{j_1},$$

we have

$$\|G^*_{[k]}(z, x) - G^*_{[k]}|_{\widetilde{x}}(z, x)\| \lesssim \|x - \widetilde{x}\|^{\beta_X} \lesssim \varepsilon_2^{\beta_X}.$$

So based on the decomposition

$$\|w - m_t G^*_{[k]}(z, x)\|^2 = \|w - m_t G^*_{[k]}|_{\widetilde{x}}(\overline{z}_j, x)\|^2 + 2\langle w - m_t G^*_{[k]}|_{\widetilde{x}}(\overline{z}_j, x), m_t G^*_{[k]}|_{\widetilde{x}}(\overline{z}_j, x) - m_t G^*_{[k]}|_{\widetilde{x}}(z, x)\rangle$$
$$+ \|m_t G^*_{[k]}|_{\widetilde{x}}(\overline{z}_j, x) - m_t G^*_{[k]}|_{\widetilde{x}}(z, x)\|^2,$$

we can obtain

$$\sum_{j=1}^{J} \int_{\mathbb{B}_{\mathbb{R}^{d_Y}}(\widetilde{z}, c_4\varepsilon_1)} \exp\left(-\frac{\left\|w - m_t G^*_{[k]}(z, x)\right\|^2}{2\sigma_t^2}\right) \cdot -\frac{\left(w - m_t G^*_{[k]}(z, x)\right)}{\sigma_t} \widetilde{\rho}_j(z) v^*_{[k]}(z|x) \, \mathrm{d}z$$

$$= \sum_{j=1}^{J} \int_{\mathbb{B}_{\mathbb{R}^{d_Y}}(\widetilde{z}, c_4\varepsilon_1)} \exp\left(-\frac{\|m_t G^*_{[k]}|_{\widetilde{x}}(\overline{z}_j, x) - m_t G^*_{[k]}|_{\widetilde{x}}(z, x)\|^2}{2\sigma_t^2}\right)$$

$$\cdot \exp\left(-\frac{\langle w - m_t G^*_{[k]}|_{\widetilde{x}}(\overline{z}_j, x), m_t G^*_{[k]}|_{\widetilde{x}}(\overline{z}_j, x) - m_t G^*_{[k]}|_{\widetilde{x}}(z, x)\rangle}{\sigma_t^2}\right)$$

$$\cdot -\frac{\left(w - m_t G^*_{[k]}(z, x)\right)}{\sigma_t} \widetilde{\rho}_j(z) v^*_{[k]}(z|x) \, \mathrm{d}z \cdot \exp\left(-\frac{\|w - m_t G^*_{[k]}|_{\widetilde{x}}(\overline{z}_j, x)\|^2}{2\sigma_t^2}\right).$$

Similarly, we have

$$\sum_{j=1}^{J} \int_{\mathbb{B}_{\mathbb{R}^{d_Y}}(\widetilde{z}, c_4\varepsilon_1)} \exp\left(-\frac{\left\|w - m_t G^*_{[k]}(z, x)\right\|^2}{2\sigma_t^2}\right) \widetilde{\rho}_j(z) v^*_{[k]}(z|x) \, \mathrm{d}z$$

$$= \sum_{j=1}^{J} \int_{\mathbb{B}_{\mathbb{R}^{d_Y}}(\widetilde{z}, c_4\varepsilon_1)} \exp\left(-\frac{\|m_t G^*_{[k]}|_{\widetilde{x}}(\overline{z}_j, x) - m_t G^*_{[k]}|_{\widetilde{x}}(z, x)\|^2}{2\sigma_t^2}\right)$$

$$\cdot \exp\left(-\frac{\langle w - m_t G^*_{[k]}|_{\widetilde{x}}(\overline{z}_j, x), m_t G^*_{[k]}|_{\widetilde{x}}(\overline{z}_j, x) - m_t G^*_{[k]}|_{\widetilde{x}}(z, x)\rangle}{\sigma_t^2}\right) \widetilde{\rho}_j(z) v^*_{[k]}(z|x) \, \mathrm{d}z$$

$$\cdot \exp\left(-\frac{\|w - m_t G^*_{[k]}|_{\widetilde{x}}(\overline{z}_j, x)\|^2}{2\sigma_t^2}\right).$$

Notice that for any $(x, w) \in \mathscr{S}_{k\widetilde{x}\widetilde{z}}$, $z_j \in \mathcal{N}_{\widetilde{\varepsilon}_1}^{\widetilde{z}}$, $\|z - z_j\| \leq \widetilde{\varepsilon}_1$ and $t \in [\underline{t}, \overline{t}]$, it holds that

$$\frac{\|w - m_t G^*_{[k]}|_{\widetilde{x}}(\overline{z}_j, x)\|^2}{2\sigma_t^2} \leq C_1 \log n,$$

$$\frac{\|m_t G^*_{[k]}|_{\widetilde{x}}(\overline{z}_j, x) - m_t G^*_{[k]}|_{\widetilde{x}}(z, x)\|^2}{2\sigma_t^2} \lesssim \frac{\widetilde{\varepsilon}_1^2}{\sigma_t^2} = \mathcal{O}(1),$$

and

$$\left|\frac{\langle w - m_t G^*_{[k]}|_{\widetilde{x}}(\overline{z}_j, x), m_t G^*_{[k]}|_{\widetilde{x}}(\overline{z}_j, x) - m_t G^*_{[k]}|_{\widetilde{x}}(z, x)\rangle}{\sigma_t^2}\right| \lesssim \frac{\widetilde{\varepsilon}_1 \sqrt{\log n}}{\sigma_t} = \mathcal{O}(1).$$

Therefore, we have

$$
\int_{\mathbb{B}_{\mathbb{R}^{d_Y}}(\widetilde{z},c_4\varepsilon_1)} \exp\left(-\frac{\|m_t G^*_{[k]}|_{\widetilde{x}}(\overline{z}_j,x) - m_t G^*_{[k]}(z,x)\|^2}{2\sigma_t^2}\right)
$$
$$
\cdot \exp\left(-\frac{\langle w - m_t G^*_{[k]}|_{\widetilde{x}}(\overline{z}_j,x), m_t G^*_{[k]}|_{\widetilde{x}}(\overline{z}_j,x) - m_t G^*_{[k]}(z,x)\rangle}{\sigma_t^2}\right) \widetilde{\rho}_j(z) v^*_{[k]}(z|x)\, \mathrm{d}z \asymp \widetilde{\varepsilon}_1^{d_Y},
$$

$$
\left|\frac{\left(w - m_t G^*_{[k]}(z,x)\right)}{\sigma_t}\right| \lesssim \sqrt{\log n},
$$

and

$$
\exp\left(-\frac{\|w - m_t G^*_{[k]}|_{\widetilde{x}}(\overline{z}_j,x)\|^2}{2\sigma_t^2}\right) \gtrsim n^{-C_1}.
$$

Therefore, if for any $j \in [J]$, there exist neural networks $\phi_j^{[1]}(w,x,t)$, $\phi_j^{[2]}(w,x,t)$ and $\phi_j^{[3]}(w,x,t)$ and number $\epsilon = o(1)$ so that for any $(x,w) \in \mathscr{S}_{k\widetilde{x}\widetilde{z}}$ and $t \in [\underline{t}, \overline{t}]$,

$$
\left\|\int_{\mathbb{B}_{\mathbb{R}^{d_Y}}(\widetilde{z},c_4\varepsilon_1)} \exp\left(-\frac{\|m_t G^*_{[k]}|_{\widetilde{x}}(\overline{z}_j,x) - m_t G^*_{[k]}(z,x)\|^2}{2\sigma_t^2}\right)\right.
$$
$$
\cdot \exp\left(-\frac{\langle w - m_t G^*_{[k]}|_{\widetilde{x}}(\overline{z}_j,x), m_t G^*_{[k]}|_{\widetilde{x}}(\overline{z}_j,x) - m_t G^*_{[k]}(z,x)\rangle}{\sigma_t^2}\right) \tag{10}
$$
$$
\left.\cdot -\frac{\left(w - m_t G^*_{[k]}(z,x)\right)}{\sigma_t} \widetilde{\rho}_j(z) v^*_{[k]}(z|x)\, \mathrm{d}z - \phi_j^{[1]}(w,x,t)\right\|_\infty \lesssim \varepsilon_1^{-d_Y} \epsilon \sqrt{\log n},
$$

$$
\left|\int_{\mathbb{B}_{\mathbb{R}^{d_Y}}(\widetilde{z},c_4\varepsilon_1)} \exp\left(-\frac{\|m_t G^*_{[k]}|_{\widetilde{x}}(\overline{z}_j,x) - m_t G^*_{[k]}(z,x)\|^2}{2\sigma_t^2}\right)\right.
$$
$$
\cdot \exp\left(-\frac{\langle w - m_t G^*_{[k]}|_{\widetilde{x}}(\overline{z}_j,x), m_t G^*_{[k]}|_{\widetilde{x}}(\overline{z}_j,x) - m_t G^*_{[k]}(z,x)\rangle}{\sigma_t^2}\right) \widetilde{\rho}_j(z) v^*_{[k]}(z|x)\, \mathrm{d}z
$$
$$
\left.- \phi_j^{[2]}(w,x,t)\right| \lesssim \varepsilon_1^{-d_Y} \epsilon,
$$

and

$$
\left|\exp\left(-\frac{\|w - m_t G^*_{[k]}|_{\widetilde{x}}(\overline{z}_j,x)\|^2}{2\sigma_t^2}\right) - \phi_j^{[3]}(w,x,t)\right| \lesssim n^{-C_1}\epsilon.
$$

Then by (2), we can obtain

$$
\left\|\nabla \log p_{t|x}(w) - \frac{1}{\sigma_t} \cdot \frac{\sum_{j=1}^J \phi_j^{[1]}(w,x,t)\phi_j^{[3]}(w,x,t)}{\sum_{j=1}^J \phi_j^{[2]}(w,x,t)\phi_j^{[3]}(w,x,t)}\right\|_\infty
$$
$$
\leq \left\|\nabla \log p_{t|x}(w) - \frac{1}{\sigma_t} \cdot \frac{\mathbb{E}_{y\sim\mu^*_{Y|x}}\left[\exp\left(-\frac{\|w-m_t y\|^2}{2\sigma_t^2}\right) \cdot -\frac{(w-m_t y)}{\sigma_t} \mathbf{1}\left(y \in \{y = G^*_{[k]}(z,x) : \|z - \widetilde{z}\| \leq c_4\varepsilon_1\}\right)\right]}{\mathbb{E}_{y\sim\mu^*_{Y|x}}\left[\exp\left(-\frac{\|w-m_t y\|^2}{2\sigma_t^2}\right) \mathbf{1}\left(y \in \{y = G^*_{[k]}(z,x) : \|z - \widetilde{z}\| \leq c_4\varepsilon_1\}\right)\right]}\right\|_\infty
$$
$$
+ \left\|\frac{1}{\sigma_t} \cdot \frac{\mathbb{E}_{y\sim\mu^*_{Y|x}}\left[\exp\left(-\frac{\|w-m_t y\|^2}{2\sigma_t^2}\right) \cdot -\frac{(w-m_t y)}{\sigma_t} \mathbf{1}\left(y \in \{y = G^*_{[k]}(z,x) : \|z - \widetilde{z}\| \leq c_4\varepsilon_1\}\right)\right]}{\mathbb{E}_{y\sim\mu^*_{Y|x}}\left[\exp\left(-\frac{\|w-m_t y\|^2}{2\sigma_t^2}\right) \mathbf{1}\left(y \in \{y = G^*_{[k]}(z,x) : \|z - \widetilde{z}\| \leq c_4\varepsilon_1\}\right)\right]}\right.
$$
$$
\left.- \frac{1}{\sigma_t} \cdot \frac{\sum_{j=1}^J \phi_j^{[1]}(w,x,t)\phi_j^{[3]}(w,x,t)}{\sum_{j=1}^J \phi_j^{[2]}(w,x,t)\phi_j^{[3]}(w,x,t)}\right\|_\infty \lesssim \frac{1}{n} + \log^2 n \cdot \epsilon.
$$

Now we construct such networks. Similar to Case 1, let

$$v^*_{[k]}|_{\widetilde{x}}(z,x) = \sum_{\substack{j\in\mathbb{N}_0^{d_X}\\ |j|<\alpha_X}} \frac{v^*_{[k](0,j)}(z,\widetilde{x})}{j!}(x-\widetilde{x})^j,$$

we have

$$
\begin{aligned}
&\left\| \int_{\mathbb{B}_{\mathbb{R}^{d_Y}}(\widetilde{z},c_4\varepsilon_1)} \exp\left( -\frac{\|m_t G^*_{[k]}|_{\widetilde{x}}(\overline{z}_j,x) - m_t G^*_{[k]}(z,x)\|^2}{2\sigma_t^2} \right) \right. \\
&\qquad\cdot \exp\left( -\frac{\langle w - m_t G^*_{[k]}|_{\widetilde{x}}(\overline{z}_j,x), m_t G^*_{[k]}|_{\widetilde{x}}(\overline{z}_j,x) - m_t G^*_{[k]}(z,x)\rangle}{\sigma_t^2} \right) \\
&\qquad\cdot -\frac{\left( w - m_t G^*_{[k]}(z,x) \right)}{\sigma_t} \widetilde{\rho}_j(z) v^*_{[k]}(z|x)\,\mathrm{d}z \\
&\quad - \int_{\mathbb{B}_{\mathbb{R}^{d_Y}}(\widetilde{z},c_4\varepsilon_1)} \exp\left( -\frac{\|m_t G^*_{[k]}|_{\widetilde{x}}(\overline{z}_j,x) - m_t G^*_{[k]}|_{\widetilde{x}}(z,x)\|^2}{2\sigma_t^2} \right) \\
&\qquad\cdot \exp\left( -\frac{\langle w - m_t G^*_{[k]}|_{\widetilde{x}}(\overline{z}_j,x), m_t G^*_{[k]}|_{\widetilde{x}}(\overline{z}_j,x) - m_t G^*_{[k]}|_{\widetilde{x}}(z,x)\rangle}{\sigma_t^2} \right) \\
&\qquad\cdot \left. -\frac{\left( w - m_t G^*_{[k]}|_{\widetilde{x}}(z,x) \right)}{\sigma_t} \widetilde{\rho}_j(z) v^*_{[k]}|_{\widetilde{x}}(z|x)\,\mathrm{d}z \right\|_\infty \lesssim (\frac{\varepsilon_2^{\beta_X}}{\sigma_{\underline{t}}}\log n + \varepsilon_2^{\alpha_X}\sqrt{\log n})\widetilde{\varepsilon}_1^{-d_Y} \lesssim \varepsilon_2^{\alpha_X}\sqrt{\log n}\,\widetilde{\varepsilon}_1^{-d_Y},
\end{aligned}
$$

and by choosing $\mathscr{L}_1 = \frac{\alpha_X \log\varepsilon_2}{\log(\varepsilon_1^2)-\log(\sigma_{\underline{t}}^2)}$ and $\mathscr{L}_2 = \frac{\alpha_X \log\varepsilon_2}{\log(\varepsilon_1\sqrt{\log n})-\log(\sigma_{\underline{t}})}$,

$$
\begin{aligned}
&\left\| \int_{\mathbb{B}_{\mathbb{R}^{d_Y}}(\widetilde{z},c_4\varepsilon_1)} \exp\left( -\frac{\|m_t G^*_{[k]}|_{\widetilde{x}}(\overline{z}_j,x) - m_t G^*_{[k]}|_{\widetilde{x}}(z,x)\|^2}{2\sigma_t^2} \right) \right. \\
&\qquad\cdot \exp\left( -\frac{\langle w - m_t G^*_{[k]}|_{\widetilde{x}}(\overline{z}_j,x), m_t G^*_{[k]}|_{\widetilde{x}}(\overline{z}_j,x) - m_t G^*_{[k]}|_{\widetilde{x}}(z,x)\rangle}{\sigma_t^2} \right) \\
&\qquad\cdot -\frac{\left( w - m_t G^*_{[k]}|_{\widetilde{x}}(z,x) \right)}{\sigma_t} \widetilde{\rho}_j(z) v^*_{[k]}|_{\widetilde{x}}(z|x)\,\mathrm{d}z \\
&\quad - \int_{\mathbb{B}_{\mathbb{R}^{d_Y}}(\widetilde{z},c_4\varepsilon_1)} \sum_{l=0}^{\mathscr{L}_1}(-1)^l \frac{\|m_t G^*_{[k]}|_{\widetilde{x}}(\overline{z}_j,x) - m_t G^*_{[k]}|_{\widetilde{x}}(z,x)\|^{2l}}{l!(\sigma_t)^{2l}} \\
&\qquad\cdot \sum_{l=0}^{\mathscr{L}_2}(-1)^l \frac{\langle w - m_t G^*_{[k]}|_{\widetilde{x}}(\overline{z}_j,x), m_t G^*_{[k]}|_{\widetilde{x}}(\overline{z}_j,x) - m_t G^*_{[k]}|_{\widetilde{x}}(z,x)\rangle^l}{l!(\sigma_t)^{2l}} \\
&\qquad\cdot \left. -\frac{\left( w - m_t G^{\omega_j}|_{\widetilde{x}}(z,x) \right)}{\sigma_t} \widetilde{\rho}_j(z) v^*_{[k]}|_{\widetilde{x}}(z|x)\,\mathrm{d}z \right\|_\infty \\
&\lesssim \varepsilon_2^{\alpha_X}\sqrt{\log n}\,\widetilde{\varepsilon}_1^{-d_Y}.
\end{aligned}
$$

Furthermore, we can write

$$
\int_{\mathbb{B}_{\mathbb{R}^{d_Y}}(\widetilde{z},c_4\varepsilon_1)} \sum_{l=0}^{\mathscr{L}_1}(-1)^l \frac{\|m_t G^*_{[k]}|_{\widetilde{x}}(\overline{z}_j,x) - m_t G^*_{[k]}|_{\widetilde{x}}(z,x)\|^{2l}}{l!(\sigma_t)^{2l}}
$$

$$
\cdot \sum_{l=0}^{\mathscr{L}_2}(-1)^l \frac{\langle w - m_t G^*_{[k]}|_{\widetilde{x}}(\overline{z}_j,x), m_t G^*_{[k]}|_{\widetilde{x}}(\overline{z}_j,x) - m_t G^*_{[k]}|_{\widetilde{x}}(z,x)\rangle^l}{l!(\sigma_t)^{2l}}
$$

$$
\cdot - \frac{(w - m_t G^{\omega_j}|_{\widetilde{x}}(z,x))}{\sigma_t} \widetilde{\rho}_j(z) v^*_{[k]}|_{\widetilde{x}}(z|x)\, \mathrm{d}z
$$

$$
= \sum_{l_1=0}^{\mathscr{L}_1} \sum_{l_2=0}^{\mathscr{L}_2} (\frac{1}{\sigma_t})^{2l_1+2l_2+1} \sum_{0\le k\le 2l_1+2l_2+1} m_t^k \sum_{i\in\mathbb{N}_0^{D_Y},|i|\le l_2+1} w^{(i)} \sum_{s\in\mathbb{N}_0^{D_X},|s|\le(2l_1+2l_2+1)\lfloor\beta_X\rfloor+\lfloor\alpha_X\rfloor} a_{l_1 l_2 k i l}\cdot x^{(s)}.
$$

So similar to Case 1, let $\mathscr{L} = \mathscr{L}_1\mathscr{L}_2 \cdot (2\mathscr{L}_1 + 2\mathscr{L}_2 + 1)\binom{\mathscr{L}_2+D_Y}{D_Y}\binom{(2\mathscr{L}_1+2\mathscr{L}_2+1)\lfloor\beta_X\rfloor+\lfloor\alpha_X\rfloor+D_X}{D_X}$, there exists a network $\phi_j^{[1]}(w,x,t) \in \Phi(H,W,R,B)$ with $H = \Theta(\log^4 n)$, $\|W\|_\infty = \Theta(\log^6 n\mathscr{L})$, $R = \Theta(\log^8 n\mathscr{L})$, $B = \exp(\Theta(\log^4 n))$ so that (10) holds with $\epsilon = \mathcal{O}(\varepsilon_2^{\alpha_X})$. The construction of $\phi_j^{[2]}(w,x,t)$ follows a similar approach. By utilizing the same analysis as in Case 1, we can establish the desired claim 2.

**Remark C.2.** *Here similar to Case 1, choosing $\delta = 0$ will result in a factor of $(\log n)^{D_Y}$ due to the orders of $\mathscr{L}_1$ and $\mathscr{L}_2$ being $\Theta(\log n)$. Instead, we can choose $\delta$ as a small positive number so that $\mathscr{L}_1$ and $\mathscr{L}_2$ will be of constant order, and the influence on the neural network size will be $\mathcal{O}(n^{\delta d_Y})$.*

### C.1.3  Case 3: $n^{-\frac{2}{2\alpha_Y+d_Y+d_X\frac{\alpha_Y}{\alpha_X}}} \le \underline{t} \le c' n^{-2\delta}\log^{-3} n$

Set $\varepsilon_1 = \sigma_{\underline{t}}\sqrt{\log n}$ and $\varepsilon_2 = n^{-\frac{1}{2\alpha_X+d_X}}(\sigma_{\underline{t}}\sqrt{\log n})^{-\frac{d_Y}{2\alpha_X+d_X}}$. Let $\mathcal{N}_{\varepsilon_2}^X$ be one of the largest $\varepsilon_2$-packing of $\mathcal{M}_X$ and $\mathcal{N}_{\varepsilon_1}^Z$ be one of the largest $\varepsilon_1$-packing of $\mathbb{B}_{\mathbb{R}^{d_Y}}(0,\overline{r})$. Then we have $J_1 := |\mathcal{N}_{\varepsilon_1}^Z| \lesssim \varepsilon_1^{-d_Y}$ and $J_2 := |\mathcal{N}_{\varepsilon_2}^X| \lesssim \varepsilon_2^{-d_X}$. Now we take an arbitrary $\widetilde{x}\in\mathcal{N}_{\varepsilon_2}^X$, $k\in\mathcal{K}_{\widetilde{x}}$ and $\widetilde{z}\in N_{\varepsilon_1}^Z$, consider set

$$
\mathscr{S}_{k\widetilde{x}\widetilde{z}} = \Big\{(x,w) : x\in\mathbb{B}_{\mathcal{M}_X}(\widetilde{x},\sqrt{2}\varepsilon_2),
$$
$$
\|w - G^*_{[k]}(\widetilde{z},x)\| \le C\,\sigma_{\underline{t}}\sqrt{\log n}, \mathrm{dist}(w,\mathcal{M}_{Y|x}) \le c_0\sigma_{\underline{t}}\sqrt{\log n}\Big\},
$$

we claim that

**Claim 3.** *Let $\mathscr{L}_1 = \Theta(\log n)$, $\mathscr{L}_2 = \lceil\frac{-2\log n}{\log\sigma_{\underline{t}}+\frac{3}{2}\log(\log n)}\rceil \vee \lceil\frac{-2\log n}{\alpha_Y\log\sigma_{\underline{t}}}\rceil$ and*

$$
\mathscr{L} = \mathscr{L}_1\mathscr{L}_2\cdot(2\mathscr{L}_1+2\mathscr{L}_2+1)\binom{(2\mathscr{L}_1+2\mathscr{L}_2+1)\lfloor\beta_Y\rfloor+d_Y}{d_Y}\binom{\mathscr{L}_2+D_Y}{D_Y}\binom{(2\mathscr{L}_1+2\mathscr{L}_2+1)\lfloor\beta_X\rfloor+\lfloor\alpha_X\rfloor+D_X}{D_X}
$$

*there exists networks $\phi^*(w,x,t)\in\Phi(H,W,R,B,\Theta(\frac{\sqrt{\log n}}{\sigma_{\underline{t}}}))$ with $H = \Theta(\log^4 n)$, $\|W\|_\infty = \Theta(\log^6 n\mathscr{L})$, $R = \Theta(\log^8 n\mathscr{L})$, $B = \exp(\Theta(\log^4 n))$, so that for any $(x,w)\in\mathscr{S}_{k\widetilde{x}\widetilde{z}}$, and $t\in[\underline{t},\overline{t}]$,*

$$
\|\phi^*(w,x,t) - \nabla\log p_{t|x}(w)\|_\infty \lesssim \log n\frac{\varepsilon_2^{\beta_X}}{\sigma_{\underline{t}}^2} + \sqrt{\log n}\frac{\varepsilon_2^{\alpha_X}}{\sigma_{\underline{t}}}.
$$

Then, similar to Case 1 and 2, by setting $\delta = 0$, the second statement of Lemma C.2 directly follows from Lemmas C.4 and C.6. Now we show Claim 3.

For any $(x,w)\in\mathscr{S}_{k\widetilde{x}\widetilde{z}}$, denote $\mathrm{Proj}_{\mathcal{M}_{Y|x}}(w)$ as the projection of $w$ to $\mathcal{M}_{Y|x}$, we have

$$
\|G^*_{[k]}(\widetilde{z},x) - \mathrm{Proj}_{\mathcal{M}_{Y|x}}(w)\| \le \|G^*_{[k]}(\widetilde{z},x) - w\| + \mathrm{dist}(w,\mathcal{M}_{Y|x}) \le c_1\sigma_{\underline{t}}\sqrt{\log n}.
$$

So

$$\{y \in \mathcal{M}_{Y|x} \ : \ \|y - w\| \leq c_2 \sigma_{\underline{t}} \sqrt{\log n}\} \subset \{y \in \mathcal{M}_{Y|x} \ : \ \|y - \mathrm{Proj}_{\mathcal{M}_{Y|x}}(w)\| \leq c_3 \sigma_{\underline{t}} \sqrt{\log n}\}$$
$$\subset \{y \in \mathcal{M}_{Y|x} \ : \ \|y - G^*_{[k]}(\widetilde{z}, x)\| \leq c_4 \sigma_{\underline{t}} \sqrt{\log n}\}$$
$$\subset \{y = G^*_{[k]}(z, x) \ : \ \|z - \widetilde{z}\| \leq c_5 \sigma_{\underline{t}} \sqrt{\log n}\}.$$

Therefore, by equation (2), we only need to approximate

$$\frac{1}{\sigma_t} \cdot \frac{\int_{\|z-\widetilde{z}\| \leq c_5 \sigma_{\underline{t}} \sqrt{\log n}} \exp\left(-\frac{\|w - m_t G^*_{[k]}(z,x)\|^2}{2\sigma_t^2}\right) \cdot \left(-\frac{w - m_t G^*_{[k]}(z,x)}{\sigma_t}\right) v^*_{[k]}(z|x)\, \mathrm{d}z}{\int_{\|z-\widetilde{z}\| \leq c_5 \sigma_{\underline{t}} \sqrt{\log n}} \exp\left(-\frac{\|w - m_t G^*_{[k]}(z,x)\|^2}{2\sigma_t^2}\right) v^*_{[k]}(z|x)\, \mathrm{d}z}. \tag{11}$$

Now we consider the polynomial approximation of $G^*_{[k]}$ at $(\widetilde{z}, \widetilde{x})$,

$$G^*_{[k]}|_{(\widetilde{z},\widetilde{x})}(z,x) = \sum_{(j_1,j_2) \in \mathcal{J}^{d_Y,D_X}_{\beta_Y,\beta_X}} \frac{G^{*(j_1,j_2)}_{[k]}(\widetilde{z},\widetilde{x})}{j_1! j_2!}(z-\widetilde{z})^{j_1}(x-\widetilde{x})^{j_2}, \tag{12}$$

we have

$$\sup_{\|z-\widetilde{z}\| \leq c_5 \sigma_{\underline{t}} \sqrt{\log n}} \sup_{x \in \mathbb{B}_{\mathcal{M}_X}(\widetilde{x},\sqrt{2}\varepsilon_2)} \|G^*_{[k]}|_{(\widetilde{z},\widetilde{x})}(z,x) - G^*_{[k]}(z,x)\| \lesssim (\sigma_{\underline{t}}\sqrt{\log n})^{\beta_Y} + \varepsilon_2^{\beta_X}.$$

Next we present the following lemma, which provides an approximation to the projection function $\mathrm{Proj}_{\mathcal{M}_{Y|x}}(w)$.

**Lemma C.11.** *If* $\tau \leq \underline{t} \leq \log^{-3} n$, *then for any fixed* $\widetilde{x} \in \mathcal{M}_X$, $k \in \mathcal{K}_{\widetilde{x}}$ *and* $\widetilde{z} \in \mathbb{B}_{\mathbb{R}^{d_Y}}(0, \overline{r})$, *there exists a neural network* $\phi_p(w, x) \in \Phi(H, W, R, B)$ *with* $H = \Theta(\log^2 n)$, $\|W\|_\infty = \Theta(\log^3 n)$, $R = \Theta(\log^4 n)$ *and* $B = \exp(\Theta(\log n))$ *so that for any* $x \in \mathcal{M}_X$ *satisfying* $\|x - \widetilde{x}\| \lesssim (\sigma_{\underline{t}}\sqrt{\log n})^{\frac{1}{\beta_X}}$, *and* $w \in \mathbb{R}^{D_Y}$ *satisfying* $\|w - G^*_{[k]}(\widetilde{z}, x)\| \lesssim (\sigma_{\underline{t}} \vee n^{-\frac{1}{2\alpha_Y + d_Y + d_X \frac{\alpha_Y}{\alpha_X}}})\sqrt{\log n}$ *and* $\mathrm{dist}(w, \mathcal{M}_{Y|x}) \leq c_0 \sigma_{\underline{t}}\sqrt{\log n}$, *it holds that*

1. $\left\|\langle J_{G^*_{[k]}|_{(\widetilde{z},\widetilde{x})}(\cdot,x)}(\phi_p(w,x)), w - G^*_{[k]}|_{(\widetilde{z},\widetilde{x})}(\phi_p(w,x),x)\rangle\right\| \lesssim \left((\sigma_{\underline{t}} \vee n^{-\frac{1}{2\alpha_Y + d_Y + d_X \frac{\alpha_Y}{\alpha_X}}})\sqrt{\log n}\right)^{2\beta_Y}.$

2. $\left\|\phi_p(w,x) - Q^*_{[k]}(\mathrm{Proj}_{\mathcal{M}_{Y|x}}(w), x)\right\| \lesssim \sigma_{\underline{t}}\sqrt{\log n}.$

Lemma C.11 suggests that that $G^*_{[k]}|_{(\widetilde{z},\widetilde{x})}(\phi_p(w,x), x)$ is a good approximation for $\mathrm{Proj}_{\mathcal{M}_{Y|x}}(w)$. Based on this, we consider the following decomposition

$$\|w - m_t G^*_{[k]}(z,x)\|^2 = \|w - G^*_{[k]}|_{(\widetilde{z},\widetilde{x})}(\phi_p(w,x),x)\|^2 + \|G^*_{[k]}|_{(\widetilde{z},\widetilde{x})}(\phi_p(w,x),x) - m_t G^*_{[k]}(z,x)\|^2$$
$$+ 2\langle w - G^*_{[k]}|_{(\widetilde{z},\widetilde{x})}(\phi_p(w,x),x), G^*_{[k]}|_{(\widetilde{z},\widetilde{x})}(\phi_p(w,x),x) - m_t G^*_{[k]}(z,x)\rangle.$$
$$\tag{13}$$

We can then substitute this expression into (11) to obtain

$$\frac{1}{\sigma_t} \cdot \frac{\int_{\|z-\widetilde{z}\|\leq c_5\sigma_{\underline{t}}\sqrt{\log n}} \exp\left(-\frac{\|w-m_tG^*_{[k]}(z,x)\|^2}{2\sigma_t^2}\right)\cdot\left(-\frac{w-m_tG^*_{[k]}(z,x)}{\sigma_t}\right)v^*_{[k]}(z|x)\,\mathrm{d}z}{\int_{\|z-\widetilde{z}\|\leq c_5\sigma_{\underline{t}}\sqrt{\log n}} \exp\left(-\frac{\|w-m_tG^*_{[k]}(z,x)\|^2}{2\sigma_t^2}\right)v^*_{[k]}(z|x)\,\mathrm{d}z}$$

$$= \frac{1}{\sigma_t}\cdot\left[\int_{\|z-\widetilde{z}\|\leq c_5\sigma_{\underline{t}}\sqrt{\log n}}\exp\left(-\frac{\|G^*_{[k]}|_{(\widetilde{z},\widetilde{x})}(\phi_p(w,x),x)-m_tG^*_{[k]}(z,x)\|^2}{2\sigma_t^2}\right)\right.$$
$$\left.\cdot\exp\left(-\frac{\langle w-G^*_{[k]}|_{(\widetilde{z},\widetilde{x})}(\phi_p(w,x),x),G^*_{[k]}|_{(\widetilde{z},\widetilde{x})}(\phi_p(w,x),x)-m_tG^*_{[k]}(z,x)\rangle}{\sigma_t^2}\right)v^*_{[k]}(z|x)\,\mathrm{d}z\right]^{-1}$$

$$\cdot\left[\int_{\|z-\widetilde{z}\|\leq c_5\sigma_{\underline{t}}\sqrt{\log n}}\exp\left(-\frac{\|G^*_{[k]}|_{(\widetilde{z},\widetilde{x})}(\phi_p(w,x),x)-m_tG^*_{[k]}(z,x)\|^2}{2\sigma_t^2}\right)\right.$$
$$\cdot\exp\left(-\frac{\langle w-G^*_{[k]}|_{(\widetilde{z},\widetilde{x})}(\phi_p(w,x),x),G^*_{[k]}|_{(\widetilde{z},\widetilde{x})}(\phi_p(w,x),x)-m_tG^*_{[k]}(z,x)\rangle}{\sigma_t^2}\right)$$
$$\left.\cdot\left(-\frac{G^*_{[k]}|_{(\widetilde{z},\widetilde{x})}(\phi_p(w,x),x)-m_tG^*_{[k]}(z,x)}{\sigma_t}\right)v^*_{[k]}(z|x)\,\mathrm{d}z\right]-\frac{w-G^*_{[k]}|_{(\widetilde{z},\widetilde{x})}(\phi_p(w,x),x)}{\sigma_t^2}$$

For the term $-\frac{w-G^*_{[k]}|_{(\widetilde{z},\widetilde{x})}(\phi_p(w,x),x)}{\sigma_t^2}$, since $G^*_{[k]}|_{(\widetilde{z},\widetilde{x})}$ is a polynomial function, using Lemmas C.7-C.10, we can obtain that there exists a neural network $\phi^{[3]}(w,x,t)\in\Phi(H,W,R,B)$ with $H=\Theta(\log^2 n)$, $\|W\|=\Theta(\log^3 n)$, $R=\Theta(\log^4 n)$ and $B=\exp(\Theta(\log^2 n))$ so that

$$\left\|\frac{w-G^*_{[k]}|_{(\widetilde{z},\widetilde{x})}(\phi_p(w,x),x))}{\sigma_t^2}-\phi^{[3]}(w,x,t)\right\|_\infty\leq\frac{1}{n}. \tag{14}$$

Then for the remaining term, notice that for any $(x,w)\in\mathscr{S}_{k\widetilde{x}\widetilde{z}}$ and $\|z-\widetilde{z}\|\lesssim\sigma_{\underline{t}}\sqrt{\log n}$,

$$\|\phi_p(w,x)-\widetilde{z}\|\leq\|\phi_p(w,x)-Q^*_{[k]}(\mathrm{Proj}_{\mathcal{M}_{Y|x}}(w),x)\|+\|Q^*_{[k]}(\mathrm{Proj}_{\mathcal{M}_{Y|x}}(w),x)-Q^*_{[k]}(G^*_{[k]}(\widetilde{z},x),x)\|\lesssim\sigma_{\underline{t}}\sqrt{\log n},$$

$$\left\|G^*_{[k]}|_{(\widetilde{z},\widetilde{x})}(\phi_p(w,x),x)-m_tG^*_{[k]}(z,x)\right\|$$
$$\leq\|G^*_{[k]}|_{(\widetilde{z},\widetilde{x})}(\phi_p(w,x),x)-G^*_{[k]}|_{(\widetilde{z},\widetilde{x})}(\widetilde{z},x)\|+\|G^*_{[k]}|_{(\widetilde{z},\widetilde{x})}(\widetilde{z},x)-G^*_{[k]}(\widetilde{z},x)\|$$
$$+\|G^*_{[k]}(\widetilde{z},x)-G^*_{[k]}(z,x)\|+\|(1-m_t)G^*_{[k]}(z,x)\| \tag{15}$$
$$\lesssim\|\phi_p(w,x)-\widetilde{z}\|+\varepsilon_2^{\beta_X}+\|z-\widetilde{z}\|+\underline{t}$$
$$\lesssim\sigma_{\underline{t}}\sqrt{\log n},$$
$$\|w-G^*_{[k]}|_{(\widetilde{z},\widetilde{x})}(\phi_p(w,x),x)\|\leq\|w-G^*_{[k]}(\widetilde{z},x)\|+\|G^*_{[k]}(\widetilde{z},x)-G^*_{[k]}|_{(\widetilde{z},\widetilde{x})}(\widetilde{z},x)\|$$
$$+\|G^*_{[k]}|_{(\widetilde{z},\widetilde{x})}(\widetilde{z},x)-G^*_{[k]}|_{(\widetilde{z},\widetilde{x})}(\phi_p(w,x),x)\|$$
$$\lesssim\sigma_{\underline{t}}\sqrt{\log n},$$

and

$$\left|\langle w-G^*_{[k]}|_{(\widetilde{z},\widetilde{x})}(\phi_p(w,x),x),G^*_{[k]}|_{(\widetilde{z},\widetilde{x})}(\phi_p(w,x),x)-m_tG^*_{[k]}(z,x)\rangle\right|$$
$$\leq\left|\langle w-G^*_{[k]}|_{(\widetilde{z},\widetilde{x})}(\phi_p(w,x),x),G^*_{[k]}|_{(\widetilde{z},\widetilde{x})}(\phi_p(w,x),x)-G^*_{[k]}|_{(\widetilde{z},\widetilde{x})}(z,x)\rangle\right|$$
$$+\left|\langle w-G^*_{[k]}|_{(\widetilde{z},\widetilde{x})}(\phi_p(w,x),x),G^*_{[k]}|_{(\widetilde{z},\widetilde{x})}(z,x)-G^*_{[k]}(z,x)\rangle\right|$$
$$+\left|\langle w-G^*_{[k]}|_{(\widetilde{z},\widetilde{x})}(\phi_p(w,x),x),G^*_{[k]}(z,x)-m_tG^*_{[k]}(z,x)\rangle\right| \tag{16}$$
$$\leq\left|\langle w-G^*_{[k]}|_{(\widetilde{z},\widetilde{x})}(\phi_p(w,x),x),J_{G^*_{[k]}|_{(\widetilde{z},\widetilde{x})}(\cdot,x)}(\phi_p(w,x))(\phi_p(w,x)-z))\rangle\right|+\mathcal{O}((\sigma_{\underline{t}}\sqrt{\log n})^3)$$
$$+\mathcal{O}((\sigma_{\underline{t}}\sqrt{\log n})^{\beta_Y+1})+\mathcal{O}(\sigma_{\underline{t}}\sqrt{\log n}\cdot\varepsilon_2^{\beta_X})$$
$$=\mathcal{O}(\sigma_{\underline{t}}^3\log n^{\frac{3}{2}}+\sigma_{\underline{t}}^{\alpha_Y+2}).$$

Therefore, denote

$$\overline{dp}_t(w,x) = \int_{\|z-\widetilde{z}\|\le c_5\sigma_{\underline{t}}\sqrt{\log n}} \exp\left(-\frac{\|G^*_{[k]}|_{(\widetilde{z},\widetilde{x})}(\phi_p(w,x),x) - m_t G^*_{[k]}(z,x)\|^2}{2\sigma_t^2}\right)$$

$$\cdot \exp\left(-\frac{\langle w - G^*_{[k]}|_{(\widetilde{z},\widetilde{x})}(\phi_p(w,x),x), G^*_{[k]}|_{(\widetilde{z},\widetilde{x})}(\phi_p(w,x),x) - m_t G^*_{[k]}(z,x)\rangle}{\sigma_t^2}\right)$$

$$\cdot \left(-\frac{G^*_{[k]}|_{(\widetilde{z},\widetilde{x})}(\phi_p(w,x),x) - m_t G^*_{[k]}(z,x)}{\sigma_t}\right) v^*_{[k]}(z|x)\,\mathrm{d}z,$$

and

$$\overline{p}_t(w,x) = \int_{\|z-\widetilde{z}\|\le c_5\sigma_{\underline{t}}\sqrt{\log n}} \exp\left(-\frac{\|G^*_{[k]}|_{(\widetilde{z},\widetilde{x})}(\phi_p(w,x),x) - m_t G^*_{[k]}(z,x)\|^2}{2\sigma_t^2}\right)$$

$$\cdot \exp\left(-\frac{\langle w - G^*_{[k]}|_{(\widetilde{z},\widetilde{x})}(\phi_p(w,x),x), G^*_{[k]}|_{(\widetilde{z},\widetilde{x})}(\phi_p(w,x),x) - m_t G^*_{[k]}(z,x)\rangle}{\sigma_t^2}\right) v^*_{[k]}(z|x)\,\mathrm{d}z,$$

we can derive

$$\left\|\frac{\overline{dp}_t(w,x)}{\overline{p}_t(w,x)}\right\| \lesssim \sqrt{\log n},$$

and

$$\overline{p}_t(w,x) \ge \int_{\|z-\phi_p(w,x)\|\le\sigma_{\underline{t}}} \exp\left(-\frac{\|G^*_{[k]}|_{(\widetilde{z},\widetilde{x})}(\phi_p(w,x),x) - m_t G^*_{[k]}(z,x)\|^2}{2\sigma_t^2}\right)$$

$$\cdot \exp\left(-\frac{\langle w - G^*_{[k]}|_{(\widetilde{z},\widetilde{x})}(\phi_p(w,x),x), G^*_{[k]}|_{(\widetilde{z},\widetilde{x})}(\phi_p(w,x),x) - m_t G^*_{[k]}(z,x)\rangle}{\sigma_t^2}\right) v^*_{[k]}(z|x)\,\mathrm{d}z$$

$$\gtrsim \int_{\|z-\phi_p(w,x)\|\le\sigma_{\underline{t}}} \exp\left(-\frac{\|G^*_{[k]}|_{(\widetilde{z},\widetilde{x})}(\phi_p(w,x),x) - G^*_{[k]}|_{(\widetilde{z},\widetilde{x})}(z,x)\|^2}{2\sigma_t^2}\right) v^*_{[k]}(z|x)\,\mathrm{d}z \gtrsim (\sigma_{\underline{t}})^{d_Y}.$$

Therefore, if there exist neural networks $\phi^{[1]}(w,x,t)$ and $\phi^{[2]}(w,x,t)$ so that for any $t \in [\underline{t},\overline{t}]$ and $(x,w) \in \mathscr{S}_{k\widetilde{x}\widetilde{z}}$,

$$\|\overline{dp}_t(w,x) - \phi^{[1]}(w,x,t)\|_\infty = \mathcal{O}\left((\sigma_{\underline{t}})^{d_Y}(\log n \cdot \frac{\varepsilon_2^{\beta_X}}{\sigma_{\underline{t}}} + \sqrt{\log n}\cdot\varepsilon_2^{\alpha_X})\right), \tag{17}$$

$$\|\overline{p}_t(w,x) - \phi^{[2]}(w,x,t)\|_\infty = \mathcal{O}\left((\sigma_{\underline{t}})^{d_Y}(\sqrt{\log n}\cdot\frac{\varepsilon_2^{\beta_X}}{\sigma_{\underline{t}}} + \varepsilon_2^{\alpha_X})\right). \tag{18}$$

Then we have

$$\left\|\frac{1}{\sigma_t}\cdot\frac{\overline{dp}_t(x)}{\overline{p}_t(x)} - \frac{1}{\sigma_t}\cdot\frac{\phi^{[1]}(x,t)}{\phi^{[2]}(x,t)}\right\|_\infty = \mathcal{O}\left(\log n\frac{\varepsilon_2^{\beta_X}}{\sigma_{\underline{t}}^2} + \sqrt{\log n}\frac{\varepsilon_2^{\alpha_X}}{\sigma_{\underline{t}}}\right). \tag{19}$$

To construct $\phi^{[1]}(w,x,t)$, we approximate $\overline{dp}_t(w,x)$ by polynomials. Let

$$G^*_{[k]}|_{\widetilde{x}}(z,x) = \sum_{\substack{j\in\mathbb{N}_0^{d_X}\\|j|<\beta_X}} \frac{G^{*(0,j)}_{[k]}(z,\widetilde{x})}{j!}(x-\widetilde{x})^j,$$

$$v^*_{[k]}|_{\widetilde{x}}(z,x) = \sum_{\substack{j \in \mathbb{N}_0^{d_X} \\ |j| < \alpha_X}} \frac{v^*_{[k](0,j)}(z,\widetilde{x})}{j!}(x-\widetilde{x})^j,$$

we have

$$\|G^*_{[k]}(z,x) - G^*_{[k]}|_{\widetilde{x}}(z,x)\| \lesssim \|x-\widetilde{x}\|^{\beta_X} \lesssim \varepsilon_2^{\beta_X}.$$

and

$$\|v^*_{[k]}(z,x) - v^*_{[k]}|_{\widetilde{x}}(z,x)\| \lesssim \|x-\widetilde{x}\|^{\alpha_X} \lesssim \varepsilon_2^{\alpha_X}.$$

Therefore,

$$\left| \|G^*_{[k]}|_{(\widetilde{z},\widetilde{x})}(\phi_p(w,x),x) - m_t G^*_{[k]}|_{\widetilde{x}}(z,x)\|^2 - \|G^*_{[k]}|_{(\widetilde{z},\widetilde{x})}(\phi_p(w,x),x) - m_t G^*_{[k]}(z,x)\|^2 \right|$$
$$\lesssim \sigma_{\underline{t}} \sqrt{\log n} \cdot \varepsilon_2^{\beta_X};$$

$$\left| \langle w - G^*_{[k]}|_{(\widetilde{z},\widetilde{x})}(\phi_p(w,x),x), G^*_{[k]}|_{(\widetilde{z},\widetilde{x})}(\phi_p(w,x),x) - m_t G^*_{[k]}|_{\widetilde{x}}(z,x) \rangle \right.$$
$$\left. - \langle w - G^*_{[k]}|_{(\widetilde{z},\widetilde{x})}(\phi_p(w,x),x), G^*_{[k]}|_{(\widetilde{z},\widetilde{x})}(\phi_p(w,x),x) - m_t G^*_{[k]}(z,x) \rangle \right|$$
$$\lesssim \sigma_{\underline{t}} \sqrt{\log n} \cdot \varepsilon_2^{\beta_X},$$

$$\int_{\|z-\widetilde{z}\| \le c_5 \sigma_{\underline{t}} \sqrt{\log n}} \exp\left( -\frac{\|G^*_{[k]}|_{(\widetilde{z},\widetilde{x})}(\phi_p(w,x),x) - m_t G^*_{[k]}(z,x)\|^2}{2\sigma_t^2} \right) \, \mathrm{d}z$$
$$\lesssim \int_{\|z-\widetilde{z}\| \le c_5 \sigma_{\underline{t}} \sqrt{\log n}} \exp\left( -\frac{\|G^*_{[k]}(\phi_p(w,x),x) - G^*_{[k]}(z,x)\|^2}{2\sigma_t^2} \right) \, \mathrm{d}z$$
$$\lesssim \int \exp\left( \frac{-L\|\phi_p(w,x) - z\|^2}{2\sigma_t^2} \right) \, \mathrm{d}z$$
$$\lesssim (\sigma_{\underline{t}})^{d_Y}.$$

So we can obtain

$$\left| \overline{dp}_t(w,x) - \int_{\|z-\widetilde{z}\| \le c_5 \sigma_{\underline{t}} \sqrt{\log n}} \exp\left( -\frac{\|G^*_{[k]}|_{(\widetilde{z},\widetilde{x})}(\phi_p(w,x),x) - m_t G^*_{[k]}|_{\widetilde{x}}(z,x)\|^2}{2\sigma_t^2} \right) \right.$$
$$\cdot \exp\left( -\frac{\langle w - G^*_{[k]}|_{(\widetilde{z},\widetilde{x})}(\phi_p(w,x),x), G^*_{[k]}|_{(\widetilde{z},\widetilde{x})}(\phi_p(w,x),x) - m_t G^*_{[k]}|_{\widetilde{x}}(z,x) \rangle}{\sigma_t^2} \right)$$
$$\left. \cdot \left( -\frac{G^*_{[k]}|_{(\widetilde{z},\widetilde{x})}(\phi_p(w,x),x) - m_t G^*_{[k]}|_{\widetilde{x}}(z,x)}{\sigma_t} \right) v^*_{[k]}|_{\widetilde{x}}(z|x) \, \mathrm{d}z \right|$$
$$\lesssim (\sigma_{\underline{t}})^{d_Y} \cdot \left( \frac{\varepsilon_2^{\beta_X}}{\sigma_{\underline{t}}} \log n + \sqrt{\log n} \cdot \varepsilon_2^{\alpha_X} \right).$$

Similarly,

$$\left| \overline{p}_t(w,x) - \int_{\|z-\widetilde{z}\| \le c_5 \sigma_{\underline{t}} \sqrt{\log n}} \exp\left( -\frac{\|G^*_{[k]}|_{(\widetilde{z},\widetilde{x})}(\phi_p(w,x),x) - m_t G^*_{[k]}|_{\widetilde{x}}(z,x)\|^2}{2\sigma_t^2} \right) \right.$$
$$\left. \cdot \exp\left( -\frac{\langle w - G^*_{[k]}|_{(\widetilde{z},\widetilde{x})}(\phi_p(w,x),x), G^*_{[k]}|_{(\widetilde{z},\widetilde{x})}(\phi_p(w,x),x) - m_t G^*_{[k]}|_{\widetilde{x}}(z,x) \rangle}{\sigma_t^2} \right) v^*_{[k]}|_{\widetilde{x}}(z|x) \, \mathrm{d}z \right|$$
$$\lesssim (\sigma_{\underline{t}})^{d_Y} \cdot \left( \frac{\varepsilon_2^{\beta_X}}{\sigma_{\underline{t}}} \sqrt{\log n} + \varepsilon_2^{\alpha_X} \right).$$

Using (15) and (16), by choosing $\mathscr{L}_1 = \Theta(\log n)$ and $\mathscr{L}_2 = \lceil \frac{-2\log n}{\log \sigma_{\underline{t}} + \frac{3}{2}\log(\log n)} \rceil \vee \lceil \frac{-2\log n}{\alpha_Y \log \sigma_{\underline{t}}} \rceil$, we have

$$\left| \exp\left( -\frac{\|G^*_{[k]}|_{(\widetilde{z},\widetilde{x})}(\phi_p(w,x),x) - m_t G^*_{[k]}|_{\widetilde{x}}(z,x)\|^2}{2\sigma_t^2} \right) \right.$$
$$\left. - \sum_{l_1=0}^{\mathscr{L}_1} (-1)^{l_1} \frac{\|G^*_{[k]}|_{(\widetilde{z},\widetilde{x})}(\phi_p(w,x),x) - m_t G^*_{[k]}|_{\widetilde{x}}(z,x)\|^{2l_1}}{2^{l_1} l_1! \sigma_t^{2l_1}} \right| \lesssim n^{-2},$$

and

$$\left| \exp\left( -\frac{\langle w - G^*_{[k]}|_{(\widetilde{z},\widetilde{x})}(\phi_p(w,x),x), G^*_{[k]}|_{(\widetilde{z},\widetilde{x})}(\phi_p(w,x),x) - m_t G^*_{[k]}|_{\widetilde{x}}(z,x) \rangle}{\sigma_t^2} \right) \right.$$
$$\left. - \sum_{l_2=0}^{\mathscr{L}_2} (-1)^{l_2} \frac{\langle w - G^*_{[k]}|_{(\widetilde{z},\widetilde{x})}(\phi_p(w,x),x), G^*_{[k]}|_{(\widetilde{z},\widetilde{x})}(\phi_p(w,x),x) - m_t G^*_{[k]}|_{\widetilde{x}}(z,x) \rangle^{l_2}}{l_2! \sigma_t^{2l_2}} \right|$$
$$\lesssim n^{-2}.$$

Therefore, we have

$$\left\| \overline{dp_t}(w,x) - \int_{\|z-\widetilde{z}\| \leq c_5 \sigma_{\underline{t}} \sqrt{\log n}} \sum_{l_1=0}^{\mathscr{L}_1} (-1)^{l_1} \frac{\|G^*_{[k]}|_{(\widetilde{z},\widetilde{x})}(\phi_p(w,x),x) - m_t G^*_{[k]}|_{\widetilde{x}}(z,x)\|^{2l_1}}{2^{l_1} l_1! \sigma_t^{2l_1}} \right.$$
$$\cdot \sum_{l_2=0}^{\mathscr{L}_2} (-1)^{l_2} \frac{\langle w - G^*_{[k]}|_{(\widetilde{z},\widetilde{x})}(\phi_p(w,x),x), G^*_{[k]}|_{(\widetilde{z},\widetilde{x})}(\phi_p(w,x),x) - m_t G^*_{[k]}|_{\widetilde{x}}(z,x) \rangle^{l_2}}{l_2! \sigma_t^{2l_2}} \qquad (20)$$
$$\cdot \left( -\frac{G^*_{[k]}|_{(\widetilde{z},\widetilde{x})}(\phi_p(w,x),x) - m_t G^*_{[k]}|_{\widetilde{x}}(z,x)}{\sigma_t} \right) v^*_{[k]}|_{\widetilde{x}}(z|x) \, \mathrm{d}z \right\|$$
$$\lesssim (\sigma_{\underline{t}})^{d_Y} n^{-1}.$$

Furthermore, notice that $G_{[k]}^*|_{(\widetilde{z},\widetilde{x})}$, $G_{[k]}^*|_{\widetilde{x}}$ and $v_{[k]}^*|_{\widetilde{x}}$ are both polynomial in $x$,

$$
\int_{\|z-\widetilde{z}\|\leq c_5\sigma_t\sqrt{\log n}}\sum_{l_1=0}^{\mathscr{L}_1}(-1)^{l_1}\frac{\|G_{[k]}^*|_{(\widetilde{z},\widetilde{x})}(\phi_p(w,x),x)-m_tG_{[k]}^*|_{\widetilde{x}}(z,x)\|^{2l_1}}{2^{l_1}l_1!\sigma_t^{2l_1}}
$$

$$
\cdot\sum_{l_2=0}^{\mathscr{L}_2}(-1)^{l_2}\frac{\langle w-G_{[k]}^*|_{(\widetilde{z},\widetilde{x})}(\phi_p(w,x),x),G_{[k]}^*|_{(\widetilde{z},\widetilde{x})}(\phi_p(w,x),x)-m_tG_{[k]}^*|_{\widetilde{x}}(z,x)\rangle^{l_2}}{l_2!\sigma_t^{2l_2}}
$$

$$
\cdot\left(-\frac{G_{[k]}^*|_{(\widetilde{z},\widetilde{x})}(\phi_p(w,x),x)-m_tG_{[k]}^*|_{\widetilde{x}}(z,x)}{\sigma_t}\right)v_{[k]}^*|_{\widetilde{x}}(z|x)\,\mathrm{d}z
$$

$$
=\int_{\|z-\widetilde{z}\|\leq c_5\sigma_t\sqrt{\log n}}\sum_{l_1=0}^{\mathscr{L}_1}\sum_{l_2=0}^{\mathscr{L}_2}\frac{(-1)^{(l_1+l_2+1)}}{2^{l_1}l_1!l_2!}(\frac{1}{\sigma_t})^{2l_1+2l_2+1}\Big\|\sum_{(j_1,j_2)\in\mathcal{J}_{\beta_Y,\beta_X}^{d_Y,D_X}}\frac{G_{[k](j_1,j_2)}^*(\widetilde{z},\widetilde{x})}{j_1!j_2!}(\phi_p(w,x)-\widetilde{z})^{j_1}(x-\widetilde{x})^{j_2}
$$

$$
-m_t\sum_{\substack{j\in\mathbb{N}_0^{d_X}\\|j|<\beta_X}}\frac{G_{[k](0,j)}^*(z,\widetilde{x})}{j!}(x-\widetilde{x})^j\Big\|^{2l_1}\cdot\Big\langle w-\sum_{(j_1,j_2)\in\mathcal{J}_{\beta_Y,\beta_X}^{d_Y,D_X}}\frac{G_{[k](j_1,j_2)}^*(\widetilde{z},\widetilde{x})}{j_1!j_2!}(\phi_p(w,x)-\widetilde{z})^{j_1}(x-\widetilde{x})^{j_2},
$$

$$
\sum_{(j_1,j_2)\in\mathcal{J}_{\beta_Y,\beta_X}^{d_Y,D_X}}\frac{G_{[k](j_1,j_2)}^*(\widetilde{z},\widetilde{x})}{j_1!j_2!}(\phi_p(w,x)-\widetilde{z})^{j_1}(x-\widetilde{x})^{j_2}-m_t\sum_{\substack{j\in\mathbb{N}_0^{d_X}\\|j|<\beta_X}}\frac{G_{[k](0,j)}^*(z,\widetilde{x})}{j!}(x-\widetilde{x})^j\Big\rangle^{l_2}
$$

$$
\cdot\Big(\sum_{(j_1,j_2)\in\mathcal{J}_{\beta_Y,\beta_X}^{d_Y,D_X}}\frac{G_{[k](j_1,j_2)}^*(\widetilde{z},\widetilde{x})}{j_1!j_2!}(\phi_p(w,x)-\widetilde{z})^{j_1}(x-\widetilde{x})^{j_2}-m_t\sum_{\substack{j\in\mathbb{N}_0^{d_X}\\|j|<\beta_X}}\frac{G_{[k](0,j)}^*(z,\widetilde{x})}{j!}(x-\widetilde{x})^j\Big)
$$

$$
\cdot\sum_{j_1\in\mathbb{N}_0^{D_X},|j_1|<\alpha_X}\frac{v_{[k](0,j_1)}^*(z|\widetilde{x})}{j_1!}(x-\widetilde{x})^{j_1}\,\mathrm{d}z
$$

$$
=\sum_{l_1=0}^{\mathscr{L}_1}\sum_{l_2=0}^{\mathscr{L}_2}(\frac{1}{\sigma_t})^{2l_1+2l_2+1}\sum_{0\leq k\leq 2l_1+l_2+1}m_t^k\sum_{s\in\mathbb{N}_0^{d_Y},|s|\leq(2l_1+2l_2+1)\lfloor\beta_Y\rfloor}(\phi_p(w,x))^{(s)}\sum_{i\in\mathbb{N}_0^{D_Y},|i|\leq l_2}w^{(i)}
$$

$$
\sum_{j\in\mathbb{N}_0^{D_X},|j|\leq(2l_1+2l_2+1)\lfloor\beta_X\rfloor+\lfloor\alpha_X\rfloor}a_{l_1l_2ksij}x^{(j)},
$$

where $a_{l_1l_2ksij}\in\mathbb{R}^{D_Y}$ are some constant coefficients. Then notice that $(\frac{1}{\sigma_t})^{2l_1+2l_2+1}a_{l_1l_2ksij}\lesssim\exp(\mathcal{O}(\log^2 n))$, we

1. Approximate $m_t$ by $\phi_m(t)\in\Phi(H,W,R,B)$ with $H=\Theta(\log^4 n)$, $\|W\|_\infty=\Theta(\log^6 n)$, $R=\Theta(\log^8 n)$ and $B=\exp(\Theta(\log^4 n))$.

2. Approximate $\sigma_t$ by $\phi_\sigma(t)\in\Phi(H,W,R,B)$ with $H=\Theta(\log^4 n)$, $\|W\|_\infty=\Theta(\log^6 n)$, $R=\Theta(\log^8 n)$ and $B=\exp(\Theta(\log^4 n))$.

3. Approximate $\frac{1}{x}$ by $\phi_{rec}(x)\in\Phi(H,W,R,B)$ with $H=\Theta(\log^4 n)$, $\|W\|_\infty=\Theta(\log^6 n)$, $R=\Theta(\log^8 n)$ and $B=\exp(\Theta(\log^4 n))$.

4. For vector $x\in\mathbb{R}^{D_Y}$, approximate $x^{(i)}$ by $\phi_{vpower}^{[D]}(x;i)\in\Phi(H,W,R,B)$ with $H=\Theta(\log^2 n\cdot\log\log n)$, $\|W\|_\infty=\Theta(\log n)$, $R=\Theta(\log^3 n)$ and $B=\exp(\Theta(\log n\cdot\log\log n))$.

5. For vector $z\in\mathbb{R}^{d_Y}$, approximate $z^{(i)}$ by $\phi_{vpower}^{[d]}(z;i)\in\Phi(H,W,R,B)$ with $H=\Theta(\log^2 n\cdot\log\log n)$, $\|W\|_\infty=\Theta(\log n)$, $R=\Theta(\log^3 n)$ and $B=\exp(\Theta(\log n\cdot\log\log n))$.

6. For $x \in \mathbb{R}$, Approximate $x^a$ by $\phi_{power}(x; a) \in \Phi(H, W, R, B)$ with $H = \Theta(\log^2 n \cdot \log \log n)$, $\|W\|_\infty = \Theta(\log n)$, $R = \Theta(\log^3 n)$ and $B = \exp(\Theta(\log n \cdot \log \log n))$.

7. For $x, y \in \mathbb{R}$, Approximate $x \cdot y$ by $\phi_{mult}(x, y) \in \Phi(H, W, R, B)$ with $H = \Theta(\log^2 n)$, $\|W\|_\infty = \Theta(1)$, $R = \Theta(\log^2 n)$ and $B = \exp(\Theta(\log^2 n))$.

We have

$$
\left\| \sum_{l_1=0}^{\mathscr{L}_1} \sum_{l_2=0}^{\mathscr{L}_2} \left(\frac{1}{\sigma_t}\right)^{2l_1+2l_2+1} \sum_{0 \le k \le 2l_1+l_2+1} m_t^k \sum_{s \in \mathbb{N}_0^{d_Y}, |s| \le (2l_1+2l_2+1)\lfloor \beta_Y \rfloor} (\phi_p(w,x))^{(s)} \sum_{i \in \mathbb{N}_0^{D_Y}, |i| \le l_2} w^{(i)} \right.
$$

$$
\sum_{j \in \mathbb{N}_0^{D_X}, |j| \le (2l_1+2l_2+1)\lfloor \beta_X \rfloor + \lfloor \alpha_X \rfloor} a_{l_1 l_2 ksij} x^{(j)}
$$

$$
- \sum_{l_1=0}^{\mathscr{L}_1} \sum_{l_2=0}^{\mathscr{L}_2} \sum_{0 \le k \le 2l_1+2l_2+1} \sum_{s \in \mathbb{N}_0^{d_Y}, |s| \le (2l_1+2l_2+1)\lfloor \beta_Y \rfloor} \sum_{i \in \mathbb{N}_0^{D_Y}, |i| \le l_2+1} \sum_{j \in \mathbb{N}_0^{D_X}, |j| \le (2l_1+2l_2+1)\lfloor \beta_X \rfloor + \lfloor \alpha_X \rfloor} a_{l_1 l_2 ksij}
$$

$$
\cdot \phi_{mult}\left( \phi_{mult}\left( \phi_{mult}\left( \phi_{mult}\left( \phi_{power}\left(\phi_{rec}(\phi_\sigma(t)); 2l_1 + 2l_2 + 1\right), \phi_{power}\left(\phi_m(t); k\right) \right) \right.\right.\right.
$$

$$
\left.\left.\left. , \phi_{vpower}^{[d_Y]}(\phi_p(w,x); s) \right), \phi_{vpower}^{[D_Y]}(\omega; i) \right), \phi_{vpower}^{[D_X]}(x; j) \right) \right\|_\infty
$$

$$
\lesssim (\sigma_{\underline{t}})^{d_Y} n^{-1}.
$$

Therefore, by concatenation and parallelization of neural networks, let $\mathscr{L} = \mathscr{L}_1 \mathscr{L}_2 \cdot (2\mathscr{L}_1 + 2\mathscr{L}_2 + 1)\binom{(2\mathscr{L}_1+2\mathscr{L}_2+1)\lfloor \beta_Y \rfloor + d_Y}{d_Y}\binom{\mathscr{L}_2+D_Y}{D_Y}\binom{(2\mathscr{L}_1+2\mathscr{L}_2+1)\lfloor \beta_X \rfloor + \lfloor \alpha_X \rfloor + D_X}{D_X}$, there exists networks $\phi_j^{[1]}(w,x,t) \in \Phi(H, W, R, B)$ with $H = \Theta(\log^4 n)$, $\|W\|_\infty = \Theta(\log^6 n\mathscr{L})$, $R = \Theta(\log^8 n\mathscr{L})$, $B = \exp(\Theta(\log^4 n))$ so that (17) holds. Similarly, there exists a neural network $\phi^{[2]}(x,t)$ with the same size as $\phi^{[1]}(x,t)$ so that (18) holds. Then using (14), (19), and Lemmas C.7-C.10, we can obtain

$$
\left\| \max\left\{ \frac{-c_2\sqrt{\log n}}{\sigma_{\underline{t}}}, \min\left\{ \frac{c_2\sqrt{\log n}}{\sigma_{\underline{t}}}, \phi_{mult}\left( \phi_{rec}\left(\phi_\sigma(t)\right), \phi_{mult}\left(\phi^{[1]}(w,x,t), \phi_{rec}\left(\phi^{[2]}(w,x,t)\right)\right) \right) \right. \right. \right.
$$

$$
\left. \left. \left. - \phi^{[3]}(w,x,t) \right\} \right\} - \nabla \log p_{t|x}(w) \right\|_\infty \lesssim \log n \cdot \frac{\varepsilon_2^{\beta_X}}{\sigma_{\underline{t}}^2} + \sqrt{\log n} \cdot \frac{\varepsilon_2^{\alpha_X}}{\sigma_{\underline{t}}}.
$$

We can then obtain Claim 3 by combining all pieces.

**Remark C.3.** *Here the neural network size has a factor of $(\log n)^{D_X}$ even with the choice of $\delta > 0$. However, we can weaken this factor to $(\log n)^{d_X}$ by assuming that $\mathcal{M}_X$ lies in a smooth $d_X$-dimensional submanifold. In this case, let $V_{\widetilde{x}} \in \mathbb{R}^{D_X \times d_X}$ be an arbitrary orthonormal basis of the tangent space of $\mathcal{M}_X$ at $\widetilde{x}$, and consider the map $\phi_{\widetilde{x}}(\cdot) = V_{\widetilde{x}}^T(\cdot - \widetilde{x})$. This map is defined on $\mathbb{B}_{\mathcal{M}_X}(\widetilde{x}, r)$ and has a smooth inverse $\phi_{\widetilde{x}}^{-1}$ for some positive $r$. By expressing $x$ in terms of $\phi_{\widetilde{x}}^{-1}(\phi_{\widetilde{x}}(x))$ and considering the Taylor expansion of $\phi_{\widetilde{x}}^{-1}$ around 0, we can approximate $\overline{p}_t(w,x)$ and $\overline{dp}_t(w,x)$ using polynomials that depend on $\phi_p(w,x)$, $w$, and $\phi_{\widetilde{x}}(x)$. Notice that $\phi_{\widetilde{x}}(x)$ is $d_X$-dimensional and can be exactly realized through a ReLU neural network. By leveraging this fact, we can change the factor of $(\log n)^{D_X}$ to $(\log n)^{d_X}$.*

### C.1.4 Case 4: $\tau \le \underline{t} \le n^{-\frac{2}{2\alpha_Y + d_Y + d_X \frac{\alpha_Y}{\alpha_X}}}$

We set $\varepsilon_1 = n^{-\frac{1}{2\alpha_Y + d_Y + d_X \frac{\alpha_Y}{\alpha_X}}}$ and $\varepsilon_2 = n^{-\frac{1}{2\alpha_X + d_X + d_Y \frac{\alpha_X}{\alpha_Y}}}$. Let $\mathcal{N}_{\varepsilon_2}^X$ be one of the largest $\varepsilon_2$-packing of $\mathcal{M}_X$ and $\mathcal{N}_{\varepsilon_1}^Z$ be one of the largest $\varepsilon_1$-packing of $\mathbb{B}_{\mathbb{R}^{d_Y}}(0, \overline{r})$. Then we have $J_1 := |\mathcal{N}_{\varepsilon_1}^Z| \lesssim \varepsilon_1^{-d_Y}$ and

$J_2 := |\mathcal{N}_{\varepsilon_2}^X| \lesssim \varepsilon_2^{-d_X}$. Then take an arbitrary $\widetilde{x} \in \mathcal{N}_{\varepsilon_2}^X$, $k \in \mathcal{K}_{\widetilde{x}}$ and $\widetilde{z} \in N_{\varepsilon_1}^Z$, Consider set

$$\mathscr{S}_{k\widetilde{x}\widetilde{z}} = \Big\{ (x, w) \ : \ x \in \mathbb{B}_{\mathcal{M}_X}(\widetilde{x}, \sqrt{2}\varepsilon_2),$$
$$\|w - G_{[k]}^*(\widetilde{z}, x)\| \leq C\,\varepsilon_1, \mathrm{dist}(w, \mathcal{M}_{Y|x}) \leq c_0 \sigma_t \sqrt{\log n} \Big\},$$

we claim that

**Claim 4.** *Let* $\mathscr{L}_1 = \Theta(\log n)$, $\mathscr{L}_2 = \lceil \frac{-2\log n}{\log(\sigma_t) - \frac{3}{2}\log(\log n)} \rceil$ *and*

$$\mathscr{L} = \mathscr{L}_1 \mathscr{L}_2 \cdot (2\mathscr{L}_1 + 2\mathscr{L}_2 + 1) \binom{(4\mathscr{L}_1 + 3\mathscr{L}_2 + 3)\lfloor \beta_Y \rfloor + 2d_Y + \lfloor \alpha_Y \rfloor}{d_Y}$$
$$\cdot \binom{\mathscr{L}_2 + D_Y}{D_Y} \binom{(2\mathscr{L}_1 + 2\mathscr{L}_2 + 1)\lfloor \beta_X \rfloor + \lfloor \alpha_X \rfloor + D_X}{D_X},$$

*there exists a network* $\phi^*(w, x, t) \in \Phi\big(H, W, R, B, \Theta(\frac{\sqrt{\log n}}{\sigma_t})\big)$ *with* $H = \Theta(\log^4 n)$, $\|W\|_\infty = \Theta\big(\log^6 n\mathscr{L}\big)$, $R = \Theta\big(\log^8 n\mathscr{L}\big)$, $B = \exp(\Theta(\log^4 n))$, *so that for any* $(x, w) \in \mathscr{S}_{k\widetilde{x}\widetilde{z}}$, *and* $t \in [\underline{t}, \overline{t}]$,

$$\|\phi^*(w, x, t) - \nabla \log p_{t|x}(w)\|_\infty = \mathcal{O}\Big( \log n \frac{\varepsilon_1^{\beta_Y} + \varepsilon_2^{\beta_X}}{\sigma_t^2} + \sqrt{\log n} \frac{\varepsilon_1^{\alpha_Y}}{\sigma_t} \Big).$$

Then similar as Case 1, 2 and 3, the first statement of Lemma C.2 directly follows from Lemmas C.4 and C.6. Now we show Claim 4. For any $(x, w) \in \mathscr{S}_{k\widetilde{x}\widetilde{z}}$, we have

$$\|G_{[k]}^*(\widetilde{z}, x) - \mathrm{Proj}_{\mathcal{M}_{Y|x}}(w)\| \lesssim \varepsilon_1,$$

and

$$\{y \in \mathcal{M}_{Y|x} \ : \ \|y - w\| \leq c_2 \sigma_t \sqrt{\log n}\} \subset \{y \in \mathcal{M}_{Y|x} \ : \ \|y - \mathrm{Proj}_{\mathcal{M}_{Y|x}}(w)\| \leq c_3 \sigma_t \sqrt{\log n}\}$$
$$\subset \{y = G_{[k]}^*(z, x) \ : \ \|z - Q_{[k]}^*(\mathrm{Proj}_{\mathcal{M}_{Y|x}}(w), x)\| \leq c_4 \sigma_t \sqrt{\log n}\}.$$

Using Lemma C.11, we have

$$\|z - \phi_p(w, x)\| \leq \|z - Q_{[k]}^*(\mathrm{Proj}_{\mathcal{M}_{Y|x}}(w), x)\| + \|\phi_p(w, x) - Q_{[k]}^*(\mathrm{Proj}_{\mathcal{M}_{Y|x}}(w), x)\|$$
$$\leq \|z - Q_{[k]}^*(\mathrm{Proj}_{\mathcal{M}_{Y|x}}(w), x)\| + \mathcal{O}\Big( \sigma_t \sqrt{\log n} \Big),$$

and thus

$$\{y = G_{[k]}^*(z, x) \ : \ \|z - Q_{[k]}^*(\mathrm{Proj}_{\mathcal{M}_{Y|x}}(w), x)\| \leq c_4 \sigma_t \sqrt{\log n}\}$$
$$\subset \{y = G_{[k]}^*(z, x) \ : \ \|z - \phi_p(w, x)\| \leq c_5 \sigma_t \sqrt{\log n}\}$$
$$\subset \{y = G_{[k]}^*(z, x) \ : \ \|z - \phi_p(w, x)\|_\infty \leq c_6 \sigma_t \sqrt{\log n}\}.$$

So based on equation (2) and decomposition (13), we only need to approximate

$$
\frac{1}{\sigma_t} \cdot \frac{\int_{\|z-\phi_p(w,x)\|_\infty \leq c_6 \sigma_{\underline{t}} \sqrt{\log n}} \exp\left(-\frac{\|w - m_t G^*_{[k]}(z,x)\|^2}{2\sigma_t^2}\right) \cdot \left(-\frac{w - m_t G^*_{[k]}(z,x)}{\sigma_t}\right) v^*_{[k]}(z|x)\, \mathrm{d}z}{\int_{\|z-\phi_p(w,x)\|_\infty \leq c_6 \sigma_{\underline{t}} \sqrt{\log n}} \exp\left(-\frac{\|w - m_t G^*_{[k]}(z,x)\|^2}{2\sigma_t^2}\right) v^*_{[k]}(z|x)\, \mathrm{d}z}
$$

$$
= \frac{1}{\sigma_t} \cdot \left[ \int_{\|z-\phi_p(w,x)\|_\infty \leq c_6 \sigma_{\underline{t}} \sqrt{\log n}} \exp\left(-\frac{\|G^*_{[k]}|_{(\widetilde{z},\widetilde{x})}(\phi_p(w,x),x) - m_t G^*_{[k]}(z,x)\|^2}{2\sigma_t^2}\right) \right.
$$

$$
\left. \cdot \exp\left(-\frac{\langle w - G^*_{[k]}|_{(\widetilde{z},\widetilde{x})}(\phi_p(w,x),x), G^*_{[k]}|_{(\widetilde{z},\widetilde{x})}(\phi_p(w,x),x) - m_t G^*_{[k]}(z,x)\rangle}{\sigma_t^2}\right) v^*_{[k]}(z|x)\, \mathrm{d}z \right]^{-1}
$$

$$
\cdot \left[ \int_{\|z-\phi_p(w,x)\|_\infty \leq c_6 \sigma_{\underline{t}} \sqrt{\log n}} \exp\left(-\frac{\|G^*_{[k]}|_{(\widetilde{z},\widetilde{x})}(\phi_p(w,x),x) - m_t G^*_{[k]}(z,x)\|^2}{2\sigma_t^2}\right) \right.
$$

$$
\cdot \exp\left(-\frac{\langle w - G^*_{[k]}|_{(\widetilde{z},\widetilde{x})}(\phi_p(w,x),x), G^*_{[k]}|_{(\widetilde{z},\widetilde{x})}(\phi_p(w,x),x) - m_t G^*_{[k]}(z,x)\rangle}{\sigma_t^2}\right)
$$

$$
\left. \cdot \left(-\frac{G^*_{[k]}|_{(\widetilde{z},\widetilde{x})}(\phi_p(w,x),x) - m_t G^*_{[k]}(z,x)}{\sigma_t}\right) v^*_{[k]}(z|x)\, \mathrm{d}z \right] - \frac{w - G^*_{[k]}|_{(\widetilde{z},\widetilde{x})}(\phi_p(w,x),x)}{\sigma_t^2}
$$

(21)

Similar to Case 3, the term $\frac{w - G^*_{[k]}|_{(\widetilde{z},\widetilde{x})}(\phi_p(w,x),x)}{\sigma_t^2}$ can be approximated by neural network $\phi^{[3]}(w,x,t) \in \Phi(H,W,R,B)$ with an error $\frac{1}{n}$ if $H = \Theta(\log^2 n)$, $\|W\|_\infty = \Theta(\log^3 n)$, $R = \Theta(\log^4 n)$ and $B = \exp(\Theta(\log^2 n))$. Then notice that $v^*_{[k]}$ is $C^{\alpha_Y,\alpha_X}$-smooth, we can write

$$
v^*_{[k]}|_{(\widetilde{z},\widetilde{x})}(z,x) = \sum_{(j_1,j_2) \in \mathcal{J}^{d_Y,D_X}_{\alpha_Y,\alpha_X}} \frac{v^*_{[k](j_1,j_2)}(\widetilde{z},\widetilde{x})}{j_1! j_2!} (z - \widetilde{z})^{j_1} (x - \widetilde{x})^{j_2}, \tag{22}
$$

where

$$
\|v^*_{[k]}|_{(\widetilde{z},\widetilde{x})}(z,x) - v^*_{[k]}(z,x)\| \lesssim \|z - \widetilde{z}\|^{\alpha_Y} + \|x - \widetilde{x}\|^{\alpha_X}.
$$

We will first build an approximation to the conditional score function by replacing $G^*_{[k]}$ and $v^*_{[k]}$ with their polynomial approximators, that is, $G^*_{[k]}|_{(\widetilde{z},\widetilde{x})}$ defined in (12) and $v^*_{[k]}|_{(\widetilde{z},\widetilde{x})}$ defined in (22). To bound the approximation error, we will consider and bound the following terms using Lemma C.11 for any $(x,w) \in \mathscr{S}_{k\widetilde{x}\widetilde{z}}$, and any $z \in \mathbb{R}^{d_Y}$ satisfying $\|z - \phi_p(w,x)\|_\infty \leq c_6 \sigma_{\underline{t}} \sqrt{\log n}$:

$$
\|\phi_p(w,x) - \widetilde{z}\| \leq \|\phi_p(w,x) - Q^*_{[k]}(\mathrm{Proj}_{\mathcal{M}_{Y|x}}(w),x)\| + \|Q^*_{[k]}(\mathrm{Proj}_{\mathcal{M}_{Y|x}}(w),x) - \widetilde{z}\|
$$

$$
= \|\phi_p(w,x) - Q^*_{[k]}(\mathrm{Proj}_{\mathcal{M}_{Y|x}}(w),x)\| + \|Q^*_{[k]}(\mathrm{Proj}_{\mathcal{M}_{Y|x}}(w),x) - Q^*_{[k]}(G^*_{[k]}(\widetilde{z},x),x)\|
$$

$$
\lesssim \varepsilon_1;
$$

$$
\left\| G^*_{[k]}|_{(\widetilde{z},\widetilde{x})}(\phi_p(w,x),x) - m_t G^*_{[k]}|_{(\widetilde{z},\widetilde{x})}(z,x) \right\|
$$

$$
\leq \left\| G^*_{[k]}|_{(\widetilde{z},\widetilde{x})}(\phi_p(w,x),x) - G^*_{[k]}|_{(\widetilde{z},\widetilde{x})}(z,x) \right\| + (1 - m_t)\|G^*_{[k]}|_{(\widetilde{z},\widetilde{x})}(z,x))\| \lesssim \sigma_{\underline{t}} \sqrt{\log n};
$$

$$
\frac{1}{\sigma_t^2} \cdot \left| \|G^*_{[k]}|_{(\widetilde{z},\widetilde{x})}(\phi_p(w,x),x) - m_t G^*_{[k]}(z,x)\|^2 - \|G^*_{[k]}|_{(\widetilde{z},\widetilde{x})}(\phi_p(w,x),x) - m_t G^*_{[k]}|_{(\widetilde{z},\widetilde{x})}(z,x)\|^2 \right|
$$

$$
\lesssim \frac{\sigma_{\underline{t}} \sqrt{\log n}(\varepsilon_1^{\beta_Y} + \varepsilon_2^{\beta_X})}{\sigma_t^2} \asymp \frac{\sqrt{\log n}(\varepsilon_1^{\beta_Y} + \varepsilon_2^{\beta_X})}{\sigma_{\underline{t}}};
$$

$$
\|w - G^*_{[k]}|_{(\widetilde{z},\widetilde{x})}(\phi_p(w,x),x)\| \leq \|w - \mathrm{Proj}_{\mathcal{M}_{Y|x}}(w)\|
$$

$$
+ \|G^*_{[k]}(Q^*_{[k]}(\mathrm{Proj}_{\mathcal{M}_{Y|x}}(w),x),x) - G^*_{[k]}(\phi_p(w,x),x)\| + \|G^*_{[k]}(\phi_p(w,x),x) - G^*_{[k]}|_{(\widetilde{z},\widetilde{x})}(\phi_p(w,x),x)\|
$$

$$
\lesssim \sigma_{\underline{t}} \sqrt{\log n};
$$

$$\frac{1}{\sigma_t^2} \cdot \left| \left\langle w - G_{[k]}^* |_{(\widetilde{z},\widetilde{x})}(\phi_p(w,x),x), G_{[k]}^* |_{(\widetilde{z},\widetilde{x})}(\phi_p(w,x),x) - m_t G_{[k]}^*(z,x) \right\rangle \right.$$
$$\left. - \left\langle w - G_{[k]}^* |_{(\widetilde{z},\widetilde{x})}(\phi_p(w,x),x), G_{[k]}^* |_{(\widetilde{z},\widetilde{x})}(\phi_p(w,x),x) - m_t G_{[k]}^* |_{(\widetilde{z},\widetilde{x})}(z,x) \right\rangle \right|$$
$$\lesssim \frac{\sigma_{\underline{t}} \sqrt{\log n} \left( \varepsilon_1^{\beta_Y} + \varepsilon_2^{\beta_X} \right)}{\sigma_t^2} \asymp \frac{\sqrt{\log n} \left( \varepsilon_1^{\beta_Y} + \varepsilon_2^{\beta_X} \right)}{\sigma_{\underline{t}}};$$

$$\left| \left\langle w - G_{[k]}^* |_{(\widetilde{z},\widetilde{x})}(\phi_p(w,x),x), G_{[k]}^* |_{(\widetilde{z},\widetilde{x})}(\phi_p(w,x),x) - m_t G_{[k]}^* |_{(\widetilde{z},\widetilde{x})}(z,x) \right\rangle \right|$$
$$\leq \left| \left\langle w - G_{[k]}^* |_{(\widetilde{z},\widetilde{x})}(\phi_p(w,x),x), J_{G_{[k]}^* |_{(\widetilde{z},\widetilde{x})}(\cdot,x)}(\phi_p(w,x))(\phi_p(w,x) - z) \right\rangle \right|$$
$$+ \left| \left\langle w - G_{[k]}^* |_{(\widetilde{z},\widetilde{x})}(\phi_p(w,x),x), G_{[k]}^* |_{(\widetilde{z},\widetilde{x})}(\phi_p(w,x),x) - G_{[k]}^* |_{(\widetilde{z},\widetilde{x})}(z,x) - J_{G_{[k]}^* |_{(\widetilde{z},\widetilde{x})}(\cdot,x)}(\phi_p(w,x))(\phi_p(w,x) - z) \right\rangle \right|$$
$$+ \left| \left\langle w - G_{[k]}^* |_{(\widetilde{z},\widetilde{x})}(\phi_p(w,x),x), (1 - m_t) G_{[k]}^* |_{(\widetilde{z},\widetilde{x})}(z,x) \right\rangle \right|$$
$$\lesssim (\varepsilon_1 \sqrt{\log n})^{2\beta_Y} \sigma_{\underline{t}} \sqrt{\log n} + \sigma_{\underline{t}}^3 (\log n)^{\frac{3}{2}} + \sigma_{\underline{t}}^3 \sqrt{\log n}$$
$$\lesssim \sigma_{\underline{t}}^3 (\log n)^{\frac{3}{2}}; \tag{23}$$

$$\left\| \frac{G_{[k]}^* |_{(\widetilde{z},\widetilde{x})}(\phi_p(w,x),x) - m_t G_{[k]}^*(z,x)}{\sigma_t} \right\|$$
$$\lesssim \left\| \frac{G_{[k]}^* |_{(\widetilde{z},\widetilde{x})}(\phi_p(w,x),x) - m_t G_{[k]}^*(z,x)}{\sigma_t} - \frac{G_{[k]}^* |_{(\widetilde{z},\widetilde{x})}(\phi_p(w,x),x) - m_t G_{[k]}^* |_{(\widetilde{z},\widetilde{x})}(z,x)}{\sigma_t} \right\| \tag{24}$$
$$+ \left\| \frac{G_{[k]}^* |_{(\widetilde{z},\widetilde{x})}(\phi_p(w,x),x) - G_{[k]}^* |_{(\widetilde{z},\widetilde{x})}(z,x)}{\sigma_t} \right\| + \frac{1 - m_t}{\sigma_t} \cdot \| G_{[k]}^* |_{(\widetilde{z},\widetilde{x})}(z,x) \|$$
$$\lesssim \sqrt{\log n}.$$

Combining all the pieces, we can obtain

$$\left| \int_{\|z - \phi_p(w,x)\|_\infty \leq c_6 \sigma_{\underline{t}} \sqrt{\log n}} \exp \left( -\frac{\| G_{[k]}^* |_{(\widetilde{z},\widetilde{x})}(\phi_p(w,x),x) - m_t G_{[k]}^*(z,x) \|^2}{2\sigma_t^2} \right) \right.$$
$$\cdot \exp \left( -\frac{\left\langle w - G_{[k]}^* |_{(\widetilde{z},\widetilde{x})}(\phi_p(w,x),x), G_{[k]}^* |_{(\widetilde{z},\widetilde{x})}(\phi_p(w,x),x) - m_t G_{[k]}^*(z,x) \right\rangle}{\sigma_t^2} \right)$$
$$\cdot \left( -\frac{G_{[k]}^* |_{(\widetilde{z},\widetilde{x})}(\phi_p(w,x),x) - m_t G_{[k]}^*(z,x)}{\sigma_t} \right) v_{[k]}^*(z|x) \, \mathrm{d}z$$
$$- \int_{\|z - \phi_p(w,x)\|_\infty \leq c_6 \sigma_{\underline{t}} \sqrt{\log n}} \exp \left( -\frac{\| G_{[k]}^* |_{(\widetilde{z},\widetilde{x})}(\phi_p(w,x),x) - m_t G_{[k]}^* |_{(\widetilde{z},\widetilde{x})}(z,x) \|^2}{2\sigma_t^2} \right) \tag{25}$$
$$\cdot \exp \left( -\frac{\left\langle w - G_{[k]}^* |_{(\widetilde{z},\widetilde{x})}(\phi_p(w,x),x), G_{[k]}^* |_{(\widetilde{z},\widetilde{x})}(\phi_p(w,x),x) - m_t G_{[k]}^* |_{(\widetilde{z},\widetilde{x})}(z,x) \right\rangle}{\sigma_t^2} \right)$$
$$\left. \cdot \left( -\frac{G_{[k]}^* |_{(\widetilde{z},\widetilde{x})}(\phi_p(w,x),x) - m_t G_{[k]}^* |_{(\widetilde{z},\widetilde{x})}(z,x)}{\sigma_t} \right) v_{[k]}^* |_{(\widetilde{z},\widetilde{x})}(z|x) \, \mathrm{d}z \right|$$
$$\lesssim \int_{\|z - \phi_p(w,x)\|_\infty \leq c_6 \sigma_{\underline{t}} \sqrt{\log n}} \exp \left( -\frac{\| G_{[k]}^* |_{(\widetilde{z},\widetilde{x})}(\phi_p(w,x),x) - m_t G_{[k]}^*(z,x) \|^2}{2\sigma_t^2} \right) \mathrm{d}z$$
$$\cdot \left( \frac{\log n \left( \varepsilon_1^{\beta_Y} + \varepsilon_2^{\beta_X} \right)}{\sigma_{\underline{t}}} + \sqrt{\log n} \cdot \varepsilon_1^{\alpha_Y} \right).$$

Similarly, we have

$$
\Bigg\| \int_{\|z - \phi_p(w,x)\|_\infty \le c_6 \sigma_{\underline{t}} \sqrt{\log n}} \exp\left( -\frac{\|G^*_{[k]}|_{(\widetilde{z},\widetilde{x})}(\phi_p(w,x),x) - m_t G^*_{[k]}|_{(\widetilde{z},\widetilde{x})}(z,x)\|^2}{2\sigma_t^2} \right)
$$

$$
\cdot \exp\left( -\frac{\langle w - G^*_{[k]}|_{(\widetilde{z},\widetilde{x})}(\phi_p(w,x),x), G^*_{[k]}|_{(\widetilde{z},\widetilde{x})}(\phi_p(w,x),x) - m_t G^*_{[k]}|_{(\widetilde{z},\widetilde{x})}(z,x) \rangle}{\sigma_t^2} \right) v^*_{[k]}|_{(\widetilde{z},\widetilde{x})}(z|x)\, \mathrm{d}z
$$

$$
- \int_{\|z - \phi_p(w,x)\|_\infty \le c_6 \sigma_{\underline{t}} \sqrt{\log n}} \exp\left( -\frac{\|G^*_{[k]}|_{(\widetilde{z},\widetilde{x})}(\phi_p(w,x),x) - m_t G^*_{[k]}(z,x)\|^2}{2\sigma_t^2} \right)
$$

$$
\cdot \exp\left( -\frac{\langle w - G^*_{[k]}|_{(\widetilde{z},\widetilde{x})}(\phi_p(w,x),x), G^*_{[k]}|_{(\widetilde{z},\widetilde{x})}(\phi_p(w,x),x) - m_t G^*_{[k]}(z,x) \rangle}{\sigma_t^2} \right) v^*_{[k]}(z|x)\, \mathrm{d}z \Bigg\|
$$

$$
\lesssim \int_{\|z - \phi_p(w,x)\|_\infty \le c_6 \sigma_{\underline{t}} \sqrt{\log n}} \exp\left( -\frac{\|G^*_{[k]}|_{(\widetilde{z},\widetilde{x})}(\phi_p(w,x),x) - m_t G^*_{[k]}(z,x)\|^2}{2\sigma_t^2} \right) \mathrm{d}z
$$

$$
\cdot \left( \frac{\sqrt{\log n}\left(\varepsilon_1^{\beta_Y} + \varepsilon_2^{\beta_X}\right)}{\sigma_{\underline{t}}} + \varepsilon_1^{\alpha_Y} \right).
$$

$$(26)$$

Denote

$$
\widetilde{dp}_t(w,x) = \int_{\|z - \phi_p(w,x)\|_\infty \le c_6 \sigma_{\underline{t}} \sqrt{\log n}} \exp\left( -\frac{\|G^*_{[k]}|_{(\widetilde{z},\widetilde{x})}(\phi_p(w,x),x) - m_t G^*_{[k]}|_{(\widetilde{z},\widetilde{x})}(z,x)\|^2}{2\sigma_t^2} \right)
$$

$$
\cdot \exp\left( -\frac{\langle w - G^*_{[k]}|_{(\widetilde{z},\widetilde{x})}(\phi_p(w,x),x), G^*_{[k]}|_{(\widetilde{z},\widetilde{x})}(\phi_p(w,x),x) - m_t G^*_{[k]}|_{(\widetilde{z},\widetilde{x})}(z,x) \rangle}{\sigma_t^2} \right)
$$

$$
\cdot \left( -\frac{G^*_{[k]}|_{(\widetilde{z},\widetilde{x})}(\phi_p(w,x),x) - m_t G^*_{[k]}|_{(\widetilde{z},\widetilde{x})}(z,x)}{\sigma_t} \right) v^*_{[k]}|_{(\widetilde{z},\widetilde{x})}(z|x)\, \mathrm{d}z,
$$

and

$$
\widetilde{p}_t(w,x)
$$

$$
= \int_{\|z - \phi_p(w,x)\|_\infty \le c_6 \sigma_{\underline{t}} \sqrt{\log n}} \exp\left( -\frac{\|G^*_{[k]}|_{(\widetilde{z},\widetilde{x})}(\phi_p(w,x),x) - m_t G^*_{[k]}|_{(\widetilde{z},\widetilde{x})}(z,x)\|^2}{2\sigma_t^2} \right)
$$

$$
\cdot \exp\left( -\frac{\langle w - G^*_{[k]}|_{(\widetilde{z},\widetilde{x})}(\phi_p(w,x),x), G^*_{[k]}|_{(\widetilde{z},\widetilde{x})}(\phi_p(w,x),x) - m_t G^*_{[k]}|_{(\widetilde{z},\widetilde{x})}(z,x) \rangle}{\sigma_t^2} \right) v^*_{[k]}|_{(\widetilde{z},\widetilde{x})}(z|x)\, \mathrm{d}z
$$

We will show that if there exist neural networks $\phi^{[1]}(w,x,t)$ and $\phi^{[2]}(w,x,t)$ so that for any $t \in [\underline{t}, \overline{t}]$ and for any $(x,w) \in \mathscr{S}_{k\widetilde{x}\widetilde{z}}$,

$$
\|\widetilde{dp}_t(w,x) - \phi^{[1]}(w,x,t)\|_\infty \lesssim (\sigma_{\underline{t}})^{d_Y} \left( \left( \frac{\log n\left(\varepsilon_1^{\beta_Y} + \varepsilon_2^{\beta_X}\right)}{\sigma_{\underline{t}}} + \sqrt{\log n}\,\varepsilon_1^{\alpha_Y} \right) \right), \tag{27}
$$

$$
\|\widetilde{p}_t(w,x) - \phi^{[2]}(w,x,t)\|_\infty \lesssim (\sigma_{\underline{t}})^{d_Y} \left( \left( \frac{\sqrt{\log n}\left(\varepsilon_1^{\beta_Y} + \varepsilon_2^{\beta_X}\right)}{\sigma_{\underline{t}}} + \varepsilon_1^{\alpha_Y} \right) \right), \tag{28}
$$

then it holds that

$$
\left\|\frac{1}{\sigma_t} \cdot \left[\int_{\|z-\phi_p(w,x)\|_\infty \leq c_6\sigma_{\underline{t}}\sqrt{\log n}} \exp\left(-\frac{\|G^*_{[k]}|_{(\widetilde{z},\widetilde{x})}(\phi_p(w,x),x) - m_t G^*_{[k]}(z,x)\|^2}{2\sigma_t^2}\right)\right.\right.
$$

$$
\left.\cdot \exp\left(-\frac{\langle w - G^*_{[k]}|_{(\widetilde{z},\widetilde{x})}(\phi_p(w,x),x), G^*_{[k]}|_{(\widetilde{z},\widetilde{x})}(\phi_p(w,x),x) - m_t G^*_{[k]}(z,x)\rangle}{\sigma_t^2}\right) v^*_{[k]}(z|x)\,\mathrm{d}z\right]^{-1}
$$

$$
\cdot\left[\int_{\|z-\phi_p(w,x)\|_\infty \leq c_6\sigma_{\underline{t}}\sqrt{\log n}} \exp\left(-\frac{\|G^*_{[k]}|_{(\widetilde{z},\widetilde{x})}(\phi_p(w,x),x) - m_t G^*_{[k]}(z,x)\|^2}{2\sigma_t^2}\right)\right.
$$

$$
\cdot \exp\left(-\frac{\langle w - G^*_{[k]}|_{(\widetilde{z},\widetilde{x})}(\phi_p(w,x),x), G^*_{[k]}|_{(\widetilde{z},\widetilde{x})}(\phi_p(w,x),x) - m_t G^*_{[k]}(z,x)\rangle}{\sigma_t^2}\right)
$$

$$
\left.\cdot\left(-\frac{G^*_{[k]}|_{(\widetilde{z},\widetilde{x})}(\phi_p(w,x),x) - m_t G^*_{[k]}(z,x)}{\sigma_t}\right) v^*_{[k]}(z|x)\,\mathrm{d}z\right]
$$

$$
\left.-\frac{1}{\sigma_t}\cdot\frac{\phi^{[1]}(w,x,t)}{\phi^{[2]}(w,x,t)}\right\|_\infty \lesssim \log n\frac{\varepsilon_1^{\beta_Y} + \varepsilon_2^{\beta_X}}{\sigma_{\underline{t}}^2} + \sqrt{\log n}\frac{\varepsilon_1^{\alpha_Y}}{\sigma_{\underline{t}}}.
$$

(29)

To show (29), we first bound $\int_{\|z-\phi_p(w,x)\|_\infty \leq c_6\sigma_{\underline{t}}\sqrt{\log n}} \exp\left(-\frac{\|G^*_{[k]}|_{(\widetilde{z},\widetilde{x})}(\phi_p(w,x),x) - m_t G^*_{[k]}(z,x)\|^2}{2\sigma_t^2}\right)\,\mathrm{d}z$.
Notice that

$$
\|\phi_p(w,x) - z\| = \|Q^*_{[k]}(G^*_{[k]}(\phi_p(w,x),x),x) - Q^*_{[k]}(G^*_{[k]}(z,x),x)\|
$$
$$
\leq L\|G^*_{[k]}(\phi_p(w,x),x) - G^*_{[k]}(z,x)\|
$$
$$
\leq \|G^*_{[k]}|_{(\widetilde{z},\widetilde{x})}(\phi_p(w,x),x) - m_t G^*_{[k]}(z,x)\| + (1-m_t)\|G^*_{[k]}(z,x)\| + \mathcal{O}(\varepsilon_1^{\beta_Y} + \varepsilon_2^{\beta_X})
$$
$$
\leq \|G^*_{[k]}|_{(\widetilde{z},\widetilde{x})}(\phi_p(w,x),x) - m_t G^*_{[k]}(z,x)\| + O(\sigma_{\underline{t}}),
$$

we have

$$
\int_{\|z-\phi_p(w,x)\|_\infty \leq c_6\sigma_{\underline{t}}\sqrt{\log n}} \exp\left(-\frac{\|G^*_{[k]}|_{(\widetilde{z},\widetilde{x})}(\phi_p(w,x),x) - m_t G^*_{[k]}(z,x)\|^2}{2\sigma_t^2}\right)\,\mathrm{d}z \lesssim \sigma_{\underline{t}}^{d_Y}.
$$

Therefore, combined with (25) and (26), we can get

$$
\left\|\int_{\|z-\phi_p(w,x)\|_\infty \leq c_6\sigma_{\underline{t}}\sqrt{\log n}} \exp\left(-\frac{\|G^*_{[k]}|_{(\widetilde{z},\widetilde{x})}(\phi_p(w,x),x) - m_t G^*_{[k]}(z,x)\|^2}{2\sigma_t^2}\right)\right.
$$

$$
\cdot \exp\left(-\frac{\langle w - G^*_{[k]}|_{(\widetilde{z},\widetilde{x})}(\phi_p(w,x),x), G^*_{[k]}|_{(\widetilde{z},\widetilde{x})}(\phi_p(w,x),x) - m_t G^*_{[k]}(z,x)\rangle}{\sigma_t^2}\right)
$$

$$
\left.\cdot\left(-\frac{G^*_{[k]}|_{(\widetilde{z},\widetilde{x})}(\phi_p(w,x),x) - m_t G^*_{[k]}(z,x)}{\sigma_t}\right) v^*_{[k]}(z|x)\,\mathrm{d}z - \phi^{[1]}(x,t)\right\|_\infty
$$

$$
\lesssim (\sigma_{\underline{t}})^{d_Y}\left(\frac{\log n(\varepsilon_1^{\beta_Y} + \varepsilon_2^{\beta_X})}{\sigma_{\underline{t}}} + \sqrt{\log n}\varepsilon_1^{\alpha_Y}\right)
$$

and

$$
\left|\int_{\|z-\phi_p(w,x)\|_\infty \leq c_6\sigma_{\underline{t}}\sqrt{\log n}} \exp\left(-\frac{\|G^*_{[k]}|_{(\widetilde{z},\widetilde{x})}(\phi_p(w,x),x) - m_t G^*_{[k]}(z,x)\|^2}{2\sigma_t^2}\right)\right.
$$

$$
\cdot \exp\left(-\frac{\langle w - G^*_{[k]}|_{(\widetilde{z},\widetilde{x})}(\phi_p(w,x),x), G^*_{[k]}|_{(\widetilde{z},\widetilde{x})}(\phi_p(w,x),x) - m_t G^*_{[k]}(z,x)\rangle}{\sigma_t^2}\right) v^*_{[k]}(z|x)\,\mathrm{d}z
$$

$$
\left.- \phi^{[2]}(x,t)\right\|_\infty \lesssim (\sigma_{\underline{t}})^{d_Y}\left(\frac{\sqrt{\log n}(\varepsilon_1^{\beta_Y} + \varepsilon_2^{\beta_X})}{\sigma_{\underline{t}}} + \varepsilon_1^{\alpha_Y}\right).
$$

Now use the fact that

$$\|G^*_{[k]}|_{(\widetilde{z},\widetilde{x})}(\phi_p(w,x),x) - m_t G^*_{[k]}(z,x)\|$$
$$\leq \|G^*_{[k]}(\phi_p(w,x),x) - G^*_{[k]}(z,x)\| + (1-m_t)\|G^*_{[k]}(z,x)\|$$
$$+ \|G^*_{[k]}|_{(\widetilde{z},\widetilde{x})}(\phi_p(w,x),x) - G^*_{[k]}(z,x)\| \lesssim \|\phi_p(x,w)-z\| + \mathcal{O}(\sigma_{\underline{t}}),$$

we have

$$\left\|\left[\int_{\|z-\phi_p(w,x)\|_\infty \leq c_6\sigma_{\underline{t}}\sqrt{\log n}} \exp\left(-\frac{\|G^*_{[k]}|_{(\widetilde{z},\widetilde{x})}(\phi_p(w,x),x) - m_t G^*_{[k]}(z,x)\|^2}{2\sigma_t^2}\right)\right.\right.$$
$$\left.\cdot \exp\left(-\frac{\langle w - G^*_{[k]}|_{(\widetilde{z},\widetilde{x})}(\phi_p(w,x),x), G^*_{[k]}|_{(\widetilde{z},\widetilde{x})}(\phi_p(w,x),x) - m_t G^*_{[k]}(z,x)\rangle}{\sigma_t^2}\right) v^*_{[k]}(z|x)\,\mathrm{d}z\right]^{-1}$$
$$\cdot\left[\int_{\|z-\phi_p(w,x)\|_\infty \leq c_6\sigma_{\underline{t}}\sqrt{\log n}} \exp\left(-\frac{\|G^*_{[k]}|_{(\widetilde{z},\widetilde{x})}(\phi_p(w,x),x) - m_t G^*_{[k]}(z,x)\|^2}{2\sigma_t^2}\right)\right.$$
$$\cdot \exp\left(-\frac{\langle w - G^*_{[k]}|_{(\widetilde{z},\widetilde{x})}(\phi_p(w,x),x), G^*_{[k]}|_{(\widetilde{z},\widetilde{x})}(\phi_p(w,x),x) - m_t G^*_{[k]}(z,x)\rangle}{\sigma_t^2}\right)$$
$$\left.\left.\cdot\left(-\frac{G^*_{[k]}|_{(\widetilde{z},\widetilde{x})}(\phi_p(w,x),x) - m_t G^*_{[k]}(z,x)}{\sigma_t}\right) v^*_{[k]}(z|x)\,\mathrm{d}z\right]\right\|$$
$$\lesssim \sqrt{\log n},$$

and

$$\int_{\|z-\phi_p(w,x)\|_\infty \leq c_6\sigma_{\underline{t}}\sqrt{\log n}} \exp\left(-\frac{\|G^*_{[k]}|_{(\widetilde{z},\widetilde{x})}(\phi_p(w,x),x) - m_t G^*_{[k]}(z,x)\|^2}{2\sigma_t^2}\right)$$
$$\cdot \exp\left(-\frac{\langle w - G^*_{[k]}|_{(\widetilde{z},\widetilde{x})}(\phi_p(w,x),x), G^*_{[k]}|_{(\widetilde{z},\widetilde{x})}(\phi_p(w,x),x) - m_t G^*_{[k]}(z,x)\rangle}{\sigma_t^2}\right) v^*_{[k]}(z|x)\,\mathrm{d}z$$
$$\geq \int_{\|z-\phi_p(w,x)\| \leq \sigma_{\underline{t}}} \exp\left(-\frac{\|G^*_{[k]}|_{(\widetilde{z},\widetilde{x})}(\phi_p(w,x),x) - m_t G^*_{[k]}(z,x)\|^2}{2\sigma_t^2}\right)$$
$$\cdot \exp\left(-\frac{\langle w - G^*_{[k]}|_{(\widetilde{z},\widetilde{x})}(\phi_p(w,x),x), G^*_{[k]}|_{(\widetilde{z},\widetilde{x})}(\phi_p(w,x),x) - m_t G^*_{[k]}(z,x)\rangle}{\sigma_t^2}\right) v^*_{[k]}(z|x)\,\mathrm{d}z$$
$$\gtrsim \int_{\|z-\phi_p(w,x)\| \leq \sigma_{\underline{t}}} \exp\left(-\frac{\langle w - G^*_{[k]}|_{(\widetilde{z},\widetilde{x})}(\phi_p(w,x),x), G^*_{[k]}|_{(\widetilde{z},\widetilde{x})}(\phi_p(w,x),x) - m_t G^*_{[k]}(z,x)\rangle}{\sigma_t^2}\right) \mathrm{d}z.$$

Moreover, when $\|z - \phi_p(w,x)\| \leq \sigma_{\underline{t}}$,

$$\left|\langle w - G^*_{[k]}|_{(\widetilde{z},\widetilde{x})}(\phi_p(w,x),x), G^*_{[k]}|_{(\widetilde{z},\widetilde{x})}(\phi_p(w,x),x) - m_t G^*_{[k]}(z,x)\rangle\right|$$
$$\leq \left|\langle w - G^*_{[k]}|_{(\widetilde{z},\widetilde{x})}(\phi_p(w,x),x), J_{G^*_{[k]}|_{(\widetilde{z},\widetilde{x})}(\cdot,x)}(\phi_p(w,x))(\phi_p(w,x)-z))\rangle\right|$$
$$+ \left|\langle w - G^*_{[k]}|_{(\widetilde{z},\widetilde{x})}(\phi_p(w,x),x), G^*_{[k]}|_{(\widetilde{z},\widetilde{x})}(\phi_p(w,x),x) - G^*_{[k]}|_{(\widetilde{z},\widetilde{x})}(z,x) - J_{G^*_{[k]}|_{(\widetilde{z},\widetilde{x})}(\cdot,x)}(\phi_p(w,x))(\phi_p(w,x)-z))\rangle\right|$$
$$+ \left|\langle w - G^*_{[k]}|_{(\widetilde{z},\widetilde{x})}(\phi_p(w,x),x), G^*_{[k]}|_{(\widetilde{z},\widetilde{x})}(z,x) - G^*_{[k]}(z,x)\rangle\right|$$
$$+ \left|\langle w - G^*_{[k]}|_{(\widetilde{z},\widetilde{x})}(\phi_p(w,x),x), (1-m_t)G^*_{[k]}(z,x)\rangle\right|$$
$$\lesssim (\varepsilon_1\sqrt{\log n})^{2\beta_Y}\sigma_{\underline{t}}\sqrt{\log n} + \sigma_{\underline{t}}^3(\log n)^{\frac{3}{2}} + \sigma_{\underline{t}}\sqrt{\log n}(\varepsilon_1^{\beta_Y} + \varepsilon_2^{\beta_X}) + \sigma_{\underline{t}}^3\sqrt{\log n}$$
$$\lesssim \sigma_{\underline{t}}^2$$

Therefore, we have

$$
\int_{\|z-\phi_p(w,x)\|_\infty \le c_6\sigma_t\sqrt{\log n}} \exp\left(-\frac{\|G^*_{[k]}|_{(\widetilde z,\widetilde x)}(\phi_p(w,x),x) - m_t G^*_{[k]}(z,x)\|^2}{2\sigma_t^2}\right)
$$
$$
\cdot \exp\left(-\frac{\langle w - G^*_{[k]}|_{(\widetilde z,\widetilde x)}(\phi_p(w,x),x), G^*_{[k]}|_{(\widetilde z,\widetilde x)}(\phi_p(w,x),x) - m_t G^*_{[k]}(z,x)\rangle}{\sigma_t^2}\right) v^*_{[k]}(z|x)\, \mathrm{d}z
$$
$$
\gtrsim (\sigma_{\underline t})^{d_Y}.
$$

We can then show (29) by combining all pieces.

Then we construct $\phi^{[1]}(w,x,t)$ by approximating $\widetilde{dp}_t(w,x)$ with polynomials. Based on statements (23) and (24), by choosing $\mathscr{L}_1 = \Theta(\log n)$ and $\mathscr{L}_2 = \lceil \frac{-2\log n}{\log(\sigma_{\underline t}) - \frac{3}{2}\log(\log n)}\rceil$, we have

$$
\left| \exp\left(-\frac{\|G^*_{[k]}|_{(\widetilde z,\widetilde x)}(\phi_p(w,x),x) - m_t G^*_{[k]}|_{(\widetilde z,\widetilde x)}(z,x)\|^2}{2\sigma_t^2}\right) \right.
$$
$$
\left. - \sum_{l_1=0}^{\mathscr{L}_1} (-1)^{l_1} \frac{\|G^*_{[k]}|_{(\widetilde z,\widetilde x)}(\phi_p(w,x),x) - m_t G^*_{[k]}|_{(\widetilde z,\widetilde x)}(z,x)\|^{2l_1}}{2^{l_1} l_1! \sigma_t^{2l_1}} \right| \lesssim n^{-2},
$$

and

$$
\left| \exp\left(-\frac{\langle w - G^*_{[k]}|_{(\widetilde z,\widetilde x)}(\phi_p(w,x),x), G^*_{[k]}|_{(\widetilde z,\widetilde x)}(\phi_p(w,x),x) - m_t G^*_{[k]}|_{(\widetilde z,\widetilde x)}(z,x)\rangle}{\sigma_t^2}\right) \right.
$$
$$
\left. - \sum_{l_2=0}^{\mathscr{L}_2} (-1)^{l_2} \frac{\langle w - G^*_{[k]}|_{(\widetilde z,\widetilde x)}(\phi_p(w,x),x), G^*_{[k]}|_{(\widetilde z,\widetilde x)}(\phi_p(w,x),x) - m_t G^*_{[k]}|_{(\widetilde z,\widetilde x)}(z,x)\rangle^{l_2}}{l_2! \sigma_t^{2l_2}} \right|
$$
$$
\lesssim n^{-2}.
$$

Therefore,

$$
\left\| \int_{\|z-\phi_p(w,x)\|_\infty \le c_6\sigma_t\sqrt{\log n}} \sum_{l_1=0}^{\mathscr{L}_1} (-1)^{l_1} \frac{\|G^*_{[k]}|_{(\widetilde z,\widetilde x)}(\phi_p(w,x),x) - m_t G^*_{[k]}|_{(\widetilde z,\widetilde x)}(z,x)\|^{2l_1}}{2^{l_1} l_1! \sigma_t^{2l_1}} \right.
$$
$$
\cdot \sum_{l_2=0}^{\mathscr{L}_2} (-1)^{l_2} \frac{\langle w - G^*_{[k]}|_{(\widetilde z,\widetilde x)}(\phi_p(w,x),x), G^*_{[k]}|_{(\widetilde z,\widetilde x)}(\phi_p(w,x),x) - m_t G^*_{[k]}|_{(\widetilde z,\widetilde x)}(z,x)\rangle^{l_2}}{l_2! \sigma_t^{2l_2}}
$$
$$
\cdot \left(-\frac{G^*_{[k]}|_{(\widetilde z,\widetilde x)}(\phi_p(w,x),x) - m_t G^*_{[k]}|_{(\widetilde z,\widetilde x)}(z,x)}{\sigma_t}\right) v^*_{[k]}|_{(\widetilde z,\widetilde x)}(z|x)\, \mathrm{d}z - \widetilde{dp}_t(x) \left\|_\infty \right.
$$
$$
\lesssim (\sigma_{\underline t})^{d_Y}\left(\frac{\log n\left(\varepsilon_1^{\beta_Y} + \varepsilon_2^{\beta_X}\right)}{\sigma_{\underline t}} + \sqrt{\log n}\,\varepsilon_1^{\alpha_Y}\right).
$$

Moreover, since $G_{[k]}^*|_{(\widetilde{z},\widetilde{x})}$ and $v_{[k]}^*|_{(\widetilde{z},\widetilde{x})}$ are polynomials in $z$ and $x$, we can write

$$
\int_{\|z-\phi_p(w,x)\|_\infty \leq c_6\sigma_t\sqrt{\log n}} \sum_{l_1=0}^{\mathscr{L}_1}(-1)^{l_1} \frac{\|G_{[k]}^*|_{(\widetilde{z},\widetilde{x})}(\phi_p(w,x),x) - m_t G_{[k]}^*|_{(\widetilde{z},\widetilde{x})}(z,x)\|^{2l_1}}{2^{l_1}l_1!\sigma_t^{2l_1}}
$$

$$
\cdot \sum_{l_2=0}^{\mathscr{L}_2}(-1)^{l_2} \frac{\langle w - G_{[k]}^*|_{(\widetilde{z},\widetilde{x})}(\phi_p(w,x),x), G_{[k]}^*|_{(\widetilde{z},\widetilde{x})}(\phi_p(w,x),x) - m_t G_{[k]}^*|_{(\widetilde{z},\widetilde{x})}(z,x)\rangle^{l_2}}{l_2!\sigma_t^{2l_2}}
$$

$$
\cdot \left(-\frac{G_{[k]}^*|_{(\widetilde{z},\widetilde{x})}(\phi_p(w,x),x) - m_t G_{[k]}^*|_{(\widetilde{z},\widetilde{x})}(z,x)}{\sigma_t}\right) v_{[k]}^*|_{(\widetilde{z},\widetilde{x})}(z|x)\,\mathrm{d}z
$$

$$
= \int_{\|z-\phi_p(w,x)\|_\infty \leq c_6\sigma_t\sqrt{\log n}} \sum_{l_1=0}^{\mathscr{L}_1}\sum_{l_2=0}^{\mathscr{L}_2} \frac{(-1)^{(l_1+l_2+1)}}{2^{l_1}l_1!l_2!}(\frac{1}{\sigma_t})^{2l_1+2l_2+1}\Big\| \sum_{(j_1,j_2)\in\mathcal{J}_{\beta_Y,\beta_X}^{d_Y,D_X}} \frac{G_{[k](j_1,j_2)}^*(\widetilde{z},\widetilde{x})}{j_1!j_2!}
$$

$$
\cdot \left((\phi_p(w,x)-\widetilde{z})^{j_1} - m_t(z-\widetilde{z})^{j_1}\right)(x-\widetilde{x})^{j_2}\Big\|^{2l_1} \cdot \Big\langle w - \sum_{(j_1,j_2)\in\mathcal{J}_{\beta_Y,\beta_X}^{d_Y,D_X}} \frac{G_{[k](j_1,j_2)}^*(\widetilde{z},\widetilde{x})}{j_1!j_2!}(\phi_p(w,x)-\widetilde{z})^{j_1}(x-\widetilde{x})^{j_2},
$$

$$
\sum_{(j_1,j_2)\in\mathcal{J}_{\beta_Y,\beta_X}^{d_Y,D_X}} \frac{G_{[k](j_1,j_2)}^*(\widetilde{z},\widetilde{x})}{j_1!j_2!}\left((\phi_p(w,x)-\widetilde{z})^{j_1} - m_t(z-\widetilde{z})^{j_1}\right)(x-\widetilde{x})^{j_2}\Big\rangle^{l_2}
$$

$$
\cdot \sum_{(j_1,j_2)\in\mathcal{J}_{\beta_Y,\beta_X}^{d_Y,D_X}} \frac{G_{[k](j_1,j_2)}^*(\widetilde{z},\widetilde{x})}{j_1!j_2!}\left((\phi_p(w,x)-\widetilde{z})^{j_1} - m_t(z-\widetilde{z})^{j_1}\right)(x-\widetilde{x})^{j_2}
$$

$$
\cdot \sum_{(j_1,j_2)\in\mathcal{J}_{\alpha_Y,\alpha_X}^{d_Y,D_X}} \frac{v_{[k](j_1,j_2)}^*(\widetilde{z},\widetilde{x})}{j_1!j_2!}(z-\widetilde{z})^{j_1}(x-\widetilde{x})^{j_2}\,\mathrm{d}z
$$

$$
= \sum_{l_1=0}^{\mathscr{L}_1}\sum_{l_2=0}^{\mathscr{L}_2}(\frac{1}{\sigma_t})^{2l_1+2l_2+1} \sum_{0\leq k\leq 2l_1+l_2+1} m_t^k \sum_{s\in\mathbb{N}_0^{d_Y},|s|\leq(4l_1+3l_2+2)\lfloor\beta_Y\rfloor+d_Y+\lfloor\alpha_Y\rfloor} (\phi_p(w,x))^{(s)} \sum_{i\in\mathbb{N}_0^{D_Y},|i|\leq l_2} w^{(i)}
$$

$$
\sum_{j\in\mathbb{N}_0^{D_X},|j|\leq(2l_1+2l_2+1)\lfloor\beta_X\rfloor+\lfloor\alpha_X\rfloor} a_{l_1l_2ksij}x^{(j)},
$$

where $a_{l_1l_2ksij} \in \mathbb{R}^{D_Y}$ are some constant coefficients. Then notice that $(\frac{1}{\sigma})^{2l_1+2l_2+1}a_{l_1l_2ksij} \lesssim \exp(\mathcal{O}(\log^2 n))$, we

1. Approximate $m_t$ by $\phi_m(t) \in \Phi(H,W,R,B)$ with $H = \Theta(\log^4 n)$, $\|W\|_\infty = \Theta(\log^6 n)$, $R = \Theta(\log^8 n)$ and $B = \exp(\Theta(\log^4 n))$.

2. Approximate $\sigma_t$ by $\phi_\sigma(t) \in \Phi(H,W,R,B)$ with $H = \Theta(\log^4 n)$, $\|W\|_\infty = \Theta(\log^6 n)$, $R = \Theta(\log^8 n)$ and $B = \exp(\Theta(\log^4 n))$.

3. Approximate $\frac{1}{x}$ by $\phi_{rec}(x) \in \Phi(H,W,R,B)$ with $H = \Theta(\log^4 n)$, $\|W\|_\infty = \Theta(\log^6 n)$, $R = \Theta(\log^8 n)$ and $B = \exp(\Theta(\log^4 n))$.

4. For vector $x \in \mathbb{R}^{D_Y}$, approximate $x^{(i)}$ by $\phi_{vpower}^{[D]}(x;i) \in \Phi(H,W,R,B)$ with $H = \Theta(\log^2 n)$, $\|W\|_\infty = \Theta(1)$, $R = \Theta(\log^2 n)$ and $B = \exp(\Theta(\log\log n))$.

5. For vector $z \in \mathbb{R}^{d_Y}$, approximate $z^{(i)}$ by $\phi_{vpower}^{[d]}(z;i) \in \Phi(H,W,R,B)$ with $H = \Theta(\log^2 n \cdot \log\log n)$, $\|W\|_\infty = \Theta(\log n)$, $R = \Theta(\log^3 n)$ and $B = \exp(\Theta(\log n \cdot \log\log n))$.

6. For $x \in \mathbb{R}$, Approximate $x^a$ by $\phi_{power}(x;a) \in \Phi(H,W,R,B)$ with $H = \Theta(\log^2 n \log\log n)$, $\|W\|_\infty = \Theta(\log n)$, $R = \Theta(\log^3 n)$ and $B = \exp(\Theta(\log n \log\log n))$.

7. For $x,y \in \mathbb{R}$, Approximate $x\cdot y$ by $\phi_{mult}(x,y) \in \Phi(H,W,R,B)$ with $H = \Theta(\log^2 n)$, $\|W\|_\infty = \Theta(1)$, $R = \Theta(\log^2 n)$ and $B = \exp(\Theta(\log^2 n))$.

We have

$$\left\| \sum_{l_1=0}^{\mathscr{L}_1}\sum_{l_2=0}^{\mathscr{L}_2} \Big(\frac{1}{\sigma_t}\Big)^{2l_1+2l_2+1} \sum_{0\le k\le 2l_1+l_2+1} m_t^k \sum_{s\in\mathbb{N}_0^{d_Y},|s|\le(4l_1+3l_2+2)\lfloor\beta_Y\rfloor+d_Y+\lfloor\alpha_Y\rfloor} (\phi_p(w,x))^{(s)} \sum_{i\in\mathbb{N}_0^{D_Y},|i|\le l_2} w^{(i)} \right.$$

$$\sum_{j\in\mathbb{N}_0^{D_X},|j|\le(2l_1+2l_2+1)\lfloor\beta_X\rfloor+\lfloor\alpha_X\rfloor} a_{l_1l_2ksij}x^{(j)}$$

$$-\sum_{l_1=0}^{\mathscr{L}_1}\sum_{l_2=0}^{\mathscr{L}_2}\sum_{0\le k\le 2l_1+2l_2+1}\sum_{s\in\mathbb{N}_0^{d_Y},|s|\le(4l_1+3l_2+2)\lfloor\beta_Y\rfloor+d_Y+\lfloor\alpha_Y\rfloor}\sum_{i\in\mathbb{N}_0^{D_Y},|i|\le l_2+1}\sum_{j\in\mathbb{N}_0^{D_X},|j|\le(2l_1+2l_2+1)\lfloor\beta_X\rfloor+\lfloor\alpha_X\rfloor} a_{l_1l_2ksij}$$

$$\cdot\phi_{mult}\bigg(\phi_{mult}\Big(\phi_{mult}\big(\phi_{mult}\big(\phi_{power}(\phi_{rec}(\phi_\sigma(t));2l_1+2l_2+1),\phi_{power}(\phi_m(t);k)\big)$$

$$\left. ,\phi_{vpower}^{[d_Y]}(\phi_p(w,x);s)\big),\phi_{vpower}^{[D_Y]}(\omega;i)\Big),\phi_{vpower}^{[D_X]}(x;j)\Big)\bigg)\right\|_\infty$$

$$\lesssim (\sigma_{\underline{t}})^{d_Y}\Big(\frac{\log n(\varepsilon_1^{\beta_Y}+\varepsilon_2^{\beta_X})}{\sigma_{\underline{t}}}+\sqrt{\log n}\varepsilon_1^{\alpha_Y}\Big).$$

Therefore, let $\mathscr{L}=\mathscr{L}_1\mathscr{L}_2\cdot(2\mathscr{L}_1+2\mathscr{L}_2+1)\binom{(4\mathscr{L}_1+3\mathscr{L}_2+3)\lfloor\beta_Y\rfloor+2d_Y+\lfloor\alpha_Y\rfloor}{d_Y}\binom{\mathscr{L}_2+D_Y}{D_Y}\binom{(2\mathscr{L}_1+2\mathscr{L}_2+1)\lfloor\beta_X\rfloor+\lfloor\alpha_X\rfloor+D_X}{D_X}$, there exists network $\phi^{[1]}(w,x,t)\in\Phi(H,W,R,B)$ with $H=\Theta(\log^4 n)$, $\|W\|_\infty=\Theta\big(\log^6 n\mathscr{L}\big)$, $R=\Theta\big(\log^8 n\mathscr{L}\big)$, $B=\exp(\Theta(\log^4 n))$ so that (27) holds. By employing same techniques, we can also obtain that there exists a neural network $\phi_j^{[2]}(w,x,t)$ with the same size as $\phi_j^{[1]}(x,t)$ so that (28) holds. Then use (29), similar as the analysis for Case 3, we can obtain Claim 4.

## C.2 Proof of Lemma C.3

Firstly we have the following lemma whose proof follows [5].

**Lemma C.12.** *The following equality holds for all $S(w_t,x,t)$ and $t>0$,*

$$\mathbb{E}_{x\sim\mu_X^*}\mathbb{E}_{y\sim\mu_{Y|x}^*}\mathbb{E}_{w_t\sim\mathcal{N}(m_ty,\sigma_t^2 I_{D_Y})}\big[\|S(w_t,x,t)-\nabla\log p_{t|x}(w_t)\|^2\big]$$

$$=\mathbb{E}_{x\sim\mu_X^*}\mathbb{E}_{y\sim\mu_{Y|x}^*}\mathbb{E}_{w_t\sim\mathcal{N}(m_ty,\sigma_t^2 I_{D_Y})}\bigg[\|S(w_t,x,t)-\frac{m_tY-w_t}{\sigma_t^2}\|^2\bigg]$$

$$+\mathbb{E}_{\mu_X^*}\mathbb{E}_{y\sim\mu_{Y|x}^*}\mathbb{E}_{w_t\sim\mathcal{N}(m_ty,\sigma_t^2 I_{D_Y})}\bigg[\big\|\nabla\log p_{t|x}(w_t)\big\|^2-\bigg\|\frac{m_tY-w_t}{\sigma_t^2}\bigg\|^2\bigg].$$

Then for any $i\in[\mathcal{I}]$, we denote

$$\ell_i(x,y,S)=\int_{t_{i-1}}^{t_i}\mathbb{E}_{w_t\sim\mathcal{N}(m_ty,\sigma_t^2 I_{D_Y})}\bigg[\|S(w_t,x,t)-\frac{m_ty-w_t}{\sigma_t^2}\|^2\bigg]\,\mathrm{d}t,$$

and $\rho_i=\sup_{(x,y)\in\mathcal{M}}\sup_{S\in\mathcal{S}_i}|\ell_i(x,y,S)|$, we have

$$\mathbb{E}_{\mu^*}[\ell_i(x,y,S)]$$

$$=\int_{t_{i-1}}^{t_i}\mathbb{E}_{x\sim\mu_X^*}\mathbb{E}_{y\sim\mu_{Y|x}^*}\mathbb{E}_{w_t\sim\mathcal{N}(m_tY,\sigma_t^2 I_{D_Y})}\bigg[\|S(w_t,x,t)-\frac{m_tY-w_t}{\sigma_t^2}\|^2\bigg]\,\mathrm{d}t$$

$$=\int_{t_{i-1}}^{t_i}\mathbb{E}_{x\sim\mu_X^*}\mathbb{E}_{y\sim\mu_{Y|x}^*}\mathbb{E}_{w_t\sim\mathcal{N}(m_ty,\sigma_t^2 I_{D_Y})}\big[\|S(w_t,x,t)-\nabla\log p_{t|x}(w_t)\|^2\big]\,\mathrm{d}t$$

$$-\int_{t_{i-1}}^{t_i}\mathbb{E}_{\mu_X^*}\mathbb{E}_{y\sim\mu_{Y|x}^*}\mathbb{E}_{w_t\sim\mathcal{N}(m_ty,\sigma_t^2 I_{D_Y})}\bigg[\big\|\nabla\log p_{t|x}(w_t)\big\|^2-\bigg\|\frac{m_tY-w_t}{\sigma_t^2}\bigg\|^2\bigg]\,\mathrm{d}t,$$

and

$$\rho_i \leq \sup_{(x,y)\in\mathcal{M}} \sup_{S\in\mathcal{S}_i} \int_{t_{i-1}}^{t_i} \mathbb{E}_{w_t\sim\mathcal{N}(m_t y,\sigma_t^2 I_{D_Y})} \left[2\|S(w_t,x,t)\|^2 + 2\|\frac{m_t y - w_t}{\sigma_t^2}\|^2\right] \mathrm{d}t$$

$$\lesssim (t_i - t_{i-1}) \cdot \frac{\log n}{t_{i-1} \wedge 1}$$

$$\lesssim (\log n)^2.$$

Let

$$S_i^* \in \arg\min_{S\in\mathcal{S}_i} \mathbb{E}_{x\sim\mu_X^*} \mathbb{E}_{y\sim\mu_{Y|x}^*}[\ell_i(x,y,S)].$$

Consider the function class

$$\overline{G}_i^* = \{g(x,y) = a(\ell_i(x,y,S) - \ell_i(x,y,S_i^*)) \;:\; a\in[0,1], S\in\mathcal{S}_i\},$$

and

$$G_i^* = \{g(x,y) = \ell_i(x,y,S) - \ell_i(x,y,S_i^*) \;:\; S\in\mathcal{S}_i\}.$$

Denote $\|g\|_2 = \sqrt{\mathbb{E}_{\mu^*}[g^2]}$, using standard symmetrization, we can get for any $r > 0$,

$$\mathcal{R}_n(\overline{G}_i^*, r) = \mathbb{E}_{\mu^*\otimes n}\left[\sup_{\substack{g\in\overline{G}_i^* \\ \|g\|_2\leq r}} \left|\frac{1}{n}\sum_{i=1}^n g(X_i,Y_i) - \mathbb{E}_{\mu^*}[g(x,y)]\right|\right] \leq \mathbb{E}_{\mu^*\otimes n}\mathbb{E}_\epsilon\left[\sup_{\substack{g\in\overline{G}^* \\ \|g\|_2\leq r}} \left|\frac{2}{n}\sum_{i=1}^n \epsilon_i g(X_i,Y_i)\right|\right],$$

where $\{\epsilon_i\}_{i=1}^n$ are $n$ i.i.d. copies from Rademacher distribution, i.e. $\mathbb{P}(\epsilon_i = 1) = \mathbb{P}(\epsilon_i = -1) = \frac{1}{2}$.

Define $d_n(g,g') = \sqrt{\frac{1}{n}\sum_{i=1}^n (g(X_i,Y_i) - g'(X_i,Y_i))^2}$, then

$$r_{ni} = \max_{\substack{g,g'\in\overline{G}_i^* \\ \|g\|_2,\|g'\|_2\leq r}} d_n(g,g') \leq 2\rho_i.$$

By equation (3.84) of [6], there exists a constant $c$ such that,

$$\mathbb{E}_{\mu^*\otimes n}[r_{ni}^2] \leq \mathbb{E}_{\mu^*\otimes n}\left[\sup_{\substack{g\in\overline{G}_i^* \\ \|g\|_2\leq r}} \frac{4}{n}\sum_{i=1}^n g^2(X_i,Y_i)\right]$$

$$\leq \mathbb{E}_{\mu^*\otimes n}\left[\sup_{\substack{g\in\overline{G}_i^* \\ \|g\|_2\leq r}} \frac{8}{n}\sum_{i=1}^n (g(X_i,Y_i) - \mathbb{E}_{\mu^*}[g(x,y)])^2\right] + 8r^2$$

$$\leq c(r^2 + \rho_i \overline{R}_n(r, \overline{G}_i^*)).$$

Then for any $g\in G_i^*$ and $a\in(0,1]$, there exists an integer $\kappa\in\mathbb{N}$, such that $\kappa\frac{\varepsilon}{2\rho_i} < a \leq (\kappa+1)\frac{\varepsilon}{2\rho_i}$ and $d_n((\kappa+1)\frac{\varepsilon}{2\rho_i}g, ag) \leq \frac{\varepsilon}{2\rho_i}\rho_i = \frac{\varepsilon}{2}$. Therefore it follows that the $\varepsilon$-covering number of $\overline{G}_i^*$ satisfies that,

$\mathbf{N}(\overline{G}_i^*, d_n, \varepsilon) \leq \mathbf{N}(G_i^*, d_n, \frac{\varepsilon}{2})\frac{2\rho_i}{\varepsilon}$ and $\log \mathbf{N}(\overline{G}_i^*, d_n, \varepsilon) \leq \log \mathbf{N}(G_i^*, d_n, \frac{\varepsilon}{2}) + \log \frac{2\rho_i}{\varepsilon}$. Moreover,

$\forall g = \ell(x, y, S) - \ell(x, y, S_i^*), g' = \ell(x, y, S') - \ell(x, y, S_i^*) \in G_i^*$,
$d_n(g, g')$

$$= \sqrt{\frac{1}{n}\sum_{i=1}^n \Big(\int_{t_{i-1}}^{t_i} \mathbb{E}_{w_t \sim \mathcal{N}(m_t Y_i, \sigma_t^2 I_{D_Y})}\Big[\|S(w_t, X_i, t) - \frac{m_t Y_i - w_t}{\sigma_t^2}\|^2 - \|S'(w_t, X_i, t) - \frac{m_t Y_i - w_t}{\sigma_t^2}\|^2\Big]\, \mathrm{d}t\Big)^2}$$

$$\leq \Big(\frac{1}{n}\sum_{i=1}^n \int_{t_{i-1}}^{t_i} \mathbb{E}_{w_t \sim \mathcal{N}(m_t Y_i, \sigma_t^2 I_{D_Y})}\|S(w_t, X_i, t) - S'(w_t, X_i, t)\|^2\, \mathrm{d}t$$

$$\cdot \int_{t_{i-1}}^{t_i} \mathbb{E}_{w_t \sim \mathcal{N}(m_t Y_i, \sigma_t^2 I_{D_Y})}\|S(w_t, X_i, t) + S'(w_t, X_i, t) - 2\frac{m_t Y_i - w_t}{\sigma_t^2}\|^2\, \mathrm{d}t\Big)^{\frac{1}{2}}$$

$$\lesssim \log n \cdot \Big(\frac{1}{n}\sum_{i=1}^n \int_{t_{i-1}}^{t_i} \mathbb{E}_{w_t \sim \mathcal{N}(m_t Y_i, \sigma_t^2 I_{D_Y})}\|S(w_t, X_i, t) - S'(w_t, X_i, t)\|^2\, \mathrm{d}t\Big)^{\frac{1}{2}}$$

$$\lesssim (\log n)^{\frac{3}{2}} \sup_{\substack{x \in \mathcal{M}_X, w \in [-c\sqrt{\log n}, c\sqrt{\log n}]^{D_Y} \\ t \in [t_{i-1}, t_i]}} \|S(w, x, t) - S'(w, x, t)\| + \frac{1}{n^2}.$$

By standard result for covering number of neural network (e.g., Lemma 3 of [3]), we have for any $\varepsilon \geq \frac{1}{n^2}$

$$\log \mathbf{N}(G_i^*, d_n, \varepsilon) \lesssim S_i H_i \log\big(\varepsilon^{-1} H_i \|W_i\|_\infty (B_i \vee 1)n\big)$$

Then by Dudley entropy integral bound, we have

$$\overline{R}_n(r, \overline{G}_i^*) \lesssim \frac{1}{n^2} + \frac{1}{\sqrt{n}}\mathbb{E}_{\mu^* \otimes n}\Big[\int_{\frac{1}{n^2}}^{r_{ni}} \sqrt{R_i H_i \log\big(\varepsilon^{-1} H_i \|W_i\|_\infty (B_i \vee 1)n\big)}\, \mathrm{d}\varepsilon\Big]$$

$$\leq \frac{1}{n^2} + \frac{1}{\sqrt{n}}\mathbb{E}_{\mu^* \otimes n}\Big[r_{ni}\int_0^1 \sqrt{R_i H_i \log\Big(\varepsilon^{-1}\frac{R_i H_i \|W_i\|_\infty (B_i \vee 1)n}{r_{ni}}\Big)}\, \mathrm{d}\varepsilon\Big]$$

$$\lesssim \sqrt{\frac{R_i H_i \log\big(H_i \|W_i\|_\infty (B_i \vee 1)n\big)}{n}}\mathbb{E}_{\mu^* \otimes n}[r_{ni}] + \sqrt{\frac{R_i H_i}{n}}\mathbb{E}_{\mu^* \otimes n}\Big[r_{ni}\sqrt{\log\frac{2\rho_i}{r_{ni}} + \frac{1}{2}}\Big]$$

$$\lesssim \sqrt{\frac{R_i H_i}{n}}\sqrt{-\frac{1}{2}\mathbb{E}_{\mu^* \otimes n}\Big[r_{ni}^2 \log \mathbb{E}_{\mu^* \otimes n}\Big(\frac{r_{ni}}{2\rho_i}\Big)^2\Big] + \frac{1}{2}\mathbb{E}_{\mu^* \otimes n}[r_{ni}^2] + \log\big(H_i \|W_i\|_\infty (B_i \vee 1)n\big)\mathbb{E}_{\mu^* \otimes n}[r_{ni}^2]},$$

where the last inequality uses that $\sqrt{-\frac{1}{2}y\log y + \frac{1}{2}y}$ is concave and non-decreasing when $y = (\frac{r_{ni}}{2\rho_i})^2 \leq 1$. Then by $\mathbb{E}_{\mu^* \otimes n}[r_{ni}^2] \leq c(r^2 + \rho_i \overline{R}_n(r, \overline{G}^*))$, we have

$$\overline{R}_n(r, \overline{G}^*) \lesssim \sqrt{\frac{R_i H_i}{n}}(r^2 + \rho_i \overline{R}_n(r, \overline{G}^*))^{\frac{1}{2}}\sqrt{\log\frac{\rho_i}{r} + \log\big(H_i \|W_i\|_\infty (B_i \vee 1)n\big)}$$

$$\lesssim \sqrt{\frac{R_i H_i}{n}}(r^2 + (\log n)^2 \overline{R}_n(r, \overline{G}^*))^{\frac{1}{2}}\sqrt{\log\frac{(\log n)^2}{r} + \log\big(H_i \|W_i\|_\infty (B_i \vee 1)n\big)}$$

Choose $\delta_{ni} = c_2 (\log n)^2 \frac{\sqrt{R_i H_i \log(R_i H_i \|W_i\|_\infty (B_i \vee 1)n)}}{\sqrt{n}}$, if $\overline{R}_n(\delta_{ni}, \overline{G}^*) > \delta_{ni}^2/(\log n)^2$, then

$$\overline{R}_n(\delta_{ni}, \overline{G}^*) \lesssim \frac{1}{\sqrt{n}}\log n \cdot \overline{R}_n(\delta_{ni}, \overline{G}^*)^{\frac{1}{2}}\sqrt{R_i H_i \log\big(R_i H_i \|W_i\|_\infty (B_i \vee 1)n\big)}$$

which means

$$\overline{R}_n(\delta_{ni}, \overline{G}^*) \lesssim \frac{(\log n)^2}{n}R_i H_i \log\big(R_i H_i \|W_i\|_\infty (B_i \vee 1)n\big) \lesssim \frac{\delta_{ni}^2}{(\log n)^2}.$$

Therefore for a large enough $c_2$, we have $\overline{R}_n(\delta_{ni}, \overline{G}^*) \leq \delta_{ni}^2/(\log n)^2$. Then denote

$$M_{ni}(S) = \frac{1}{n}\sum_{i=1}^{n}\ell_i(X_i, Y_i, S)$$

and

$$M_i^*(S) = \mathbb{E}_{\mu^*}[\ell_i(X, Y, S)],$$

we have the following lemma,

**Lemma C.13.** *There exist some constants $(c_0, c_1, c_2)$ such that it holds with probability larger than $1 - \frac{1}{n^2}$ that,*

$$\forall S \in \mathcal{S}_i,$$

$$\frac{|M_{ni}(S) - M_{ni}(S_i^*) - M_i^*(S) + M_i^*(S_i^*)|}{\delta_{ni} + \|\ell_i(x, y, S) - \ell_i(x, y, S_i^*)\|_2}$$
$$\leq c_2\delta_{ni}/(\log n)^2.$$

Therefore, we have

$$\|\ell_i(x, y, S) - \ell_i(x, y, S_i^*)\|_2^2$$
$$= \mathbb{E}_{\mu^*}[(\ell_i(x, y, S) - \ell_i(x, y, S_i^*))^2]$$
$$= \mathbb{E}_{\mu^*}\left[\left(\int_{t_{i-1}}^{t_i} \mathbb{E}_{w_t \sim \mathcal{N}(m_t y, \sigma_t^2 I_{D_Y})}\left[\|S(w_t, x, t) - \frac{m_t y - w_t}{\sigma_t^2}\|^2 - \|S_i^*(w_t, x, t) - \frac{m_t y - w_t}{\sigma_t^2}\|^2\right]dt\right)^2\right]$$
$$\leq \mathbb{E}_{\mu^*}\left[\int_{t_{i-1}}^{t_i} \mathbb{E}_{w_t \sim \mathcal{N}(m_t y, \sigma_t^2 I_{D_Y})}\|S(w_t, x, t) - S_i^*(w_t, x, t)\|^2 dt\right.$$
$$\left.\cdot \int_{t_{i-1}}^{t_i} \mathbb{E}_{w_t \sim \mathcal{N}(m_t y, \sigma_t^2 I_{D_Y})}\|S(w_t, x, t) + S_i^*(w_t, x, t) - 2\frac{m_t y - w_t}{\sigma_t^2}\|^2 dt\right]$$
$$\lesssim (\log n)^2 \cdot \mathbb{E}_{\mu^*}\left[\int_{t_{i-1}}^{t_i} \mathbb{E}_{w_t \sim \mathcal{N}(m_t y, \sigma_t^2 I_{D_Y})}\|S(w_t, x, t) - S_i^*(w_t, x, t)\|^2 dt\right]$$
$$\lesssim (\log n)^2 \cdot \mathbb{E}_{\mu^*}\left[\int_{t_{i-1}}^{t_i} \mathbb{E}_{w_t \sim \mathcal{N}(m_t y, \sigma_t^2 I_{D_Y})}\|S(w_t, x, t) - \nabla \log p_{t|x}(w_t)\|^2 dt\right]$$
$$+ (\log n)^2 \cdot \min_{S \in \mathcal{S}_i}\mathbb{E}_{\mu^*}\left[\int_{t_{i-1}}^{t_i} \mathbb{E}_{w_t \sim \mathcal{N}(m_t y, \sigma_t^2 I_{D_Y})}\|S(w_t, x, t) - \nabla \log p_{t|x}(w_t)\|^2 dt\right].$$

Then notice that

$$\mathbb{E}_{\mu^*}\Big[\int_{t_{i-1}}^{t_i}\mathbb{E}_{w_t\sim\mathcal{N}(m_t y,\sigma_t^2 I_{D_Y})}\|\widehat{S}(w_t,x,t)-\nabla\log p_{t|x}(w_t)\|^2\,\mathrm{d}t\Big]$$

$$-\min_{S\in\mathcal{S}_i}\mathbb{E}_{\mu^*}\Big[\int_{t_{i-1}}^{t_i}\mathbb{E}_{w_t\sim\mathcal{N}(m_t y,\sigma_t^2 I_{D_Y})}\|S(w_t,x,t)-\nabla\log p_{t|x}(w_t)\|^2\,\mathrm{d}t\Big]$$

$$=\mathbb{E}_{\mu^*}\Big[\int_{t_{i-1}}^{t_i}\mathbb{E}_{w_t\sim\mathcal{N}(m_t y,\sigma_t^2 I_{D_Y})}\|\widehat{S}(w_t,x,t)-\nabla\log p_{t|x}(w_t)\|^2\,\mathrm{d}t\Big]$$

$$-\mathbb{E}_{\mu^*}\Big[\int_{t_{i-1}}^{t_i}\mathbb{E}_{w_t\sim\mathcal{N}(m_t y,\sigma_t^2 I_{D_Y})}\|S_i^*(w_t,x,t)-\nabla\log p_{t|x}(w_t)\|^2\,\mathrm{d}t\Big]$$

$$=M_i^*(\widehat{S})-M^*(S_i^*)$$

$$\leq M_i^*(\widehat{S})-M^*(S_i^*)+M_{ni}(S_i^*)-M_{ni}(\widehat{S})$$

$$\leq c_2\frac{\delta_{ni}^2}{(\log n)^2}+\frac{\delta_{ni}}{\log n}\cdot\Big(\sqrt{\mathbb{E}_{\mu^*}\Big[\int_{t_{i-1}}^{t_i}\mathbb{E}_{w_t\sim\mathcal{N}(m_t y,\sigma_t^2 I_{D_Y})}\|\widehat{S}(w_t,x,t)-\nabla\log p_{t|x}(w_t)\|^2\,\mathrm{d}t\Big]}$$

$$+\sqrt{\min_{S\in\mathcal{S}_i}\mathbb{E}_{\mu^*}\Big[\int_{t_{i-1}}^{t_i}\mathbb{E}_{w_t\sim\mathcal{N}(m_t y,\sigma_t^2 I_{D_Y})}\|S(w_t,x,t)-\nabla\log p_{t|x}(w_t)\|^2\,\mathrm{d}t\Big]}\Big)$$

So it holds with probability larger than $1-\frac{1}{n}$ that,

$$\mathbb{E}_{\mu^*}\Big[\int_{t_{i-1}}^{t_i}\mathbb{E}_{w_t\sim\mathcal{N}(m_t y,\sigma_t^2 I_{D_Y})}\|\widehat{S}(w_t,x,t)-\nabla\log p_{t|x}(w_t)\|^2\,\mathrm{d}t\Big]$$

$$\lesssim\frac{\delta_{ni}^2}{(\log n)^2}+\min_{S\in\mathcal{S}_i}\mathbb{E}_{\mu^*}\Big[\int_{t_{i-1}}^{t_i}\mathbb{E}_{w_t\sim\mathcal{N}(m_t y,\sigma_t^2 I_{D_Y})}\|S(w_t,x,t)-\nabla\log p_{t|x}(w_t)\|^2\,\mathrm{d}t\Big]$$

$$\lesssim(\log n)^2\frac{R_i H_i\log\left(R_i H_i\|W_i\|_\infty(B_i\vee 1)n\right)}{n}$$

$$+\min_{S\in\mathcal{S}_i}\mathbb{E}_{\mu^*}\Big[\int_{t_{i-1}}^{t_i}\mathbb{E}_{w_t\sim\mathcal{N}(m_t y,\sigma_t^2 I_{D_Y})}\|S(w_t,x,t)-\nabla\log p_{t|x}(w_t)\|^2\,\mathrm{d}t\Big].$$

## C.3 Proof of Technical Results

### C.3.1 Proof of Lemma B.2

For the first statement, denote $\mathrm{vol}_{\mathcal{M}_{Y|x}}$ as the volume measure of $\mathcal{M}_{Y|x}$. Then notice that

$$\mathbb{E}_{y\sim\mu_{Y|x}^*}[g(y)\cdot\mathbf{1}(y\in\mathbb{B}_{\mathcal{M}_{Y|x}}(y^*,r))]=\int g(y)\cdot\mathbf{1}(y\in\mathbb{B}_{\mathcal{M}_{Y|x}}(y^*,r))f(y|x)\,\mathrm{dvol}_{\mathcal{M}_{Y|x}}(y)$$

$$=\int_{U_{Y|x}^{(x^*,y^*)}}g(y)\cdot\mathbf{1}(y\in\mathbb{B}_{\mathcal{M}_{Y|x}}(y^*,r))f(y|x)\,\mathrm{dvol}_{\mathcal{M}_{Y|x}}(y)$$

$$=\int_{\mathbb{B}_{\mathbb{R}^{d_Y}}(0,r_1)}g(G^*(z,x))\mathbf{1}(y\in\mathbb{B}_{\mathcal{M}_{Y|x}}(y^*,r))f(G^*(z,x)|x)\sqrt{\det(J_{G^*(\cdot,x)}(z)^T J_{G^*(\cdot,x)}(z))}\,\mathrm{d}z$$

$$=\int_{\mathbb{B}_{\mathbb{R}^{d_Y}}(0,r_1)}g(G^*(z,x))\mathbf{1}(G^*(z,x)\in\mathbb{B}_{\mathcal{M}_{Y|x}}(y^*,r)))v^*(z|x)\,\mathrm{d}z.$$

For the second statement, since $\widetilde{r}\geq r$ and $\{(x_k^*,y_k^*)\}_{k=1}^K\subset\mathcal{M}$ is a $\frac{r}{2}$-cover of $\mathcal{M}$, for any $(x,y)\in\mathcal{M}$, there exists $k'\in[K]$ so that $\|(x_{k'}^*,y_{k'}^*)-(x,y)\|\leq\frac{r}{2}\leq\frac{\widetilde{r}}{2}$. Therefore, $\sum_{k=1}^K\rho_k(x,y)\geq\rho_{k'}(x,y)=1$

and $\sum_{k=1}^{K} \widetilde{\rho}_k(x,y) = 1$. So based on $\widetilde{r} \leq r_0$ and the first statement, we have

$$
\mathbb{E}_{y \sim \mu_{Y|x}^*}[g(y)] = \sum_{k=1}^{K} \mathbb{E}_{y \sim \mu_{Y|x}^*}[g(y)\widetilde{\rho}_k(x,y)]
$$
$$
= \sum_{k=1}^{K} \mathbb{E}_{y \sim \mu_{Y|x}^*}[g(y)\widetilde{\rho}_k(x,y) \cdot \mathbf{1}(y \in \mathbb{B}_{\mathcal{M}_{Y|x}}(y_k, \widetilde{r}))]
$$
$$
= \sum_{k=1}^{K} \int_{\mathbb{B}_{\mathbb{R}^{d_Y}}(0,r_1)} g(G_{[k]}^*(z,x))\widetilde{\rho}_k(x, G_{[k]}^*(z,x))\mathbf{1}(G_{[k]}^*(z,x) \in \mathbb{B}_{\mathcal{M}_{Y|x}}(y_k, \widetilde{r}))v_{[k]}^*(z|x)\, \mathrm{d}z
$$
$$
= \sum_{k=1}^{K} \int_{\mathbb{B}_{\mathbb{R}^{d_Y}}(0,r_1)} g(G_{[k]}^*(z,x))\widetilde{\rho}_k(x, G_{[k]}^*(z,x))v_{[k]}^*(z|x)\, \mathrm{d}z.
$$

## C.4 Proof of Lemma C.4

Without loss of generality, we assume for any $x \in \mathcal{M}_X$, $\mathcal{M}_{Y|x} \subset \mathbb{B}_{\mathbb{R}^{D_Y}}(0,1)$. Then for any $w \in \mathbb{R}^{D_Y}$,

$$
\|\nabla \log p_{t|x}(w)\| = \left\|\frac{\nabla p_{t|x}(w)}{p_{t|x}(w)}\right\| = \left\|\frac{\mathbb{E}_{y \sim \mu_{Y|x}^*}\left[\exp\left(-\frac{\|w-m_t y\|^2}{2\sigma_t^2}\right) \cdot -\frac{(w-m_t y)}{\sigma_t^2}\right]}{\mathbb{E}_{y \sim \mu_{Y|x}^*}\left[\exp\left(-\frac{\|w-m_t y\|^2}{2\sigma_t^2}\right)\right]}\right\|
$$
$$
\leq \sqrt{\sum_{l=1}^{D_Y}\left(\frac{|w_l|+1}{\sigma_t^2}\right)^2} \leq \frac{\|w\|+\sqrt{D_Y}}{\sigma_t^2}.
$$

Furthermore, by Lemma B.2, it holds with a large enough constant $c$ that

$$
\int_{\mathbb{R}^{D_Y}} \|\nabla \log p_{t|x}(w)\|^2 p_{t|x}(w)\mathbf{1}\left(\mathrm{dist}(w, \mathcal{M}_{Y|x}) \geq c_0 \sigma_{t_{i-1}}\sqrt{\log n}\right)\mathrm{d}w
$$
$$
\leq \int_{\mathbb{R}^{D_Y}} \frac{\|w\|+\sqrt{D_Y}}{\sigma_t^2} \cdot \mathbb{E}_{y \sim \mu_{Y|x}^*}\left[\frac{1}{(2\pi\sigma_t^2)^{\frac{D_Y}{2}}}\exp\left(-\frac{\|w-m_t y\|^2}{2\sigma_t^2}\right)\right]\mathbf{1}\left(\mathrm{dist}(w, \mathcal{M}_{Y|x}) \geq c_0 \sigma_{t_{i-1}}\sqrt{\log n}\right)\mathrm{d}w
$$
$$
= \sum_{k=1}^{K} \int_{\mathbb{R}^{D_Y}} \int_{\mathbb{B}_{\mathbb{R}^{d_Y}}(0,r_1)} \frac{\|w\|+\sqrt{D_Y}}{\sigma_t^2}\frac{1}{(2\pi\sigma_t^2)^{\frac{D_Y}{2}}} \cdot \mathbf{1}\left(\mathrm{dist}(w, \mathcal{M}_{Y|x}) \geq c_0 \sigma_{t_{i-1}}\sqrt{\log n}\right)
$$
$$
\cdot \exp\left(-\frac{\|w-m_t G_{[k]}^*(z,x)\|^2}{2\sigma_t^2}\right) \cdot \widetilde{\rho}_k(x, G_{[k]}^*(z,x))v_{[k]}^*(z|x)\, \mathrm{d}z\, \mathrm{d}w
$$
$$
\leq \sum_{k=1}^{K} \int_{\mathbb{R}^{D_Y}} \int_{\mathbb{B}_{\mathbb{R}^{d_Y}}(0,r_1)} \frac{\|w\|+\sqrt{D_Y}}{\sigma_t^2}\frac{1}{(2\pi\sigma_t^2)^{\frac{D_Y}{2}}} \cdot \mathbf{1}\left(\mathrm{dist}(w, \mathcal{M}_{Y|x}) \geq c_0 \sigma_{t_{i-1}}\sqrt{\log n}, \|w\| \leq c\sqrt{\log n}\right)
$$
$$
\cdot \exp\left(-\frac{\|w-m_t G_{[k]}^*(z,x)\|^2}{2\sigma_t^2}\right) \cdot \widetilde{\rho}_k(x, G_{[k]}^*(z,x))v_{[k]}^*(z|x)\, \mathrm{d}z\, \mathrm{d}w
$$
$$
+ \sum_{k=1}^{K} \int_{\mathbb{R}^{D_Y}} \int_{\mathbb{B}_{\mathbb{R}^{d_Y}}(0,r_1)} \frac{\|w\|+\sqrt{D_Y}}{\sigma_t^2}\frac{1}{(2\pi\sigma_t^2)^{\frac{D_Y}{2}}} \cdot \mathbf{1}\left(\|w\| > c\sqrt{\log n}\right)
$$
$$
\cdot \exp\left(-\frac{\|w-m_t G_{[k]}^*(z,x)\|^2}{2\sigma_t^2}\right) \cdot \widetilde{\rho}_k(x, G_{[k]}^*(z,x))v_{[k]}^*(z|x)\, \mathrm{d}z\, \mathrm{d}w
$$
$$
\leq \sum_{k=1}^{K} \int_{\mathbb{R}^{D_Y}} \int_{\mathbb{B}_{\mathbb{R}^{d_Y}}(0,r_1)} \frac{\|w\|+\sqrt{D_Y}}{\sigma_t^2}\frac{1}{(2\pi\sigma_t^2)^{\frac{D_Y}{2}}} \cdot \mathbf{1}\left(\mathrm{dist}(w, \mathcal{M}_{Y|x}) \geq c_0 \sigma_{t_{i-1}}\sqrt{\log n}, \|w\| \leq c\sqrt{\log n}\right)
$$
$$
\cdot \exp\left(-\frac{\|w-m_t G_{[k]}^*(z,x)\|^2}{2\sigma_t^2}\right) \cdot \widetilde{\rho}_k(x, G_{[k]}^*(z,x))v_{[k]}^*(z|x)\, \mathrm{d}z\, \mathrm{d}w + \frac{1}{n^2}.
$$

Moreover, for large enough constant $c_0$, we have

$$\sum_{k=1}^{K} \int_{\mathbb{R}^{D_Y}} \int_{\mathbb{B}_{\mathbb{R}^{d_Y}}(0,r_1)} \frac{\|w\| + \sqrt{D_Y}}{\sigma_t^2} \frac{1}{(2\pi\sigma_t^2)^{\frac{D_Y}{2}}} \cdot \mathbf{1}\Big(\text{dist}(w, \mathcal{M}_{Y|x}) \geq c_0 \sigma_{t_{i-1}} \sqrt{\log n}, \|w\| \leq c\sqrt{\log n}\Big)$$

$$\cdot \exp\left(-\frac{\|w - m_t G_{[k]}^*(z,x)\|^2}{2\sigma_t^2}\right) \cdot \widetilde{\rho}_k(x, G_{[k]}^*(z,x)) v_{[k]}^*(z|x) \, dz \, dw$$

$$\leq \sum_{k=1}^{K} \frac{c\sqrt{\log n} + \sqrt{D_Y}}{\sigma_t^2} \exp\left(-\frac{c_0^2 \sigma_{t_{i-1}}^2 \log n}{4\sigma_t^2}\right) \int_{\mathbb{R}^{D_Y}} \int_{\mathbb{B}_{\mathbb{R}^{d_Y}}(0,r_1)} v_{[k]}^*(z|x) \cdot \mathbf{1}\Big(\|w\| \leq c\sqrt{\log n}\Big) \, dz \, dw \leq \frac{1}{n^2}$$

Therefore, we have

$$\int \|\nabla \log p_{t|x}(w)\|^2 p_{t|x}(w) \cdot \mathbf{1}\Big(\text{dist}(w, \mathcal{M}_{Y|x}) \geq c_0 \sigma_{t_{i-1}} \sqrt{\log n}\Big) \, dw \leq c_1 \frac{1}{n^2}.$$

Similarly, we can show

$$\int \|S(w,x,t)\|^2 p_{t|x}(w) \cdot \mathbf{1}\Big(\text{dist}(w, \mathcal{M}_{Y|x}) \geq c_0 \sigma_{t_{i-1}} \sqrt{\log n}\Big) \, dx$$

$$\leq \int c^2 \frac{\log n}{\sigma_t^2} p_{t|x}(w) \cdot \mathbf{1}\Big(\text{dist}(x, \mathcal{M}_{Y|x}) \geq c_0 \sigma_{t_{i-1}} \sqrt{\log n}\Big) \, dx \leq c^2 c_1 \frac{1}{n^2}.$$

The first statement is then proved. For the second statement. Denote $\text{Proj}_{\mathcal{M}_{Y|x}}(w)$ as any point inside $\arg\min_{y \in \mathcal{M}_{Y|x}} \|w - y\|$. Then for any $x \in \mathcal{M}_X$, $w \in \mathbb{R}^{D_Y}$ with $\text{dist}(w, \mathcal{M}_{Y|x}) \leq c_0 \sigma_{t_{i-1}} \sqrt{\log n}$, we denote $\omega = (x, \text{Proj}_{\mathcal{M}_{Y|x}}(w))$, and use the notation $G^\omega : \mathbb{B}_{\mathbb{R}^{d_Y}}(0,r_1) \times U_X^\omega \to U_Y^\omega$ and $v^\omega$ in Assumption D. Then there exists a constant $L_1$ so that for any $z \in \mathbb{B}_{\mathbb{R}^{d_Y}}(0,r_1)$,

$$\|G^\omega(z,x) - G^\omega(0,x)\| \leq L_1 \|z\|,$$

and by Lemma B.2, we have

$$(2\pi\sigma_t^2)^{\frac{D}{2}} p_{t|x}(w) = \mathbb{E}_{y \sim \mu_{Y|x}^*}\left[\exp\left(-\frac{\|w - m_t y\|^2}{2\sigma_t^2}\right)\right]$$

$$\geq \mathbb{E}_{y \sim \mu_{Y|x}^*}\left[\exp\left(-\frac{\|w - m_t y\|^2}{2\sigma_t^2}\right) \cdot \mathbf{1}\Big(y \in \mathbb{B}_{\mathcal{M}_{Y|x}}(\text{Proj}_{\mathcal{M}_{Y|x}}(w), r_0 \sigma_t)\Big)\right]$$

$$= \int_{\mathbb{B}_{\mathbb{R}^{d_Y}}(0,r_1)} \exp\left(-\frac{\|w - m_t G^\omega(z,x)\|^2}{2\sigma_t^2}\right) \cdot \mathbf{1}\Big(G^\omega(z,x) \in \mathbb{B}_{\mathcal{M}_{Y|x}}(G^\omega(0,x), r_0 \sigma_t)\Big) v^\omega(z|x) \, dz$$

$$\geq \int_{\mathbb{B}_{\mathbb{R}^{d_Y}}(0,r_1)} \exp\left(-\frac{(c_0 \sigma_{t_{i-1}} \sqrt{\log n} + r_0 \sigma_t + (1 - m_t))^2}{2\sigma_t}\right) \cdot \mathbf{1}(G^\omega(z,x) \in \mathbb{B}_{\mathcal{M}_{Y|x}}(G^\omega(0,x), r_0 \sigma_t)) v^\omega(z|x) \, dz$$

$$\geq \int_{\mathbb{B}_{\mathbb{R}^{d_Y}}(0, r_1 \wedge \frac{r_0 \sigma_t}{L_1})} \exp\left(-\frac{(c_0 \sigma_{t_{i-1}} \sqrt{\log n} + r_0 \sigma_t + (1 - m_t))^2}{2\sigma_t}\right) v^\omega(z|x) \, dz$$

$$\geq n^{-c_2}.$$

Therefore, for any $x \in \mathcal{M}_X$ and $w \in \mathbb{R}^{D_Y}$ with $\text{dist}(w, \mathcal{M}_{Y|x}) \leq c_0 \sigma_{t_{i-1}} \sqrt{\log n}$,

$$\|\nabla \log p_{t|x}(w)\| = \left\| \frac{\mathbb{E}_{y \sim \mu_{Y|x}^*}\left[\exp\left(-\frac{\|w - m_t y\|^2}{2\sigma_t^2}\right) \cdot -\frac{(w - m_t y)}{\sigma_t^2}\right]}{\mathbb{E}_{y \sim \mu_{Y|x}^*}\left[\exp\left(-\frac{\|w - m_t y\|^2}{2\sigma_t^2}\right)\right]} \right\|$$

$$\leq \left\| \frac{\mathbb{E}_{y \sim \mu_{Y|x}^*}\left[\exp\left(-\frac{\|w - m_t y\|^2}{2\sigma_t^2}\right) \cdot -\frac{(w - m_t y)}{\sigma_t^2} \cdot \mathbf{1}(\|w - m_t y\| \leq c_3 \sigma_t \sqrt{\log n})\right]}{\mathbb{E}_{y \sim \mu_{Y|x}^*}\left[\exp\left(-\frac{\|w - m_t y\|^2}{2\sigma_t^2}\right)\right]} \right\|$$

$$+ \left\| \frac{\mathbb{E}_{y \sim \mu_{Y|x}^*}\left[\exp\left(-\frac{\|w - m_t y\|^2}{2\sigma_t^2}\right) \cdot -\frac{(w - m_t y)}{\sigma_t^2} \cdot \mathbf{1}(\|w - m_t y\| > c_3 \sigma_t \sqrt{\log n})\right]}{\mathbb{E}_{y \sim \mu_{Y|x}^*}\left[\exp\left(-\frac{\|w - m_t y\|^2}{2\sigma_t^2}\right)\right]} \right\|,$$

when $c_3$ is large enough, we have

$$\left\| \frac{\mathbb{E}_{y \sim \mu^*_{Y|x}} \left[ \exp\left(-\frac{\|w - m_t y\|^2}{2\sigma_t^2}\right) \cdot -\frac{(w - m_t y)}{\sigma_t^2} \cdot \mathbf{1}(\|w - m_t y\| > c_3 \sigma_t \sqrt{\log n}) \right]}{\mathbb{E}_{y \sim \mu^*_{Y|x}} \left[ \exp\left(-\frac{\|w - m_t y\|^2}{2\sigma_t^2}\right) \right]} \right\|$$

$$\leq n^{c_2} \left\| \mathbb{E}_{y \sim \mu^*_{Y|x}} \left[ \exp\left(-\frac{\|w - m_t y\|^2}{2\sigma_t^2}\right) \cdot -\frac{(w - m_t y)}{\sigma_t^2} \cdot \mathbf{1}(\|w - m_t y\| > c_3 \sigma_t \sqrt{\log n}) \right] \right\| \lesssim \frac{1}{n},$$

so

$$\|\nabla \log p_{t|x}(w)\| \leq \left\| \frac{\mathbb{E}_{y \sim \mu^*_{Y|x}} \left[ \exp\left(-\frac{\|w - m_t y\|^2}{2\sigma_t^2}\right) \cdot -\frac{(w - m_t y)}{\sigma_t^2} \cdot \mathbf{1}(\|w - m_t y\| \leq c_3 \sigma_t \sqrt{\log n}) \right]}{\mathbb{E}_{y \sim \mu^*_{Y|x}} \left[ \exp\left(-\frac{\|w - m_t y\|^2}{2\sigma_t^2}\right) \cdot \mathbf{1}(\|w - m_t y\| \leq c_3 \sigma_t \sqrt{\log n}) \right]} \right\| + \frac{1}{n}$$

$$\lesssim \frac{\sqrt{\log n}}{\sigma_t} \asymp \frac{\sqrt{\log n}}{\sigma_{t_{i-1}}}.$$

We can then get the desired statement by combining all pieces.

### C.4.1 Proof of Lemma C.5

By Assumption D, there exists constant $L, L_1, L_1'$ that for any $\omega = (x^*, y^*) \in \mathcal{M}$ and any $x, x' \in U_X^\omega$, $z, z' \in \mathbb{B}_{\mathbb{R}^{d_Y}}(0, r_1)$,

$$\|G^\omega(z, x) - G^\omega(z', x')\| \leq L_1(\|z - z'\| + \|x - x'\|^{\beta_X \wedge 1}),$$

$$\|G^\omega(z, x) - G^\omega(z', x) - J_{G^\omega(\cdot, x)}(z')(z - z')\| \leq L_1 \|z - z'\|^2,$$

and

$$\|z - z'\| = \|Q_x^\omega(G^\omega(z, x)) - Q_x^\omega(G^\omega(z', x))\| \leq L\|G^\omega(z, x) - G^\omega(z', x)\|.$$

Therefore, for any $z' \in \mathbb{B}_{\mathbb{R}^{d_Y}}(0, r_1)$ and unit vector $h \in \mathbb{R}^{d_Y}$, there exists a number $a_0$ so that for any $0 < a \leq a_0$ and $z = z' + ah \in \mathbb{B}_{\mathbb{R}^{d_Y}}(0, r_1)$, it holds that

$$a\|J_{G^\omega(\cdot, x)}(z')h\| \geq \|G^\omega(z, x) - G^\omega(z', x)\| - L_1\|z - z'\|^2$$

$$\geq \frac{1}{L}\|z - z'\| - L_1\|z - z'\|^2$$

$$= \frac{a}{L} - L_1 a^2.$$

By setting $a \to 0$, we have for any unit vector $h$

$$\|J_{G^\omega(\cdot, x)}(z')h\| \geq \frac{1}{L}.$$

The proof for the first statement is then completed. For the second and third statement, without loss of generality, we assume $L, L_1 \geq 1$ and $r_0, r \leq 1$, then we choose

$$r = \left(\frac{r_0}{8L_1^2 L}\right)^{\frac{1}{\beta_X \wedge 1}} \wedge \left(\frac{r_1}{4L_1 L}\right)^{\frac{1}{\beta_X \wedge 1}} \wedge \frac{1}{3}\left(\frac{r_0}{4L_1}\right)^{\frac{1}{\beta_X \wedge 1}},$$

and $\bar{r} = Lr + LL_1 r^{\beta_X \wedge 1}$. For the second statement, for any $k \in [K]$, if $\|x - x_k^*\| \leq r$ and $\|y - y_k^*\| \leq r$, we have

$$\begin{aligned}
\|Q_x^{(x_k^*, y_k^*)}(y)\| &= \|Q_x^{(x_k^*, y_k^*)}(y) - Q_x^{(x_k^*, y_k^*)}(G^{(x_k^*, y_k^*)}(0, x))\| \\
&\leq L\|y - G^{(x_k^*, y_k^*)}(0, x)\| \\
&\leq L\|y - y_k^*\| + L\|G^{(x_k^*, y_k^*)}(0, x_k^*) - G^{(x_k^*, y_k^*)}(0, x)\| \\
&\leq Lr + LL_1 r^{\beta_X \wedge 1} = \bar{r} \leq \frac{r_1}{2} \wedge \frac{r_0}{4L_1}.
\end{aligned} \tag{30}$$

For the last statement, notice that for any $(x, y) \in \mathcal{M}$, there exists $k \in [K]$, so that $\|x - x_k^*\| \le r$, and $\|y - y_k^*\| \le r$, and there exists $x^* \in \mathcal{N}_{\varepsilon_2}^X$ so that $\|x - x^*\| \le \varepsilon_2 \le r$. So we have

$$\|x^* - x_k^*\| \le \|x - x^*\| + \|x - x_k^*\| \le 2r,$$

which implies $k \in \mathcal{K}_{x^*}$. Then by equation (30), there exists $z^* \in \mathcal{N}_{\varepsilon_1}^Z$ so that $\|z^* - Q_x^{(x_k^*, y_k^*)}(y)\| \le \varepsilon_1$, and thus

$$\|G^{(x_k^*, y_k^*)}(z^*, x) - y\| = \|G^{(x_k^*, y_k^*)}(z^*, x) - G^{(x_k^*, y_k^*)}(Q_x^{(x_k^*, y_k^*)}(y), x)\| \le L_1 \varepsilon_1.$$

Proof of the second and third statement is then completed. For the last statement, if $z \in \mathbb{B}_{\mathbb{R}^{d_Y}}(0, \bar{r})$ and

$$\|x - x_k^*\| \le \|x - x^*\| + \|x^* - x_k^*\| \le \varepsilon_2 + 2r \le 3r,$$

then

$$
\begin{aligned}
\|G^{(x_k^*, y_k^*)}(z, x) - y_k^*\| &= \|G^{(x_k^*, y_k^*)}(z, x) - G^{(x_k^*, y_k^*)}(0, x_k^*)\| \\
&\le L_1 \|z\| + L_1 \|x - x_k^*\|^{\beta_X \wedge 1} \\
&\le L_1 \bar{r} + L_1 (3r)^{\beta_Y \wedge 1} \\
&\le \frac{r_0}{2}.
\end{aligned}
$$

### C.4.2   Proof of Lemma C.6

Consider $x \in \mathcal{M}_X$ and $w \in \mathbb{R}^{D_Y}$ so that $\operatorname{dist}(w, \mathcal{M}_{Y|x}) \le c_0 \sigma_t \sqrt{\log n}$. Then there exists $y \in \mathcal{M}_{Y|x}$, $j_2 \in [J_2]$, $j_1 \in [J_1]$, $k \in \mathcal{K}_{\widetilde{x}_{j_2}}$, so that $\|\widetilde{x}_{j_2} - x\| \le \varepsilon_2$, $\|G_{[k]}^*(\widetilde{z}_{j_1}, x) - y\| \le L_1 \varepsilon_1$, and $\|w - y\| \le c_0 \sigma_t \sqrt{\log n}$. So we have

$$\|w - G_{[k]}^*(\widetilde{z}_{j_1}, x)\| \le L_1 \varepsilon_1 + c_0 \sigma_t \sqrt{\log n}.$$

Then notice that $G_{[k]}^*(z, x)$ is $C_{D_Y}^{\beta_Y, \beta_X}$-smooth, let

$$G_{[k]}^* |_{\widetilde{x}}(z, x) = \sum_{\substack{j \in \mathbb{N}_0^{d_X} \\ |j| < \beta_X}} \frac{G_{[k](0,j)}^*(z, \widetilde{x})}{j!} (x - \widetilde{x})^j,$$

we have

$$\|G_{[k]}^* |_{\widetilde{x}_{j_2}}(z, x) - G_{[k]}^*(z, x)\| \le L_1 \|\widetilde{x}_{j_2} - x\|^{\beta_X} \le L_1 \varepsilon_2^{\beta_X},$$

and therefore,

$$\|w - G_{[k]}^* |_{\widetilde{x}_{j_2}}(\widetilde{z}_{j_1}, x)\| \le L_1 \varepsilon_1 + c_0 \sigma_t \sqrt{\log n} + L_1 \varepsilon_2^{\beta_X} \le 2L_1 \varepsilon_1 + c_0 \sigma_t \sqrt{\log n}.$$

Then define

$$\widetilde{\rho}(x) = \begin{cases} 1 & |x| < 1 \\ 0 & |x| > 2 \\ 2 - |x| & 1 < |x| \le 2 \end{cases}$$

For any $j_1 \in [J_1]$ and $j_2 \in [J_2]$, $k \in \mathcal{K}_{\widetilde{x}_{j_2}}$, define

$$\widetilde{\rho}_{kj_1 j_2}(x, w) = \widetilde{\rho}\left(\frac{\|x - \widetilde{x}_{j_2}\|^2}{\varepsilon_2^2}\right) \widetilde{\rho}\left(\frac{\|w - G_{[k]}^* |_{\widetilde{x}_{j_2}}(\widetilde{z}_{j_1}, x)\|^2}{(c_0 \sigma_t \sqrt{\log n} + 2L_1 \varepsilon_1)^2}\right),$$

$$\rho_{kj_1 j_2}(x, w) = \frac{\widetilde{\rho}_{kj_1 j_2}(x, w)}{\sum_{j_1 \in [J_1]} \sum_{j_2 \in [J_2]} \sum_{k \in \mathcal{K}_{\widetilde{x}_{j_2}}} \widetilde{\rho}_{kj_1 j_2}(x, w)}.$$

Then if $\rho_{kj_1j_2}(x, w) \neq 0$ and if $\rho_{kj'_1j'_2}(x, w) \neq 0$, we have

$$\|\widetilde{x}_{j_2} - \widetilde{x}_{j'_2}\| \leq \|x - \widetilde{x}_{j'_2}\| + \|x - \widetilde{x}_{j_2}\| \leq 2\sqrt{2}\varepsilon_2,$$

and

$$\begin{aligned}
\|\widetilde{z}_{j_1} - \widetilde{z}_{j'_1}\| &\leq L\|G^*_{[k]}(\widetilde{z}_{j_1}, x) - G^*_{[k]}(\widetilde{z}_{j'_1}, x)\| \\
&\leq L\|G^*_{[k]}|_{\widetilde{x}_{j_2}}(\widetilde{z}_{j_1}, x) - G^*_{[k]}|_{\widetilde{x}_{j'_2}}(\widetilde{z}_{j'_1}, x)\| + 2LL_1\varepsilon_2^{\beta_X} \\
&\leq L\|w - G^*_{[k]}|_{\widetilde{x}_{j_2}}(\widetilde{z}_{j_1}, x)\| + L\|w - G^*_{[k]}|_{\widetilde{x}_{j'_2}}(\widetilde{z}_{j'_1}, x)\| + 2LL_1\varepsilon_2^{\beta_X} \\
&\leq 2\sqrt{2}Lc_0\sigma_t\sqrt{\log n} + 2L_1\varepsilon_1 + 2LL_1\varepsilon_2^{\beta_X} \\
&\leq (2\sqrt{2}Lc_0 + 2L_1 + 2LL_1)\varepsilon_1.
\end{aligned}$$

So for any $x \in \mathcal{M}_X$ and $w \in \mathbb{R}^{D_Y}$, there are only constant-order number of $k, j_1, j_2$ so that $\rho_{kj_1j_2}(x, w) \neq 0$. Then we can write

$$\nabla \log p_{t|x}(w) = \sum_{j_1 \in [J_1]} \sum_{j_2 \in [J_2]} \sum_{k \in \mathcal{K}_{\widetilde{x}_{j_2}}} \nabla \log p_{t|x}(w) \cdot \rho_{kj_1j_2}(x, w).$$

By Lemma C.9 and C.10, and the fact that $G^*_{[k]}|_{\widetilde{x}_{j_2}}(\widetilde{z}_{j_1}, x)$ is polynomial function of $x$, we construct the following neural networks:

1. For $j_1 \in [J_1]$, $j_2 \in [J_2]$ and $k \in \mathcal{K}_{\widetilde{x}_{j_2}}$, we approximate $\widetilde{\rho}_{kj_1j_2}(x, w)$ by $\phi_{\widetilde{\rho}_{kj_1j_2}}(x, w) \in \Phi(H, W, R, B)$ with $H = \Theta(\log n)$, $\|W\|_\infty = \Theta(\log n)$, $R = \Theta(\log n)$ and $B = \exp(\Theta(\log n))$.

2. We approximate $\frac{1}{x}$ by $\phi_{rec}(x) \in \Phi(H, W, R, B)$ with $H = \Theta(\log^2 n)$, $\|W\|_\infty = \Theta(\log^3 n)$, $R = \Theta(\log^4 n)$ and $B = \exp(\Theta(\log^2 n))$.

3. We approximate $x \cdot y$ by $\phi_{mult}(x, y) \in \Phi(H, W, R, B)$ with $H = \Theta(\log n)$, $\|W\|_\infty = \Theta(\log n)$, $R = \Theta(\log n)$ and $B = \exp(\Theta(\log n))$.

We have for any $x \in \mathcal{M}_X$ and $w \in \mathbb{R}^{D_Y}$ with $\text{dist}(w, \mathcal{M}_{Y|x}) \leq c_0\sigma_t\sqrt{\log n}$,

$$\left\| \sum_{j_1 \in [J_1]} \sum_{j_2 \in [J_2]} \sum_{k \in \mathcal{K}_{\widetilde{x}_{j_2}}} \nabla \log p_{t|x}(w) \cdot \rho_{kj_1j_2}(x, w) \right.$$

$$\left. - \phi_{muti} \left( \sum_{j_1 \in [J_1]} \sum_{j_2 \in [J_2]} \sum_{k \in \mathcal{K}_{\widetilde{x}_{j_2}}} \phi_{muti} \left( \phi^*_{kj_1j_2}(w, x, t), \phi_{\widetilde{\rho}_{kj_1j_2}}(x, w) \right), \phi_{rec} \left( \sum_{j_1 \in [J_1]} \sum_{j_2 \in [J_2]} \sum_{k \in \mathcal{K}_{\widetilde{x}_{j_2}}} \phi_{\widetilde{\rho}_{kj_1j_2}}(x, w) \right) \right) \right\|_\infty$$

$$\leq \left\| \sum_{j_1 \in [J_1]} \sum_{j_2 \in [J_2]} \sum_{k \in \mathcal{K}_{\widetilde{x}_{j_2}}} \nabla \log p_{t|x}(w) \cdot \rho_{kj_1j_2}(x, w) - \sum_{j_1 \in [J_1]} \sum_{j_2 \in [J_2]} \sum_{k \in \mathcal{K}_{\widetilde{x}_{j_2}}} \phi^*_{kj_1j_2}(w, x, t) \cdot \rho_{kj_1j_2}(x, w) \right\|_\infty$$

$$+ \left\| \sum_{j_1 \in [J_1]} \sum_{j_2 \in [J_2]} \sum_{k \in \mathcal{K}_{\widetilde{x}_{j_2}}} \phi^*_{kj_1j_2}(w, x, t) \cdot \widetilde{\rho}_{kj_1j_2} \left( \sum_{j_1 \in [J_1]} \sum_{j_2 \in [J_2]} \sum_{k \in \mathcal{K}_{\widetilde{x}_{j_2}}} \widetilde{\rho}_{kj_1j_2}(x, w) \right)^{-1} \right.$$

$$\left. - \sum_{j_1 \in [J_1]} \sum_{j_2 \in [J_2]} \sum_{k \in \mathcal{K}_{\widetilde{x}_{j_2}}} \phi^*_{kj_1j_2}(w, x, t) \cdot \widetilde{\rho}_{kj_1j_2}(x, w) \cdot \phi_{rec} \left( \sum_{j_1 \in [J_1]} \sum_{j_2 \in [J_2]} \sum_{k \in \mathcal{K}_{\widetilde{x}_{j_2}}} \widetilde{\rho}_{kj_1j_2}(x, w) \right) \right\|_\infty$$

$$+ \left\| \sum_{j_1 \in [J_1]} \sum_{j_2 \in [J_2]} \sum_{k \in \mathcal{K}_{\widetilde{x}_{j_2}}} \phi^*_{kj_1j_2}(w, x, t) \cdot \widetilde{\rho}_{kj_1j_2}(x, w) \cdot \phi_{rec} \left( \sum_{j_1 \in [J_1]} \sum_{j_2 \in [J_2]} \sum_{k \in \mathcal{K}_{\widetilde{x}_{j_2}}} \widetilde{\rho}_{kj_1j_2}(x, w) \right) \right.$$

$$\left. - \sum_{j_1 \in [J_1]} \sum_{j_2 \in [J_2]} \sum_{k \in \mathcal{K}_{\widetilde{x}_{j_2}}} \phi^*_{kj_1j_2}(w, x, t) \cdot \phi_{\widetilde{\rho}_{kj_1j_2}}(x, w) \cdot \phi_{rec} \left( \sum_{j_1 \in [J_1]} \sum_{j_2 \in [J_2]} \sum_{k \in \mathcal{K}_{\widetilde{x}_{j_2}}} \widetilde{\rho}_{kj_1j_2}(x, w) \right) \right\|_\infty$$

$$+ \left\| \sum_{j_1 \in [J_1]} \sum_{j_2 \in [J_2]} \sum_{k \in \mathcal{K}_{\widetilde{x}_{j_2}}} \phi^*_{kj_1j_2}(w, x, t) \cdot \phi_{\widetilde{\rho}_{kj_1j_2}}(x, w) \cdot \phi_{rec} \left( \sum_{j_1 \in [J_1]} \sum_{j_2 \in [J_2]} \sum_{k \in \mathcal{K}_{\widetilde{x}_{j_2}}} \widetilde{\rho}_{kj_1j_2}(x, w) \right) \right.$$

$$\left. - \phi_{muti} \left( \sum_{j_1 \in [J_1]} \sum_{j_2 \in [J_2]} \sum_{k \in \mathcal{K}_{\widetilde{x}_{j_2}}} \phi_{muti} \left( \phi^*_{kj_1j_2}(w, x, t), \phi_{\widetilde{\rho}_{kj_1j_2}}(x, w) \right), \phi_{rec} \left( \sum_{j_1 \in [J_1]} \sum_{j_2 \in [J_2]} \sum_{k \in \mathcal{K}_{\widetilde{x}_{j_2}}} \phi_{\widetilde{\rho}_{kj_1j_2}}(x, w) \right) \right) \right\|_\infty$$

$$\lesssim \varepsilon + \frac{1}{n}.$$

Finally, by concatenation and parallelization of neural networks (see for example, Lemmas F.1-F.3 in [2]), there exists $\phi_{score}(x) \in \Phi(H_1, W_1, S_1, B_1, \Theta(\frac{\sqrt{\log n}}{\sigma_t}))$ with $H_1 = \Theta(H + \log^2 n)$, $\|W_1\|_\infty = \Theta(J_1 J_2(\|W\|_\infty + \log n) + \log^3 n)$, $S_1 = \Theta(J_1 J_2(S + \log n) + \log^4 n)$ and $B_1 = \exp(\Theta(\log^2 n)) \vee B$ so that

$$\phi_{score}(x) = \max \left( - c_2 \frac{\sqrt{\log n}}{\sigma_t}, \min \left( c_2 \frac{\sqrt{\log n}}{\sigma_t}, \right. \right.$$

$$\phi_{muti} \left( \sum_{j_1 \in [J_1]} \sum_{j_2 \in [J_2]} \sum_{k \in \mathcal{K}_{\widetilde{x}_{j_2}}} \phi_{muti} \left( \phi^*_{kj_1j_2}(w, x, t), \phi_{\widetilde{\rho}_{kj_1j_2}}(x, w) \right), \phi_{rec} \left( \sum_{j_1 \in [J_1]} \sum_{j_2 \in [J_2]} \sum_{k \in \mathcal{K}_{\widetilde{x}_{j_2}}} \phi_{\widetilde{\rho}_{kj_1j_2}}(x, w) \right) \right) \right) \right),$$

where the max and min functions are applied elementwise to vectors. The result is then follows from the fact that $\|\nabla \log p_{t|x}(w)\|_\infty \leq c_2 \frac{\sqrt{\log n}}{\sigma_t}$ when $x \in \mathcal{M}_X$ and $\text{dist}(w, \mathcal{M}_{Y|x}) \leq c_0 \sigma_t \sqrt{\log n}$.

### C.4.3 Proof of Lemma C.11

Recall

$$G_{[k]}^*|_{(\widetilde{z},\widetilde{x})}(z,x) = \sum_{(j_1,j_2)\in\mathcal{J}_{\beta_Y,\beta_X}^{d_Y,D_X}} \frac{G_{[k](j_1,j_2)}^*(\widetilde{z},\widetilde{x})}{j_1!j_2!}(z-\widetilde{z})^{j_1}(x-\widetilde{x})^{j_2},$$

For any $l \in [d_Y]$, denotes $\mathbf{1}_l$ as the $d_Y$-dimensional vector in which the $l$th element being 1 and other elements being 0, by $\beta_Y \geq 2$, we have

$$\left\|\frac{\partial G_{[k]}^*|_{(\widetilde{z},\widetilde{x})}(z,x)}{\partial z_l} - G_{[k](\mathbf{1}_l,0)}^*(\widetilde{z},\widetilde{x})\right\| \lesssim \|z-\widetilde{z}\| + \|x-\widetilde{x}\|^{\beta_X/2}.$$

Then notice that for any $z \in \mathbb{B}_{\mathbb{R}^{d_Y}}(0,r_1)$,

$$\lambda_{\min}(J_{G_{[k]}^*(\cdot,x)}(z)^T J_{G_{[k]}^*(\cdot,x)}(z)) \geq \frac{1}{L},$$

there exists a constant $r_2$ so that for any $z \in \mathbb{B}_{\mathbb{R}^{d_Y}}(\widetilde{z},r_2)$ and $x \in \mathbb{B}_{\mathcal{M}_X}(\widetilde{x},(\sigma_{\underline{t}}\sqrt{\log n})^{\frac{1}{\beta_X}})$,

$$\lambda_{\min}(J_{G_{[k]}^*|_{(\widetilde{z},\widetilde{x})}(\cdot,x)}(z)^T J_{G_{[k]}^*|_{(\widetilde{z},\widetilde{x})}(\cdot,x)}(z)) \geq \frac{1}{2L}.$$

Let $h(w,x,z) = (J_{G_{[k]}^*|_{(\widetilde{z},\widetilde{x})}(\cdot,x)}(z))^T(w - G_{[k]}^*|_{(\widetilde{z},\widetilde{x})}(z,x))$. Then we can write the Jacobian of $h$ with respect to $z$ as

$$J_{h(w,x,\cdot)}(z) = -J_{G_{[k]}^*|_{(\widetilde{z},\widetilde{x})}(\cdot,x)}(z)^T J_{G_{[k]}^*|_{(\widetilde{z},\widetilde{x})}(\cdot,x)}(z) + \sum_{l=1}^{D_Y}(w_l - G_{[k],l}^*|_{(\widetilde{z},\widetilde{x})}(z,x))\mathcal{H}_l(z,x),$$

where $G_{[k]}^*|_{(\widetilde{z},\widetilde{x})}(z,x) = (G_{[k],1}^*|_{(\widetilde{z},\widetilde{x})}(z,x),\cdots,G_{[k],D_Y}^*|_{(\widetilde{z},\widetilde{x})}(z,x))$ and $\mathcal{H}_l(z,x)$ denotes the Hessian matrix of $G_{[k],l}^*|_{(\widetilde{z},\widetilde{x})}(\cdot,x)$ at $z$. Then denote

$$g(w,x,z) = z - (J_{h(w,x,\cdot)}(z))^{-1}h(w,x,z).$$

Note that for any $w \in \mathbb{R}^{D_Y}$ satisfying $\|w - G_{[k]}^*(\widetilde{z},x)\| \lesssim (\sigma_{\underline{t}} \vee n^{-\frac{1}{2\alpha_Y+d_Y+d_X\frac{\alpha_Y}{\alpha_X}}})\sqrt{\log n}$ and $x \in \mathbb{B}_{\mathcal{M}_X}(\widetilde{x},(\sigma_{\underline{t}}\sqrt{\log n})^{\frac{1}{\beta_X}})$, we have

$$\|w - G_{[k]}^*|_{(\widetilde{z},\widetilde{x})}(\widetilde{z},x)\| \leq \|w - G_{[k]}^*(\widetilde{z},x)\| + C\|x-\widetilde{x}\|^{\beta_X} \lesssim (\sigma_{\underline{t}} \vee n^{-\frac{1}{2\alpha_Y+d_Y+d_X\frac{\alpha_Y}{\alpha_X}}})\sqrt{\log n}.$$

So by $\lambda_{\min}(J_{G_{[k]}^*|_{(\widetilde{z},\widetilde{x})}(\cdot,x)}(z)^T J_{G_{[k]}^*|_{(\widetilde{z},\widetilde{x})}(\cdot,x)}(z)) \geq \frac{1}{2L}$ and the $C^\infty$-smoothness of $h$, when $n$ is large enough, the exist positive constants $r_3, L_2, L_3$ so that when $\|z-\widetilde{z}\| \leq r_3$ and $\|x-\widetilde{x}\| \leq (\sigma_{\underline{t}}\sqrt{\log n})^{\frac{1}{\beta_X}}$,

$$-L_2 I_{d_Y} \preccurlyeq J_{h(w,x,\cdot)}(z) \preccurlyeq -L_3 I_{d_Y}.$$

Furthermore, by

$$\|h(w,x,\widetilde{z})\| \lesssim (\sigma_{\underline{t}} \vee n^{-\frac{1}{2\alpha_Y+d_Y+d_X\frac{\alpha_Y}{\alpha_X}}})\sqrt{\log n}.$$

we have

$$\|g(w,x,\widetilde{z}) - \widetilde{z}\| = \mathcal{O}(\|h(w,x,\widetilde{z})\|) \lesssim (\sigma_{\underline{t}} \vee n^{-\frac{1}{2\alpha_Y+d_Y+d_X\frac{\alpha_Y}{\alpha_X}}})\sqrt{\log n},$$

and

$$
\begin{aligned}
\|h(w,x,g(w,x,\widetilde{z}))\| &= \left\|h\left(w,x,\widetilde{z}-(J_{h(w,x,\cdot)}(\widetilde{z}))^{-1}h(w,x,\widetilde{z})\right)\right\| \\
&= \left\|h(w,x,\widetilde{z}) - J_{h(w,x,\cdot)}(\widetilde{z})(J_{h(w,x,\cdot)}(\widetilde{z}))^{-1}h(w,x,\widetilde{z})\right\| + \mathcal{O}(\|h(w,x,\widetilde{z})\|^2) \\
&\lesssim \left((\sigma_{\underline{t}} \vee n^{-\frac{1}{2\alpha_Y+d_Y+d_X\frac{\alpha_Y}{\alpha_X}}})\sqrt{\log n}\right)^2.
\end{aligned}
$$

Similarly, define

$$\overline{g}(w,x) = \underbrace{g(w,x,g(w,x,\circ g(w,x,\circ \cdots \circ g(w,x,g(w,x,g(}_{\lceil \log_2(2\beta_Y)\rceil}w,x,\widetilde{z})))))),$$

we can obtain

$$\|\overline{g}(w,x) - \widetilde{z}\| \lesssim (\sigma_{\underline{t}} \vee n^{-\frac{1}{2\alpha_Y + d_Y + d_X \frac{\alpha_Y}{\alpha_X}}})\sqrt{\log n},$$

and

$$\|h(w,x,\overline{g}(w,x))\| \lesssim \left((\sigma_{\underline{t}} \vee n^{-\frac{1}{2\alpha_Y + d_Y + d_X \frac{\alpha_Y}{\alpha_X}}})\sqrt{\log n}\right)^{2\beta_Y}.$$

Then we approximate $\overline{g}(w,x)$ by the neural network. Notice that by Cayley-Hamilton theorem, for $A \in \mathbb{R}^{d\times d}$, denote $S_k$ as the trace of $A^k$ and $B_k$ as the $k$th complete exponential Bell polynomial.[2] We can write

$$\det(A) = \frac{1}{d!}B_d(S_1, -1!S_2, \cdots, (-1)^{d-1}(n-1)!S_d)$$

$$A^{-1} = \frac{1}{\det(A)}\sum_{k=0}^{d-1}(-1)^{d+k-1}\frac{A^{d-k-1}}{k!}B_i(S_1, -1!S_2, \cdots, (-1)^{k-1}(k-1)!S_k).$$

By Lemmas C.9 and C.10, there exists $\phi_g(w,x,z) \in \Phi(H,W,R,B)$ and $\phi_{g(\cdot,\widetilde{z})}(w,x) \in \Phi(H_1, W_1, R_1, B_1)$ with $H \asymp H_1 = \Theta(\log^2 n)$, $\|W\|_\infty \asymp \|W_1\|_\infty = \Theta(\log^3 n)$, $R \asymp R_1 = \Theta(\log^4 n)$ and $B \asymp B_1 = \exp(\Theta(\log^2 n))$ so that for any $x \in \mathcal{M}_X$ satisfying $\|x - \widetilde{x}\| \lesssim (\sigma_{\underline{t}}\sqrt{\log n})^{\frac{1}{\beta_X}}$, $w \in \mathbb{R}^{D_Y}$ satisfying $\|w - G^*_{[k]}(\widetilde{z}, x)\| \lesssim (\sigma_{\underline{t}} \vee n^{-\frac{1}{2\alpha_Y + d_Y + d_X \frac{\alpha_Y}{\alpha_X}}})\sqrt{\log n}$ and $\|z\| \leq r_3$,

$$\|\phi_g(w,x,z) - g(w,x,z)\| \lesssim \left((\sigma_{\underline{t}} \vee n^{-\frac{1}{2\alpha_Y + d_Y + d_X \frac{\alpha_Y}{\alpha_X}}})\sqrt{\log n}\right)^{2\beta_Y}.$$

and

$$\|\phi_{g(\cdot,\widetilde{z})}(w,x) - g(w,x,\widetilde{z})\| \lesssim \left((\sigma_{\underline{t}} \vee n^{-\frac{1}{2\alpha_Y + d_Y + d_X \frac{\alpha_Y}{\alpha_X}}})\sqrt{\log n}\right)^{2\beta_Y}.$$

Furthermore,

$$\left\|\underbrace{g(w,x,g(w,x,\circ g(w,x,\circ \cdots \circ g(w,x,g(w,x,g(}_{\lceil \log_2(2\beta_Y)\rceil}w,x,\widetilde{z}))))))\right.$$
$$\left. - \underbrace{\phi_g(w,x,\phi_g(w,x,\circ\phi_g(w,x,\circ \cdots \circ \phi_g(w,x,\phi_g(w,x,\phi_{g(\cdot,\widetilde{z})}(}_{\lceil \log_2(2\beta_Y)\rceil}w,x))))))\right\|$$
$$\lesssim \left((\sigma_{\underline{t}} \vee n^{-\frac{1}{2\alpha_Y + d_Y + d_X \frac{\alpha_Y}{\alpha_X}}})\sqrt{\log n}\right)^{2\beta_Y}.$$

So by concatenation and parallelization of neural networks, there exists $\phi_p(w,x) \in \Phi(H,W,R,B)$ with $H = \Theta(\log^2 n)$, $\|W\|_\infty = \Theta(\log^3 n)$, $R = \Theta(\log^4 n)$ and $B = \exp(\Theta(\log^2 n))$ so that for

---

[2] $B_k(x_1,\ldots,x_k) = \sum_{w=1}^{k} B_{k,w}(x_1, x_2, \ldots, x_{k-w+1})$ with $B_{k,w}(x_1, x_2, \ldots x_{k-w+1})$
$= \sum_{\substack{j_1 + \ldots + j_{k-w+1} = w \\ j_1 + 2j_2 + \ldots + (k-w+1)j_{k-w+1} = k}} \frac{k!}{j_1! j_2! \ldots j_{k-w+1}!}\left(\frac{x_1}{1!}\right)^{j_1}\left(\frac{x_2}{2!}\right)^{j_2}\cdots\left(\frac{x_{k-w+1}}{k-w+1!}\right)^{j_{k-w+1}}$

any $x \in \mathcal{M}_X$ satisfying $\|x - \widetilde{x}\| \lesssim (\sigma_{\underline{t}} \sqrt{\log n})^{\frac{1}{\beta_X}}$, $w \in \mathbb{R}^{D_Y}$ satisfying $\|w - G^*_{[k]}(\widetilde{z}, x)\| \lesssim (\sigma_{\underline{t}} \vee n^{-\frac{1}{2\alpha_Y + d_Y + d_X \frac{\alpha_Y}{\alpha_X}}}) \sqrt{\log n}$,

$$\|\phi_p(w, x) - \overline{g}(w, x)\| \lesssim \left( (\sigma_{\underline{t}} \vee n^{-\frac{1}{2\alpha_Y + d_Y + d_X \frac{\alpha_Y}{\alpha_X}}}) \sqrt{\log n} \right)^{2\beta_Y}.$$

So we have

$$\left\| \langle J_{G^*_{[k]}|_{(\widetilde{z}, \widetilde{x})}(\cdot, x)}(\phi_p(w, x)), w - G^*_{[k]}|_{(\widetilde{z}, \widetilde{x})}(\phi_p(w, x), x) \rangle \right\|$$

$$= \|h(w, x, \phi_p(w, x))\| \lesssim \left( (\sigma_{\underline{t}} \vee n^{-\frac{1}{2\alpha_Y + d_Y + d_X \frac{\alpha_Y}{\alpha_X}}}) \sqrt{\log n} \right)^{2\beta_Y}.$$

The proof of the first statement is completed. Then for the second statement, define

$$f(w, x, z) = \|w - G^*_{[k]}|_{(\widetilde{z}, \widetilde{x})}(z, x)\|^2.$$

Then we have

$$J_{f(w, x, \cdot)}(z) = -2h(w, x, z)$$

and the Hessian matrix of $f(w, x, \cdot)$ at $z$ is $-2J_{h(w, x, \cdot)}(z)$. Then we denote

$$\overline{g}_k(w, x) = \underbrace{g(w, x, g(w, x, \circ g(w, x, \circ \cdots \circ g(w, x, g(w, x, g(w, x, \widetilde{z}))))))}_{k}.$$

Then for any $x \in \mathcal{M}_X$ satisfying $\|x - \widetilde{x}\| \lesssim (\sigma_{\underline{t}} \sqrt{\log n})^{\frac{1}{\beta_X}}$, $w \in \mathbb{R}^{D_Y}$ satisfying $\|w - G^*_{[k]}(\widetilde{z}, x)\| \lesssim (\sigma_{\underline{t}} \vee n^{-\frac{1}{2\alpha_Y + d_Y + d_X \frac{\alpha_Y}{\alpha_X}}}) \sqrt{\log n}$, we have $\overline{z} = \lim_{k \to \infty} g_k(w, x)$ exists and

$$\|\overline{z} - \phi_p(w, x)\| \lesssim \left( (\sigma_{\underline{t}} \vee n^{-\frac{1}{2\alpha_Y + d_Y + d_X \frac{\alpha_Y}{\alpha_X}}}) \sqrt{\log n} \right)^{2\beta_Y} \lesssim \sigma_{\underline{t}} \sqrt{\log n}$$

$$\|\overline{z} - \widetilde{z}\| \lesssim (\sigma_{\underline{t}} \vee n^{-\frac{1}{2\alpha_Y + d_Y + d_X \frac{\alpha_Y}{\alpha_X}}}) \sqrt{\log n},$$

$$J_{f(w, x, \cdot)}(\overline{z}) = 0.$$

Therefore, for any $\|z - \widetilde{z}\| \le r_3$,

$$f(w, x, z) - f(w, x, \overline{z}) \ge 2L_3 \|z - \overline{z}\|^2.$$

Then if

$$\|Q^*_{[k]}(\operatorname{Proj}_{\mathcal{M}_{Y|x}}(\omega), x) - \overline{z}\| \ge C_2 \sigma_{\underline{t}} \sqrt{\log n},$$

for a large enough $C_2$, we have,

$$\|w - G^*_{[k]}|_{(\widetilde{z}, \widetilde{x})}(Q^*_{[k]}(\operatorname{Proj}_{\mathcal{M}_{Y|x}}(\omega), x), x)\|^2 - \|w - G^*_{[k]}|_{(\widetilde{z}, \widetilde{x})}(\overline{z}, x)\|^2 \ge 2L_3 C_2^2 \sigma_{\underline{t}}^2 \log n.$$

Moreover, notice that $\|w - G^*_{[k]}(\widetilde{z}, x)\| \lesssim (\sigma_{\underline{t}} \vee n^{-\frac{1}{2\alpha_Y + d_Y + d_X \frac{\alpha_Y}{\alpha_X}}}) \sqrt{\log n}$ and $\operatorname{dist}(w, \mathcal{M}_{Y|x}) \le c_0 \sigma_{\underline{t}} \sqrt{\log n}$, we have

$$\|\operatorname{Proj}_{\mathcal{M}_{Y|x}}(\omega) - G^*_{[k]}(\widetilde{z}, x)\| \lesssim (\sigma_{\underline{t}} \vee n^{-\frac{1}{2\alpha_Y + d_Y + d_X \frac{\alpha_Y}{\alpha_X}}}) \sqrt{\log n},$$

and

$$\|Q^*_{[k]}(\operatorname{Proj}_{\mathcal{M}_{Y|x}}(\omega), x) - \widetilde{z}\| \lesssim (\sigma_{\underline{t}} \vee n^{-\frac{1}{2\alpha_Y + d_Y + d_X \frac{\alpha_Y}{\alpha_X}}}) \sqrt{\log n}.$$

Therefore

$$\|w - G^*_{[k]}|_{(\widetilde z,\widetilde x)}(\overline z, x)\|$$

$$\leq \|w - G^*_{[k]}|_{(\widetilde z,\widetilde x)}(Q^*_{[k]}(\mathrm{Proj}_{\mathcal{M}_{Y|x}}(\omega), x), x)\|$$

$$\leq \|w - G^*_{[k]}(Q^*_{[k]}(\mathrm{Proj}_{\mathcal{M}_{Y|x}}(\omega), x), x)\| + ((\sigma_{\underline t} \vee n^{-\frac{1}{2\alpha_Y + d_Y + d_X \frac{\alpha_Y}{\alpha_X}}})\sqrt{\log n})^{\beta_Y} + \sigma_{\underline t}\sqrt{\log n}$$

$$\leq C_3 \sigma_{\underline t}\sqrt{\log n}.$$

So we have

$$\|w - G^*_{[k]}|_{(\widetilde z,\widetilde x)}(Q^*_{[k]}(\mathrm{Proj}_{\mathcal{M}_{Y|x}}(\omega), x), x)\| - \|w - G^*_{[k]}|_{(\widetilde z,\widetilde x)}(\overline z, x)\|$$

$$= \frac{\|w - G^*_{[k]}|_{(\widetilde z,\widetilde x)}(Q^*_{[k]}(\mathrm{Proj}_{\mathcal{M}_{Y|x}}(\omega), x), x)\|^2 - \|w - G^*_{[k]}|_{(\widetilde z,\widetilde x)}(\overline z, x)\|^2}{\|w - G^*_{[k]}|_{(\widetilde z,\widetilde x)}(Q^*_{[k]}(\mathrm{Proj}_{\mathcal{M}_{Y|x}}(\omega), x), x)\| + \|w - G^*_{[k]}|_{(\widetilde z,\widetilde x)}(\overline z, x)\|} \geq \frac{L_3 C_2^2}{C_3}\sigma_{\underline t}\sqrt{\log n}.$$

Then notice that

$$\|w - G^*_{[k]}(Q^*_{[k]}(\mathrm{Proj}_{\mathcal{M}_{Y|x}}(\omega), x), x)\| \geq \|w - G^*_{[k]}|_{(\widetilde z,\widetilde x)}(Q^*_{[k]}(\mathrm{Proj}_{\mathcal{M}_{Y|x}}(\omega), x), x)\| - C\sigma_{\underline t}\sqrt{\log n},$$

and

$$\|w - G^*_{[k]}(\overline z, x)\| \leq \|w - G^*_{[k]}|_{(\widetilde z,\widetilde x)}(\overline z, x)\| + C\sigma_{\underline t}\sqrt{\log n},$$

when $C_2$ is large enough, we have

$$\|w - G^*_{[k]}(\overline z, x)\| < \|w - G^*_{[k]}(Q^*_{[k]}(\mathrm{Proj}_{\mathcal{M}_{Y|x}}(\omega), x), x)\|,$$

which cause contradiction. So

$$\|Q^*_{[k]}(\mathrm{Proj}_{\mathcal{M}_{Y|x}}(\omega), x) - \phi_p(w, x)\|$$

$$\leq \|Q^*_{[k]}(\mathrm{Proj}_{\mathcal{M}_{Y|x}}(\omega), x) - \overline z\| + \|\phi_p(w, x) - \overline z\| \lesssim \sigma_{\underline t}\sqrt{\log n}.$$

### C.4.4 Proof of lemma C.12

Let $\phi(x, \mu, \Sigma)$ denotes the density function of $\mathcal{N}(\mu, \Sigma)$, we have

$$\mathbb{E}_{x \sim \mu_X^*} \mathbb{E}_{y \sim \mu_{Y|x}^*} \mathbb{E}_{w_t \sim \mathcal{N}(m_t y, \sigma_t^2 I_{D_Y})} \left[ \|S(w_t, x, t) - \nabla \log p_{t|x}(w_t)\|^2 \right]$$

$$= \mathbb{E}_{x \sim \mu_X^*} \left[ \int \|S(w_t, x, t) - \nabla \log p_{t|x}(w_t)\|^2 p_{t|x}(w_t) \, dw_t \right]$$

$$= \mathbb{E}_{x \sim \mu_X^*} \left[ \int \|S(w_t, x, t)\|^2 p_{t|x}(w_t) \, dw_t + \int \|\nabla \log p_{t|x}(w_t)\|^2 p_{t|x}(w_t) \, dw_t \right.$$
$$\left. - 2 \int S(w_t, x, t)^T \nabla \log p_{t|x}(w_t) \, p_{t|x}(w_t) \, dw_t \right]$$

$$= \mathbb{E}_{x \sim \mu_X^*} \left[ \int \|S(w_t, x, t)\|^2 p_{t|x}(w_t) \, dw_t + \int \|\nabla \log p_{t|x}(w_t)\|^2 p_{t|x}(w_t) \, dw_t \right.$$
$$\left. - 2 \int S(w_t, x, t)^T \nabla p_{t|x}(w_t) \, dw_t \right]$$

$$= \mathbb{E}_{x \sim \mu_X^*} \left[ \int \|S(w_t, x, t)\|^2 p_{t|x}(w_t) \, dw_t + \int \|\nabla \log p_{t|x}(w_t)\|^2 p_{t|x}(w_t) \, dw_t \right.$$
$$\left. - 2 \int S(w_t, x, t)^T \nabla \left( \mathbb{E}_{y \sim \mu_{Y|x}^*} [\phi(w_t, m_t y, \sigma_t^2 I_{D_y})] \right) dw_t \right]$$

$$= \mathbb{E}_{x \sim \mu_X^*} \left[ \int \|S(w_t, x, t)\|^2 p_{t|x}(w_t) \, dw_t + \int \|\nabla \log p_{t|x}(w_t)\|^2 p_{t|x}(w_t) \, dw_t \right.$$
$$\left. - 2 \int S(w_t, x, t)^T \mathbb{E}_{y \sim \mu_{Y|x}^*} [\nabla \phi(w_t, m_t y, \sigma_t^2 I_{D_y})] dw_t \right]$$

$$= \mathbb{E}_{x \sim \mu_X^*} \left[ \int \|S(w_t, x, t)\|^2 p_{t|x}(w_t) \, dw_t + \int \|\nabla \log p_{t|x}(w_t)\|^2 p_{t|x}(w_t) \, dw_t \right.$$
$$\left. - 2 \int S(w_t, x, t)^T \mathbb{E}_{y \sim \mu_{Y|x}^*} [\nabla \log \phi(w_t, m_t y, \sigma_t^2 I_{D_y}) \cdot \phi(w_t, m_t y, \sigma_t^2 I_{D_y})] dw_t \right]$$

$$= \mathbb{E}_{x \sim \mu_X^*} \mathbb{E}_{y \sim \mu_{Y|x}^*} \mathbb{E}_{w_t \sim \mathcal{N}(m_t y, \sigma_t^2 I_{D_Y})} \left[ \|S(w_t, x, t)\|^2 + \|\nabla \log p_{t|x}(w_t)\|^2 \right.$$
$$\left. - 2 S(w_t, x, t)^T \nabla \log \phi(w_t, m_t y, \sigma_t^2 I_{D_y}) \right]$$

$$= \mathbb{E}_{x \sim \mu_X^*} \mathbb{E}_{y \sim \mu_{Y|x}^*} \mathbb{E}_{w_t \sim \mathcal{N}(m_t y, \sigma_t^2 I_{D_Y})} \left[ \|S(w_t, x, t) - \frac{m_t Y - w_t}{\sigma_t^2}\|^2 \right]$$
$$+ \mathbb{E}_{\mu_X^*} \mathbb{E}_{y \sim \mu_{Y|x}^*} \mathbb{E}_{w_t \sim \mathcal{N}(m_t y, \sigma_t^2 I_{D_Y})} \left[ \|\nabla \log p_{t|x}(w_t)\|^2 - \left\| \frac{m_t Y - w_t}{\sigma_t^2} \right\|^2 \right].$$

### C.4.5 Proof of Lemma C.13

For $G_i^* = \{g(x, y) = \ell_i(x, y, S) - \ell_i(x, y, S_i^*) \; : \; S \in \mathcal{S}_i\}$, it holds that

$$\sup_{g \in G_i^*} \sup_{(x,y) \in \mathcal{M}} |g(x, y)| \lesssim (\log n)^2.$$

Define

$$Z_n(\delta, G_i^*) = \sup_{\substack{g \in G_i^* \\ \|g\|_2 \leq \delta}} \left| \frac{1}{n} \sum_{i=1}^n g(X_i, Y_i) - \mathbb{E}_{\mu^*}[g(x, y)] \right|.$$

Since $\frac{1}{n} \sup_{\substack{g \in G_i^* \\ \|g\|_2 \leq \delta}} \sum_{i=1}^n var(g(x_i)) \leq \delta^2$, by the tail inequality for suprema of bounded empirical processes (see for example, Theorem 3.27 of [6]), it holds that

$$P(Z_n(\delta, G_i^*) \geq \mathbb{E}_{\mu^* \otimes n}[Z_n(\delta, G_i^*)] + c_0(\delta + (\log n)\sqrt{\mathbb{E}_{\mu^* \otimes n}[Z_n(\delta, G_i^*)]})\sqrt{t} + c_1(\log n)^2 t) \leq \exp(-nt).$$

$$(31)$$

Using the standard symmetrization (see, for example, Proposition 4.11 of [6]), we can get

$$\mathbb{E}_{\mu^* \otimes n}\left[Z_n(\delta, G_i^*)\right] \leq \mathbb{E}_{\mu^* \otimes n} \mathbb{E}_\epsilon \left[ \sup_{\substack{g \in G_i^* \\ \|g\|_2 \leq \delta}} \left| \frac{2}{n} \sum_{i=1}^n \epsilon_i g(X_i, Y_i) \right| \right]$$

$$= 2\overline{R}_n(\delta, G_i^*) \leq 2\overline{R}_n(\delta, \overline{G}_i^*),$$

where recall that $\overline{G}_i^* = \{ag \mid a \in (0,1], g \in G_i^*\}$ and $\{\epsilon_i\}_{i=1}^n$ are $n$ i.i.d. copies from Rademacher distribution, i.e. $\mathbb{P}(\epsilon_i = 1) = \mathbb{P}(\epsilon_i = -1) = \frac{1}{2}$. Therefore by $\overline{R}_n(\delta_{ni}, \overline{G}_i^*) \leq \delta_{ni}^2/(\log n)^2$, it holds that

$$\forall r \geq \delta_{ni}, \quad \mathbb{E}_{\mu^* \otimes n}\left[Z_n(r, G_i^*)\right] \leq 2\overline{R}_n(r, \overline{G}_i^*)$$

$$= 2\mathbb{E}_{\mu^* \otimes n} \mathbb{E}_\varepsilon \left[ \sup_{\substack{g \in \overline{G}_i^* \\ \|\frac{\delta_{ni}}{r} g\|_2 \leq \delta_{ni}}} \frac{r}{\delta_{ni}} \left| \frac{1}{n} \sum_{i=1}^n \varepsilon_i \frac{\delta_{ni}}{r} g(x_i) \right| \right]$$

$$\leq 2 \frac{r}{\delta_{ni}} \overline{R}_n(\delta_{ni}, \overline{G}_i^*)$$

$$\leq 2r\delta_{ni}/(\log n)^2.$$

Define the events

$$\mathcal{A}_0 = \{Z_n(\delta_{ni}, G_i^*) \geq c_2 \delta_{ni}^2/(\log n)^2\};$$

$$\mathcal{A}_1 = \left\{ \exists g \in G_i^*, \text{ such that } \left| \frac{1}{n} \sum_{i=1}^n g(X_i, Y_i) - \mathbb{E}_{\mu^* \otimes n}[g(x, y)] \right| \geq c_2 \delta_{ni} \|g\|_2/(\log n)^2 \right.$$

$$\left. \text{and } \|g\|_2 \geq \delta_{ni} \right\}.$$

Using equation (31), there exist some constants $(c_0', c_1', c_2)$ such that

$$P(\mathcal{A}_0) \leq \frac{1}{n^2}.$$

Define $\mathcal{S}_m = \{2^{m-1}\delta_{ni} \leq \|g\|_2 \leq 2^m \delta_{ni}\}$ with $m = 1, \cdots M$, since $\|g\|_2 \lesssim (\log n)^2$, we have $M \lesssim \log(\frac{1}{\delta_{ni}})$.

Under $\mathcal{A}_1 \cap \mathcal{S}_m$, it holds that $Z_n(2^m \delta_{ni}, G_i^*) \geq c_2 2^{m-1} \delta_{ni}^2/(\log n)^2$. Therefore,

$$P(\mathcal{A}_1) = \sum_{m=1}^M P(\mathcal{A}_1 \cap \mathcal{S}_m) \leq \frac{1}{n^2}.$$

Moreover, under $\mathcal{A}_0^c \cap \mathcal{A}_1^c$, we have

$$\sup_{g \in G_i^*} \frac{\left| \frac{1}{n} \sum_{i=1}^n g(X_i, Y_i) - \mathbb{E}_{\mu^*}[g(x, y)] \right|}{\delta_{ni} + \|g\|_2} \leq c_2 \delta_{ni}/(\log n)^2.$$

We can then get the desired conclusion.

# D    Proof of Theorem 1

Under Assumptions A and B, we can derive the following theorem for controlling the conditional score approximation error.

**Lemma D.1.** *For any $t \in [\underline{t}, \overline{t}]$ with $1 < \frac{\overline{t}}{\underline{t}} \leq 2$:*

1. *If* $\tau \leq \underline{t} < n^{-\frac{2}{2\alpha_Y + d_Y + d_X \frac{\alpha_Y}{\alpha_X}}}$, *there exists a neural network* $\phi_{score}(w, x, t) \in \Phi(H, W, R, B, V)$ *satisfying*

$$\mathbb{E}_{\mu_X^*}\left[\int_{\underline{t}}^{\overline{t}}\int_{\mathbb{R}^{D_Y}}\left\|\phi_{score}(w, x, t) - \nabla \log p_{t|x}(w)\right\|^2 p_{t|x}(w)\, \mathrm{d}w \mathrm{d}t\right]$$
$$= \widetilde{\mathcal{O}}(\varepsilon_1^{2\alpha_Y} + \varepsilon_2^{2\alpha_X}),$$

   *where* $\varepsilon_1 = n^{-\frac{1}{2\alpha_Y + d_Y + \frac{\alpha_Y}{\alpha_X}d_X}}$ *and* $\varepsilon_2 = \varepsilon_1^{\frac{\alpha_Y}{\alpha_X}}$. *Here* $H$, $W$, $R$, $B$ *and* $V$ *are evaluated as* $H = \Theta\left(\log^4 n\right)$, $\|W\|_\infty = \widetilde{\Theta}\left(\varepsilon_1^{-d_Y}\varepsilon_2^{-d_X}\right)$, $R = \widetilde{\Theta}\left(\varepsilon_1^{-d_Y}\varepsilon_2^{-d_X}\right)$, $B = \exp\left(\Theta(\log^4 n)\right)$ *and* $V = \Theta(\sqrt{\frac{\log n}{\underline{t}}})$.

2. *If* $n^{-\frac{2}{2\alpha_Y + d_Y + d_X \frac{\alpha_Y}{\alpha_X}}} \leq \underline{t} \leq T$, *there exists a neural network* $\phi_{score}(w, x, t) \in \Phi(H, W, R, B, V)$ *satisfying*

$$\mathbb{E}_{\mu_X^*}\left[\int_{\underline{t}}^{\overline{t}}\int_{\mathbb{R}^{D_Y}}\left\|\phi_{score}(w, x, t) - \nabla \log p_{t|x}(w)\right\|^2 p_{t|x}(w)\, \mathrm{d}w \mathrm{d}t\right] = \widetilde{\mathcal{O}}(\varepsilon_2^{2\alpha_X}),$$

   *with* $\varepsilon_2 = \widetilde{\Theta}(n^{-\frac{1}{2\alpha_X + d_X}}\underline{t}^{-\frac{d_Y}{4\alpha_X + 2d_X}})$. *Here* $H$, $W$, $R$, $B$ *and* $V$ *are evaluated as* $H = \Theta\left(\log^4 n\right)$, $\|W\|_\infty = \widetilde{\Theta}\left(\underline{t}^{-\frac{d_Y}{2}}\varepsilon_2^{-d_X}\right)$, $R = \widetilde{\Theta}\left(\underline{t}^{-\frac{d_Y}{2}}\varepsilon_2^{-d_X}\right)$, $B = \exp\left(\Theta(\log^4 n)\right)$ *and* $V = \Theta(\sqrt{\frac{\log n}{\underline{t}\wedge 1}})$.

Then similarly to the proof of Theorem 2, combined with Lemma C.3 for the estimation error, we can obtain the desired result.

## D.1 Proof of Lemma D.1

Similar as Lemma C.4, we have the following lemma for addressing the unboundedness of the space and the score function.

**Lemma D.2.** *If* $\sup_{w \in \mathbb{R}^{D_Y}, x \in [-1,1]^{D_X}} \sup_{t \in [\tau, T]} [\|S(w, x, t)\|_\infty \sigma_t] \leq c\sqrt{\log n}$. *Then, there exist constants* $(c_0, c_1, c_2, c_3)$ *so that for any* $i \in [\mathcal{I}]$ *and* $t \in [t_{i-1}, t_i]$ *with* $1 < \frac{t_i}{t_{i-1}} \leq 2$,

1. *Denote* $\mathrm{dist}(w, [-1,1]^{D_Y})$ *as the distance of point* $w \in \mathbb{R}^{D_Y}$ *to set* $[-1,1]^{D_Y}$. *Then for any* $x \in [-1,1]^{D_X}$,

$$\int_{\mathbb{R}^{D_Y}}\left\|\nabla \log p_{t|x}(w) - S(w, x, t)\right\|^2 p_{t|x}(w)\, \mathrm{d}w$$
$$\leq \int_{\mathbb{R}^{D_Y}}\left\|\nabla \log p_{t|x}(w) - S(w, x, t)\right\|^2 p_{t|x}(w) \cdot \mathbf{1}\left(\mathrm{dist}(w, [-1,1]^{D_Y}) \leq c_0 \sigma_{t_{i-1}}\sqrt{\log n}\right)\, \mathrm{d}w$$
$$+ (1 + c^2) \cdot c_1 \frac{1}{n^2}.$$

2. *For any* $x \in [-1,1]^{D_X}$ *and* $w \in \mathbb{R}^{D_Y}$ *satisfying* $\mathrm{dist}(w, [-1,1]^{D_Y}) \leq c_0 \sigma_{t_{i-1}}\sqrt{\log n}$, *we have*

   (a) $\|\nabla \log p_{t|x}(w)\|_\infty \leq c_2 \frac{\sqrt{\log n}}{\sigma_{t_{i-1}}}$.

   (b) $(2\pi\sigma_t^2)^{\frac{D}{2}} p_{t|x}(w) \geq n^{-c_3}$.

The proof of Lemma D.2 directly follows from Lemma A.2-A.4 of [2]. Then we fix a time interval $t \in [\underline{t}, \overline{t}]$ where $1 < \frac{\overline{t}}{\underline{t}} \leq 2$. Then similar as lemma C.6, we demonstrate that it is enough to provide local approximations to the score function.

**Lemma D.3.** *Suppose $\tau \leq \underline{t} \leq T$, $\varepsilon_2 > 0$, and $\varepsilon_1 \geq \sigma_{\underline{t}}\sqrt{\log n}$. Let $\mathcal{N}^X_{\varepsilon_2} = (\widetilde{x}_1, \widetilde{x}_2, \cdots, \widetilde{x}_{J_2})$ be one of the largest $\varepsilon_2$-packing of $[-1,1]^{D_X}$, and let $\mathcal{N}^Y_{\varepsilon_1} = (\widetilde{y}_1, \widetilde{y}_2, \cdots, \widetilde{y}_{J_1})$ be one of the largest $\varepsilon_1$-packing of $[-1,1]^{D_Y}$. Then if for any $j_1 \in [J_1]$ and $j_2 \in [J_2]$, there exists a neural network $\phi^*_{j_1 j_2}(x, w, t) \in \Phi\big(H, W, R, B, \Theta(\frac{\sqrt{\log n}}{\sigma_{\underline{t}}})\big)$ so that for any $t \in [\underline{t}, \overline{t}]$, $x \in \mathbb{B}_{[-1,1]^{D_X}}(\widetilde{x}_{j_2}, \sqrt{2}\varepsilon_2)$ and $w \in \mathbb{R}^{D_Y}$ satisfying $\|w - \widetilde{y}_{j_1}\| \leq \sqrt{2}(2\varepsilon_1 + c_0\sigma_{\underline{t}}\sqrt{\log n})$ and $\mathrm{dist}(w, [-1,1]^{D_X}) \leq c_0\sigma_{\underline{t}}\sqrt{\log n}$,*

$$\|\phi^*_{j_1 j_2}(w, x, t) - \nabla \log p_{t|x}(w)\|_\infty \leq \varepsilon.$$

*Then there exists a neural network $\phi_{\mathrm{score}}(w, x, t) \in \big(H_1, W_1, R_1, B_1, \Theta(\frac{\sqrt{\log n}}{\sigma_{\underline{t}}})\big)$ with $H_1 = \Theta(H + \log^2 n)$, $\|W_1\|_\infty = \Theta(J_1 J_2(\|W\|_\infty + \log n) + \log^3 n)$, $R_1 = \Theta(J_1 J_2(R + \log n) + \log^4 n)$ and $B_1 = \exp(\Theta(\log^2 n)) \vee B$, so that for any $t \in [\underline{t}, \overline{t}]$, $x \in [-1, 1]^{D_X}$ and $w \in \mathbb{R}^{D_Y}$ satisfying $\mathrm{dist}(x, [-1,1]^{D_Y}) \leq c_0\sigma_{\underline{t}}\sqrt{\log n}$,*

$$\|\phi_{\mathrm{score}}(w, x, t) - \nabla \log p_{t|x}(w)\|_\infty \lesssim \varepsilon + \frac{1}{n}.$$

The proof of Lemma D.3 can be conducted similarly to the proof of Lemma C.6. Then similar as (2), by statement 2 of Lemma D.2, there exists a large enough constant $c_2$, so that for any $t \in [\underline{t}, \overline{t}]$, $x \in [-1,1]^{D_X}$, $w \in \mathbb{R}^{D_Y}$ with $\mathrm{dist}(w, [-1,1]^{D_Y}) \leq c_0\sigma_{\underline{t}}\sqrt{\log n}$, and any partition $\{\mathcal{A}_{(x,w)}, [-1,1]^{D_Y} \backslash \mathcal{A}_{(x,w)}\}$ of $[-1,1]^{D_Y}$ satisfying $\{y \in [-1,1]^{D_Y} : \|y - w\| \leq c_2\sigma_{\underline{t}}\sqrt{\log n}\} \subset \mathcal{A}_{w,x}$, it holds that

$$\left\| \nabla \log p_{t|x}(w) - \frac{1}{\sigma_t} \cdot \frac{\mathbb{E}_{y \sim \mu^*_{Y|x}}\left[\exp\left(-\frac{\|w - m_t y\|^2}{2\sigma_t^2}\right) \cdot -\frac{(w - m_t y)}{\sigma_t}\mathbf{1}(y \in \mathcal{A}_{(x,w)})\right]}{\mathbb{E}_{y \sim \mu^*_{Y|x}}\left[\exp\left(-\frac{\|w - m_t y\|^2}{2\sigma_t^2}\right)\mathbf{1}(y \in \mathcal{A}_{(x,w)})\right]} \right\|_\infty \leq \frac{1}{n}. \qquad (32)$$

We will approximate $\nabla \log p_{t|x}(w)$ by constructing suitable sets $\mathcal{A}_{(x,w)}$ for small $\underline{t}$ and large $\underline{t}$.

### D.1.1 Case 1: $n^{-\frac{2}{2\alpha_Y + D_Y + D_X \frac{\alpha_Y}{\alpha_X}}} \leq \underline{t} \leq T$

Set $\varepsilon_1 = \sigma_{\underline{t}}\sqrt{\log n}$, $\varepsilon_2 = n^{-\frac{1}{2\alpha_X + D_X}}(\sigma_{\underline{t}}\sqrt{\log n})^{-\frac{D_Y}{2\alpha_X + D_X}}$. Let $\mathcal{N}^X_{\varepsilon_2}$ be one of the largest $\varepsilon_2$-packing of $[-1,1]^{D_X}$ and $\mathcal{N}^Y_{\varepsilon_1}$ be one of the largest $\varepsilon_1$-packing of $[-1,1]^{D_Y}$. Then we have $J_1 = |\mathcal{N}^Y_{\varepsilon_1}| \lesssim \varepsilon_1^{-D_Y}$ and $J_2 = |\mathcal{N}^X_{\varepsilon_2}| \lesssim \varepsilon_2^{-D_X}$. Now we take an arbitrary $\widetilde{x} \in \mathcal{N}^X_{\varepsilon_2}$ and $\widetilde{y} \in N^Y_{\varepsilon_1}$, consider set

$$\mathscr{S}_{\widetilde{x}\widetilde{y}} = \Big\{(x, w) : x \in \mathbb{B}_{[-1,1]^{D_X}}(\widetilde{x}, \sqrt{2}\varepsilon_2),$$
$$\|w - \widetilde{y}\| \leq c_3\,\sigma_{\underline{t}}\sqrt{\log n}, \mathrm{dist}(w, [-1,1]^{D_Y}) \leq c_0\sigma_{\underline{t}}\sqrt{\log n}\Big\},$$

we claim that

**Claim 5.** *There exists $\phi^*(w, x, t) \in \Phi\big(H, W, R, B, \Theta(\frac{\sqrt{\log n}}{\sigma_{\underline{t}}})\big)$ with $H = \Theta(\log^4 n)$, $\|W\|_\infty = \widetilde{\Theta}(1)$, $R = \widetilde{\Theta}(1)$, $B = \exp(\Theta(\log^4 n))$, so that for any $(x, w) \in \mathscr{S}_{\widetilde{x}\widetilde{y}}$, and $t \in [\underline{t}, \overline{t}]$,*

$$\|\phi^*(w, x, t) - \nabla \log p_{t|x}(w)\|_\infty = \widetilde{\mathcal{O}}(\frac{\varepsilon_2^{\alpha_X}}{\sigma_{\underline{t}}}).$$

Then the second statement of Lemma D.1 directly follows from Lemmas D.2 and D.3. Now we show Claim 5.

Firstly, notice that for any $(x, w) \in \mathscr{S}_{\widetilde{x}\widetilde{y}}$,

$$\{y \in [-1,1]^{D_Y} : \|y - w\| \leq c_2\sigma_{\underline{t}}\sqrt{\log n}\}$$
$$\subset \{y \in [-1,1]^{D_Y} : \|y - \widetilde{y}\| \leq c_2\sigma_{\underline{t}}\sqrt{\log n} + \|w - \widetilde{y}\|\}$$
$$\subset \{y \in [-1,1]^{D_Y} : \|y - \widetilde{y}\|_\infty \leq c_4\sigma_{\underline{t}}\sqrt{\log n}\}.$$

Therefore, by equation ([32](#)), we only need to approximate

$$\frac{1}{\sigma_t} \cdot \frac{\int_{\|y-\widetilde{y}\|_\infty \leq c_4 \sigma_t \sqrt{\log n}} \mathbf{1}(y \in [-1,1]^{D_X}) \exp\left(-\frac{\|w-m_t y\|^2}{2\sigma_t^2}\right) \cdot \left(-\frac{w-m_t y}{\sigma_t}\right) \mu^*(y|x)\,\mathrm{d}y}{\int_{\|y-\widetilde{y}\|_\infty \leq c_4 \sigma_t \sqrt{\log n}} \mathbf{1}(y \in [-1,1]^{D_X}) \exp\left(-\frac{\|w-m_t y\|^2}{2\sigma_t^2}\right) \mu^*(y|x)\,\mathrm{d}y}. \tag{33}$$

Let

$$\mu^*|_{\widetilde{x}}(y|x) = \sum_{\substack{j \in \mathbb{N}_0^{d_X} \\ |j| < \alpha_X}} \frac{\mu^*_{(0,j)}(y|\widetilde{x})}{j!}(x-\widetilde{x})^j.$$

Then when $y \in [-1,1]^{D_Y}$ and $x \in [-1,1]^{D_X}$,

$$|\mu^*(y|x) - \mu^*|_{\widetilde{x}}(y|x)| \lesssim \|x-\widetilde{x}\|^{\alpha_X} \lesssim \varepsilon_2^{\alpha_X}.$$

So we have

$$\left\| \frac{\int_{\|y-\widetilde{y}\|_\infty \leq c_4 \sigma_t \sqrt{\log n}} \mathbf{1}(y \in [-1,1]^{D_X}) \exp\left(-\frac{\|w-m_t y\|^2}{2\sigma_t^2}\right) \cdot \left(-\frac{w-m_t y}{\sigma_t}\right) \mu^*(y|x)\,\mathrm{d}y}{\int_{\|y-\widetilde{y}\|_\infty \leq c_4 \sigma_t \sqrt{\log n}} \mathbf{1}(y \in [-1,1]^{D_X}) \exp\left(-\frac{\|w-m_t y\|^2}{2\sigma_t^2}\right) \mu^*(y|x)\,\mathrm{d}y} \right.$$
$$\left. - \frac{\int_{\|y-\widetilde{y}\|_\infty \leq c_4 \sigma_t \sqrt{\log n}} \mathbf{1}(y \in [-1,1]^{D_X}) \exp\left(-\frac{\|w-m_t y\|^2}{2\sigma_t^2}\right) \cdot \left(-\frac{w-m_t y}{\sigma_t}\right) \mu^*|_{\widetilde{x}}(y|x)\,\mathrm{d}y}{\int_{\|y-\widetilde{y}\|_\infty \leq c_4 \sigma_t \sqrt{\log n}} \mathbf{1}(y \in [-1,1]^{D_X}) \exp\left(-\frac{\|w-m_t y\|^2}{2\sigma_t^2}\right) \mu^*|_{\widetilde{x}}(y|x)\,\mathrm{d}y} \right\|$$

$$\leq \left\| \frac{\int_{\|y-\widetilde{y}\|_\infty \leq c_4 \sigma_t \sqrt{\log n}} \mathbf{1}(y \in [-1,1]^{D_X}) \exp\left(-\frac{\|w-m_t y\|^2}{2\sigma_t^2}\right) \cdot \left(-\frac{w-m_t y}{\sigma_t}\right) \mu^*(y|x)\,\mathrm{d}y}{\int_{\|y-\widetilde{y}\|_\infty \leq c_4 \sigma_t \sqrt{\log n}} \mathbf{1}(y \in [-1,1]^{D_X}) \exp\left(-\frac{\|w-m_t y\|^2}{2\sigma_t^2}\right) \mu^*(y|x)\,\mathrm{d}y} \right.$$
$$\left. - \frac{\int_{\|y-\widetilde{y}\|_\infty \leq c_4 \sigma_t \sqrt{\log n}} \mathbf{1}(y \in [-1,1]^{D_X}) \exp\left(-\frac{\|w-m_t y\|^2}{2\sigma_t^2}\right) \cdot \left(-\frac{w-m_t y}{\sigma_t}\right) \mu^*|_{\widetilde{x}}(y|x)\,\mathrm{d}y}{\int_{\|y-\widetilde{y}\|_\infty \leq c_4 \sigma_t \sqrt{\log n}} \mathbf{1}(y \in [-1,1]^{D_X}) \exp\left(-\frac{\|w-m_t y\|^2}{2\sigma_t^2}\right) \mu^*(y|x)\,\mathrm{d}y} \right\|$$

$$+ \left\| \frac{\int_{\|y-\widetilde{y}\|_\infty \leq c_4 \sigma_t \sqrt{\log n}} \mathbf{1}(y \in [-1,1]^{D_X}) \exp\left(-\frac{\|w-m_t y\|^2}{2\sigma_t^2}\right) \cdot \left(-\frac{w-m_t y}{\sigma_t}\right) \mu^*(y|x)\,\mathrm{d}y}{\int_{\|y-\widetilde{y}\|_\infty \leq c_4 \sigma_t \sqrt{\log n}} \mathbf{1}(y \in [-1,1]^{D_X}) \exp\left(-\frac{\|w-m_t y\|^2}{2\sigma_t^2}\right) \mu^*(y|x)\,\mathrm{d}y} \right.$$
$$\left. - \frac{\int_{\|y-\widetilde{y}\|_\infty \leq c_4 \sigma_t \sqrt{\log n}} \mathbf{1}(y \in [-1,1]^{D_X}) \exp\left(-\frac{\|w-m_t y\|^2}{2\sigma_t^2}\right) \cdot \left(-\frac{w-m_t y}{\sigma_t}\right) \mu^*(y|x)\,\mathrm{d}y}{\int_{\|y-\widetilde{y}\|_\infty \leq c_4 \sigma_t \sqrt{\log n}} \mathbf{1}(y \in [-1,1]^{D_X}) \exp\left(-\frac{\|w-m_t y\|^2}{2\sigma_t^2}\right) \mu^*|_{\widetilde{x}}(y|x)\,\mathrm{d}y} \right\|$$

$$\lesssim \sup_{\substack{y \in [-1,1]^{D_X} \\ \|y-\widetilde{y}\|_\infty \leq c_4 \sigma_t \sqrt{\log n}}} \left( \left\| \frac{w-m_t y}{\sigma_t} \right\| \cdot \frac{|\mu^*(y|x) - \mu^*|_{\widetilde{x}}(y|x)|}{\mu^*(y|x)} \right)$$

$$+ \left\| \frac{\int_{\|y-\widetilde{y}\|_\infty \leq c_4 \sigma_t \sqrt{\log n}} \mathbf{1}(y \in [-1,1]^{D_X}) \exp\left(-\frac{\|w-m_t y\|^2}{2\sigma_t^2}\right) \cdot \left(-\frac{w-m_t y}{\sigma_t}\right) \mu^*(y|x)\,\mathrm{d}y}{\int_{\|y-\widetilde{y}\|_\infty \leq c_4 \sigma_t \sqrt{\log n}} \mathbf{1}(y \in [-1,1]^{D_X}) \exp\left(-\frac{\|w-m_t y\|^2}{2\sigma_t^2}\right) \mu^*(y|x)\,\mathrm{d}y} \right\|$$

$$\cdot \left\| \frac{\int_{\|y-\widetilde{y}\|_\infty \leq c_4 \sigma_t \sqrt{\log n}} \mathbf{1}(y \in [-1,1]^{D_X}) \exp\left(-\frac{\|w-m_t y\|^2}{2\sigma_t^2}\right) (\mu^*|_{\widetilde{x}}(y|x) - \mu^*(y|x))\,\mathrm{d}y}{\int_{\|y-\widetilde{y}\|_\infty \leq c_4 \sigma_t \sqrt{\log n}} \mathbf{1}(y \in [-1,1]^{D_X}) \exp\left(-\frac{\|w-m_t y\|^2}{2\sigma_t^2}\right) \mu^*|_{\widetilde{x}}(y|x)\,\mathrm{d}y} \right\|$$

$$\lesssim \sup_{\substack{y \in [-1,1]^{D_X} \\ \|y-\widetilde{y}\|_\infty \leq c_4 \sigma_t \sqrt{\log n}}} \left( \left\| \frac{w-m_t y}{\sigma_t} \right\| \cdot \frac{|\mu^*(y|x) - \mu^*|_{\widetilde{x}}(y|x)|}{\mu^*(y|x)} \right)$$

$$+ \sup_{\substack{y \in [-1,1]^{D_X} \\ \|y-\widetilde{y}\|_\infty \leq c_4 \sigma_t \sqrt{\log n}}} \left\| \frac{w-m_t y}{\sigma_t} \right\| \cdot \sup_{\substack{y \in [-1,1]^{D_X} \\ \|y-\widetilde{y}\|_\infty \leq c_4 \sigma_t \sqrt{\log n}}} \frac{|\mu^*(y|x) - \mu^*|_{\widetilde{x}}(y|x)|}{\mu^*|_{\widetilde{x}}(y|x)}$$

$$\lesssim \sqrt{\log n} \cdot \varepsilon_2^{\alpha_X},$$

where the last inequality uses the lower boundedness of $\mu^*(y|x)$ over $y \in [-1,1]^{D_Y}$. Then notice that for any $(x,w) \in \mathscr{S}_{\widetilde{x}\widetilde{y}}$ and $\|y - \widetilde{y}\| \lesssim \sigma_{\underline{t}}\sqrt{\log n}$,

$$\|w - m_t y\| \leq \|w - \widetilde{y}\| + \|\widetilde{y} - y\| + \|y - m_t y\| \leq C\,\sigma_{\underline{t}}\sqrt{\log n}. \tag{34}$$

Therefore, denote

$$\overline{dp}_t(w,x) = \int_{\|y-\widetilde{y}\|_\infty \leq c_4\sigma_{\underline{t}}\sqrt{\log n}} \mathbf{1}(y \in [-1,1]^{D_X}) \exp\left(-\frac{\|w - m_t y\|^2}{2\sigma_t^2}\right) \cdot \left(-\frac{w - m_t y}{\sigma_t}\right) \mu^*|_{\widetilde{x}}(y|x)\,\mathrm{d}y,$$

and

$$\overline{p}_t(w,x) = \int_{\|y-\widetilde{y}\|_\infty \leq c_4\sigma_{\underline{t}}\sqrt{\log n}} \mathbf{1}(y \in [-1,1]^{D_X}) \exp\left(-\frac{\|w - m_t y\|^2}{2\sigma_t^2}\right) \mu^*|_{\widetilde{x}}(y|x)\,\mathrm{d}y.$$

We can derive

$$\left\|\frac{\overline{dp}_t(w,x)}{\overline{p}_t(w,x)}\right\| \lesssim \sqrt{\log n},$$

and

$$\overline{p}_t(w,x) \gtrsim n^{-C}\sigma_{\underline{t}}^{D_Y}.$$

Therefore, if there exist neural networks $\phi^{[1]}(w,x,t)$ and $\phi^{[2]}(w,x,t)$ so that for any $t \in [\underline{t}, \overline{t}]$ and $(x,w) \in \mathscr{S}_{\widetilde{x}\widetilde{y}}$,

$$\|\overline{dp}_t(w,x) - \phi^{[1]}(w,x,t)\|_\infty = \widetilde{\mathcal{O}}\left((\sigma_{\underline{t}})^{D_Y}\varepsilon_2^{\alpha_X}n^{-C}\right), \tag{35}$$

$$\|\overline{p}_t(w,x) - \phi^{[2]}(w,x,t)\|_\infty = \widetilde{\mathcal{O}}\left((\sigma_{\underline{t}})^{D_Y}\varepsilon_2^{\alpha_X}n^{-C}\right). \tag{36}$$

Then we have

$$\left\|\frac{1}{\sigma_t}\cdot\frac{\overline{dp}_t(x)}{\overline{p}_t(x)} - \frac{1}{\sigma_t}\cdot\frac{\phi^{[1]}(x,t)}{\phi^{[2]}(x,t)}\right\|_\infty = \widetilde{\mathcal{O}}\left(\frac{\varepsilon_2^{\alpha_X}}{\sigma_t}\right). \tag{37}$$

To construct $\phi^{[1]}(w,x,t)$, use (34), by choosing $\mathscr{L} = \Theta(\log n)$, we have

$$\left|\exp\left(-\frac{\|w - m_t y\|^2}{2\sigma_t^2}\right) - \sum_{l_1=0}^{\mathscr{L}}(-1)^{l_1}\frac{\|w - m_t y\|^{2l_1}}{2^{l_1}l_1!\sigma_t^{2l_1}}\right|$$
$$\lesssim n^{-2-C}.$$

Therefore, we have

$$\left\|\overline{dp}_t(w,x) - \int_{\|y-\widetilde{y}\|_\infty \leq c_4\sigma_{\underline{t}}\sqrt{\log n}}\mathbf{1}(y \in [-1,1]^{D_X})\sum_{l_1=0}^{\mathscr{L}}(-1)^{l_1}\frac{\|w - m_t y\|^{2l_1}}{2^{l_1}l_1!\sigma_t^{2l_1}}\cdot\left(-\frac{w - m_t y}{\sigma_t}\right)\mu^*|_{\widetilde{x}}(y|x)\,\mathrm{d}y\right\|$$
$$\lesssim (\sigma_t)^{D_Y}n^{-1-C}. \tag{38}$$

Then notice that $\mu^*|_{\widetilde{x}}(y|x)$ is polynomial in $x$,

$$\int_{\|y-\widetilde{y}\|_\infty \leq c_4\sigma_{\underline{t}}\sqrt{\log n}}\mathbf{1}(y \in [-1,1]^{D_X})\sum_{l_1=0}^{\mathscr{L}}(-1)^{l_1}\frac{\|w - m_t y\|^{2l_1}}{2^{l_1}l_1!\sigma_t^{2l_1}}\cdot\left(-\frac{w - m_t y}{\sigma_t}\right)\mu^*|_{\widetilde{x}}(y|x)\,\mathrm{d}y$$
$$= \sum_{l_1=0}^{\mathscr{L}}\left(\frac{1}{\sigma_t}\right)^{2l_1+1}\sum_{0\leq k\leq 2l_1+1}m_t^k\sum_{i\in\mathbb{N}_0^{D_Y},|i|\leq 2l_1+1}w^{(i)}\sum_{j\in\mathbb{N}_0^{D_X},|j|\leq\lfloor\alpha_X\rfloor}a_{l_1kij}x^{(j)},$$

where $a_{l_1kij} \in \mathbb{R}^{D_Y}$ are some constant coefficients. So similar as the analysis for the manifold setting, there exists networks $\phi^{[1]}(w,x,t) \in \Phi(H,W,R,B)$ with $H = \Theta(\log^4 n)$, $\|W\|_\infty = \widetilde{\Theta}(1)$, $R = \widetilde{\Theta}(1)$, $B = \exp(\Theta(\log^4 n))$ so that (35) holds. Similarly, there exists a neural network $\phi^{[2]}(x,t)$ with the same size as $\phi^{[1]}(x,t)$ so that (36) holds. Then using (37) and Lemmas C.7-C.10, we can obtain Claim 5.

### D.1.2  Case 2: $\tau \le \underline{t} \le n^{-\frac{2}{2\alpha_Y + d_Y + d_X \frac{\alpha_Y}{\alpha_X}}}$

We set $\varepsilon_1 = n^{-\frac{1}{2\alpha_Y + d_Y + d_X \frac{\alpha_Y}{\alpha_X}}}$ and $\varepsilon_2 = n^{-\frac{1}{2\alpha_X + d_X + d_Y \frac{\alpha_X}{\alpha_Y}}}$. Let $\mathcal{N}^X_{\varepsilon_2}$ be one of the largest $\varepsilon_2$-packing of $[-1,1]^{D_X}$ and $\mathcal{N}^Y_{\varepsilon_1}$ be one of the largest $\varepsilon_1$-packing of $[-1,1]^{D_X}$. Then we have $J_1 = |\mathcal{N}^Y_{\varepsilon_1}| \lesssim \varepsilon_1^{-d_Y}$ and $J_2 = |\mathcal{N}^X_{\varepsilon_2}| \lesssim \varepsilon_2^{-d_X}$. Then take an arbitrary $\widetilde{x} \in \mathcal{N}^X_{\varepsilon_2}$ and $\widetilde{y} \in N^Y_{\varepsilon_1}$, Consider set

$$
\begin{aligned}
\mathscr{S}_{\widetilde{x}\widetilde{z}} = \Big\{ (x,w) \ : \ & x \in \mathbb{B}_{[-1,1]^{D_X}}(\widetilde{x}, \sqrt{2}\varepsilon_2), \\
& \|w - \widetilde{y}\| \le C\,\varepsilon_1, \mathrm{dist}(w, [-1,1]^{D_Y}) \le c_0 \sigma_{\underline{t}} \sqrt{\log n} \Big\},
\end{aligned}
$$

we claim that

**Claim 6.** *There exists $\phi^*(w, x, t) \in \Phi\big(H, W, R, B, \Theta(\frac{\sqrt{\log n}}{\sigma_{\underline{t}}})\big)$ with $H = \Theta(\log^4 n)$, $\|W\|_\infty = \widetilde{\Theta}(1)$, $R = \widetilde{\Theta}(1)$, $B = \exp(\Theta(\log^4 n))$, so that for any $(x,w) \in \mathscr{S}_{\widetilde{x}\widetilde{y}}$, and $t \in [\underline{t}, \overline{t}]$,*

$$
\|\phi^*(w, x, t) - \nabla \log p_{t|x}(w)\|_\infty = \widetilde{\mathcal{O}}\Big(\frac{\varepsilon_1^{\alpha_Y}}{\sigma_{\underline{t}}}\Big).
$$

Then the desired result directly follows from Lemmas D.2 and D.3. Now we show Claim 6. For any $(x,w) \in \mathscr{S}_{\widetilde{x}\widetilde{y}}$, we have

$$
\{y \in [-1,1]^{D_Y} \ : \ \|y - w\| \le c_2 \sigma_{\underline{t}} \sqrt{\log n}\} \subset \{y \in [-1,1]^{D_Y} \ : \ \|y - w\|_\infty \le c_2 \sigma_{\underline{t}} \sqrt{\log n}\}.
$$

Moreover, notice that $\mu^*$ is $C^{\alpha_Y, \alpha_X}$-smooth, we can write

$$
\mu^*|_{(\widetilde{y},\widetilde{x})}(y|x) = \sum_{(j_1,j_2) \in \mathcal{J}^{D_Y, D_X}_{\alpha_Y, \alpha_X}} \frac{\mu^*_{(j_1,j_2)}(\widetilde{y}, \widetilde{x})}{j_1! j_2!} (y - \widetilde{y})^{j_1} (x - \widetilde{x})^{j_2}, \tag{39}
$$

where for any $x \in [-1,1]^{D_X}$ and $y \in [-1,1]^{D_Y}$,

$$
\|\mu^*|_{(\widetilde{y},\widetilde{x})}(y|x) - \mu^*(y, x)\| \lesssim \|y - \widetilde{y}\|^{\alpha_Y} + \|x - \widetilde{x}\|^{\alpha_X}.
$$

So we have

$$
\left\|\frac{\int_{\|y-w\|_\infty \le c_2\sigma_t\sqrt{\log n}} \mathbf{1}(y\in[-1,1]^{D_X}) \exp\left(-\frac{\|w-m_t y\|^2}{2\sigma_t^2}\right)\cdot\left(-\frac{w-m_t y}{\sigma_t}\right)\mu^*(y|x)\,\mathrm{d}y}{\int_{\|y-w\|_\infty \le c_2\sigma_t\sqrt{\log n}} \mathbf{1}(y\in[-1,1]^{D_X}) \exp\left(-\frac{\|w-m_t y\|^2}{2\sigma_t^2}\right)\mu^*(y|x)\,\mathrm{d}y}\right.
$$

$$
\left.-\frac{\int_{\|y-w\|_\infty \le c_2\sigma_t\sqrt{\log n}} \mathbf{1}(y\in[-1,1]^{D_X}) \exp\left(-\frac{\|w-m_t y\|^2}{2\sigma_t^2}\right)\cdot\left(-\frac{w-m_t y}{\sigma_t}\right)\mu^*|_{(\widetilde{y},\widetilde{x})}(y|x)\,\mathrm{d}y}{\int_{\|y-w\|_\infty \le c_2\sigma_t\sqrt{\log n}} \mathbf{1}(y\in[-1,1]^{D_X}) \exp\left(-\frac{\|w-m_t y\|^2}{2\sigma_t^2}\right)\mu^*|_{(\widetilde{y},\widetilde{x})}(y|x)\,\mathrm{d}y}\right\|
$$

$$
\le \left\|\frac{\int_{\|y-w\|_\infty \le c_2\sigma_t\sqrt{\log n}} \mathbf{1}(y\in[-1,1]^{D_X}) \exp\left(-\frac{\|w-m_t y\|^2}{2\sigma_t^2}\right)\cdot\left(-\frac{w-m_t y}{\sigma_t}\right)\mu^*(y|x)\,\mathrm{d}y}{\int_{\|y-w\|_\infty \le c_2\sigma_t\sqrt{\log n}} \mathbf{1}(y\in[-1,1]^{D_X}) \exp\left(-\frac{\|w-m_t y\|^2}{2\sigma_t^2}\right)\mu^*(y|x)\,\mathrm{d}y}\right.
$$

$$
\left.-\frac{\int_{\|y-w\|_\infty \le c_2\sigma_t\sqrt{\log n}} \mathbf{1}(y\in[-1,1]^{D_X}) \exp\left(-\frac{\|w-m_t y\|^2}{2\sigma_t^2}\right)\cdot\left(-\frac{w-m_t y}{\sigma_t}\right)\mu^*|_{(\widetilde{y},\widetilde{x})}(y|x)\,\mathrm{d}y}{\int_{\|y-w\|_\infty \le c_2\sigma_t\sqrt{\log n}} \mathbf{1}(y\in[-1,1]^{D_X}) \exp\left(-\frac{\|w-m_t y\|^2}{2\sigma_t^2}\right)\mu^*(y|x)\,\mathrm{d}y}\right\|
$$

$$
+\left\|\frac{\int_{\|y-w\|_\infty \le c_2\sigma_t\sqrt{\log n}} \mathbf{1}(y\in[-1,1]^{D_X}) \exp\left(-\frac{\|w-m_t y\|^2}{2\sigma_t^2}\right)\cdot\left(-\frac{w-m_t y}{\sigma_t}\right)\mu^*(y|x)\,\mathrm{d}y}{\int_{\|y-w\|_\infty \le c_2\sigma_t\sqrt{\log n}} \mathbf{1}(y\in[-1,1]^{D_X}) \exp\left(-\frac{\|w-m_t y\|^2}{2\sigma_t^2}\right)\mu^*(y|x)\,\mathrm{d}y}\right.
$$

$$
\left.-\frac{\int_{\|y-w\|_\infty \le c_2\sigma_t\sqrt{\log n}} \mathbf{1}(y\in[-1,1]^{D_X}) \exp\left(-\frac{\|w-m_t y\|^2}{2\sigma_t^2}\right)\cdot\left(-\frac{w-m_t y}{\sigma_t}\right)\mu^*(y|x)\,\mathrm{d}y}{\int_{\|y-w\|_\infty \le c_2\sigma_t\sqrt{\log n}} \mathbf{1}(y\in[-1,1]^{D_X}) \exp\left(-\frac{\|w-m_t y\|^2}{2\sigma_t^2}\right)\mu^*|_{(\widetilde{y},\widetilde{x})}(y|x)\,\mathrm{d}y}\right\|
$$

$$
\lesssim \sup_{\substack{y\in[-1,1]^{D_X} \\ \|y-w\|_\infty \le c_2\sigma_t\sqrt{\log n}}} \left(\left\|\frac{w-m_t y}{\sigma_t}\right\|\cdot\frac{|\mu^*(y|x)-\mu^*|_{(\widetilde{y},\widetilde{x})}(y|x)|}{\mu^*(y|x)}\right)
$$

$$
+\left\|\frac{\int_{\|y-w\|_\infty \le c_2\sigma_t\sqrt{\log n}} \mathbf{1}(y\in[-1,1]^{D_X}) \exp\left(-\frac{\|w-m_t y\|^2}{2\sigma_t^2}\right)\cdot\left(-\frac{w-m_t y}{\sigma_t}\right)\mu^*(y|x)\,\mathrm{d}y}{\int_{\|y-w\|_\infty \le c_2\sigma_t\sqrt{\log n}} \mathbf{1}(y\in[-1,1]^{D_X}) \exp\left(-\frac{\|w-m_t y\|^2}{2\sigma_t^2}\right)\mu^*(y|x)\,\mathrm{d}y}\right\|
$$

$$
\cdot\left\|\frac{\int_{\|y-w\|_\infty \le c_2\sigma_t\sqrt{\log n}} \mathbf{1}(y\in[-1,1]^{D_X}) \exp\left(-\frac{\|w-m_t y\|^2}{2\sigma_t^2}\right)(\mu^*|_{(\widetilde{y},\widetilde{x})}(y|x)-\mu^*(y|x))\,\mathrm{d}y}{\int_{\|y-w\|_\infty \le c_2\sigma_t\sqrt{\log n}} \mathbf{1}(y\in[-1,1]^{D_X}) \exp\left(-\frac{\|w-m_t y\|^2}{2\sigma_t^2}\right)\mu^*|_{(\widetilde{y},\widetilde{x})}(y|x)\,\mathrm{d}y}\right\|
$$

$$
\lesssim \sup_{\substack{y\in[-1,1]^{D_X} \\ \|y-w\|_\infty \le c_2\sigma_t\sqrt{\log n}}} \left(\left\|\frac{w-m_t y}{\sigma_t}\right\|\cdot\frac{|\mu^*(y|x)-\mu^*|_{(\widetilde{y},\widetilde{x})}(y|x)|}{\mu^*(y|x)}\right)
$$

$$
+\sup_{\substack{y\in[-1,1]^{D_X} \\ \|y-w\|_\infty \le c_2\sigma_t\sqrt{\log n}}} \left\|\frac{w-m_t y}{\sigma_t}\right\|\cdot\sup_{\substack{y\in[-1,1]^{D_X} \\ \|y-w\|_\infty \le c_2\sigma_t\sqrt{\log n}}}\frac{|\mu^*(y|x)-\mu^*|_{(\widetilde{y},\widetilde{x})}(y|x)|}{\mu^*|_{(\widetilde{y},\widetilde{x})}(y|x)}
$$

$$
\lesssim \sqrt{\log n}\cdot\varepsilon_1^{\alpha_Y}.
$$

(40)

For any $s\in[D_Y]$, denote $\mathbf{1}^s=(\mathbf{1}_1^s,\mathbf{1}_2^s,\cdots,\mathbf{1}_{D_Y}^s)$ as the $D_Y$-dimensional vector where the $s$th element

being 1 and other elements being 0 (i.e., $\mathbf{1}_k^s = \mathbf{1}(s = k)$), and denote

$$\widetilde{dp}_{ts}(w, x)$$

$$= \int_{\|y-w\|_\infty \leq c_2 \sigma_{\underline{t}}\sqrt{\log n}} \mathbf{1}(y \in [-1,1]^{D_X}) \exp\left(-\frac{\|w - m_t y\|^2}{2\sigma_t^2}\right) \cdot \left(-\frac{w_s - m_t y_s}{\sigma_t}\right) \mu^*|_{(\widetilde{y},\widetilde{x})}(y|x)\, dy$$

$$= -\frac{1}{\sigma_t} \cdot \sum_{(j_1,j_2)\in\mathcal{J}_{\alpha_Y,\alpha_X}^{D_Y,D_X}} \frac{\mu^*_{(j_1,j_2)}(\widetilde{y},\widetilde{x})}{j_1! j_2!}(x - \widetilde{x})^{j_2}$$

$$\cdot \int_{\|y-w\|_\infty \leq c_2 \sigma_{\underline{t}}\sqrt{\log n}} \mathbf{1}(y \in [-1,1]^{D_X}) \prod_{s_1=1}^{D_Y} \exp(-\frac{(w_{s_1} - m_t y_{s_1})^2}{2\sigma_t^2})(w_{s_1} - m_t y_{s_1})^{\mathbf{1}_{s_1}^s}(y_{s_1} - \widetilde{y}_{s_1})^{j_{1 s_1}}\, dy$$

$$= -\frac{1}{\sigma_t} \cdot \sum_{(j_1,j_2)\in\mathcal{J}_{\alpha_Y,\alpha_X}^{D_Y,D_X}} \frac{\mu^*_{(j_1,j_2)}(\widetilde{y},\widetilde{x})}{j_1! j_2!}(x - \widetilde{x})^{j_2}$$

$$\cdot \prod_{s_1=1}^{D_Y} \int_{-1\vee(w_{s_1}-c_2\sigma_{\underline{t}}\sqrt{\log n})}^{1\wedge(w_{s_1}+c_2\sigma_{\underline{t}}\sqrt{\log n})} \exp(-\frac{(w_{s_1} - m_t y_{s_1})^2}{2\sigma_t^2})(w_{s_1} - m_t y_{s_1})^{\mathbf{1}_{s_1}^s}(y_{s_1} - \widetilde{y}_{s_1})^{j_{1 s_1}}\, dy_{s_1},$$

and

$$\widetilde{p}_t(w, x) = \int_{\|y-w\|_\infty \leq c_2 \sigma_{\underline{t}}\sqrt{\log n}} \mathbf{1}(y \in [-1,1]^{D_X}) \exp\left(-\frac{\|w - m_t y\|^2}{2\sigma_t^2}\right) \mu^*|_{(\widetilde{y},\widetilde{x})}(y|x)\, dy$$

$$= \sum_{(j_1,j_2)\in\mathcal{J}_{\alpha_Y,\alpha_X}^{D_Y,D_X}} \frac{\mu^*_{(j_1,j_2)}(\widetilde{y},\widetilde{x})}{j_1! j_2!}(x - \widetilde{x})^{j_2} \cdot \prod_{s_1=1}^{D_Y} \int_{-1\vee(w_{s_1}-c_2\sigma_{\underline{t}}\sqrt{\log n})}^{1\wedge(w_{s_1}+c_2\sigma_{\underline{t}}\sqrt{\log n})} \exp(-\frac{(w_{s_1} - m_t y_{s_1})^2}{2\sigma_t^2})(y_{s_1} - \widetilde{y}_{s_1})^{j_{1 s_1}}\, dy_{s_1}.$$

Then notice that for any $(x, w) \in \mathscr{S}_{\widetilde{x}\widetilde{y}}$, and any $y \in [-1,1]^{D_Y}$ satisfying $\|y - w\|_\infty \leq c_2 \sigma_{\underline{t}}\sqrt{\log n}$:

$$\|w - m_t y\| \leq \|w - y\| + \|y - m_t y\| \leq C\, \sigma_{\underline{t}}\sqrt{\log n},$$

we can derive

$$\left\| \frac{\widetilde{dp}_t(w, x)}{\widetilde{p}_t(w, x)} \right\| \lesssim \sqrt{\log n},$$

and

$$\widetilde{p}_t(w, x) \gtrsim n^{-C}(\sigma_{\underline{t}})^{D_Y}.$$

Therefore, if there exist neural networks $\phi^{[1]}(w, x, t)$ and $\phi^{[2]}(w, x, t)$ so that for any $t \in [\underline{t}, \overline{t}]$ and $(x, w) \in \mathscr{S}_{\widetilde{x}\widetilde{y}}$,

$$\|\widetilde{dp}_t(w, x) - \phi^{[1]}(w, x, t)\|_\infty = \widetilde{\mathcal{O}}\left((\sigma_{\underline{t}})^{D_Y}(\varepsilon_1^{\alpha_Y})n^{-C}\right), \tag{41}$$

$$\|\widetilde{p}_t(w, x) - \phi^{[2]}(w, x, t)\|_\infty = \widetilde{\mathcal{O}}\left((\sigma_{\underline{t}})^{D_Y}\varepsilon_1^{\alpha_Y}n^{-C}\right). \tag{42}$$

Then we have

$$\left\| \frac{1}{\sigma_t}\cdot\frac{\widetilde{dp}_t(x)}{\widetilde{p}_t(x)} - \frac{1}{\sigma_t}\cdot\frac{\phi^{[1]}(x,t)}{\phi^{[2]}(x,t)} \right\|_\infty = \widetilde{\mathcal{O}}\left(\frac{\varepsilon_2^{\alpha_X} + \varepsilon_1^{\alpha_Y}}{\sigma_{\underline{t}}}\right). \tag{43}$$

Then we construct $\phi^{[1]}(w, x, t)$ by approximating $\widetilde{dp}_t(w, x)$ with polynomials. By choosing $\mathscr{L} = \Theta(\log n)$, we have for any $s_1 \in [D_Y]$,

$$\left| \exp\left(-\frac{(w_{s_1} - m_t y_{s_1})^2}{2\sigma_t^2}\right) - \sum_{l_1=0}^{\mathscr{L}} (-1)^{l_1}\frac{(w_{s_1} - m_t y_{s_1})^{2l_1}}{2^{l_1} l_1! \sigma_t^{2l_1}} \right| \lesssim n^{-2-C}.$$

Then for any $s \in [D_Y]$, we can write

$$-\frac{1}{\sigma_t} \cdot \sum_{(j_1,j_2) \in \mathcal{J}_{\alpha_Y,\alpha_X}^{D_Y,D_X}} \frac{\mu^*{}_{(j_1,j_2)}(\widetilde{y},\widetilde{x})}{j_1!j_2!}(x-\widetilde{x})^{j_2}$$

$$\cdot \prod_{s_1=1}^{D_Y} \int_{-1 \vee (w_{s_1}-c_2\sigma_t\sqrt{\log n})}^{1 \wedge (w_{s_1}+c_2\sigma_t\sqrt{\log n})} \sum_{l_1=0}^{\mathscr{L}} (-1)^{l_1} \frac{(w_{s_1}-m_t y_{s_1})^{2l_1}}{2^{l_1}l_1!\sigma_t^{2l_1}}(w_{s_1}-m_t y_{s_1})^{\mathbf{1}_{s_1}^s}(y_{s_1}-\widetilde{y}_{s_1})^{j_{1s_1}}\,\mathrm{d}y_{s_1}$$

$$= -\frac{1}{\sigma_t} \cdot \sum_{(j_1,j_2) \in \mathcal{J}_{\alpha_Y,\alpha_X}^{D_Y,D_X}} \frac{\mu^*{}_{(j_1,j_2)}(\widetilde{y},\widetilde{x})}{j_1!j_2!}(x-\widetilde{x})^{j_2}$$

$$\cdot \prod_{s_1=1}^{D_Y} \sum_{l_1=0}^{\mathscr{L}_1} (\frac{1}{\sigma_t})^{2l_1} \sum_{0 \le k \le 2l_1+1} m_t^k$$

$$\sum_{0 \le s_2 \le 2+j_{1s_1}} (-1 \vee (w_{s_1}-c_2\sigma_t\sqrt{\log n}))^{s_2} \sum_{0 \le s_3 \le 2+j_{1s_1}} (1 \wedge (w_{s_1}+c_2\sigma_t\sqrt{\log n}))^{s_3} \sum_{0 \le i \le 2l_1+1} a_{sj_1 s_1 l_1 k s_2 s_3 i} w_{s_1}^i.$$

So similar as the analysis for the manifold setting, there exists networks $\phi^{[1]}(w,x,t) \in \Phi(H,W,R,B)$ with $H = \Theta(\log^4 n)$, $\|W\|_\infty = \widetilde{\Theta}(1)$, $R = \widetilde{\Theta}(1)$, $B = \exp(\Theta(\log^4 n))$ so that (41) holds. Similarly, there exists a neural network $\phi^{[2]}(x,t)$ with the same size as $\phi^{[1]}(x,t)$ so that (42) holds. Then using (40), (43), and Lemmas C.7-C.10, we can obtain Claim 6.