# OpenReview forum: "Conditional Diffusion Models are Minimax-Optimal and Manifold-Adaptive for Conditional Distribution Estimation"
_ICLR.cc/2025/Conference — ICLR 2025 Poster_

### Official Review · Reviewer_YRRd · 2024-11-01

**Soundness:** 3
**Presentation:** 3
**Contribution:** 3
**Rating:** 6
**Confidence:** 4

**Summary:**

This paper uses distribution regression to estimate the conditional distribution induced by conditional forward-backward diffusion models. Authors utilize neural network to estimate the key points and also allow for high-dimensional X and Y with low-dimensional structures. Solid theoretical results are provided.

**Strengths:**

1. Authors introduce SDE in a simple way and made it easy to connect SDE to distribution regression.

2. The paper extends the unconditional distribution estimator to the conditional one and provides theoretical results. Theories are given in an elegant way that they clearly explain the connections to the theories of unconditional distribution estimator, mean regression problems, and nonparametric regression under l2 error. Theoretical results are interesting and inspiring.

**Weaknesses:**

The paper mentions that it provides practical guidance to design the neural network, including the network size, smoothness level and so on. Though it is a theoretical work, some empirical results would significantly demonstrate the usefulness of the paper. I would recommend authors run some simulations to show how the theoretical results guide the design of neural networks. For example, under what scenarios, the errors are optimally controlled?

The theoretical results are interesting.  Can the authors highlight the difficulties and summarize the new ways of proof if there are? This could be a huge contribution to the statistical learning community. I didn't see the challenges from my end and all seem to be standard.

**Questions:**

What is p_data on Line 122?

Authors need to check all the parenthetical and narrative citations throughout the whole paper. Many were mis-used. For example, Line 316 should be parenthetical citation.

Definition of neural network class is already given in Tang and Yang, 2024 (AISTATS). Authors need to reference it or directly call it from the citation.

In remark 1, can authors discuss the case when alpha_X>1? What does this mean to the minimax rate?

In Remark 2, can the authors clarify why D_Y need to satisfy this condition to have the first term dominate the second one?




Rong Tang and Yun Yang. Adaptivity of diffusion models to manifold structures. 27th International Conference on Artificial Intelligence and Statistics, 2024.

---

> ### Author Response · Authors · 2024-11-19
>
> **Point 1: simulation and practical guidance.**
>
> Reply: Thank you for your valuable suggestions.  We have included a simulation in our paper, which can be found in Appendix A of the supplementary material. Please see our global comment for more details.
>
> **Point 2: Technical difficulties.**
>
> Reply:    One main technical challenge in extending the analysis from the unconditional case to the conditional case is developing new analytical approaches to bound the approximation error of the conditional score function using neural networks due to the inclusion of the extra covariate variable $X\in\mathcal M_X$. In particular, as we assume the support of $Y|X=x$ to be a general, unknown nonlinear submanifold $\mathcal M_{Y|X=x}$   dependent on the covariate value $X=x$,      the task of handling unknown supports for the
> conditional distributions requires us to deal with infinitely many unknown and nonlinear submanifolds   $\mathcal M_{Y|X=x}$ with $x\in \mathcal{M}_X$. This poses a significant technical challenge that we tackle in our work. In particular, we need to partition the joint space $\mathcal{M}$ of $(X,Y)$ into small pieces with varying resolution levels in $X$ and $Y$, tailored to times $t$: the resolution level in $Y$ should decrease with increasing $t$, reflecting the diminished importance of finer details in $Y$ due to noise injection over time; while the resolution level in $X$ should increase with $t$, emphasizing the growing significance of the global dependence on $X$ as finer details in $Y$ are smoothed out.
>
> **Point 3: What is $p_{data}$ on Line 122?**
>
> Reply: Apologies for the confusion; there was a typo. The term $p_{\rm data}$ should refer to the target distribution, which is correctly denoted as $\mu^*$.
>
> **Point 4: In remark 1, can authors discuss the case when $\alpha_X>1$? What does this mean to the minimax rate?**
>
> Reply: The rate is strictly decreasing in terms of $\alpha_X$, so allowing for a larger $\alpha_X$ can lead to a faster convergence rate. We expect that a similar phenomenon to occur in the minimax rate as well.
>
> **Point 5: In Remark 2, can the authors clarify why $D_Y$ need to satisfy this condition to have the first term dominate the second one?**
>
> Reply: Since the two terms are
>          $$n^{-\frac{1}{2+\frac{D_X}{\alpha_X}}}\quad \text{and}\quad n^{-\frac{1+\frac{1}{\alpha_Y}}{2+\frac{D_X}{\alpha_X}+\frac{D_Y}{\alpha_Y}}},$$  the transition boundary is $\frac{\frac{1}{\alpha_Y}}{\frac{D_Y}{\alpha_Y}}=\frac{1}{D_Y}=\frac{1}{2+\frac{D_X}{\alpha_X}}$. Thus the dominant term is $n^{-\frac{1+\frac{1}{\alpha_Y}}{2+\frac{D_X}{\alpha_X}+\frac{D_Y}{\alpha_Y}}}$ if $D_Y \geq 2+\frac{D_X}{\alpha_X}$; and the dominant is $n^{-\frac{1}{2+\frac{D_X}{\alpha_X}}}$ if $D_Y\leq 2+\frac{D_X}{\alpha_X}$.   The underlying reason for this transition is that  a larger $D_Y$ increase the complexity of estimating the conditional density of $Y$, making the task of conditional distribution estimation (distribution regression) in Wasserstein distance  more challenging than nonparametric mean regression.
>
>
> **Other comments.**
>
>   Thank you for pointing out these important issues. We will address them in the revised version of our main text.

---

### Official Review · Reviewer_2asE · 2024-11-03

**Soundness:** 4
**Presentation:** 4
**Contribution:** 3
**Rating:** 6
**Confidence:** 4

**Summary:**

- The paper presents a derivation of the estimation error bound for the diffusion models with sampling guided by the covariate regressor. The authors closely build upon the methodology from Oko et al 2023 https://proceedings.mlr.press/v202/oko23a/oko23a.pdf who explored the statistical learning theory for diffusion models via score matching estimators and proved nearly minimax optimal estimation rates in total variation and Wasserstein metric of order 1 for fully connected neural-networks (including suggestions on generalisation to U-nets and ConvNets).

- The presented paper aims to introduce the statistical learning theory for conditional diffusion models using the distribution regression of the covariate structure guiding the sampling. Authors investigate the properties of the estimators using the finite sample analysis of the statistical error bound under various metrics, the relations of latent lower dimensional data structures and the smoothness properties of the data.

- The main contribution is:
   1) Given the assumptions on the size of the architecture,  the estimation error bound of the conditional diffusion estimator is minimax optimal under total variation and Wasserstein distance of order 1, authors include the discussion on the generalization to the unconditional case as a special case,
   2) if there is a low-dimensional latent space structure within the high-dimensional data manifold (say within the feature space or feature and label space), the resulting conditional distribution estimators adapts to the lower-dimensional latent space.

**Strengths:**

- The paper is mathematically well written. The notation is consistent and precise. Definitions and proofs are provided in the extensive Appendix. These can be useful for deriving and investigating other novel properties and algorithms.
- The problem is relevant and well chosen and the previous work is well mapped.
- There is an interesting potential applications for measuring the quality of the conditional diffusion estimators and guided  sampling. The paper also  provides a background for the reasoning about the selection of the model architecture.
- The interesting contribution is in analyzing the latent dimension representations of the data and the adaptability of the higher dimensional estimator.
- The results provide insight into the impact of the smoothness of the underlying data manifold on the performance of the estimator.

**Weaknesses:**

1) Paper is missing motivational examples and practical experiments to demonstrate how the presented theory can be used on real-life problems. The results are theoretically interesting and there is no issue with the soundness of the arguments. Nevertheless, without a practical demonstration on how to use the derived results, the main contribution of the paper may be harder to access to the wider machine learning community.

2) While it is great to see the methodology related to the proofs as a part of the main paper, it would be great to see particular direct applications in the main body of the paper.

3) The usual approach to estimate the gradients of log p_t(y_t|y) (see line 200) is to use the gradient of unconditional model and add gradient of log-likelihood of the ‘reverse predictor' which detects ‘the covariate values’ x based on the features constructed from whom the unconditional values ‘y_t’. Authors decided to use marginal forward diffusion which results in empirical estimation of the joint density between the covariate X (the guiding covariates) and Y (the unconditional diffusion). The discussion on incorporating computational efficiency of this approach in comparison to the 'reverse predictor' approach is missing.
4) The proven Theorems assume the initial constraints of the sizes of the architecture such as depth, width, norm of the parameters, etc. These constraints are set as functionals of the order of magnitude of the data sample $n$. It is not well discussed what happens if these constraints are not satisfied.

**Questions:**

1) Are the covariates X also located in the Besov space support?
2) The marginals M_X and M_Y are assumed to be  [-1,1]  D-X, resp. D_Y dimensional cube. How does the scaling of the data to the [-1, 1] cube impact some other intrinsic features, e.g. latent translations? What type of scaling transformations would you recommend?
3) Can you provide guidelines on how you estimate $\alpha_X$ and $\alpha_Y$ or do you assume these values?
4) How does presented results in Theorem 2 relate to latent diffusion models?

5) p.7 Remark 1: Does this remark imply that the kernel estimator can well approximate the performance of the conditional diffusion based estimator?

6) It would be great if authors can  include experiments for different embedded lower dimensional structures, and demonstrate how the presented theory supports and navigates the experimentally obtained results. Using even simple synthetic geometric objects would be great.

7) Can you provide more discussion on the implications of the different regimes discussed in Remark 3? In particular how would you select the smoothness parameters $\alpha_x$ to investigate the transition phases?

8 )   How does Lemma 2 helps to assess the generalization error?

---

> ### Author Response · Authors · 2024-11-19
>
> Point 1:  Practical experiments.
>
> Thank you for your valuable suggestions. We have included a simulation in our paper, which can be found in Appendix A of the updated supplementary material.  Please see our global comments for more details.
>
> Point 2: The discussion on "reverse predictor" approach.
>
> Given that the training objective we are considering differs from that of the unconditional diffusion model only in the input of the score network—by the addition of covariates—it should not increase much on the computational efficiency. However, the classifier-guidance method requires extra effort to learn the external classifier $x|y_t$. Furthermore, from a theoretical perspective, our goal is not to learn the score function of the joint density of $X$ and $Y$, but rather the conditional density of $Y$ given $X$. Therefore, we **do not** need to impose any smoothness assumptions to the marginal density of $X$, but only need to assume that $Y$ smoothly depends on $X$. However, for learning the classifier $p(x|y_t)=p(x)p(y_t|x)/p(y_t)$ in the classifier-guidance method, we might need to assume that the marginal density $p(x)$ of $X$ is also smooth, an assumption that can be avoided in the classifier-free method we are considering.
>
> Point 3: Constraints to NN and implication of Lemma 2.
>
> Reply: Thank you for raising this important point.  This architecture is designed so that the estimation error and approximation can be well balanced.  If we are choose a NN larger  than  the specified size, we will achieve a smaller approximation error, i.e., the term $\min_{S\in \mathcal S_i} \mathbb{E} \int_{t_{i-1}}^{t_i} \int_{\mathbb{R}^{D_Y}} || S(w,x,t)-\nabla \log p_t(\cdot\,|\,x)(w)||^2 p_t(\cdot\,|\,x)(w) dwdt$ in Lemma 2, but will also lead to a larger estimation error (or excess risk), i.e., the term  $\frac{R_iH_i \log \big(R_iH_i\|W_i\|_{\infty}(B_i \vee 1) n\big) (\log n)^2}{n}$ in Lemma 2.   Consequently, this results in a larger total generalization error when these terms are summed. Similarly, selecting a NN smaller than the specified size will lead to a smaller excess risk but a larger approximation error due to the reduced capacity of the NN. This trade-off also culminates in a larger total generalization error. In summary, if these constraints are not satisfied,  we may end up with an estimator that exhibits  a larger error in terms of the expected Wasserstein distance compared with Theorem 2.
>
> Point 4:  Assumption on data space and smoothness.
>
> Sorry for the confusion,  we only need the space $\mathcal M_X$  to have a upper Minkowski dimension $d$; it is not necessary to impose any smoothness conditions on $\mathcal M_X$ (e.g., being a smooth manifold).  The assumption of $\mathcal M_X=[-1,1]^{D_X}$ and $\mathcal M_Y=[-1,1]^{D_Y}$ in Theorem 1 is only for technical simplicity. The similar technique may also apply to, for example, unit ball, or more generally, compact set with smooth boundary.  We assume that we know $\alpha_X$ and $\alpha_Y$ to set the architecture of NN. Theoretically, it may also possible to construct estimators that adaptive to smoothness level using, for example, the  Lepski method.
>
> Point 5: Relate to latent diffusion models.
>
> In Theorem 2,  we consider the diffusion model that operates directly within the ambient space. Compared with the latent diffusion, it is perhaps less intuitive why this model can effectively adapt to the manifold structure of the data, which serves as a key motivation for the current research described in Theorem 2. For the latent diffusion model, the analysis may be divided into two part: one is the analysis for the encoder-decoder learning, and the second one is the analysis of the score matching in the encoded space, which may be treated as the Euclidean case we considered in Theorem 1.
>
> Point 6: KDE estimator
>
> The KDE is demonstrated to achieve the same rate as our equation (10) only within the Euclidean space,  where both $X$ and $Y$ lack any manifold structure. While our Theorem 2 allows both $X$ and $Y$ to have low-dimensional structures. Therefore, compared to KDE, the conditional diffusion model can adapt to the underlying manifold structures, leading to better performance in such scenarios.
>
> Point 7: transition phases.
>
> For the transition phase in terms of smoothness parameters, since the transition boundary is $\frac{\frac{1}{\alpha_Y}}{\frac{D_Y}{\alpha_Y}}=\frac{1}{D_Y}=\frac{1}{2+\frac{D_X}{\alpha_X}}$, the transition boundary is independent of $\alpha_Y$. In terms of $\alpha_X$, the dominant term is $n^{-\frac{1+\frac{1}{\alpha_Y}}{2+\frac{D_X}{\alpha_X}+\frac{D_Y}{\alpha_Y}}}$ if $\alpha_X\geq \frac{D_X}{D_Y-2}$ with $D_Y\geq 2$; and the dominant is $n^{-\frac{1}{2+\frac{D_X}{\alpha_X}}}$ if $\alpha_X\leq \frac{D_X}{D_Y-2}$ or $D_Y\leq 2$.  The reason is that when $\alpha_X$ is small, the bottleneck will be estimating the global dependence of $Y$ on $X$, so the dominant term will be the minimax rate of nonparametric regression.

---

> > ### Comment · Reviewer_2asE · 2024-11-27
> >
> > I would like to thank authors for their response. I will keep my score unchanged as I would appreciate experimental section. I however believe the paper provides good theoretical background for conditional diffusion models, its contribution is novel, potentially impactful and deserves to be presented to the wider machine learning community at the ICLR conference.

---

### Official Review · Reviewer_x4xs · 2024-11-03

**Soundness:** 3
**Presentation:** 3
**Contribution:** 2
**Rating:** 8
**Confidence:** 3

**Summary:**

Previous work has studied the min-max optimality of score-based diffusion models in estimating the density of an unknown distribution, and in estimating the intrinsic dimension of data lying on a submanifold of a sample space.

This work extends these results to the conditional modelling setting, where a conditioning variate changes the distribution being modeled. The distribution is assumed to vary smoothly with the conditioning variate.

These results show that very similar rates of convergence of the total variation and Wasserstein distances between the true and  estimated data distributions hold to the unconditional modeling setting, with additional terms depending on the dimensionality of the conditioning variate. In the limit of the conditioning variate having dimension zero, the same rates as previous works on the unconditional setting are obtained.

**Strengths:**

- The contributions are, while possibly not surprising, are meaningful and interesting results extending the theoretical analysis of diffusion models.
- The paper is generally very well written and easy to follow, given the technical nature of the content. In particular the remarks on the theorems make understanding the implications of them more accessible.

**Weaknesses:**

- The practical implications of the theoretical work presented are not completely clear to me. The authors mention that the results suggest how to design networks, but from my understanding on theorem 2, this suggests simply that larger networks produce better bounds?
- The implications of assumption D are unclear to me. The variable $r_0$ is undefined. Is this condition meant to hold for any $r_0$? If so then this implies a need for the decoding to be smooth globally, with I believe would limit topological structure of the low-dimension data manifold. If this is local condition, the is this not a consequence of the smooth manifold structure of $\mathcal{M}$?

**Questions:**

Please see the weaknesses section.

---

> ### Author Response · Authors · 2024-11-19
>
> Point 1: practical implications of the theoretical work
>
> Reply: One guidance for designing the neural network is to utilize a smaller-sized neural network for approximating score functions corresponding to larger values of  $t$. This is motivated by the smoothing effect of Gaussian noise injection during the forward diffusion. So, we may consider a neural network class whose size diminishes over time, such as the piece-wise constant complexity neural network class described in equation (1):
>
> $$S(y,x,t)=\sum_{i=1}^{\mathcal{I}} S_i(y,x, t) \cdot \mathbf{1}\left(t_{i-1} \leq t<t_{i}\right) ,$$
>
> where for each $i \in [\mathcal{I}] $, $S_i $ is a ReLU neural network whose size decreases as $i$ increases. A simulation study included in Appendix A examines the performance of this piecewise NN structure in toy examples. Please see our global comments for details. The size of each ReLU neural network should be carefully chosen to achieve a balance between approximation error and estimation error: larger sizes increase the capacity of the NN, thus reducing the approximation error, but they also introduce larger random fluctuation in the training objective and increase the estimation error. The optimal size of each ReLU neural network should achieve a balance between these two errors, and will grow with  $n$. Explicit expressions for the optimal sizes are provided in Theorem 2.
>
>
> Point 2:   Implications of assumption D.
>
> Reply: Sorry for the confusion. Here the assumption means that there **exists** a positive constant $r_0$, so that for any $x_0\in \mathcal{M}$,  an encoder-decoder structure can be **locally** defined over $x \in \mathbb{B}_{\mathcal{M}_X}\left(x_0, r_0\right)$.  Here, the incorporation of the terminologies "encoder" and  "decoder"  is primarily intended to enhance clarity.  The formal mathematical definition of a manifold is defined through local charts/parametrizations, which can be interpreted as an encoder-decoder pair and enable us to define the manifold smoothness with respect to the response and covariate ($\beta_Y$ and $\beta_X$).

---

### Author Response · Authors · 2024-11-19
**Global comments**

**Examples to illustrate the contribution.**  We have included a simulation in our paper, which can be found in Appendix A of the supplementary material. In this toy example, we consider a   response $Y$ generated by the process $$Y = ((1 + 1.5 \cos(x)) \cos(\phi), (1 + 1.5 \sin(x)) \sin(\phi), 1.5 \sin(x)),$$  and  aim at estimating  $Y|x$ and $Y|\phi$.   We employ two neural network classes to approximate the conditional score functions: a standard ReLU NN and a piecewise ReLU NN. For the piecewise ReLU NN, the total time horizon is partitioned into $5$ intervals, with a ReLU NN used to model the conditional score within each interval (c.f. Section 2.3), which aligns with the structure suggested and employed in our theoretical analysis. The width of the hidden layers is selected to ensure comparability of training parameters between the two types of conditional score families.

Through simulations, we observe that for different $x$ (or $\phi$) values, the generated response $Y$ from the fitted conditional diffusion model will concentrate around different ellipse (or tilted ellipse). This indicates that the conditional diffusion model effectively captures both the covariate information and the underlying manifold structure. Additionally, we assess the Maximum Mean Discrepancy (MMD) distance between the generated data and the true conditional distribution across various covariate values, as well as the average across all covariate values. Consistent with our theoretical findings, incorporating the piecewise structure into the neural network results in a noticeably reduced MMD distance. Below, we provide the MMD results for your convenience.


 | Condition          | Piecewise NN | NN    | Condition            | Piecewise NN | NN    |
|--------------------|--------------|-------|----------------------|--------------|-------|
| **$x = 0$**        | 0.0023       | 0.0032| **$\phi = 0$**       | 0.0002       | 0.0034|
| **$x = 0.5$**      | 0.0013       | 0.0014| **$\phi = 0.5$**     | 0.0009       | 0.0013|
| **$x = 1$**        | 0.0007       | 0.0017| **$\phi = 1$**       | 0.0014       | 0.0028|
| **$x \sim Unif[0,1]$** | 0.0021   | 0.0031| **$\phi \sim Unif[0, 2\pi]$** | 0.0008 | 0.0015|

*Table: The table presents the MMD between the conditional diffusion model estimator and the target conditional distribution. The first three rows show the MMD for the conditional distributions $Y \mid x = 0$, $Y \mid x = 0.5$, and $Y \mid x = 1$ as well as $Y \mid \phi = 0$, $Y \mid \phi = 0.5$, and $Y \mid \phi = 1$. The last row displays the expected conditional MMD, where the expectation is taken with respect to $x$ sampled from a uniform distribution on $[0,1]$ and $\phi$ sampled from uniform over $[0,2\pi]$.*



We will also add a discussion in our main text to emphasize this simulation:



*We have also conducted a simulation study (see Appendix A) to demonstrate the effectiveness of this theoretically guided neural network architecture compared to a standard single ReLU neural network (across both space and time). In this experiments,  we consider cases where, given the covariate $X$, the response $Y$ is supported on different (tilted) ellipses depending on the values of the covariate.   Consistent with our theoretical findings,  the simulations show that incorporating  the piecewise structure into the neural network results in a more accurate estimation of the conditional distribution.*

---

### Meta-Review · Area_Chair_Wr8A · 2024-12-21

**Metareview:**

This paper studies conditional diffusion models for generative modeling, focusing on generating data given a covariate. Using a statistical framework of distribution regression, the authors analyze the large-sample properties of conditional distribution estimators under the assumption that the conditional distribution changes smoothly with the covariate. They derive a minimax-optimal convergence rate under the total variation metric, consistent with existing literature. The analysis is further extended to scenarios where both data and covariates lie on low-dimensional manifolds, showing that the model adapts to these structures. The resulting error bounds under the Wasserstein metric depend only on the intrinsic dimensionalities of the data and covariates. This paper has received unanimous support from the reviewers. Therefore, I recommend acceptance.

**Additional Comments On Reviewer Discussion:**

Even before the rebuttal, this paper received unanimous support from the reviewers.

---

### Decision · Program_Chairs · 2025-01-22

Accept (Poster)